# Conditional Distribution Compression via the Kernel Conditional Mean Embedding

**Dominic Broadbent**
School of Mathematics
University of Bristol
Bristol, United Kingdom
dominic.broadbent@bristol.ac.uk

**Nick Whiteley**
School of Mathematics
University of Bristol
Bristol, United Kingdom

**Robert Allison**
School of Mathematics
University of Bristol
Bristol, United Kingdom

**Tom Lovett**
Mathematical Institute
University of Oxford
Oxford, United Kingdom

## Abstract

Existing distribution compression methods, like Kernel Herding (KH), were originally developed for unlabelled data. However, no existing approach directly compresses the conditional distribution of *labelled* data. To address this gap, we first introduce the *Average Maximum Conditional Mean Discrepancy* (AMCMD), a metric for comparing conditional distributions, and derive a closed form estimator. Next, we make a key observation: in the context of distribution compression, the cost of constructing a compressed set targeting the AMCMD can be reduced from $\mathcal{O}(n^3)$ to $\mathcal{O}(n)$. Leveraging this, we extend KH to propose *Average Conditional Kernel Herding* (ACKH), a linear-time greedy algorithm for constructing compressed sets that target the AMCMD. To better understand the advantages of *directly* compressing the conditional distribution rather than doing so via the joint distribution, we introduce *Joint Kernel Herding* (JKH), an adaptation of KH designed to compress the joint distribution of labelled data. While herding methods provide a simple and interpretable selection process, they rely on a greedy heuristic. To explore alternative optimisation strategies, we also propose *Joint Kernel Inducing Points* (JKIP) and *Average Conditional Kernel Inducing Points* (ACKIP), which *jointly* optimise the compressed set while maintaining linear complexity. Experiments show that directly preserving conditional distributions with ACKIP outperforms both joint distribution compression and the greedy selection used in ACKH. Moreover, we see that JKIP consistently outperforms JKH.

## 1 Introduction

Given a large unlabelled dataset $\mathcal{D} := \{\boldsymbol{x}_i\}_{i=1}^n \subset \mathcal{X}$ sampled i.i.d. from the distribution $\mathbb{P}_X$, a major challenge is constructing a compressed set, $\mathcal{C} := \{\tilde{\boldsymbol{x}}_j\}_{j=1}^m$ with $m \ll n$, that preserves the essential statistical properties of the original data. This compressed set can then replace the full dataset in downstream tasks, significantly reducing computational costs while maintaining statistical fidelity. Existing distribution compression algorithms leverage the theory of *Reproducing Kernel*

39th Conference on Neural Information Processing Systems (NeurIPS 2025).

*Hilbert Spaces* (RKHS) [1] to embed distributions into function space. Specifically, these methods minimise the *Maximum Mean Discrepancy* (MMD) [2] between the true *kernel mean embedding* of the distribution $\mathbb{P}_X$, denoted $\mu_X$, and the kernel mean embedding estimated with the compressed set, denoted $\tilde{\mu}_X$. This ensures that the empirical distribution of the compressed set, $\tilde{\mathbb{P}}_X$, remains close to the true distribution $\mathbb{P}_X$ in terms of MMD.

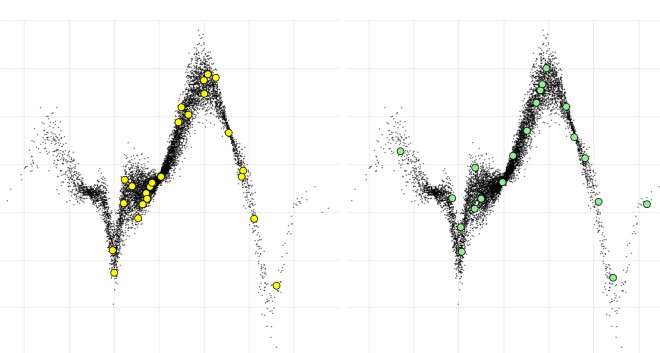

Figure 1: Compressed set of size $m = 25$ generated by ACKIP (green), initialised with uniformly at random subsample (yellow).

Several well-known distribution compression algorithms include Kernel Herding [3], Kernel Quadrature [4–9], Support Points [10], Gradient Flow [11], and Kernel Thinning [12–14]. Kernel Herding was the first method proposed for distribution compression and remains one of the most intuitive approaches. It is a greedy algorithm that iteratively constructs a compressed set via gradient descent, optimising *super-samples* that are not part of the original dataset. Kernel Herding was originally designed to compress distributions over unlabelled data. However, we demonstrate that it can be adapted to compress the joint distribution $\mathbb{P}_{X,Y}$ of a *labelled* dataset $\mathcal{D} := \{(\boldsymbol{x}_i, \boldsymbol{y}_i)\}_{i=1}^n \subset \mathcal{X} \times \mathcal{Y}$ using the theory of tensor product RKHSs. This is achieved by optimising a compressed set $\mathcal{C} := \{(\tilde{\boldsymbol{x}}_j, \tilde{\boldsymbol{y}}_j)\}_{j=1}^m$ targeting the *Joint Maximum Mean Discrepancy* (JMMD) [15]. In a similar fashion, we also adapt the gradient flow approach of [16], which optimises all points in the compressed set jointly.

No existing distribution compression method targets the family of conditional distributions $\mathbb{P}_{Y|X}$ of a labelled dataset. To address the gap, we require the *kernel conditional mean embedding* (KCME), denoted $\mu_{Y|X}$. The KCME provides a way to embed the family of conditional distributions $\mathbb{P}_{Y|X}$ into an RKHS, and is a widely used technique for non-parametric modelling of complex conditional distributions. Given $n$ labelled samples, it can be consistently estimated with a computational cost of $\mathcal{O}(n^3)$ [17, 18]. Despite this high cost, the KCME has been successfully applied in various fields, including conditional distribution testing [18, 19], conditional independence testing [18, 20], conditional density estimation [21–23], likelihood-free inference [24], Bayesian optimisation [25], probabilistic inference [17, 22, 26–28], calibration of neural networks [29], reinforcement learning [30–32], and as a consistent multi-class classifier [33]. Intuitively, we posit that directly compressing the conditional distribution should be preferable to indirect joint compression, much as direct conditional density estimation outperforms approaches based on separate joint and marginal estimates [34]. Proofs of all our theoretical results are in Section B.

**Our key contributions are**:

- In Section 4.1, we define the *Average Maximum Conditional Mean Discrepancy* (AMCMD), show that it satisfies the properties of a proper metric on the space of conditional distributions, and derive a closed form estimate.
- In Section 4.2, we make a crucial observation: the cost of estimating the AMCMD, excluding terms irrelevant for distribution compression, can be reduced from $\mathcal{O}(n^3)$ to $\mathcal{O}(n)$ via application of the tower property.
- This observation enables the development of *Average Conditional Kernel Herding* (ACKH), a linear-time algorithm which constructs a compressed set such that $\tilde{\mathbb{P}}_{Y|X=\boldsymbol{x}} \approx \mathbb{P}_{Y|X=\boldsymbol{x}}$ a.e. $\boldsymbol{x}$ wrt $\mathbb{P}_X$. Furthermore, in Section 4.3, we propose *Average Conditional Kernel Inducing Points* (ACKIP) as a *non-greedy*, linear-time alternative that *jointly* optimises the compressed set to the same end.
- For comparison purposes, in Section 3, we propose *Joint Kernel Herding* (JKH) and *Joint Kernel Inducing Points* (JKIP), extending existing compression algorithms to target the joint distribution.
- In Section 5, across various datasets and evaluation metrics, we show that directly targeting the conditional distribution via ACKIP is preferable to compressing the joint distribution via JKH or JKIP. We also demonstrate the limitations of the greedy heuristic used by JKH and ACKH, with JKIP and ACKIP outperforming their counterparts.

## 2 Preliminaries

In this section we briefly introduce the relevant theory of RKHSs, for a more thorough treatment see the various detailed surveys which exist in the literature [35–37].

Throughout this work, we consider $(\Omega, \mathcal{F}, \mathbb{P})$ as the underlying probability space. Let $(\mathcal{X}, \mathscr{X})$ and $(\mathcal{Y}, \mathscr{Y})$ be separable measure spaces, and let $X : \Omega \to \mathcal{X}$ and $Y : \Omega \to \mathcal{Y}$ be random variables with distributions $\mathbb{P}_X$ and $\mathbb{P}_Y$, respectively. We denote the joint distribution of $X$ and $Y$ by $\mathbb{P}_{X,Y}$ and the conditional distribution, in the measure-theoretic sense of [18], by $\mathbb{P}_{Y|X}$. Given a labelled dataset $\mathcal{D} := \{(\boldsymbol{x}_i, \boldsymbol{y}_i)\}_{i=1}^n$, we refer to the empirical distribution of $\mathcal{D}$ as $\hat{\mathbb{P}}_{X,Y}$. For a compressed set $\mathcal{C} := \{(\tilde{\boldsymbol{x}}_i, \tilde{\boldsymbol{y}}_i)\}_{i=1}^m$, $m \ll n$, we instead denote the empirical distribution as $\tilde{\mathbb{P}}_{X,Y}$.

**Reproducing Kernel Hilbert Spaces**: Each positive definite kernel function $k : \mathcal{X} \times \mathcal{X} \to \mathbb{R}$ induces a vector space of functions from $\mathcal{X}$ to $\mathbb{R}$, known as a *Reproducing Kernel Hilbert Space* (RKHS) [1], denoted by $\mathcal{H}_k$. An RKHS $\mathcal{H}_k$ is defined by two key properties: (i) For all $\boldsymbol{x} \in \mathcal{X}$, the function $k(\boldsymbol{x}, \cdot) : \mathcal{X} \to \mathbb{R}$ belongs to $\mathcal{H}_k$; (ii) The kernel function $k$ satisfies the *reproducing property*, meaning that for all $f \in \mathcal{H}_k$ and $\boldsymbol{x} \in \mathcal{X}$, we have $f(\boldsymbol{x}) = \langle f(\cdot), k(\boldsymbol{x}, \cdot) \rangle_{\mathcal{H}_k}$, where $\langle \cdot, \cdot \rangle_{\mathcal{H}_k}$ denotes the inner product in $\mathcal{H}_k$. A key property of kernels is the concept of *universality* [38]. Intuitively, if a kernel $k \in C_0(\mathcal{X})$ is universal, then for every function $f \in C_0(\mathcal{X})$, there exists a function $g \in \mathcal{H}_k$ that approximates it arbitrarily well. This property is satisfied for many common kernel functions such as the Gaussian Laplacian, and Matérn, for example [38].

**Tensor Products of Reproducing Kernel Hilbert Spaces**: Let $l : \mathcal{Y} \times \mathcal{Y} \to \mathbb{R}$ be the reproducing kernel inducing the RKHS $\mathcal{H}_l$, and denote $\mathcal{H}_k \otimes \mathcal{H}_l$ to be the tensor product of the RKHSs $\mathcal{H}_k$ and $\mathcal{H}_l$, consisting of functions $g : \mathcal{X} \times \mathcal{Y} \to \mathbb{R}$. Then, for $h, h' \in \mathcal{H}_k$ and $f, f' \in \mathcal{H}_l$, the inner product in $\mathcal{H}_k \otimes \mathcal{H}_l$ is given by $\langle f \otimes g, f' \otimes g' \rangle_{\mathcal{H}_k \otimes \mathcal{H}_l} := \langle g, g' \rangle_{\mathcal{H}_k} \langle f, f' \rangle_{\mathcal{H}_l}$. Under the integrability condition $\mathbb{E}_{\mathbb{P}_X}[k(X, X)] < \infty$, $\mathbb{E}_{\mathbb{P}_Y}[l(Y, Y)] < \infty$, one can define the *joint* kernel mean embedding $\mu_{X,Y} := \mathbb{E}_{\mathbb{P}_{X,Y}}[k(X, \cdot)l(Y, \cdot)] \in \mathcal{H}_k \otimes \mathcal{H}_l$ such that $\mathbb{E}_{\mathbb{P}_{X,Y}}[g(X, Y)] = \langle \mu_{X,Y}, g \rangle_{\mathcal{H}_k \otimes \mathcal{H}_l}$ for all $g \in \mathcal{H}_k \otimes \mathcal{H}_l$ [18]. The joint kernel mean embedding can be estimated straightforwardly as $\hat{\mu}_{X,Y} := \sum_{i=1}^n k(\boldsymbol{x}_i, \cdot)l(\boldsymbol{y}_i, \cdot)$ with i.i.d. samples from the joint distribution. The tensor product structure is advantageous as it permits the natural construction of a tensor product kernel from kernels defined on $\mathcal{X}$ and $\mathcal{Y}$. This insight is particularly important when $\mathcal{X}$ and $\mathcal{Y}$ have distinct characteristics that make a direct definition of a p.d. kernel difficult, for example, if $\mathcal{X} = \mathbb{R}^d$ and $\mathcal{Y} = \mathbb{N}^p$.

Given additional random variables $X' : \Omega \to \mathcal{X}$, $Y' : \Omega \to \mathcal{Y}$, and the embedding $\mu_{X',Y'}$ of $\mathbb{P}_{X',Y'}$, one can define the *Joint Maximum Mean Discrepancy* (JMMD) [15] as

$$\mathrm{JMMD}(\mathbb{P}_{X,Y}, \mathbb{P}_{X',Y'}) := \|\mu_{X,Y} - \mu_{X',Y'}\|_{\mathcal{H}_k \otimes \mathcal{H}_l}.$$

For a particular class of *characteristic* tensor product kernels [39], the mapping $\mathbb{P}_{X,Y} \mapsto \mu_{X,Y}$ is *injective* [40]. Hence, by the virtue of Theorem 5 [2], it is the case that $\mathrm{JMMD}(\mathbb{P}_{X,Y}, \mathbb{P}_{X,Y}) = \|\mu_{X,Y} - \mu_{X',Y'}\|_{\mathcal{H}_k \otimes \mathcal{H}_l} = 0$ if and only if $\mathbb{P}_{X,Y} = \mathbb{P}_{X',Y'}$.

**Conditional Kernel Mean Embedding**: Under the integrability condition $\mathbb{E}_{\mathbb{P}_Y}[\sqrt{l(Y, Y)}] < \infty$, the kernel *conditional* mean embedding (KCME) of $\mathbb{P}_{Y|X}$ is defined as $\mu_{Y|X} := \mathbb{E}_{\mathbb{P}_{Y|X}}[l(Y, \cdot) \mid X]$ where $\mu_{Y|X} : \Omega \to \mathcal{H}_l$ is an $X$-measurable random variable outputting *functions* in $\mathcal{H}_l$. Similar to the unconditional case, for any $f \in \mathcal{H}_l$, it can be shown that $\mathbb{E}_{\mathbb{P}_{Y|X}}[f(Y) \mid X] \overset{\text{a.s.}}{=} \langle f, \mu_{Y|X} \rangle_{\mathcal{H}_l}$ [18].

The KCME can be written as the composition of a deterministic function $F_{Y|X} : \mathcal{X} \to \mathcal{H}_l$ and the random variable $X : \Omega \to \mathcal{X}$, i.e. $\mu_{Y|X} = F_{Y|X} \circ X$ (Theorem 4.1, [18]). For consistency in notation, throughout the remainder of this work, whenever we refer to the KCME $\mu_{Y|X}$, we mean $F_{Y|X}$. The KCME can be estimated directly [18, 30] using i.i.d. samples from the joint distribution $\mathcal{D}$ as

$$\hat{\mu}_{Y|X}^{\mathcal{D}} := \sum_{i,j=1}^n k(\boldsymbol{x}_i, \cdot) W_{ij} l(\boldsymbol{y}_j, \cdot) \tag{1}$$

where the superscript $\mathcal{D}$ refers to the data used to estimate $\mu_{Y|X}$, we define $W := (K + \lambda I)^{-1}$, $[K]_{ij} = k(\boldsymbol{x}_i, \boldsymbol{x}_j)$, $i, j = 1, \ldots, n$, and $\lambda > 0$ is a regularisation parameter.

**Maximum Conditional Mean Discrepancy**: Given two conditional distributions $\mathbb{P}_{Y|X}$ and $\mathbb{P}_{Y'|X'}$, with KCMEs $\mu_{Y|X}$ and $\mu_{Y'|X'}$, the *Maximum Conditional Mean Discrepancy* (MCMD) was defined

by [18] as a function of the conditioning variable $\boldsymbol{x} \in \mathcal{X}$, which returns a metric on $\mathbb{P}_{Y|X=\boldsymbol{x}}$ and $\mathbb{P}_{Y'|X'=\boldsymbol{x}}$, that is

$$\text{MCMD}\left[\mathbb{P}_{Y|X}, \mathbb{P}_{Y'|X'}\right](\boldsymbol{x}) := \|\mu_{Y|X=\boldsymbol{x}} - \mu_{Y'|X'=\boldsymbol{x}}\|_{\mathcal{H}_l}.$$

In the following sense sense, the MCMD is a natural metric on the space of conditional distributions:

**Theorem 2.1.** *(Theorem 5.2. [18]) Suppose $l : \mathcal{Y} \times \mathcal{Y} \to \mathbb{R}$ is characteristic, that $\mathbb{P}_X$ and $\mathbb{P}_{X'}$ are absolutely continuous with respect to each other, and that $\mathbb{P}(\cdot \mid X)$ and $\mathbb{P}(\cdot \mid X')$ admit regular versions. Then $MCMD\left[\mathbb{P}_{Y|X}, \mathbb{P}_{Y'|X'}\right](\cdot) = 0$ almost everywhere $\boldsymbol{x}$ wrt $\mathbb{P}_X$ (or $\mathbb{P}_{X'}$) if and only if, for almost all $\boldsymbol{x} \in \mathcal{X}$ wrt $\mathbb{P}_X$ (or $\mathbb{P}_{X'}$), we have $\mathbb{P}_{Y|X=\boldsymbol{x}}(B) = \mathbb{P}_{Y'|X'=\boldsymbol{x}}(B)$ for all $B \in \mathscr{Y}$.*

**Related Work**: Existing distribution compression methods focus on unlabelled data, where the goal is to approximate $\mathbb{P}_X$ using a smaller representative set that minimises the Maximum Mean Discrepancy (MMD). Perhaps the most intuitive of these is Kernel Herding [3, 41], which greedily constructs a compressed set by iteratively optimising points that minimise the MMD. While simple and interpretable, its greedy nature can lead to suboptimal solutions, motivating alternatives such as Gradient Flow [11], which jointly optimises all points via gradient descent, and Kernel Thinning [13, 14], which restricts the compressed set to a subset of the original data to support theoretical guarantees. Despite these advances, existing approaches target unlabelled distributions. In contrast, this work introduces algorithms for *joint* and *conditional* distribution compression. For additional discussion of related work see Section A.1.

## 3 Joint Distribution Compression

Given a labelled dataset $\mathcal{D}$, one may be interested in compressing the joint distribution $\mathbb{P}_{X,Y}$, rather than the marginals $\mathbb{P}_X, \mathbb{P}_Y$.

### 3.1 Joint Kernel Herding

By inducing an RKHS $\mathcal{H}_k \otimes \mathcal{H}_l$ with the tensor product kernel $k(\cdot, \cdot)l(\cdot, \cdot)$, one can modify the Kernel Herding [3] algorithm to instead target the joint distribution. First, we assume that $\|k(\boldsymbol{x}, \cdot)l(\boldsymbol{y}, \cdot)\|_{\mathcal{H}_k \otimes \mathcal{H}_l} = R$ for all $\boldsymbol{x} \in \mathcal{X}$ and $\boldsymbol{y} \in \mathcal{Y}$, where $R$ is a constant. This condition holds for commonly used stationary kernels such as the Gaussian, Laplace, and Matérn kernels. Then, assuming we are at the $(m + 1)^{\text{th}}$ iteration, having already constructed a compressed set of size $m$, $\mathcal{C} = \{(\tilde{\boldsymbol{x}}_j, \tilde{\boldsymbol{y}}_j)\}_{j=1}^m$, the next pair is chosen as the solution to the optimisation problem

$$\underset{(\boldsymbol{x}, \boldsymbol{y}) \in \mathcal{X} \times \mathcal{Y}}{\arg\min} \quad \frac{1}{m+1} \sum_{j=1}^m k(\boldsymbol{x}, \tilde{\boldsymbol{x}}_j)k(\boldsymbol{y}, \tilde{\boldsymbol{y}}_j) - \mathbb{E}_{(\boldsymbol{x}', \boldsymbol{y}') \sim \mathbb{P}_{X,Y}}\left[k(\boldsymbol{x}, \boldsymbol{x}')l(\boldsymbol{y}, \boldsymbol{y}')\right]. \tag{2}$$

We refer to this algorithm as *Joint Kernel Herding* (JKH). The optimisation problem in (2) can be interpreted as reducing at each iteration the $\text{JMMD}(\mathbb{P}_{X,Y}, \tilde{\mathbb{P}}_{X,Y})$; see Section B.2.

### 3.2 Joint Kernel Inducing Points

The greedy optimisation approach of JKH is a convenient heuristic which focuses computational effort on optimising one new pair at a time while previously selected pairs are fixed. However, this strategy may give poor solutions, as it never revisits or adjusts earlier selections. Instead, we can first select an initial compressed set of $m$ pairs through uniformly at random subsampling of $\mathcal{D}$, followed by refining all pairs *jointly*. Going forward, we refer to this algorithm as *Joint Kernel Inducing Points* (JKIP), noting that it may be viewed as a discretised Wasserstein gradient flow [11].

We target the $\text{JMMD}(\mathbb{P}_{X,Y}, \tilde{\mathbb{P}}_{X,Y})$ by solving the optimisation problem

$$\underset{(\tilde{\boldsymbol{X}}, \tilde{\boldsymbol{Y}}) \subset \mathcal{X} \times \mathcal{Y}}{\arg\min} \quad \frac{1}{m^2} \sum_{i,j=1}^m k(\tilde{\boldsymbol{x}}_i, \tilde{\boldsymbol{x}}_j)l(\tilde{\boldsymbol{y}}_i, \tilde{\boldsymbol{y}}_j) - \frac{2}{m} \sum_{i=1}^m \mathbb{E}_{(\boldsymbol{x}', \boldsymbol{y}') \sim \mathbb{P}_{X,Y}}\left[k(\tilde{\boldsymbol{x}}_i, \boldsymbol{x}')l(\tilde{\boldsymbol{y}}_i, \boldsymbol{y}')\right] \tag{3}$$

via gradient descent. In the general case, one cannot compute the expectation above, hence we instead target its empirical alternative, namely $\text{JMMD}(\hat{\mathbb{P}}_{X,Y}, \tilde{\mathbb{P}}_{X,Y})$. Defining $\boldsymbol{X} := [\boldsymbol{x}_1, \boldsymbol{x}_2, \ldots, \boldsymbol{x}_n]^\top$,

$\boldsymbol{Y} := [\boldsymbol{y}_1, \boldsymbol{y}_2, \ldots, \boldsymbol{y}_n]^\top$, $\tilde{\boldsymbol{X}} := [\tilde{\boldsymbol{x}}_1, \tilde{\boldsymbol{x}}_2, \ldots, \tilde{\boldsymbol{x}}_m]^\top$, and $\tilde{\boldsymbol{Y}} := [\tilde{\boldsymbol{y}}_1, \tilde{\boldsymbol{y}}_2, \ldots, \tilde{\boldsymbol{y}}_m]^\top$, this reduces to targeting

$$\mathcal{L}^{\mathcal{D}}(\tilde{\boldsymbol{X}}, \tilde{\boldsymbol{Y}}) := \frac{1}{m^2}\text{Tr}\left(K_{\tilde{\boldsymbol{X}}\tilde{\boldsymbol{X}}}L_{\tilde{\boldsymbol{Y}}\tilde{\boldsymbol{Y}}}\right) - \frac{2}{mn}\text{Tr}\left(K_{\tilde{\boldsymbol{X}}\boldsymbol{X}}L_{\boldsymbol{Y}\tilde{\boldsymbol{Y}}}\right), \tag{4}$$

where $[K_{\tilde{\boldsymbol{X}}\tilde{\boldsymbol{X}}}]_{ij} := k(\tilde{\boldsymbol{x}}_i, \tilde{\boldsymbol{x}}_j)$, $[K_{\tilde{\boldsymbol{X}}\boldsymbol{X}}]_{ij} := k(\tilde{\boldsymbol{x}}_i, \boldsymbol{x}_j)$, $[L_{\tilde{\boldsymbol{Y}}\tilde{\boldsymbol{Y}}}]_{ij} := l(\tilde{\boldsymbol{y}}_i, \tilde{\boldsymbol{y}}_j)$, and $[L_{\tilde{\boldsymbol{Y}}\boldsymbol{Y}}]_{ij} := l(\tilde{\boldsymbol{y}}_i, \boldsymbol{y}_j)$. One might initially assume that JKIP is more computationally expensive than JKH due to its higher cost per gradient step, but this is not the case. In fact, to construct a compressed set of size $m$, both JKH and JKIP have time complexity of $\mathcal{O}(mn + m^2)$; see Section D.2.1 and D.2.2.

## 4 From Joint to Conditional Distribution Compression

### 4.1 The Average Maximum Conditional Mean Discrepancy

In Section 2, we recalled the MCMD, which is a function of $\boldsymbol{x} \in \mathcal{X}$ that outputs a metric on the conditional distributions $\mathbb{P}_{Y|X=\boldsymbol{x}}$, $\mathbb{P}_{Y'|X'=\boldsymbol{x}}$ with fixed conditioning values. However, for the purposes of distribution compression, we require a discrepancy measure that applies to entire families of conditional distributions $\mathbb{P}_{Y|X}$, $\mathbb{P}_{Y'|X'}$. Prior work has introduced the discrepancy

$$\mathbb{E}_{\boldsymbol{x} \sim \mathbb{P}_X}\left[\|\mu_{Y|X=\boldsymbol{x}} - \mu_{Y'|X=\boldsymbol{x}}\|_{\mathcal{H}_l}^2\right], \tag{5}$$

which was independently proposed as the *Average Maximum Mean Discrepancy* (AMMD) in [42] and the *Kernel Conditional Discrepancy* (KCD) in [19]. Throughout this work, we will refer to (5) as the KCD. We introduce a more general alternative, and show it is a proper metric: given an additional random variable $X^* : \Omega \to \mathcal{X}$ with distribution $\mathbb{P}_{X^*}$, we define the *Average Maximum Conditional Mean Discrepancy* (AMCMD) as

$$\text{AMCMD}\left[\mathbb{P}_{X^*}, \mathbb{P}_{Y|X}, \mathbb{P}_{Y'|X'}\right] := \sqrt{\mathbb{E}_{\boldsymbol{x} \sim \mathbb{P}_{X^*}}\left[\|\mu_{Y|X=\boldsymbol{x}} - \mu_{Y'|X'=\boldsymbol{x}}\|_{\mathcal{H}_l}^2\right]}. \tag{6}$$

In the above definition, the expectation is taken with respect to a distinct probability measure $\mathbb{P}_{X^*} \neq \mathbb{P}_X, \mathbb{P}_{x'}$, which broadens its applicability compared to the KCD. See Section C.2 for an illustrative example.

The following theorem establishes that the AMCMD is indeed a proper metric.

**Theorem 4.1.** *Suppose that $l : \mathcal{Y} \times \mathcal{Y} \to \mathbb{R}$ is a characteristic kernel, that $\mathbb{P}_X$, $\mathbb{P}_{X'}$, and $\mathbb{P}_{X^*}$ are absolutely continuous with respect to each other, and that $\mathbb{P}(\cdot \mid X)$ and $\mathbb{P}(\cdot \mid X')$ admit regular versions. Then, AMCMD $\left[\mathbb{P}_{X^*}, \mathbb{P}_{Y|X}, \mathbb{P}_{Y'|X'}\right] = 0$ if and only if, for almost all $\boldsymbol{x} \in \mathcal{X}$ wrt $\mathbb{P}_{X^*}$, $\mathbb{P}_{Y|X=\boldsymbol{x}}(B) = \mathbb{P}_{Y'|X'=\boldsymbol{x}}(B)$ for all $B \in \mathscr{Y}$.*

*Moreover, assuming the Radon-Nikodym derivatives $\frac{d\mathbb{P}_{X^*}}{d\mathbb{P}_X}$, $\frac{d\mathbb{P}_{X^*}}{d\mathbb{P}_{X'}}$, $\frac{d\mathbb{P}_{X^*}}{d\mathbb{P}_{X''}}$ are bounded, then the triangle inequality is satisfied, i.e. AMCMD $\left[\mathbb{P}_{X^*}, \mathbb{P}_{Y|X}, \mathbb{P}_{Y''|X''}\right] \leq$ AMCMD $\left[\mathbb{P}_{X^*}, \mathbb{P}_{Y|X}, \mathbb{P}_{Y'|X'}\right] +$ AMCMD $\left[\mathbb{P}_{X^*}, \mathbb{P}_{Y'|X'}, \mathbb{P}_{Y''|X''}\right]$.*

**Remark 4.2.** The boundedness condition on the Radon-Nikodym derivative may be intuitively understood as a condition on the relative heaviness of the tails of the distribution $\mathbb{P}_{X^*}$ compared to $\mathbb{P}_X$. For example, if $\mathbb{P}_{X^*} = \mathcal{N}(\mu, \sigma_*^2)$ and $\mathbb{P}_X = \mathcal{N}(\mu, \sigma^2)$, then $\frac{d\mathbb{P}_{X^*}}{d\mathbb{P}_X}$ is bounded whenever $\sigma^2 > \sigma_*^2$.

Given sets of i.i.d. samples $\{\boldsymbol{x}_i^*\}_{i=1}^q \sim \mathbb{P}_{X^*}$, $\mathcal{N} := \{(\boldsymbol{x}_i, \boldsymbol{y}_i)\}_{i=1}^n \sim \mathbb{P}_{X,Y}$, and $\mathcal{M} := \{(\boldsymbol{x}_i', \boldsymbol{y}_i')\}_{i=1}^m \sim \mathbb{P}_{X',Y'}$, we define a plug-in estimate of the AMCMD$^2$ as

$$\widehat{\text{AMCMD}}^2\left[\mathbb{P}_{X^*}, \mathbb{P}_{Y|X}, \mathbb{P}_{Y'|X'}\right] := \frac{1}{q}\sum_{i=1}^q \left\|\hat{\mu}_{Y|X=\boldsymbol{x}_i^*}^{\mathcal{N}} - \hat{\mu}_{Y'|X'=\boldsymbol{x}_i^*}^{\mathcal{M}}\right\|_{\mathcal{H}_l}^2, \tag{7}$$

and derive a closed form expression as follows:

**Lemma 4.3.**

$$\widehat{AMCMD}^2\left[\mathbb{P}_{X^*}, \mathbb{P}_{Y|X}, \mathbb{P}_{Y'|X'}\right]$$
$$= \frac{1}{q}Tr\left(K_{\boldsymbol{X}^*\boldsymbol{X}}W_{\boldsymbol{X}\boldsymbol{X}}L_{\boldsymbol{Y}\boldsymbol{Y}}W_{\boldsymbol{X}\boldsymbol{X}}K_{\boldsymbol{X}\boldsymbol{X}^*}\right) - \frac{2}{q}Tr\left(K_{\boldsymbol{X}^*\boldsymbol{X}}W_{\boldsymbol{X}\boldsymbol{X}}L_{\boldsymbol{Y}\boldsymbol{Y}'}W_{\boldsymbol{X}'\boldsymbol{X}'}K_{\boldsymbol{X}'\boldsymbol{X}^*}\right)$$
$$+ \frac{1}{q}Tr\left(K_{\boldsymbol{X}^*\boldsymbol{X}'}W_{\boldsymbol{X}'\boldsymbol{X}'}L_{\boldsymbol{Y}'\boldsymbol{Y}'}W_{\boldsymbol{X}'\boldsymbol{X}'}K_{\boldsymbol{X}'\boldsymbol{X}^*}\right),$$

*where, for example, we have defined* $[K_{\boldsymbol{X'X^*}}]_{ij} := k(\boldsymbol{x'}, \boldsymbol{x^*})$, $[L_{\boldsymbol{YY'}}]_{ij} := l(\boldsymbol{y}_i, \boldsymbol{y'}_j)$, *and* $W_{\boldsymbol{X'X'}} := (K_{\boldsymbol{X'X'}} + \lambda_m I)^{-1}$.

This estimate is $\mathcal{O}(n^3 + m^3 + q(n^2 + m^2))$ to compute. As a corollary of Theorem 4.5 in [19], we establish its consistency in the special case that $X^* = X' = X$, which corresponds to the regime under which our conditional distribution compression algorithms will operate.

**Corollary 4.4.** *Assume that $k(\cdot, \cdot)$ and $l(\cdot, \cdot)$ are bounded, $k(\cdot, \cdot)$ is universal, and let the regularisation parameters $\lambda_n$ and $\lambda_m$ decay at slower rates than $\mathcal{O}(n^{-1/2})$ and $\mathcal{O}(m^{-1/2})$ respectively. Then, $\widehat{AMCMD}\left[\mathbb{P}_X, \mathbb{P}_{Y|X}, \mathbb{P}_{Y'|X}\right] \xrightarrow{p} AMCMD\left[\mathbb{P}_X, \mathbb{P}_{Y|X}, \mathbb{P}_{Y'|X}\right]$ as $n, m, q \to \infty$.*

**Remark 4.5.** The conditions in Corollary 4.4 ensure that $\hat{\mu}_{Y|X}$ and $\hat{\mu}_{Y'|X}$ converge to $\mu_{Y|X}$ and $\mu_{Y'|X}$ respectively in $L^2(\mathcal{X}, \mathbb{P}_X; \mathcal{H}_l)$ norm, that is, the norm of the space of square $\mathbb{P}_X$-integrable $\mathcal{H}_l$-valued functions [19].

**Remark 4.6.** The conditions on $k$ and $l$ are satisfied by many common choices. For example, with continuous conditional distributions, both can be Gaussian kernels; for discrete conditional distributions, $l$ can be replaced with an indicator kernel [33].

## 4.2 Average Conditional Kernel Herding

We now introduce *Average Conditional Kernel Herding* (ACKH), a greedy algorithm that constructs a compressed set targeting the $\text{AMCMD}^2[\mathbb{P}_X, \mathbb{P}_{Y|X}, \hat{\mathbb{P}}_{Y|X}]$. The ACKH objective is derived by expanding the squared norm in (6), and ignoring the term which is invariant wrt the compressed set.

Assuming we are at the $(m+1)^{\text{th}}$ iteration, having already constructed the compressed set $\mathcal{C} = \{(\tilde{\boldsymbol{x}}_j, \tilde{\boldsymbol{y}}_j)\}_{j=1}^m$, we expand the compressed set $\mathcal{C} = \mathcal{C} \cup (\boldsymbol{x}, \boldsymbol{y})$ by solving

$$\underset{\boldsymbol{x}, \boldsymbol{y}}{\arg\min} \left\{ \mathbb{E}_{\boldsymbol{x'} \sim \mathbb{P}_X}\left[\left\|\tilde{\mu}_{Y|X=\boldsymbol{x'}}^{\mathcal{C}}\right\|_{\mathcal{H}_l}^2\right] - 2\mathbb{E}_{\boldsymbol{x'} \sim \mathbb{P}_X}\left[\left\langle \mu_{Y|X=\boldsymbol{x'}}, \tilde{\mu}_{Y|X=\boldsymbol{x'}}^{\mathcal{C}}\right\rangle_{\mathcal{H}_l}\right] \right\}. \tag{8}$$

As we do not have access to $\mu_{Y|X}$ and $\mathbb{P}_X$, we must estimate this objective; this is equivalent to targeting the $\text{AMCMD}^2[\hat{\mathbb{P}}_X, \hat{\mathbb{P}}_{Y|X}, \tilde{\mathbb{P}}_{Y|X}]$. Naïvely, one might assume that we must estimate $\mu_{Y|X}$ using $\mathcal{D}$, incurring a high computational cost of $\mathcal{O}(n^3)$. The following result allows us to avoid this:

**Lemma 4.7.** *Let $h : \mathcal{X} \to \mathcal{H}_l$ be a vector-valued function, then*

$$\mathbb{E}_{\boldsymbol{x} \sim \mathbb{P}_X}\left[\left\langle \mu_{Y|X=\boldsymbol{x}}, h(\boldsymbol{x})\right\rangle_{\mathcal{H}_l}\right] = \mathbb{E}_{(\boldsymbol{x}, \boldsymbol{y}) \sim \mathbb{P}_{X,Y}}\left[h(\boldsymbol{x})(\boldsymbol{y})\right].$$

**Remark 4.8.** If we let $h(\boldsymbol{x}) = \tilde{\mu}_{Y|X=\boldsymbol{x}}^{\mathcal{C}}$, then Lemma 4.7 eliminates the need to explicitly estimate $\mu_{Y|X}$ in (8), and hence, the cost of estimating the objective is reduced from $\mathcal{O}(n^3)$ to $\mathcal{O}(n)$.

Even after applying Lemma 4.7 we still have an expectation in (8) which we may not be able to compute in general. Estimating this expectation using $\mathcal{D}$, we obtain the closed form objective

$$\mathcal{G}^{\mathcal{D}}(\boldsymbol{x}, \boldsymbol{y}) := \frac{1}{n}\text{Tr}(\bar{K}(\boldsymbol{x})\tilde{W}(\boldsymbol{x})\tilde{L}(\boldsymbol{y})\tilde{W}(\boldsymbol{x})\bar{K}(\boldsymbol{x})^\top) - \frac{2}{n}\text{Tr}(\bar{K}(\boldsymbol{x})\tilde{W}(\boldsymbol{x})\bar{L}(\boldsymbol{y})^\top), \tag{9}$$

where we define $[\bar{K}(\boldsymbol{x})]_{ij} := k(\boldsymbol{x}_i, \tilde{\boldsymbol{x}}_j)$, $i = 1, \ldots, n$, $j = 1, \ldots, m$, $[\tilde{K}(\boldsymbol{x})]_{ij} := k(\tilde{\boldsymbol{x}}_i, \tilde{\boldsymbol{x}}_j)$, $i, j = 1, \ldots, m$, and $\tilde{W}(\boldsymbol{x}) := (\tilde{K}(\boldsymbol{x}) + \lambda I)^{-1}$. Analogous definitions hold for the response kernel $l(\cdot, \cdot)$, yielding matrices $\bar{L}(\boldsymbol{y}) \in \mathbb{R}^{n \times m}$ and $\tilde{L}(\boldsymbol{y}) \in \mathbb{R}^{m \times m}$. To construct a compressed set of size $m$, ACKH can be shown to have an overall time complexity of $\mathcal{O}(m^4 + m^3 n)$; see Section D.2.3.

## 4.3 Average Conditional Kernel Inducing Points

Unlike JKH, where updates depend only on the newest pair in the compressed set, the presence of an inverse in the objective (9) prevents a similar simplification. [1] This leads to a quartic dependence of ACKH on the compressed set size $m$, which is a fairly significant limitation. By instead optimising each pair in the compressed set simultaneously, we can reduce this to just cubic dependence on $m$.

---

[1] Through the method of bordering [43], it is technically possible to update the inverse of a growing matrix, minimising re-computation. Unfortunately, bordering is a highly numerically unstable procedure, and kernel matrices are often very ill-conditioned in practice.

We refer to this algorithm as *Average Conditional Kernel Inducing Points* (ACKIP), solving the optimisation problem

$$\arg\min_{\mathcal{C}} \left\{ \mathbb{E}_{\boldsymbol{x}'\sim\mathbb{P}_X}\left[\left\|\tilde{\mu}_{Y|X=\boldsymbol{x}'}^{\mathcal{C}}\right\|_{\mathcal{H}_l}^2\right] - 2\mathbb{E}_{(\boldsymbol{x}',\boldsymbol{y}')\sim\mathbb{P}_{X,Y}}\left[\tilde{\mu}_{Y|X=\boldsymbol{x}'}^{\mathcal{C}}(\boldsymbol{y}')\right] \right\} \tag{10}$$

over each pair in $\mathcal{C} = (\tilde{\boldsymbol{X}}, \tilde{\boldsymbol{Y}})$ via gradient descent. Once again, even after applying Lemma 4.7 to (10) we still retain an expectation which in general is difficult to compute. Estimating this ecoectatuion using $\mathcal{D}$, we obtain the closed form objective

$$\mathcal{J}^{\mathcal{D}}(\tilde{\boldsymbol{X}}, \tilde{\boldsymbol{Y}}) := \frac{1}{n}\text{Tr}\left(K_{\boldsymbol{X}\tilde{\boldsymbol{X}}}W_{\tilde{\boldsymbol{X}}\tilde{\boldsymbol{X}}}L_{\tilde{\boldsymbol{Y}}\tilde{\boldsymbol{Y}}}W_{\tilde{\boldsymbol{X}}\tilde{\boldsymbol{X}}}K_{\tilde{\boldsymbol{X}}\boldsymbol{X}}\right) - \frac{2}{n}\text{Tr}\left(L_{\boldsymbol{Y}\tilde{\boldsymbol{Y}}}W_{\tilde{\boldsymbol{X}}\tilde{\boldsymbol{X}}}K_{\tilde{\boldsymbol{X}}\boldsymbol{X}}\right), \tag{11}$$

where $W_{\tilde{\boldsymbol{X}}\tilde{\boldsymbol{X}}} := (K_{\tilde{\boldsymbol{X}}\tilde{\boldsymbol{X}}} + \lambda I)^{-1}$. To construct a compressed set of size $m$, ACKIP has an overall complexity of $\mathcal{O}(m^3 + m^2 n)$, i.e. a factor of $m$ *faster* than ACKH; see Section D.2.4.

## 5 Experiments

Building on the experimental setup of Kernel Herding [3], we demonstrate how the methods proposed in this paper can be applied to compress the conditional distribution. We report the root mean square error $\text{RMSE}(\mathcal{C}) := \sqrt{\frac{1}{n}\sum_{i=1}^{n}\left(\mathbb{E}[h(Y) \mid X = \boldsymbol{x}_i] - \langle\hat{\mu}_{Y|X=\boldsymbol{x}_i}^{\mathcal{C}}, h\rangle_{\mathcal{H}_l}\right)^2}$, across a range of test functions $h : \mathcal{Y} \to \mathbb{R}$. This aligns with the standard applications of the KCME [17–33], where one estimates the conditional expectation of a function of interest $h$. When the exact value of the conditional expectation is unavailable, we approximate $\mathbb{E}[h(Y) \mid X = \boldsymbol{x}_i]$ via its full-data estimate, $\langle\hat{\mu}_{Y|X=\boldsymbol{x}_i}^{\mathcal{D}}, h\rangle_{\mathcal{H}_l}$. Note that when $h$ is the identity function, the estimate reduces to the familiar regression setting i.e. $\mathbb{E}[Y \mid X]$, and when $h$ is an indicator function, it corresponds to estimating class-conditional probabilities e.g. $\mathbb{P}(Y = 0 \mid X)$. For full details of the experiments, including results on additional test functions omitted from the main text due to space constraints, see Section C.

### 5.1 Matching the True Conditional Distribution

In general, the expectations in (2), (3), (8), and (10) must be estimated. However, when the kernels $k$ and $l$ are Gaussian, and we let $\mathbb{P}_X = \mathcal{N}(\mu, \sigma^2)$ and $\mathbb{P}_{Y|X} = \mathcal{N}(a_0 + a_1 X, \sigma_\epsilon^2)$ for $\mu, a_0, a_1 \in \mathbb{R}$ and $\sigma^2, \sigma_\epsilon^2 > 0$, the integrals can be evaluated analytically. See Section C.3 for details. We construct compressed sets of size $m = 500$, and compute the $\text{AMCMD}^2\left[\mathbb{P}_X, \mathbb{P}_{Y|X}, \tilde{\mathbb{P}}_{Y|X}\right]$ achieved by each method. Figures 2 and 3 highlight the advantages of directly targeting the conditional distribution, with ACKH and ACKIP achieving lower AMCMD compared to JKH and JKIP. Additionally, in the case of ACKIP versus ACKH, it demonstrates the superiority of joint optimisation over herding, where the inability to revisit previous selections limits ACKH's performance in comparison to ACKIP. Moreover, we can see that the reduced AMCMD achieved by ACKH and ACKIP translates to improved performance in estimating conditional expectations across a variety of test functions.

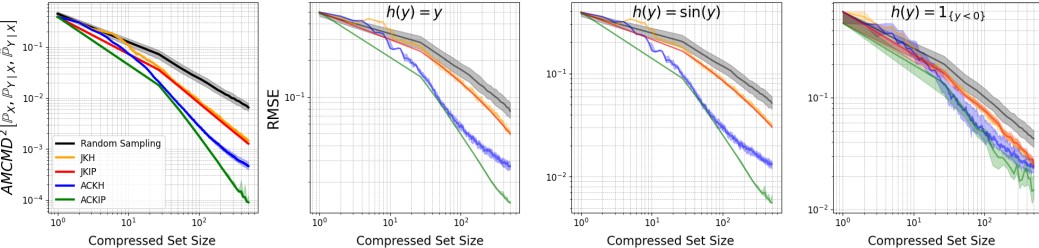

Figure 2: Results for the true conditional distribution compression task with parameters set as $a_0 = -0.5$, $a_1 = 0.5$, $\mu = 1$, $\sigma^2 = 1$, and $\sigma_\epsilon^2 = 0.5$. The $\text{AMCMD}^2$ (first plot), and the RMSE across three test functions, versus the size of the compressed set is reported. For JKH (orange), JKIP (red), ACKH (blue), and ACKIP (green), we display the median performance (bold line) with the 25th-75th percentiles (shaded region) over 20 runs. The error of random sampling (black) over 500 runs is also plotted for comparison.

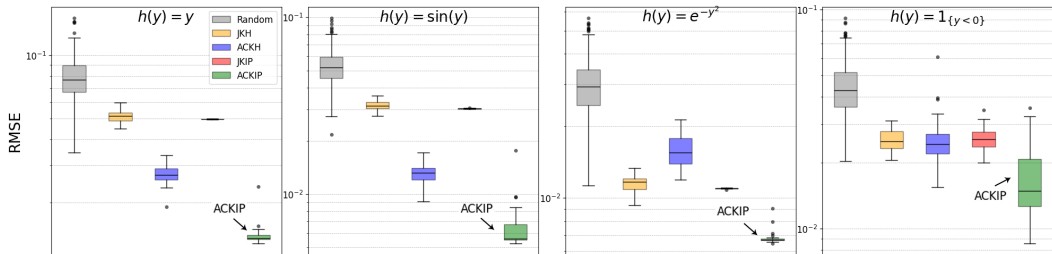

Figure 3: Results of the true conditional distribution compression task for compressed sets of size $m = 500$. The RMSE across a variety of test functions is reported, with the IQR highlighted for each method. Outliers are calculated as being above $Q_3 + 1.5\text{IQR}$ and below $Q_1 - 1.5\text{IQR}$.

## 5.2 Matching the Empirical Conditional Distribution

In this section, we present experiments targeting the empirical conditional distribution of synthetic and real-data. Across all datasets, we generate or subsample down to $n = 10,000$ pairs, split off 10% for validation, 10% for testing, and construct compressed sets up to size $m = 250$.

### 5.2.1 Continuous Conditional Distributions

**Real**: We use the *Superconductivity* dataset from UCI [44]. *Superconductivity* is composed of $d = 81$ features relating to the chemical composition of superconductors with the target being its critical temperature [45]. In Figure 4 and 5 we see that ACKIP achieves the lowest RMSE across each of the test functions, with ACKH in second for all but one. We also note that JKIP achieves favourable performance versus JKH.

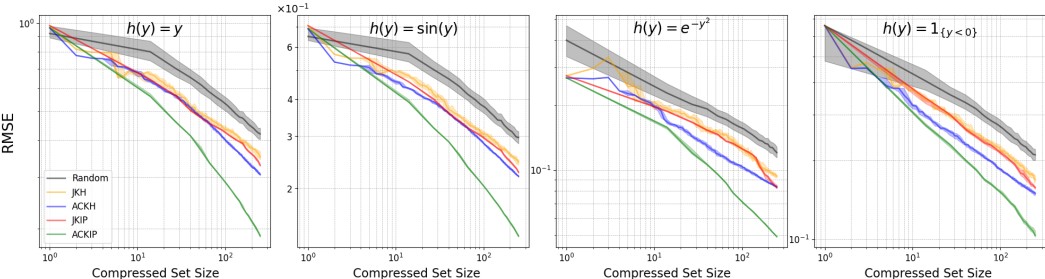

Figure 4: RMSE versus size of compressed set for the *Superconductivity* data; the RMSE is calculated against the full data estimates of $\mathbb{E}[h(Y) \mid X = \boldsymbol{x}_i]$ as the true values are not available.

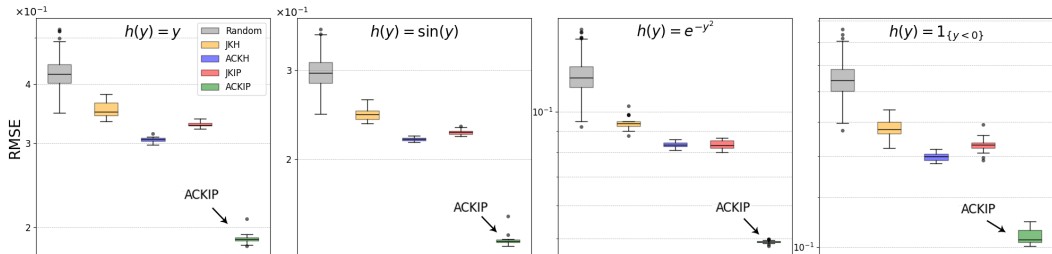

Figure 5: RMSE achieved by compressed sets of size $m = 250$ constructed by each method for the *Superconductivity* data. The IQR is highlighted for each method with outliers calculated as being above $Q_3 + 1.5\text{IQR}$ and below $Q_1 - 1.5\text{IQR}$.

**Synthetic**: We design a challenging dataset with a highly non-linear feature-response relationship and pronounced heteroscedastic noise, referring to it as *Heteroscedastic* going forward. Let

$\mathbb{P}_X = \mathcal{N}(0, 2^2)$ and $\mathbb{P}_{Y|X=\boldsymbol{x}} = \mathcal{N}(f(\boldsymbol{x}), \sigma^2(\boldsymbol{x}))$, with $f(\boldsymbol{x}) := \sum_{i=1}^{4} a_i \exp\left(-\frac{1}{b_i}(\boldsymbol{x} - c_i)^2\right)$, and $\sigma^2(\boldsymbol{x}) := \sigma_1^2 + |\sigma_2^2 \sin(\boldsymbol{x})|$. Figure 1 compares a compressed set constructed by ACKIP, with the random sample which initialised the optimisation procedure. It demonstrates how random sampling fails to adequately represent key areas of the data cloud, and how ACKIP can construct a better representation. In Figures 6 and 7 we see that ACKIP attains the lowest RMSE across three of the four test functions. For the remaining function, all methods exhibit relatively similar performance. We also note that JKIP outperforms or achieves similar performance versus JKH across the test functions.

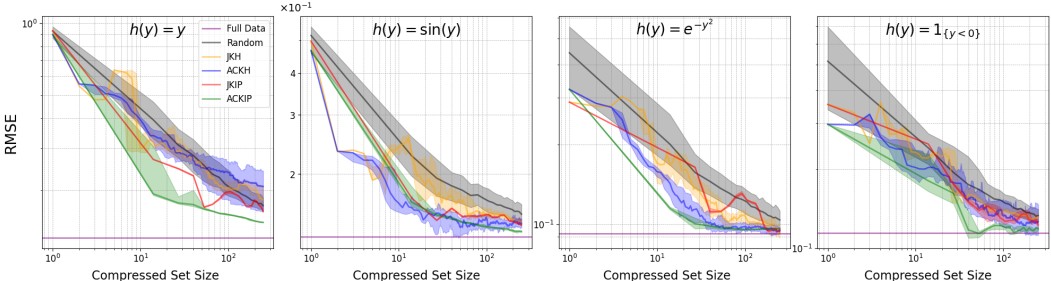

Figure 6: RMSE versus size of compressed set for the *Heteroscedastic* data with parameters set as $\boldsymbol{a} := [3, -3, 6, -6]^\top$, $\boldsymbol{b} := [1, 0.1, 2, 0.5]^\top$, $\boldsymbol{c} := [-5, -2, 2, 5]^\top$, $\sigma_1^2 = 0.1$ and $\sigma_2^2 = 0.75$. The RMSE is calculated against the true value of the conditional expectations: the performance of the full data (purple) is hence also highlighted here.

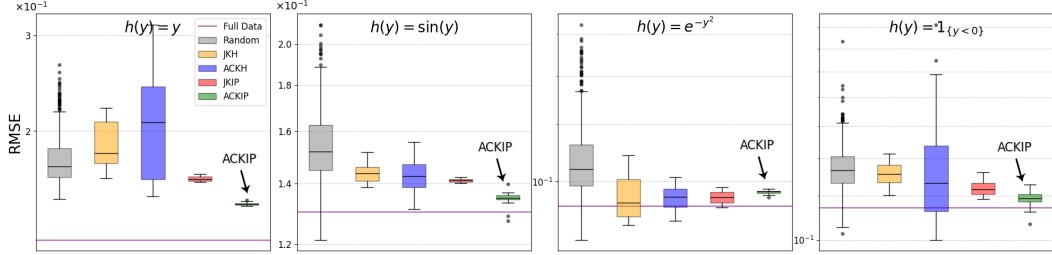

Figure 7: RMSE achieved by compressed sets of size $m = 250$ constructed by each method for the *Heteroscedastic* data. The IQR is highlighted for each method with outliers calculated as being above $Q_3 + 1.5\text{IQR}$ and below $Q_1 - 1.5\text{IQR}$.

### 5.2.2 Discrete Conditional Distributions

Conditional distributions may also be discrete, as in classification settings where responses take one of $C$ distinct values $\boldsymbol{y} \in \{0, 1, \ldots, C\}$. In such cases, applying an indicator kernel on the response space enables the KCME to serve as a consistent multi-class classifier, in contrast to methods like SVCs and GPCs [33]. The use of the indicator kernel renders standard gradient-based optimisation inapplicable on the response space, necessitating an alternative optimisation strategy (see Section C.1.3 for details). Figure 8 illustrates the strong performance of ACKIP on a challenging, synthetic, imbalanced four-class classification dataset, which we refer to as *Imbalanced*. We see that the KCME, trained with the compressed set constructed by ACKIP, estimates the class-conditional probabilities with accuracy that very closely matches that of the full dataset at just 3% of the size. In contrast, the figure also exposes the limitations of the herding approach: ACKH performs worse than random on three of the four classes, and JKIP outperforms JKH on three out of the four classes. For full details on *Imbalanced*, and additional experimental results, including on MNIST, see see Section C.1.3.

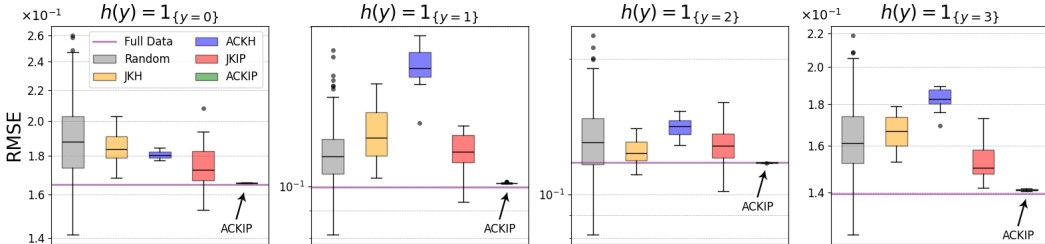

Figure 8: RMSE achieved by compressed sets of size $m = 250$ constructed by each method for the *Imbalanced* data. The RMSE is calculated against the true value of the conditional expectations: the performance of the full data (purple) is hence also highlighted here. The IQR is highlighted for each method with outliers calculated as being above $Q_3 + 1.5\text{IQR}$ and below $Q_1 - 1.5\text{IQR}$.

## 6 Discussion

**Applicability**: ACKIP generates a compressed set that enables efficient estimation (from $\mathcal{O}(n^3)$ to $\mathcal{O}(m^3 + m^2 n)$) and evaluation (from $\mathcal{O}(n^2)$ to $\mathcal{O}(m^2)$) of the KCME while maintaining a close approximation to the true KCME in terms of the AMCMD. The KCME is widely used across various applications [17–33], despite its original $\mathcal{O}(n^3)$ computational cost. By reducing this to $\mathcal{O}(n)$, whilst impacting empirical performance minimally, our approach may significantly expand the range of scenarios where the KCME can be practically applied. In particular, wherever one may have used a random subsample of size $m$ to estimate the KCME with cost $\mathcal{O}(m^3)$, ACKIP can be inserted, where running even just a few iterations of the algorithm increases the efficacy of the random sample at just $\mathcal{O}(nm^2)$ additional cost.

**Limitations**: Our algorithms currently lack a formal convergence guarantee. Although empirical results consistently show convergence across a wide range of experimental settings, including both real-world and synthetic conditional distributions, continuous and discrete, a rigorous theoretical proof remains open. In practical terms, our approach depends on computing kernel gradients, which may be unsuitable for data domains where gradients are ill-defined or difficult to interpret, such as graphs or text [46, 47]. In such cases, gradient-free alternatives inspired by Kernel Thinning [12–14] may be preferable to versions of JKH and ACKH that greedily select optimal sample pairs directly from the existing dataset (see Algorithms 5 and 6).

## 7 Conclusions

We showed that existing distribution compression methods can be extended to target the joint distribution, introducing JKH and JKIP. In particular, JKIP removes the heuristic limitations of greedy optimisation by jointly optimising the compressed set, while preserving the same computational cost. We then presented the AMCMD, an extension of the MCMD that defines a proper metric on families of conditional distributions. We derive a closed form estimate of the AMCMD and demonstrate that it can be consistently estimated in the regime of our compression algorithms. Then, leveraging the AMCMD, we proposed ACKH and ACKIP, two linear-time conditional distribution compression algorithms that are the first of their kind. Experimentation across a range of scenarios indicates that it is preferable to compress the conditional distribution directly using ACKIP or ACKH, rather than through the joint distribution via JKH or JKIP. Moreover, we see that the greedy optimisation approach of ACKH limits its empirical performance, and increases its computational cost, versus ACKIP. Finally, we also note that JKIP consistently outperforms JKH across our experimental settings. This work opens up numerous promising avenues for future research; for a detailed discussion of potential directions, see Section A.2.

## 8 Acknowledgements

We would like to thank the EPSRC Centre for Doctoral Training in Computational Statistics and Data Science (COMPASS), EP/S023569/1 for funding Dominic Broadbent's PhD studentship.

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

# Appendix

## Table of Contents

## A   Further Discussion

In this section we include further discussions and conclusions that could not fit in the main body, including related work, applications, limitations and future work.

### A.1   Related Work

#### A.1.1   Standard Distribution Compression

As noted in the introduction, numerous distribution compression methods exist which target the distribution of unlabelled data. Most of these are kernel-based methods that target the MMD, with one notable exception (Support Points), which can actually be shown to be equivalent to a kernel-based method for a specific choice of kernel. To the best of our knowledge, no existing method performs joint or conditional distribution compression as defined in this work. We now briefly summarise the existing approaches.

**Kernel Herding** [3, 41] constructs a compressed set by greedily minimising the MMD, optimising one point at a time. Let $\boldsymbol{X} := [\boldsymbol{x}_1, \boldsymbol{x}_2, \ldots, \boldsymbol{x}_n]^\top \subset \mathbb{R}^d$ be an i.i.d. sample from the target distribution $\mathbb{P}_X$, and let $\boldsymbol{Z} := [\boldsymbol{z}_1, \boldsymbol{z}_2, \ldots, \boldsymbol{z}_m]^\top \subset \mathbb{R}^d$ denote the current compressed set. At each iteration, the next point is chosen by solving

$$\boldsymbol{z}_{m+1} = \underset{\boldsymbol{z} \in \mathbb{R}^d}{\arg\min} \ \frac{1}{m+1} \sum_{j=1}^{m} k(\boldsymbol{z}, \boldsymbol{z}_j) - \mathbb{E}_{\boldsymbol{x} \sim \mathbb{P}_X} \left[ k(\boldsymbol{z}, \boldsymbol{x}) \right],$$

after which the compressed set is updated to $\boldsymbol{Z} := [\boldsymbol{z}_1, \boldsymbol{z}_2, \ldots, \boldsymbol{z}_m, \boldsymbol{z}_{m+1}]^\top$, and the process is repeated. This greedy strategy has the advantage of focusing computational effort on optimising one new point at a time while leaving previously selected points fixed. However, it may yield suboptimal solutions, as earlier selections are never revisited or refined. This optimisation approach is used to target the joint distribution in Joint Kernel Herding, and the conditional distribution in Average Conditional Kernel Herding.

**Gradient Flow** [11] methods address the problem of distribution compression by solving a discretised Wasserstein gradient flow of the MMD. In practice, this corresponds to performing gradient descent on all points in the compressed set simultaneously, i.e., solving

$$\arg\min_{\boldsymbol{Z} \subset \mathbb{R}^d} \quad \frac{1}{m^2} \sum_{i,j=1}^m k(\boldsymbol{z}_i, \boldsymbol{z}_j) - \frac{2}{m} \sum_{i=1}^m \mathbb{E}_{\boldsymbol{x} \sim \mathbb{P}_X} [k(\boldsymbol{z}_i, \boldsymbol{x})].$$

Under certain technical convexity assumptions on the objective [11], this method can be shown to converge to the global optimum; in practice, however, one should only expect to obtain a locally optimal solution. We take this optimisation approach to target the joint distribution in Joint Kernel Inducing Points, and the conditional distribution in Average Conditional Kernel Inducing Points.

**Kernel Thinning** [12–14] constructs a compressed set that is a proper subset of the original dataset, i.e., all points in the compressed set are drawn directly from the input data. This restriction stems from the method's original motivation of thinning the output of Markov Chain Monte Carlo (MCMC) methods, where subsampling is commonly referred to as standard thinning. Kernel thinning proceeds via a two-stage procedure targeting the MMD. First, the input dataset is probabilistically split into $2^m$ MMD-balanced candidate sets, each of size $\lfloor n/2^m \rfloor$, discarding the remaining $n - 2^m \lfloor n/2^m \rfloor$ points if $n$ is not evenly divisible. Second, the best candidate set from this partitioning is selected, and then greedily refined by iteratively replacing points with others from the original dataset whenever doing so improves the MMD. This construction enables the derivation of convergence rates that are state of the art in the literature, however practically it has the significant disadvantage of throwing away potentially significant amounts of information in the $n - 2^m \lfloor n/2^m \rfloor$ points.

**Support Points** [10] is a method for constructing compressed sets that takes a similar optimisation-based approach to Gradient Flow methods, but targets the *Energy Distance* (ED) rather than the MMD. The ED between distributions $\mathbb{P}_X$ and $\mathbb{Q}_X$ is defined as

$$\text{ED}(\mathbb{P}_X, \mathbb{Q}_X) := 2\,\mathbb{E}[\|\boldsymbol{a} - \boldsymbol{b}\|_2] - \mathbb{E}[\|\boldsymbol{a} - \boldsymbol{a}'\|_2] - \mathbb{E}[\|\boldsymbol{b} - \boldsymbol{b}'\|_2],$$

where $\boldsymbol{a}, \boldsymbol{a}' \sim \mathbb{P}_X$ and $\boldsymbol{b}, \boldsymbol{b}' \sim \mathbb{Q}_X$. Much like the MMD, the energy distance is zero if and only if $\mathbb{P}_X = \mathbb{Q}_X$ [10, Theorem 1]. Although Support Points is not initially expressed in kernel form, the energy distance is in fact equivalent to the MMD for a particular choice of negative-definite kernel [48].

### A.1.2 Dataset Distillation

Dataset distillation produces a compressed set which attempts to replicate the performance of the original data on a downstream task, most typically image classification [49]. However, these algorithms are model-dependent and preserve task-specific performance. In contrast, distribution compression is model-agnostic: a distributional discrepancy is targeted, independent of any downstream model. The aim is not to preserve performance on a particular model, but rather to preserve the distribution itself under compression. Therefore, the approaches introduced in this work are *task-agnostic*: the compressed set could be reused across diverse downstream applications, as discussed in the introduction.

### A.1.3 Maximum Mean Discrepancies for Conditional Distributions

The Maximum Mean Discrepancy (MMD) was introduced as a metric on the space of distributions $\mathbb{P}_X$ and has become widely used in machine learning [2]. More recently, there has been growing interest in developing MMD-like discrepancy measures for conditional distributions [18, 19, 42, 50]. One such discrepancy:

$$\mathbb{E}_{\boldsymbol{x} \sim \mathbb{P}_X} \left[ \|\mu_{Y|X=\boldsymbol{x}} - \mu_{Y'|X=\boldsymbol{x}}\|_{\mathcal{H}_l}^2 \right],$$

was introduced as the Kernel Conditional Discrepancy (KCD) by [19] to measure conditional distributional treatment effects. Independently, [42] proposed the same object under the name

Average Maximum Mean Discrepancy (AMMD) in the context of generative modelling. They are limited to cases the outer expectation must be taken with respect to the distribution of the shared conditioning variable $\mathbb{P}_X$. In contrast, we introduce the Average Maximum Conditional Mean Discrepancy (AMCMD):

$$\sqrt{\mathbb{E}_{\boldsymbol{x} \sim \mathbb{P}_{X^*}} \left[ \|\mu_{Y|X=\boldsymbol{x}} - \mu_{Y'|X'=\boldsymbol{x}}\|^2_{\mathcal{H}_l} \right]},$$

which allows the outer expectation to be taken with respect to a distinct distribution over the conditioning variable. This generalisation extends the applicability of AMCMD beyond KCD and AMMD while still recovering their results when setting $X = X' = X^*$. Moreover, we show that the AMCMD satisfies the identity of indiscernibles, and the triangle inequality under the conditions of Theorem 4.1, therefore satisfying the properties of a proper metric over the space of conditional distributions. Furthermore, unlike in [42], we show that the AMCMD can be consistently estimated using i.i.d. samples from the joint distribution, rather than requiring access to i.i.d. features $\boldsymbol{x}_i$ with conditionally independent responses $\boldsymbol{y}_{i,j}$ at each $\boldsymbol{x}_i$—a highly restrictive assumption. We derive a closed-form estimator for the AMCMD that is similar in form to the one proposed for KCD by [19], and show consistency of the estimate in the case that $X^* = X' = X$ (Corollary 4.4), which is the case under which our compression algorithms lie.

The *Conditional Maximum Mean Discrepancy* (CMMD) [50], defined as

$$\text{CMMD}(\mathbb{P}_{Y|X}, \mathbb{P}_{Y'|X} := \|\mu_{Y|X} - \mu_{Y'|X}\|_{\text{HS}(\mathcal{H}_k, \mathcal{H}_l)},$$

measures the Hilbert–Schmidt norm of the operator difference $\mu_{Y|X} - \mu_{Y'|X}$. Originally developed for use in moment-matching networks, the CMMD has since been applied to domains such as domain adaptation [51] and stochastic differential equations (SDEs) [52]. However, as noted in [42, 18], the strong assumptions required to ensure that $\mu_{Y|X}$ and $\mu_{Y'|X}$ exist as Hilbert–Schmidt operators imply that CMMD may not even be well-defined at the population level in many settings, unlike the AMCMD.

### A.1.4  Accelerating the Kernel Conditional Mean Embedding

The most closely related work is that of [53], which leverages the equivalence between the operator-theoretic estimate of the KCME and the solution to a vector-valued kernel ridge regression problem. They develop an operator-valued stochastic gradient descent algorithm to learn the KCME operator from streaming data. While both our approach and theirs utilise gradient-based methods, our work is fundamentally different. Instead of learning the *operator*, we learn the *compressed set* itself via gradient descent. Moreover, by identifying an MMD-based objective function, we establish connections with the distribution compression literature. This shift in perspective leads to a significantly different formulation and set of theoretical insights.

Beyond this, other approaches exist that are less similar. Some methods aim to speed up *evaluation* of the trained KCME at arbitrary input, rather than the training process itself: [30] and [54] use LASSO regression to construct sparse KCME estimates for efficient repeated queries. Working in Bayesian Optimisation, [25] introduce a greedy algorithm that sequentially optimises the conditional expectation of a fixed function $f \in \mathcal{H}_k$ using the KCME. Meanwhile, [55] apply sketching techniques [56] to approximate the KCME, however they do not deliver a compressed set. Finally, [57] propose a decentralised approach where a network of agents collaboratively approximates the KCME by optimising sparse covariance operators and exchanging them across the network.

In contrast to these methods, by framing the problem through an MMD-based objective function and directly optimising the compressed set, we introduce a new perspective that enables both theoretical advancements and practical improvements in scalable conditional distribution compression.

### A.1.5  Supervised Kernel Thinning

In [58], the authors apply the method of Kernel Thinning [12–14] in order to accelerate the training of two non-parametric regression models: Nadaraya-Watson (NW) kernel regression, and kernel ridge regression (KRR). That is, given a labelled dataset $\{(\boldsymbol{x}_i, y_i)\}_{i=1}^n$, $\boldsymbol{x} \in \mathbb{R}^d$, $y \in \mathbb{R}$, they construct a compressed set with Kernel Thinning, using a specialised input kernel, and then derive better-than-iid-subsampling bounds on the MSE achieved by the model trained with the compressed set.

More specifically, they use

$$k_{NW}((\boldsymbol{x}, y), (\boldsymbol{x}', y')) := k(\boldsymbol{x}, \boldsymbol{x}') + k(\boldsymbol{x}, \boldsymbol{x}') \cdot \langle y, y' \rangle_{\mathbb{R}}$$

for the construction of the compressed set targeting the Nadaraya-Watson model, and

$$k_{KRR}((\boldsymbol{x}, y), (\boldsymbol{x}', y')) := k(\boldsymbol{x}, \boldsymbol{x}')^2 + k(\boldsymbol{x}, \boldsymbol{x}') \cdot \langle y, y' \rangle_{\mathbb{R}}$$

for the construction of the compressed set targeting the kernel ridge regression model. For NW regression the feature kernel $k$ can be infinite dimensional, and their results still hold, however for KRR, their better-than-iid bounds hold only for finite dimensional feature kernels $k$.

Very importantly, they make no claims about compression of the joint or conditional distribution, and indeed it is straightforward to see that $k_{NW}$ and $k_{KRR}$ are **not** characteristic, as the linear kernel $l(y, y') := \langle y, y' \rangle_{\mathbb{R}}$ applied in both $k_{NW}$ and $k_{KRR}$ is not characteristic, and can recover changes only in the first moment between distributions.

### A.2 Future Work

**Distribution Shift**: The closely related fields of Covariate Shift [59], Distribution Shift [60], Transfer Learning [61], and Domain Adaptation [62] have been the focus of significant research in recent years. Notably, the MMD has become widely used across these areas, as evidenced by works such as [15, 63–66], among others. In this context, the AMCMD metric introduced in this work has natural applications, e.g. in covariate shift scenarios, as the choice of the distribution used for the outer expectation, denoted by $\mathbb{P}_{X^*}$, can differ from the distributions from which the observed features arise, $\mathbb{P}_X$ and $\mathbb{P}_{X'}$. Furthermore, the AMCMD is especially relevant when one encounters *Conditional Shift* [67–69], where the conditional distribution of the data changes across domains.

**Two-Sample Testing**: The MMD was originally introduced as a metric for two-sample testing—that is, for determining whether two datasets are drawn from the same underlying distribution [2]. In that work, the authors propose an MMD-based hypothesis test and analyse its statistical properties. It would be natural to undertake a similar investigation for the AMCMD, and additionally to study how conditional distribution compression affects the resulting test.

**Estimator Consistency**: The consistency of the AMCMD estimator is established—via Corollary 4.4, which follows from Theorem 4.5 of [19]—in the special case where $X \neq X' \neq X^*$. This setting aligns with our application of the AMCMD, as it corresponds to the regime in which our compression algorithms operate. However, it would be interesting to extend the consistency result to the most general case, where $X \neq X' \neq X^*$. Moreover, a promising direction for future work is to characterise the conditions under which convergence rates can be guaranteed. This includes both the well-specified case, where $\mu_{Y|X} \in \mathcal{H}_\Gamma$ [18, 19, 70], and the more general misspecified setting, where $\mu_{Y|X}$ is not assumed to lie in $\mathcal{H}_\Gamma$ [71, 72].

**Differential Privacy**: In the work of [73], an optimisation procedure similar to that used in JKIP/ACKIP is used to compress a dataset for training a Kernel Ridge Regression model. Notably, they demonstrate that test performance remains strong even with significant corruption of the compressed set, suggesting potential applications in privacy preservation. In the context of *Bayesian Coresets* [74], [75] introduced the concept of *pseudocoresets*, where they apply stochastic gradient descent to optimise a compressed set targeting a Bayesian posterior, and further establish differential privacy guarantees by corrupting gradients. It would be interesting to investigate the impact of corruption on the performance of compressed sets in our setting and derive corresponding differential privacy guarantees.

**Global Conditional Distribution Compression**: The compressed sets generated by ACKH and ACKIP focus on compressing the conditional distribution in regions where $\mathbb{P}_X$ has high density, due to the use of the AMCMD objective function. An interesting direction would be to develop methods that provide a more uniform weighting across the entire conditioning space, ensuring balanced compression regardless of feature density variations. This may be particularly valuable for cases where data observed at the tails of the feature distribution are especially important e.g. in health related scenarios.

**Alternative Optimisation Strategies**: Further exploration of alternative optimisation strategies targeting the AMCMD could also be valuable. Potential approaches include algorithms inspired by Kernel Thinning, second-order methods such as Newton or Quasi-Newton techniques, and metaheuristic strategies like simulated annealing for identifying global optima.

**Real-world Applications**: Finally, it would be interesting to see how the compressed sets generated by ACKIP perform when used to estimate a KCME applied in the important real-world downstream tasks [17–32], beyond multi-class classification [33] which we have explored in this work.

# B  Proofs

In this section, we provide technical proofs for the results in the main paper, and rigorously describe and discuss the assumptions that we adopt throughout the paper.

## B.1  Technical Assumptions

The assumptions that we work under are laid out in this section, alongside commentary on their restrictiveness.

In order to guarantee the existence of the kernel conditional mean embedding $\mu_{Y|X} := \mathbb{E}_{\mathbb{P}_{Y|X}}[l(Y, \cdot)]$, [18] require that $l : \mathcal{Y} \times \mathcal{Y} \to \mathbb{R}$ is measurable and $\mathbb{E}_{\mathbb{P}_Y}[\sqrt{l(Y,Y)}] < \infty$. However, to ensure that $\mu_{Y|X}$ is an element of the space of equivalence classes of measurable functions $L^2(\mathcal{X}, \mathbb{P}_X; \mathcal{H}_l)$, we actually require that $\mathbb{E}_{\mathbb{P}_Y}[l(Y,Y)] < \infty$ [18]. This fact is needed for the proof of Theorem 4.1, hence we make this slightly stronger integrability assumption.

For there to exist a deterministic function $F_{Y|X} : \mathcal{X} \to \mathcal{H}_l$ such that $\mu_{Y|X} = F_{Y|X} \circ X$, we require that $\mathcal{H}_l$ is separable, which is not a restrictive assumption, and is guaranteed if the kernel $l : \mathcal{X} \times \mathcal{X} \to \mathbb{R}$ is continuous [18], or if $\mathcal{H}_l$ is finite dimensional, e.g. when $l$ is the indicator kernel defined on a subset of the natural numbers. Recall that in the main body of this work, and for the remainder of this section, when we refer to $\mu_{Y|X}$, we mean $F_{Y|X}$.

In various theorems throughout this section we will make assumptions that kernels are *bounded*, *characteristic* or *universal*. Assuming a kernel $k$ defined on $\mathcal{X}$ is

1. *bounded* ensures that there exists some constant $B > 0$ such that $\sup_{\boldsymbol{x} \in \mathcal{X}} k(\boldsymbol{x}, \boldsymbol{x}) \leq B$. This assumption is trivially satisfied for many commonly used kernels such as the Gaussian, Laplacian and indicator kernels.

2. *characteristic* ensures that the corresponding kernel mean embedding $\mu_X$ is injective, and hence the corresponding $\text{MMD}(\mathbb{P}_X, \mathbb{P}_{X'})$ is a proper metric. On $\mathbb{R}^d$, the Gaussian, Laplacian, B-spline, inverse multi-quadratics, and the Matérn class of kernels can be shown to be characteristic [40], and on $\mathbb{N}^C := \{0, 1, \dots, C\}$ the indicator kernel is characteristic [33]. Note that $k \otimes l : (\mathcal{X} \times \mathcal{Y}) \times (\mathcal{X} \times \mathcal{Y}) \to \mathbb{R}$ is characteristic in the case that $k$ and $l$ are continuous, bounded and translation-invariant kernels (e.g. the Gaussian, Matérn and Laplace kernels) (Theorem 4 [40]).

3. *universal* ensures it is continuous and that the RKHS $\mathcal{H}_k$ is dense on the space of continuous functions $C(\mathcal{X})$. That is, for every function $f \in C(\mathcal{X})$, and every $\epsilon > 0$, there exists a function $g \in \mathcal{H}_k$ such that $\|f - g\|_\infty \leq \epsilon$ [76]. This assumption is satisfied for many common kernel functions, e.g. the Gaussian of the Laplacian [18].

## B.2  Equivalence of Optimising the JMMD and Joint Kernel Herding

In this subsection, in order to guarantee the existence of the joint kernel mean embedding $\mu_{X,Y} \in \mathcal{H}_k \otimes \mathcal{H}_l$, we must assume that $k : \mathcal{X} \times \mathcal{X} \to \mathbb{R}$ is measurable with $\mathbb{E}_{\mathbb{P}_X}[k(X,X)] < \infty$, and that $l : \mathcal{Y} \times \mathcal{Y} \to \mathbb{R}$ is measurable with $\mathbb{E}_{\mathbb{P}_Y}[l(Y,Y)] < \infty$. We also reiterate the assumption from the main body that $\|k(\boldsymbol{x}, \cdot), l(\boldsymbol{y}, \cdot)\| = R$ for all $\boldsymbol{x} \in \mathcal{X}$ and $\boldsymbol{y} \in \mathcal{Y}$, where $R$ is a constant. This of course implies that

$$\begin{aligned}
\|k(\boldsymbol{x}, \cdot)l(\boldsymbol{y}, \cdot)\|^2_{\mathcal{H}_k \otimes \mathcal{H}_l} &= \langle k(\boldsymbol{x}, \cdot)l(\boldsymbol{y}, \cdot), k(\boldsymbol{x}, \cdot)l(\boldsymbol{y}, \cdot) \rangle_{\mathcal{H}_k \otimes \mathcal{H}_l} \\
&= \langle k(\boldsymbol{x}, \cdot), k(\boldsymbol{x}, \cdot) \rangle_{\mathcal{H}_k} \cdot \langle l(\boldsymbol{y}, \cdot), l(\boldsymbol{y}, \cdot) \rangle_{\mathcal{H}_l} \\
&= k(\boldsymbol{x}, \boldsymbol{x})l(\boldsymbol{y}, \boldsymbol{y}) = R^2 \text{ for all } \boldsymbol{x} \in \mathcal{X} \text{ and } \boldsymbol{y} \in \mathcal{Y}.
\end{aligned}$$

Now, assuming we are at the $(m+1)^{\text{th}}$ iteration of the Joint Kernel Herding algorithm, having already constructed a compressed set of size $m$, $\mathcal{C}^m := \{(\tilde{\boldsymbol{x}}_j, \tilde{\boldsymbol{y}}_j)\}_{j=1}^m$, the next pair is chosen as the solution

to the optimisation problem

$$\arg\min_{(\boldsymbol{x},\boldsymbol{y})\in\mathcal{X}\times\mathcal{Y}} \frac{1}{m+1}\sum_{j=1}^{m}k(\boldsymbol{x},\tilde{\boldsymbol{x}}_j)l(\boldsymbol{y},\tilde{\boldsymbol{y}}_j) - \mathbb{E}_{\mathbb{P}_{X,Y}}\left[k(\boldsymbol{x},X)l(\boldsymbol{y},Y)\right].$$

We now show that these updates greedily optimise the JMMD$(\mathbb{P}_{X,Y},\tilde{\mathbb{P}}_{X,Y})$ between the joint kernel mean embedding estimated with the compressed set, and the true joint kernel mean embedding. Adding $(\boldsymbol{x},\boldsymbol{y})$ to the compressed set $\mathcal{C}^m$, we compute the JMMD$(\mathbb{P}_{X,Y},\tilde{\mathbb{P}}_{X,Y})$ as

$$\left\|\mu_{X,Y} - \frac{1}{m+1}\left(\sum_{j=1}^{m}k(\tilde{\boldsymbol{x}}_j,\cdot)l(\tilde{\boldsymbol{y}}_j,\cdot) + k(\boldsymbol{x},\cdot)l(\boldsymbol{y},\cdot)\right)\right\|_{\mathcal{H}_k\otimes\mathcal{H}_l}^2$$

$$\underset{(a)}{=} \langle\mu_{X,Y},\mu_{X,Y}\rangle_{\mathcal{H}_k\otimes\mathcal{H}_l}$$

$$- 2\left\langle\mu_{X,Y}, \frac{1}{m+1}\left(\sum_{j=1}^{m}k(\tilde{\boldsymbol{x}}_j,\cdot)l(\tilde{\boldsymbol{y}}_j,\cdot) + k(\boldsymbol{x},\cdot)l(\boldsymbol{y},\cdot)\right)\right\rangle_{\mathcal{H}_k\otimes\mathcal{H}_l}$$

$$+ \frac{1}{(m+1)^2}\left\langle\left(\sum_{j=1}^{m}k(\tilde{\boldsymbol{x}}_j,\cdot)l(\tilde{\boldsymbol{y}}_j,\cdot) + k(\boldsymbol{x},\cdot)l(\boldsymbol{y},\cdot)\right), \left(\sum_{j=1}^{m}k(\tilde{\boldsymbol{x}}_j,\cdot)l(\tilde{\boldsymbol{y}}_j,\cdot) + k(\boldsymbol{x},\cdot)l(\boldsymbol{y},\cdot)\right)\right\rangle_{\mathcal{H}_k\otimes\mathcal{H}_l}$$

$$\underset{(b)}{=} \langle\mu_{X,Y},\mu_{X,Y}\rangle_{\mathcal{H}_k\otimes\mathcal{H}_l}$$

$$- \frac{2}{m+1}\left(\sum_{j=1}^{m}\mathbb{E}_{(\boldsymbol{x}',\boldsymbol{y}')\sim\mathbb{P}_{X,Y}}\left[k(\tilde{\boldsymbol{x}}_j,\boldsymbol{x}')l(\tilde{\boldsymbol{y}}_j,\boldsymbol{y}')\right] + \mathbb{E}_{(\boldsymbol{x}',\boldsymbol{y}')\sim\mathbb{P}_{X,Y}}\left[k(\boldsymbol{x},\boldsymbol{x}')l(\boldsymbol{y},\boldsymbol{y}')\right]\right)$$

$$+ \frac{1}{(m+1)^2}\left(\sum_{i,j=1}^{m}k(\tilde{\boldsymbol{x}}_i,\tilde{\boldsymbol{x}}_j)l(\tilde{\boldsymbol{y}}_i,\tilde{\boldsymbol{y}}_j) + \sum_{i=1}^{m}k(\tilde{\boldsymbol{x}}_i,\boldsymbol{x})l(\tilde{\boldsymbol{y}}_i,\boldsymbol{y}) + \sum_{j=1}^{m}k(\boldsymbol{x},\tilde{\boldsymbol{x}}_j)l(\boldsymbol{y},\tilde{\boldsymbol{y}}_j) + k(\boldsymbol{x},\boldsymbol{x})l(\boldsymbol{y},\boldsymbol{y})\right)$$

$$\underset{(c)}{=} C_1 + C_2 + C_3 + C_4 - \frac{2}{m+1}\mathbb{E}_{(\boldsymbol{x}',\boldsymbol{y}')\sim\mathbb{P}_{X,Y}}\left[k(\boldsymbol{x},\boldsymbol{x}')l(\boldsymbol{y},\boldsymbol{y}')\right] + \frac{2}{(m+1)^2}\sum_{j=1}^{m}k(\boldsymbol{x},\tilde{\boldsymbol{x}}_j)l(\boldsymbol{y},\tilde{\boldsymbol{y}}_j),$$

where (a) follows from expanding the squared norm; (b) follows from linearity of inner products and the definition of the joint kernel mean embedding; and (c) follows from setting $C_1 := \langle\mu_{X,Y},\mu_{X,Y}\rangle_{\mathcal{H}_k\otimes\mathcal{H}_l}$, $C_2 := -\frac{2}{m+1}\sum_{j=1}^{m}\mathbb{E}_{(\boldsymbol{x}',\boldsymbol{y}')\sim\mathbb{P}_{X,Y}}\left[k(\tilde{\boldsymbol{x}}_j,\boldsymbol{x}')l(\tilde{\boldsymbol{y}}_j,\boldsymbol{y}')\right]$, $C_3 := \sum_{i,j=1}^{m}k(\tilde{\boldsymbol{x}}_i,\tilde{\boldsymbol{x}}_j)l(\tilde{\boldsymbol{y}}_i,\tilde{\boldsymbol{y}}_j)$, and $C_4 = \frac{1}{(m+1)^2}k(\boldsymbol{x},\boldsymbol{x})l(\boldsymbol{y},\boldsymbol{y})$, where we have assumed $C_4$ is constant. Now, as we are optimising with respect to $(\boldsymbol{x},\boldsymbol{y})$, we can ignore those invariant terms, and solve the optimisation problem

$$\arg\min_{(\boldsymbol{x},\boldsymbol{y})\in\mathcal{X}\times\mathcal{Y}} \frac{2}{(m+1)^2}\sum_{j=1}^{m}k(\boldsymbol{x},\tilde{\boldsymbol{x}}_j)l(\boldsymbol{y},\tilde{\boldsymbol{y}}_j) - \frac{2}{m+1}\mathbb{E}_{(\boldsymbol{x}',\boldsymbol{y}')\sim\mathbb{P}_{X,Y}}\left[k(\boldsymbol{x},\boldsymbol{x}')l(\boldsymbol{y},\boldsymbol{y}')\right]$$

which is the change in JMMD$(\mathbb{P}_{X,Y},\tilde{\mathbb{P}}_{X,Y})$ from adding the point $(\boldsymbol{x},\boldsymbol{y})$ to the compressed set. Note that this is exactly equivalent to solving the optimisation problem

$$\arg\min_{(\boldsymbol{x},\boldsymbol{y})\in\mathcal{X}\times\mathcal{Y}} \frac{1}{m+1}\sum_{j=1}^{m}k(\boldsymbol{x},\tilde{\boldsymbol{x}}_j)l(\boldsymbol{y},\tilde{\boldsymbol{y}}_j) - \mathbb{E}_{\mathbb{P}_{X,Y}}\left[k(\boldsymbol{x},X)l(\boldsymbol{y},Y)\right],$$

which is precisely the update used in Joint Kernel Herding.

**Remark B.1.** In general, one can not usually evaluate the joint expectation in (2), and hence this is estimated using the samples from $\mathcal{D}$, corresponding to optimising the JMMD$(\hat{\mathbb{P}}_{X,Y},\tilde{\mathbb{P}}_{X,Y})$.

## B.3 Derivation of JKH Objective and Gradients

We denote $\mathcal{L}_m^{\mathcal{D}} : \mathbb{R}^d \times \mathbb{R}^p \to \mathbb{R}$ to be the estimate of the objective in (2), using the entire dataset $\mathcal{D}$, then

$$
\mathcal{L}_m^{\mathcal{D}}(\boldsymbol{x}, \boldsymbol{y}) := \frac{1}{m+1} \sum_{j=1}^m k(\boldsymbol{x}, \tilde{\boldsymbol{x}}_j) k(\boldsymbol{y}, \tilde{\boldsymbol{y}}_j) - \frac{1}{n} \sum_{i=1}^n k(\boldsymbol{x}, \boldsymbol{x}_i) k(\boldsymbol{y}, \boldsymbol{y}_i)
$$

$$
= \frac{1}{m+1} \tilde{\boldsymbol{K}}_m(\boldsymbol{x})^\top \tilde{\boldsymbol{L}}_m(\boldsymbol{y}) - \frac{1}{n} \boldsymbol{K}_n(\boldsymbol{x})^\top \boldsymbol{L}_n(\boldsymbol{y}) \tag{12}
$$

where $\tilde{\boldsymbol{K}}_m(\boldsymbol{x}) := [k(\boldsymbol{x}, \tilde{\boldsymbol{x}}_1), \ldots, k(\boldsymbol{x}, \tilde{\boldsymbol{x}}_m)]^\top$, and $\boldsymbol{K}_n(\boldsymbol{x}) := [k(\boldsymbol{x}, \boldsymbol{x}_1), \ldots, k(\boldsymbol{x}, \boldsymbol{x}_n)]^\top$, $\tilde{\boldsymbol{L}}_m(\boldsymbol{y}) := [l(\boldsymbol{y}, \tilde{\boldsymbol{y}}_1), \ldots, l(\boldsymbol{y}, \tilde{\boldsymbol{y}}_m)]$, and $\boldsymbol{L}_n(\boldsymbol{y}) := [l(\boldsymbol{y}, \boldsymbol{y}_1), \ldots, l(\boldsymbol{y}, \boldsymbol{y}_n)]^\top$.

We solve the optimisation problem in (2) using gradient descent, hence we need to derive gradients of (12) with respect to both $\boldsymbol{x}$ and $\boldsymbol{y}$. It is straightforward to see that

$$
\nabla_{\boldsymbol{x}} \mathcal{L}_m^{\mathcal{D}}(\boldsymbol{x}, \boldsymbol{y}) = \frac{1}{m+1} \nabla_{\boldsymbol{x}} \tilde{\boldsymbol{K}}_m(\boldsymbol{x})^\top \tilde{\boldsymbol{L}}_m(\boldsymbol{y}) - \frac{1}{n} \nabla_{\boldsymbol{x}} \boldsymbol{K}_n(\boldsymbol{x})^\top \boldsymbol{L}_n(\boldsymbol{y}) \tag{13}
$$

and

$$
\nabla_{\boldsymbol{y}} \mathcal{L}_m^{\mathcal{D}}(\boldsymbol{x}, \boldsymbol{y}) = \frac{1}{m+1} \tilde{\boldsymbol{K}}_m(\boldsymbol{x})^\top \nabla_{\boldsymbol{y}} \tilde{\boldsymbol{L}}_m(\boldsymbol{y}) - \frac{1}{n} \boldsymbol{K}_n(\boldsymbol{x})^\top \nabla_{\boldsymbol{y}} \boldsymbol{L}_n(\boldsymbol{y}) \tag{14}
$$

where we have defined

$$
\nabla_{\boldsymbol{x}} \tilde{\boldsymbol{K}}_m(\boldsymbol{x}) := [\nabla_{\boldsymbol{x}} k(\boldsymbol{x}, \tilde{\boldsymbol{x}}_1), \ldots, \nabla_{\boldsymbol{x}} k(\boldsymbol{x}, \tilde{\boldsymbol{x}}_m)]^\top \in \mathbb{R}^{m \times p},
$$

$$
\nabla_{\boldsymbol{x}} \boldsymbol{K}_n(\boldsymbol{x}) := [\nabla_{\boldsymbol{x}} k(\boldsymbol{x}, \boldsymbol{x}_1), \ldots, \nabla_{\boldsymbol{x}} k(\boldsymbol{x}, \boldsymbol{x}_n)]^\top \in \mathbb{R}^{n \times p},
$$

$$
\nabla_{\boldsymbol{y}} \tilde{\boldsymbol{L}}_m(\boldsymbol{y}) := [\nabla_{\boldsymbol{y}} l(\boldsymbol{y}, \tilde{\boldsymbol{y}}_1), \ldots, \nabla_{\boldsymbol{y}} l(\boldsymbol{y}, \tilde{\boldsymbol{y}}_m)]^\top \in \mathbb{R}^{m \times p},
$$

$$
\nabla_{\boldsymbol{y}} \boldsymbol{L}_n(\boldsymbol{y}) := [\nabla_{\boldsymbol{y}} l(\boldsymbol{y}, \boldsymbol{y}_1), \ldots, \nabla_{\boldsymbol{y}} l(\boldsymbol{y}, \boldsymbol{y}_n)]^\top \in \mathbb{R}^{n \times p}.
$$

Hence, it is easy to see that the gradient can be computed with $\mathcal{O}(m+n)$ time and storage complexity.

## B.4 Derivation of JKIP Objective and Gradients

Before we state and prove our lemma, we first recall some properties of tensor calculus.

### B.4.1 Tensor Calculus

Let $F : \mathbb{R}^{m \times d} \to \mathbb{R}^{m \times m}$ be a matrix-valued function taking matrix-valued inputs with

$$
K(\boldsymbol{X}) := \begin{bmatrix} k(\boldsymbol{x}_1, \boldsymbol{x}_1) & \ldots & k(\boldsymbol{x}_1, \boldsymbol{x}_m) \\ \vdots & \ddots & \vdots \\ k(\boldsymbol{x}_m, \boldsymbol{x}_1) & \ldots & k(\boldsymbol{x}_m, \boldsymbol{x}_m) \end{bmatrix} \in \mathbb{R}^{m \times m}
$$

where $k : \mathbb{R}^d \times \mathbb{R}^d \to \mathbb{R}$ will be some kernel, and $\boldsymbol{X} = [\boldsymbol{x}_1, \ldots, \boldsymbol{x}_m]^\top \in \mathbb{R}^{m \times d}$, $\boldsymbol{x}_i \in \mathbb{R}^d$ for $i = 1, \ldots, m$. Then, we have

$$
[\nabla_{\boldsymbol{X}} K(\boldsymbol{X})]_{ijl} := \nabla_{\boldsymbol{x}_l} k(\boldsymbol{x}_i, \boldsymbol{x}_j) \in \mathbb{R}^d, \quad i, j, l = 1, \ldots, m,
$$

such that $\nabla_{\boldsymbol{X}} K(\boldsymbol{X}, \boldsymbol{X}) \in \mathbb{R}^{m \times m \times m \times d}$, i.e. a fourth-order tensor. With some abuse of notation, we reproduce the usual derivative identities below, for a more in-depth review see, for example, [77, 78]:

**Trace Rule**: Given some matrix $A \in \mathbb{R}^{m \times m}$ we have

$$
\nabla_{\boldsymbol{X}} (\mathrm{Tr}(K(\boldsymbol{X})A)) = \mathrm{Tr}(\nabla_{\boldsymbol{X}} K(\boldsymbol{X})A) \in \mathbb{R}^{m \times d} \tag{15}
$$

which establishes the linearity of the gradient operator with respect to the trace. Note that the trace on the RHS is understood here to be a partial trace over the last two dimensions of the fourth-order tensor, i.e.

$$
[\mathrm{Tr}(\nabla_{\boldsymbol{X}} K(\boldsymbol{X})A)]_l = \sum_{i,j=1}^m [\nabla_{\boldsymbol{X}} K(\boldsymbol{X})]_{ijl} A_{ji} \in \mathbb{R}^d, \; l = 1, \ldots, m \tag{16}
$$

**Inverse Rule**: We also have

$$\nabla_{\boldsymbol{X}}(K(\boldsymbol{X})^{-1}) = -K(\boldsymbol{X})^{-1}\nabla_{\boldsymbol{X}}K(\boldsymbol{X})K(\boldsymbol{X})^{-1} \in \mathbb{R}^{m \times m \times m \times d}. \tag{17}$$

assuming $K(\boldsymbol{X})^{-1} \in \mathbb{R}^{m \times m}$ exists, where

$$[\nabla_{\boldsymbol{X}}(K(\boldsymbol{X})^{-1})]_{ijpq} = -\sum_{s,t=1}^{m}[K(\boldsymbol{X})^{-1}]_{is}\,[\nabla_{\boldsymbol{X}}K(\boldsymbol{X})]_{stpq}\,[K(\boldsymbol{X})^{-1}]_{tj}.$$

**Product Rule**: Given a second function $L : \mathbb{R}^{m \times d} \to \mathbb{R}^{m \times m}$, defined similarly to $K$ with kernel $l : \mathbb{R}^d \times \mathbb{R}^d \to \mathbb{R}$, then we have

$$\nabla_{\boldsymbol{X}}(K(\boldsymbol{X})L(\boldsymbol{X})) = \nabla_{\boldsymbol{X}}(K(\boldsymbol{X}))L(\boldsymbol{X}) + K(\boldsymbol{X})\nabla_{\boldsymbol{X}}(L(\boldsymbol{X})) \in \mathbb{R}^{m \times m \times m \times d}, \tag{18}$$

where we have

$$[\nabla_{\boldsymbol{X}}(K(\boldsymbol{X})L(\boldsymbol{X}))]_{ijpq} = \sum_{s=1}^{m}\Big([\nabla_{\boldsymbol{X}}K(\boldsymbol{X})]_{ispq}[L(\boldsymbol{X})]_{sj} + [K(\boldsymbol{X})]_{is}[\nabla_{\boldsymbol{X}}L(\boldsymbol{X})]_{sjpq}\Big)$$

### B.4.2 Statement and Proof of Lemma

Letting $\mathcal{L}^{\mathcal{D}} : \mathbb{R}^{m \times d} \times \mathbb{R}^{m \times p} \to \mathbb{R}$ be the estimate of the objective in (3) using the entire dataset $\mathcal{D}$, then we have

$$\mathcal{L}^{\mathcal{D}}(\tilde{\boldsymbol{X}}, \tilde{\boldsymbol{Y}}) := \frac{1}{m^2}\sum_{i,j=1}^{m}k(\tilde{\boldsymbol{x}}_i, \tilde{\boldsymbol{x}}_j)l(\tilde{\boldsymbol{y}}_i, \tilde{\boldsymbol{y}}_j) - \frac{2}{mn}\sum_{i,j=1}^{m,n}k(\tilde{\boldsymbol{x}}_i, \boldsymbol{x}_j)l(\tilde{\boldsymbol{y}}_i, \boldsymbol{y}_j)$$

$$= \frac{1}{m^2}\mathrm{Tr}\left(K_{\tilde{\boldsymbol{X}},\tilde{\boldsymbol{X}}}L_{\tilde{\boldsymbol{Y}},\tilde{\boldsymbol{Y}}}\right) - \frac{2}{mn}\mathrm{Tr}\left(K_{\tilde{\boldsymbol{X}},\boldsymbol{X}}L_{\boldsymbol{Y},\tilde{\boldsymbol{Y}}}\right)$$

where $[K_{\tilde{\boldsymbol{X}}\tilde{\boldsymbol{X}}}]_{ij} := k(\tilde{\boldsymbol{x}}_i, \tilde{\boldsymbol{x}}_j)$, $[K_{\tilde{\boldsymbol{X}}\boldsymbol{X}}]_{iq} := k(\tilde{\boldsymbol{x}}_i, \boldsymbol{x}_q)$, $[L_{\tilde{\boldsymbol{Y}}\tilde{\boldsymbol{Y}}}]_{ij} := l(\tilde{\boldsymbol{y}}_i, \tilde{\boldsymbol{y}}_j)$, and $[L_{\tilde{\boldsymbol{Y}}\boldsymbol{Y}}]_{iq} := l(\tilde{\boldsymbol{y}}_i, \boldsymbol{y}_q)$, for $i, j = 1, \ldots, m$ and $q = 1, \ldots, n$.

Note that optimising this objective with respect to the compressed set $\mathcal{C} = (\tilde{\boldsymbol{X}}, \tilde{\boldsymbol{Y}})$ is equivalent to optimising the JMMD between $\hat{\mathbb{P}}_{X,Y}$ and $\tilde{\mathbb{P}}_{X,Y}$, that is, the empirical joint distributions of the full dataset $\mathcal{D}$ and the compressed set $\mathcal{C}$ respectively:

$$\mathrm{JMMD}^2\left(\hat{\mathbb{P}}_{X,Y}, \tilde{\mathbb{P}}_{X,Y}\right) = \|\hat{\mu}_{X,Y} - \tilde{\mu}_{\mathbb{P}_{X,Y}}\|^2_{\mathcal{H}_k}$$

$$= \langle\hat{\mu}_{X,Y}, \hat{\mu}_{X,Y}\rangle_{\mathcal{H}_k} - 2\langle\hat{\mu}_{X,Y}, \tilde{\mu}_{\mathbb{P}_{X,Y}}\rangle_{\mathcal{H}_k} + \langle\tilde{\mu}_{\mathbb{P}_{X,Y}}, \tilde{\mu}_{\mathbb{P}_{X,Y}}\rangle_{\mathcal{H}_k}$$

$$= \langle\hat{\mu}_{X,Y}, \hat{\mu}_{X,Y}\rangle_{\mathcal{H}_k} + \mathcal{L}^{\mathcal{D}}(\tilde{\boldsymbol{X}}, \tilde{\boldsymbol{Y}}).$$

**Lemma B.2.** *We compute the gradients of the objective function $\mathcal{L}^{\mathcal{D}} : \mathbb{R}^{m \times d} \times \mathbb{R}^{m \times p} \to \mathbb{R}$ as*

$$\nabla_{\tilde{\boldsymbol{X}}}\mathcal{L}^{\mathcal{D}}(\tilde{\boldsymbol{X}}, \tilde{\boldsymbol{Y}}) = \frac{1}{m^2}Tr\left(\nabla_{\tilde{\boldsymbol{X}}}K_{\tilde{\boldsymbol{X}},\tilde{\boldsymbol{X}}}L_{\tilde{\boldsymbol{Y}},\tilde{\boldsymbol{Y}}}\right) - \frac{2}{mn}Tr\left(\nabla_{\tilde{\boldsymbol{X}}}K_{\tilde{\boldsymbol{X}},\boldsymbol{X}}L_{\boldsymbol{Y},\tilde{\boldsymbol{Y}}}\right) \tag{19}$$

$$\nabla_{\tilde{\boldsymbol{Y}}}\mathcal{L}^{\mathcal{D}}(\tilde{\boldsymbol{X}}, \tilde{\boldsymbol{Y}}) = \frac{1}{m^2}Tr\left(K_{\tilde{\boldsymbol{X}},\tilde{\boldsymbol{X}}}\nabla_{\tilde{\boldsymbol{Y}}}L_{\tilde{\boldsymbol{Y}},\tilde{\boldsymbol{Y}}}\right) - \frac{2}{mn}Tr\left(K_{\tilde{\boldsymbol{X}},\boldsymbol{X}}\nabla_{\tilde{\boldsymbol{Y}}}L_{\boldsymbol{Y},\tilde{\boldsymbol{Y}}}\right) \tag{20}$$

*where*

$$\nabla_{\tilde{\boldsymbol{X}}}K_{\tilde{\boldsymbol{X}},\tilde{\boldsymbol{X}}} \in \mathbb{R}^{m \times m \times m \times d}, \;\; with \;\; \left[\nabla_{\tilde{\boldsymbol{X}}}K_{\tilde{\boldsymbol{X}},\tilde{\boldsymbol{X}}}\right]_{ijq} := \nabla_{\tilde{\boldsymbol{x}}_q}k(\tilde{\boldsymbol{x}}_i, \tilde{\boldsymbol{x}}_j) \in \mathbb{R}^d,$$

$$\nabla_{\tilde{\boldsymbol{X}}}K_{\tilde{\boldsymbol{X}},\boldsymbol{X}} \in \mathbb{R}^{m \times n \times m \times d}, \;\; with \;\; \left[\nabla_{\tilde{\boldsymbol{X}}}K_{\tilde{\boldsymbol{X}},\boldsymbol{X}}\right]_{ijq} := \nabla_{\tilde{\boldsymbol{x}}_q}k(\tilde{\boldsymbol{x}}_i, \boldsymbol{x}_j) \in \mathbb{R}^d,$$

$$\nabla_{\tilde{\boldsymbol{Y}}}L_{\tilde{\boldsymbol{Y}},\tilde{\boldsymbol{Y}}} \in \mathbb{R}^{m \times m \times m \times p}, \;\; with \;\; \left[\nabla_{\tilde{\boldsymbol{Y}}}L_{\tilde{\boldsymbol{Y}},\tilde{\boldsymbol{Y}}}\right]_{ijq} := \nabla_{\tilde{\boldsymbol{y}}_q}l(\tilde{\boldsymbol{y}}_i, \tilde{\boldsymbol{y}}_j) \in \mathbb{R}^p,$$

$$\nabla_{\tilde{\boldsymbol{Y}}}K_{\boldsymbol{Y},\tilde{\boldsymbol{Y}}} \in \mathbb{R}^{m \times n \times m \times p}, \;\; with \;\; \left[\nabla_{\tilde{\boldsymbol{Y}}}L_{\tilde{\boldsymbol{Y}},\boldsymbol{Y}}\right]_{ijq} := \nabla_{\tilde{\boldsymbol{y}}_q}l(\tilde{\boldsymbol{y}}_i, \boldsymbol{y}_j) \in \mathbb{R}^p,$$

*with $\mathcal{O}(mn + m^2)$ time and storage complexity, i.e. linear with respect to the size of the full dataset $\mathcal{D}$.*

**Proof**: We have

$$\mathcal{L}^{\mathcal{D}}(\tilde{\boldsymbol{X}}, \tilde{\boldsymbol{Y}}) = \frac{1}{m^2} \text{Tr}\left(K_{\tilde{\boldsymbol{X}}, \tilde{\boldsymbol{X}}} L_{\tilde{\boldsymbol{Y}}, \tilde{\boldsymbol{Y}}}\right) - \frac{2}{mn} \text{Tr}\left(K_{\tilde{\boldsymbol{X}}, \boldsymbol{X}} L_{\boldsymbol{Y}, \tilde{\boldsymbol{Y}}}\right).$$

Applying rules (15) and (18), it is straightforward to see that

$$\nabla_{\tilde{\boldsymbol{X}}} \mathcal{L}^{\mathcal{D}}(\tilde{\boldsymbol{X}}, \tilde{\boldsymbol{Y}}) = \frac{1}{m^2} \text{Tr}\left(\nabla_{\tilde{\boldsymbol{X}}} K_{\tilde{\boldsymbol{X}}, \tilde{\boldsymbol{X}}} L_{\tilde{\boldsymbol{Y}}, \tilde{\boldsymbol{Y}}}\right) - \frac{2}{mn} \text{Tr}\left(\nabla_{\tilde{\boldsymbol{X}}} K_{\tilde{\boldsymbol{X}}, \boldsymbol{X}} L_{\boldsymbol{Y}, \tilde{\boldsymbol{Y}}}\right),$$

and

$$\nabla_{\tilde{\boldsymbol{Y}}} \mathcal{L}^{\mathcal{D}}(\tilde{\boldsymbol{X}}, \tilde{\boldsymbol{Y}}) = \frac{1}{m^2} \text{Tr}\left(K_{\tilde{\boldsymbol{X}}, \tilde{\boldsymbol{X}}} \nabla_{\tilde{\boldsymbol{Y}}} L_{\tilde{\boldsymbol{Y}}, \tilde{\boldsymbol{Y}}}\right) - \frac{2}{mn} \text{Tr}\left(K_{\tilde{\boldsymbol{X}}, \boldsymbol{X}} \nabla_{\tilde{\boldsymbol{Y}}} L_{\boldsymbol{Y}, \tilde{\boldsymbol{Y}}}\right).$$

Now, in order to show that these gradients can be computed with $\mathcal{O}(mn + m^2)$ time and storage complexity, the critical observation is that the *majority* of the elements of these fourth-order tensors will be equal to zero, i.e.

$$\left[\nabla_{\tilde{\boldsymbol{X}}} K_{\tilde{\boldsymbol{X}}, \tilde{\boldsymbol{X}}}\right]_{ijq} := \nabla_{\tilde{\boldsymbol{x}}_q} k(\tilde{\boldsymbol{x}}_i, \tilde{\boldsymbol{x}}_j) = 0 \text{ when } i \neq q \text{ and } j \neq q,$$

$$\left[\nabla_{\tilde{\boldsymbol{X}}} K_{\tilde{\boldsymbol{X}}, \boldsymbol{X}}\right]_{ijq} := \nabla_{\tilde{\boldsymbol{x}}_q} k(\tilde{\boldsymbol{x}}_i, \boldsymbol{x}_j) = 0 \text{ when } i \neq q$$

and

$$\left[\nabla_{\tilde{\boldsymbol{Y}}} L_{\tilde{\boldsymbol{Y}}, \tilde{\boldsymbol{Y}}}\right]_{ijq} := \nabla_{\tilde{\boldsymbol{y}}_q} l(\tilde{\boldsymbol{y}}_i, \tilde{\boldsymbol{y}}_j) = 0 \text{ when } i \neq q \text{ and } j \neq q,$$

$$\left[\nabla_{\tilde{\boldsymbol{Y}}} L_{\tilde{\boldsymbol{Y}}, \boldsymbol{Y}}\right]_{ijq} := \nabla_{\tilde{\boldsymbol{y}}_q} l(\tilde{\boldsymbol{y}}_i, \boldsymbol{y}_j) = 0 \text{ when } i \neq q.$$

Then, by using the identity in (16), we have that,

$$\left[\text{Tr}\left(\nabla_{\tilde{\boldsymbol{X}}} K_{\tilde{\boldsymbol{X}}, \tilde{\boldsymbol{X}}} L_{\tilde{\boldsymbol{Y}}, \tilde{\boldsymbol{Y}}}\right)\right]_{qr} = \sum_{i=1}^{m} \sum_{j=1}^{m} \left[\nabla_{\tilde{\boldsymbol{X}}} K_{\tilde{\boldsymbol{X}}, \tilde{\boldsymbol{X}}}\right]_{ijqr} \left[L_{\tilde{\boldsymbol{Y}}, \tilde{\boldsymbol{Y}}}\right]_{ji}$$

$$= \sum_{i=1}^{m} \left[\nabla_{\tilde{\boldsymbol{X}}} K_{\tilde{\boldsymbol{X}}, \tilde{\boldsymbol{X}}}\right]_{iqqr} \left[L_{\tilde{\boldsymbol{Y}}, \tilde{\boldsymbol{Y}}}\right]_{qi} + \sum_{j=1}^{m} \left[\nabla_{\tilde{\boldsymbol{X}}} K_{\tilde{\boldsymbol{X}}, \tilde{\boldsymbol{X}}}\right]_{qjqr} \left[L_{\tilde{\boldsymbol{Y}}, \tilde{\boldsymbol{Y}}}\right]_{jq}$$

$$\underset{\text{(a)}}{=} 2 \sum_{i=1}^{m} \left[\nabla_{\tilde{\boldsymbol{X}}} K_{\tilde{\boldsymbol{X}}, \tilde{\boldsymbol{X}}}\right]_{iqqr} \left[L_{\tilde{\boldsymbol{Y}}, \tilde{\boldsymbol{Y}}}\right]_{iq} \tag{21}$$

for $q = 1 \ldots, m$ and $r = 1, \ldots, d$, and where (a) follows trivially from the symmetry of the kernel functions $l(\cdot, \cdot)$ and $k(\cdot, \cdot)$. Hence, here, the *only* terms we have to compute and store are

$$\nabla_{\tilde{\boldsymbol{x}}_q} k(\tilde{\boldsymbol{x}}_i, \tilde{\boldsymbol{x}}_q) \in \mathbb{R}^d, \quad i = 1, \ldots, m, \quad q = 1, \ldots, m$$

and $L_{\tilde{\boldsymbol{Y}}, \tilde{\boldsymbol{Y}}} \in \mathbb{R}^{m \times m}$, which can be accomplished with cost $\mathcal{O}(m^2)$ in *both* storage and time, ignoring any dependence on the dimension of the feature space $d$. A very similar derivation holds for the term

$$\text{Tr}\left(K_{\tilde{\boldsymbol{X}}, \tilde{\boldsymbol{X}}} \nabla_{\tilde{\boldsymbol{Y}}} L_{\tilde{\boldsymbol{Y}}, \tilde{\boldsymbol{Y}}}\right).$$

Now, tackling the cross term, we can use (16) to get that

$$\left[\text{Tr}\left(\nabla_{\tilde{\boldsymbol{X}}} K_{\tilde{\boldsymbol{X}}, \boldsymbol{X}} L_{\boldsymbol{Y}, \tilde{\boldsymbol{Y}}}\right)\right]_{qr} = \sum_{i=1}^{m} \sum_{j=1}^{n} \left[\nabla_{\tilde{\boldsymbol{X}}} K_{\tilde{\boldsymbol{X}}, \boldsymbol{X}}\right]_{ijqr} \left[L_{\boldsymbol{Y}, \tilde{\boldsymbol{Y}}}\right]_{ji}$$

$$= \sum_{j=1}^{n} \left[\nabla_{\tilde{\boldsymbol{X}}} K_{\tilde{\boldsymbol{X}}, \boldsymbol{X}}\right]_{qjqr} \left[L_{\boldsymbol{Y}, \tilde{\boldsymbol{Y}}}\right]_{jq} \tag{22}$$

with $q = 1 \ldots, m$ and $r = 1, \ldots, d$. Hence the only terms we must compute and store are

$$\nabla_{\tilde{\boldsymbol{x}}_q} k(\tilde{\boldsymbol{x}}_q, \boldsymbol{x}_j) \in \mathbb{R}^d, \quad q = 1, \ldots, m, \quad j = 1, \ldots, n,$$

and $L_{\boldsymbol{Y},\tilde{\boldsymbol{Y}}} \in \mathbb{R}^{n \times m}$, which can be accomplished with cost $\mathcal{O}(nm)$ in *both* storage and time, ignoring any dependence on the dimension of the feature space $d$. Again, a very similar derivation holds for the term

$$\mathrm{Tr}\left(K_{\tilde{\boldsymbol{X}},\boldsymbol{X}}\nabla_{\tilde{\boldsymbol{Y}}}L_{\boldsymbol{Y},\tilde{\boldsymbol{Y}}}\right).$$

Hence, the final computation and storage cost of computing the gradients is $\mathcal{O}(nm + m^2)$, i.e. *linear* in the size of the target dataset $\mathcal{D}$. ∎

**Remark B.3.** Above we have derived analytical gradients of the objective function, and shown they can be computed in linear time. In practice one computes the gradients using JAX's [79] auto-differentiation capabilities. The authors observed minimal slowdown from using auto-differentiation.

## B.5 Proof of Theorem 4.1

**Theorem B.4.** *Suppose that $l : \mathcal{Y} \times \mathcal{Y} \to \mathbb{R}$ is a characteristic kernel, that $\mathbb{P}_X$, $\mathbb{P}_{X'}$, and $\mathbb{P}_{X^*}$ are absolutely continuous with respect to each other, and that $\mathbb{P}(\cdot \mid X)$ and $\mathbb{P}(\cdot \mid X')$ admit regular versions. Then, AMCMD $\left[\mathbb{P}_{X^*},\mathbb{P}_{Y|X},\mathbb{P}_{Y'|X'}\right] = 0$ if and only if, for almost all $\boldsymbol{x} \in \mathcal{X}$ wrt $\mathbb{P}_{X^*}$, $\mathbb{P}_{Y|X=\boldsymbol{x}}(B) = \mathbb{P}_{Y'|X'=\boldsymbol{x}}(B)$ for all $B \in \mathscr{Y}$.*

*Moreover, assuming the Radon-Nikodym derivatives $\frac{d\mathbb{P}_{X^*}}{d\mathbb{P}_X}$, $\frac{d\mathbb{P}_{X^*}}{d\mathbb{P}_{X'}}$, $\frac{d\mathbb{P}_{X^*}}{d\mathbb{P}_{X''}}$ are bounded, then the triangle inequality is satisfied, i.e. AMCMD $\left[\mathbb{P}_{X^*},\mathbb{P}_{Y|X},\mathbb{P}_{Y''|X''}\right] \leq$ AMCMD $\left[\mathbb{P}_{X^*},\mathbb{P}_{Y|X},\mathbb{P}_{Y'|X'}\right] +$ AMCMD $\left[\mathbb{P}_{X^*},\mathbb{P}_{Y'|X'},\mathbb{P}_{Y''|X''}\right]$.*

**Proof**:

It is clear that

$$\mathrm{AMCMD}\left[\mathbb{P}_{X^*},\mathbb{P}_{Y|X},\mathbb{P}_{Y'|X'}\right] := \sqrt{\mathbb{E}_{\boldsymbol{x}\sim\mathbb{P}_{X^*}}\left[\|\mu_{Y|X=\boldsymbol{x}} - \mu_{Y'|X'=\boldsymbol{x}}\|_{\mathcal{H}_l}^2\right]}$$

is non-negative and symmetric in $\mathbb{P}_{Y|X}$ and $\mathbb{P}_{Y'|X'}$.

We first prove the equivalence result:

( $\Longrightarrow$ ) Assume that AMCMD $\left[\mathbb{P}_{X^*},\mathbb{P}_{Y|X},\mathbb{P}_{Y'|X'}\right] := \sqrt{\mathbb{E}_{\boldsymbol{x}\sim\mathbb{P}_{X^*}}\left[\|\mu_{Y|X=\boldsymbol{x}} - \mu_{Y'|X'=\boldsymbol{x}}\|_{\mathcal{H}_l}^2\right]} = 0$ . This implies that $\mu_{Y|X=\boldsymbol{x}} = \mu_{Y'|X'=\boldsymbol{x}}$ almost everywhere $\boldsymbol{x}$ wrt $\mathbb{P}_{X^*}$. Now, by the fact that $\mathbb{P}_{X^*}$ and $\mathbb{P}_X$ (or $\mathbb{P}_{X'}$) are absolutely continuous with respect to each other, we also have that $\mu_{Y|X=\boldsymbol{x}} = \mu_{Y'|X'=\boldsymbol{x}}$ almost everywhere $\boldsymbol{x}$ wrt $\mathbb{P}_X$ (or $\mathbb{P}_{X'}$). Hence, we must have $\mathrm{MCMD}\left(\mathbb{P}_{Y|X=\cdot},\mathbb{P}_{Y'|X'=\cdot}\right) := \left\|\mu_{Y|X=\cdot} - \mu_{Y'|X'=\cdot}\right\|_{\mathcal{H}_l} = 0$ almost everywhere $\boldsymbol{x} \in \mathcal{X}$ wrt $\mathbb{P}_X$ (or $\mathbb{P}_{X'}$). Thus, by Theorem 2.1, we have that for almost all $\boldsymbol{x} \in \mathcal{X}$ wrt $\mathbb{P}_X$ (or $\mathbb{P}_{X'}$) , $\mathbb{P}_{Y|X=\boldsymbol{x}}(B) = \mathbb{P}_{Y'|X'=\boldsymbol{x}}(B)$ for all $B \in \mathscr{Y}$. However, again by absolute continuity of measures, this is equivalent to stating that for almost all $\boldsymbol{x} \in \mathcal{X}$ wrt $\mathbb{P}_{X^*}$, $\mathbb{P}_{Y|X=\boldsymbol{x}}(B) = \mathbb{P}_{Y'|X'=\boldsymbol{x}}(B)$ for all $B \in \mathscr{Y}$.

( $\Longleftarrow$ ) Assume that for almost all $\boldsymbol{x} \in \mathcal{X}$ wrt $\mathbb{P}_{X^*}$, $\mathbb{P}_{Y|X=\boldsymbol{x}}(B) = \mathbb{P}_{Y'|X'=\boldsymbol{x}}(B)$ for all $B \in \mathscr{Y}$. Then, by the fact that $\mathbb{P}_{X^*}$ and $\mathbb{P}_X$ (or $\mathbb{P}_{X'}$) are absolutely continuous with respect to each other, and Theorem 2.1, we have that $\mathrm{MCMD}\left(\mathbb{P}_{Y|X=\cdot},\mathbb{P}_{Y'|X'=\cdot}\right) := \left\|\mu_{Y|X=\cdot} - \mu_{Y'|X'=\cdot}\right\|_{\mathcal{H}_l} = 0$ almost everywhere $\boldsymbol{x}$ wrt $\mathbb{P}_X$ (or $\mathbb{P}_{X'}$). This implies that $\mu_{Y|X=\boldsymbol{x}} = \mu_{Y'|X'=\boldsymbol{x}}$ almost everywhere $\boldsymbol{x} \in \mathcal{X}$ wrt $\mathbb{P}_{X'}$ (or $\mathbb{P}_{X'}$), and by absolutely continuity, $\mathbb{P}_{X^*}$ also. Hence, we must have AMCMD $\left[\mathbb{P}_{X^*},\mathbb{P}_{Y|X},\mathbb{P}_{Y'|X'}\right] := \sqrt{\mathbb{E}_{\boldsymbol{x}\sim\mathbb{P}_{X^*}}\left[\|\mu_{Y|X=\boldsymbol{x}} - \mu_{Y'|X'=\boldsymbol{x}}\|_{\mathcal{H}_l}^2\right]} = 0$.

Finally, we show the triangle inequality, that is, given additional random variables $X'' : \Omega \to \mathcal{X}$, $Y'' : \Omega \to \mathcal{Y}$, with conditional distribution $\mathbb{P}_{Y''|X''}$ and KCME $\mu_{Y''|X''}$, we show (suppressing the first argument $\mathbb{P}_{X^*}$)

$$\mathrm{AMCMD}\left[\mathbb{P}_{Y|X},\mathbb{P}_{Y''|X''}\right] \leq \mathrm{AMCMD}\left[\mathbb{P}_{Y|X},\mathbb{P}_{Y'|X'}\right] + \mathrm{AMCMD}\left[\mathbb{P}_{Y'|X'},\mathbb{P}_{Y''|X''}\right].$$

Firstly, denote $L^2(\mathcal{X}, \mathbb{P}_{X^*}; \mathcal{H}_l)$ to be the Banach space of (equivalence classes of) measurable functions $f : \mathcal{X} \to \mathcal{H}_l$ such that $\|f\|_{\mathcal{H}_l}^2$ is $\mathbb{P}_{X^*}$-integrable with norm defined by

$$\|f\|_2 := \left( \int_{\mathcal{X}} \|f(\boldsymbol{x})\|_{\mathcal{H}_l}^2 \, \mathrm{d}\mathbb{P}_{X^*}(\boldsymbol{x}) \right)^{\frac{1}{2}}.$$

Now, it is shown in [18], that $\mu_{Y|X}$ belongs to $L^2(\mathcal{X}, \mathbb{P}_X; \mathcal{H}_l)$, where we stress that the measure is $\mathbb{P}_X$, *not* $\mathbb{P}_{X^*}$. They arrive at this conclusion by the measurability of $\mu_{Y|X}$, and by noting that

$$\int_{\mathcal{X}} \|\mu_{Y|X=\boldsymbol{x}}\|_{\mathcal{H}_l}^2 \mathrm{d}\mathbb{P}_X(\boldsymbol{x}) \underset{(a)}{=} \mathbb{E}_{\mathbb{P}_X} \left[ \| \mathbb{E}_{\mathbb{P}_{Y|X}} \left[ l(Y, \cdot) \right] \|_{\mathcal{H}_l}^2 \right]$$

$$\underset{(b)}{\leq} \mathbb{E}_{\mathbb{P}_X} \left[ \mathbb{E}_{\mathbb{P}_{Y|X}} \left[ \| l(Y, \cdot) \|_{\mathcal{H}_l}^2 \right] \right]$$

$$\underset{(c)}{=} \mathbb{E}_{\mathbb{P}_Y} \left[ \| l(Y, \cdot) \|_{\mathcal{H}_l}^2 \right] = \mathbb{E}_{\mathbb{P}_Y} \left[ l(Y, Y) \right] < \infty$$

where (a) follows by the definition of the KCME; (b) follows by the Generalised Conditional Jensen's Inequality (Theorem A.2 [18]); and (c) by the tower property, the reproducing property, and by our integrability assumption.

Therefore, by further assuming that the Radon-Nikodym derivative $\frac{\mathrm{d}\mathbb{P}_{X^*}}{\mathrm{d}\mathbb{P}_X}$ is bounded, i.e. there exists some constant $M > 0$ such that $\frac{\mathrm{d}\mathbb{P}_{X^*}}{\mathrm{d}\mathbb{P}_X}(\boldsymbol{x}) \leq M$ for all $\boldsymbol{x} \in \mathcal{X}$, we have

$$\int_{\mathcal{X}} \|\mu_{Y|X=\boldsymbol{x}}\|_{\mathcal{H}_l}^2 \mathrm{d}\mathbb{P}_{X^*}(\boldsymbol{x}) = \mathbb{E}_{\mathbb{P}_{X^*}} \left[ \| \mathbb{E}_{\mathbb{P}_{Y|X}} \left[ l(Y, \cdot) \right] \|_{\mathcal{H}_l}^2 \right]$$

$$\underset{(a)}{\leq} M \cdot \mathbb{E}_{\mathbb{P}_X} \left[ \| \mathbb{E}_{\mathbb{P}_{Y|X}} \left[ l(Y, \cdot) \right] \|_{\mathcal{H}_l}^2 \right] < \infty$$

where (a) follows directly from the boundedness condition. Thus, it is now clear that $\mu_{Y|X} \in L^2(\mathcal{X}, \mathbb{P}_{X^*}; \mathcal{H}_l)$. Hence, assuming that $\frac{\mathrm{d}\mathbb{P}_{X^*}}{\mathrm{d}\mathbb{P}_{X'}}$ and $\frac{\mathrm{d}\mathbb{P}_{X^*}}{\mathrm{d}\mathbb{P}_{X''}}$ are also bounded, the functions $f, g : \mathcal{X} \to \mathcal{H}_l$ defined by

$$f(\boldsymbol{x}) := \mu_{Y|X=\boldsymbol{x}} - \mu_{Y'|X'=\boldsymbol{x}}, \quad g(\boldsymbol{x}) := \mu_{Y'|X'=\boldsymbol{x}} - \mu_{Y''|X''=\boldsymbol{x}}$$

belong to $L^2(\mathcal{X}, \mathbb{P}_{X^*}; \mathcal{H}_l)$. The triangle inequality then follows by a straightforward application of the Minkowski inequality, i.e.

$$\|f + g\|_p \leq \|f\|_p + \|g\|_p$$

for the special case where $p = 2$. ∎

**Remark B.5.** Assuming the Radon-Nikodym derivatives $\frac{\mathrm{d}\mathbb{P}_{X^*}}{\mathrm{d}\mathbb{P}_X}$ and $\frac{\mathrm{d}\mathbb{P}_{X^*}}{\mathrm{d}\mathbb{P}_{X'}}$ are bounded, by the arguments above it is clear that

$$\mathrm{AMCMD} \left[ \mathbb{P}_{X^*}, \mathbb{P}_{Y|X}, \mathbb{P}_{Y'|X'} \right] := \sqrt{\mathbb{E}_{\boldsymbol{x} \sim \mathbb{P}_{X^*}} \left[ \|\mu_{Y|X=\boldsymbol{x}} - \mu_{Y'|X'=\boldsymbol{x}}\|_{\mathcal{H}_l}^2 \right]}$$

$$= \|\mu_{Y|X} - \mu_{Y'|X'}\|_{L^2(\mathcal{X}, \mathbb{P}_{X^*}; \mathcal{H}_l)},$$

that is, the AMCMD can be understood as the norm of the difference of $\mu_{Y|X}$ and $\mu_{Y'|X'}$ in $L^2(\mathcal{X}, \mathbb{P}_X^*; \mathcal{H}_l)$ space.

## B.6 Proof of Lemma 4.3

**Lemma B.6.**

$$\widehat{\mathrm{AMCMD}}^2 \left[ \mathbb{P}_{X^*}, \mathbb{P}_{Y|X}, \mathbb{P}_{Y'|X'} \right]$$

$$= \frac{1}{q} Tr \left( K_{\boldsymbol{X}^* \boldsymbol{X}} W_{\boldsymbol{X}\boldsymbol{X}} L_{\boldsymbol{Y}\boldsymbol{Y}} W_{\boldsymbol{X}\boldsymbol{X}} K_{\boldsymbol{X}\boldsymbol{X}^*} \right) - \frac{2}{q} Tr \left( K_{\boldsymbol{X}^* \boldsymbol{X}} W_{\boldsymbol{X}\boldsymbol{X}} L_{\boldsymbol{Y}\boldsymbol{Y}'} W_{\boldsymbol{X}'\boldsymbol{X}'} K_{\boldsymbol{X}'\boldsymbol{X}^*} \right)$$

$$+ \frac{1}{q} Tr \left( K_{\boldsymbol{X}^* \boldsymbol{X}'} W_{\boldsymbol{X}'\boldsymbol{X}'} L_{\boldsymbol{Y}'\boldsymbol{Y}'} W_{\boldsymbol{X}'\boldsymbol{X}'} K_{\boldsymbol{X}'\boldsymbol{X}^*} \right),$$

*where we have defined* $W_{\boldsymbol{X}'\boldsymbol{X}'} := (K_{\boldsymbol{X}'\boldsymbol{X}'} + \lambda_m I)^{-1}$ *with* $[K_{\boldsymbol{X}'\boldsymbol{X}'}]_{ij} := k(\boldsymbol{x}'_i, \boldsymbol{x}'_j)$, $W_{\boldsymbol{X}\boldsymbol{X}} := (K_{\boldsymbol{X}'\boldsymbol{X}'} + \lambda_m I)^{-1}$ *with* $[K_{\boldsymbol{X}\boldsymbol{X}}]_{ij} := k(\boldsymbol{x}_i, \boldsymbol{x}_j)$, $[K_{\boldsymbol{X}'\boldsymbol{X}^*}]_{ij} := k(\boldsymbol{x}', \boldsymbol{x}^*)$, $K_{\boldsymbol{X}^* \boldsymbol{X}} := K_{\boldsymbol{X}'\boldsymbol{X}^*}^\top$, $[L_{\boldsymbol{Y}\boldsymbol{Y}}]_{ij} := l(\boldsymbol{y}_i, \boldsymbol{y}_j)$, $[L_{\boldsymbol{Y}\boldsymbol{Y}'}]_{ij} := l(\boldsymbol{y}_i, \boldsymbol{y}'_j)$, *and* $[L_{\boldsymbol{Y}'\boldsymbol{Y}'}]_{ij} := l(\boldsymbol{y}'_i, \boldsymbol{y}'_j)$.

**Proof**: We have defined

$$\widehat{\text{AMCMD}}^2 \left[ \mathbb{P}_{X^*}, \mathbb{P}_{Y|X}, \mathbb{P}_{Y'|X'} \right] := \frac{1}{q} \sum_{i=1}^{q} \left\| \hat{\mu}_{Y|X=\boldsymbol{x}_i^*} - \hat{\mu}_{Y'|X'=\boldsymbol{x}_i^*} \right\|_{\mathcal{H}_l}^2 .$$

Using equation (1), we have

$$\hat{\mu}_{Y|X=\boldsymbol{x}} := \sum_{i,j=1}^{n} k(\boldsymbol{x}, \boldsymbol{x}_i) W_{ij} l(\boldsymbol{y}_j, \cdot), \quad \hat{\mu}_{Y'|X'=\boldsymbol{x}} := \sum_{s,t=1}^{m} k(\boldsymbol{x}, \boldsymbol{x}_s') W_{st}' l(\boldsymbol{y}_t', \cdot),$$

then we can expand the MCMD$^2$ as

$$\| \hat{\mu}_{Y|X=\boldsymbol{x}} - \hat{\mu}_{Y'|X'\boldsymbol{x}} \|_{\mathcal{H}_l}^2 = \left\langle \hat{\mu}_{Y|X=\boldsymbol{x}}, \hat{\mu}_{Y|X=\boldsymbol{x}} \right\rangle_{\mathcal{H}_l} - 2 \left\langle \hat{\mu}_{Y|X=\boldsymbol{x}}, \hat{\mu}_{Y'|X'=\boldsymbol{x}} \right\rangle_{\mathcal{H}_l} \qquad (23)$$
$$+ \left\langle \hat{\mu}_{Y'|X'=\boldsymbol{x}}, \hat{\mu}_{Y'|X'=\boldsymbol{x}} \right\rangle_{\mathcal{H}_l} .$$

Now,

$$\left\langle \hat{\mu}_{Y|X=\boldsymbol{x}}, \hat{\mu}_{Y'|X'=\boldsymbol{x}} \right\rangle_{\mathcal{H}_l} = \left\langle \sum_{i,j=1}^{n} k(\boldsymbol{x}, \boldsymbol{x}_i) W_{ij} l(\boldsymbol{y}_j, \cdot), \sum_{s,t=1}^{m} k(\boldsymbol{x}, \boldsymbol{x}_s') W_{st}' l(\boldsymbol{y}_t', \cdot) \right\rangle_{\mathcal{H}_l}$$

$$\underset{\text{(a)}}{=} \sum_{i,j=1}^{n} \sum_{s,t=1}^{m} k(\boldsymbol{x}, \boldsymbol{x}_i) W_{ij} \left\langle l(\boldsymbol{y}_j, \cdot), l(\boldsymbol{y}_t', \cdot) \right\rangle_{\mathcal{H}_l} k(\boldsymbol{x}, \boldsymbol{x}_s') W_{st}'$$

$$\underset{\text{(b)}}{=} \sum_{i,j=1}^{n} \sum_{s,t=1}^{m} k(\boldsymbol{x}, \boldsymbol{x}_i) W_{ij} l(\boldsymbol{y}_j, \boldsymbol{y}_t') k(\boldsymbol{x}, \boldsymbol{x}_s') W_{st}'$$

$$\underset{\text{(c)}}{=} \sum_{i,j=1}^{n} \sum_{s,t=1}^{m} k(\boldsymbol{x}, \boldsymbol{x}_i) W_{ij} l(\boldsymbol{y}_j, \boldsymbol{y}_t') W_{ts}' k(\boldsymbol{x}_s', \boldsymbol{x})$$

where (a) follows from the linearity of inner products; (b) follows from the reproducing property on $\mathcal{H}_l$; and (c) follows from symmetry of the kernel $k(\cdot, \cdot)$. Therefore,

$$\frac{1}{q} \sum_{r=1}^{q} \left\langle \hat{\mu}_{Y|X=\boldsymbol{x}_r^*}, \hat{\mu}_{Y'|X'=\boldsymbol{x}_r^*} \right\rangle_{\mathcal{H}_l} = \frac{1}{q} \sum_{r=1}^{q} \sum_{i,j=1}^{n} \sum_{s,t=1}^{m} k(\boldsymbol{x}_r^*, \boldsymbol{x}_i) W_{ij} l(\boldsymbol{y}_j, \boldsymbol{y}_t') W_{ts}' k(\boldsymbol{x}_s', \boldsymbol{x}_r^*)$$

$$= \frac{1}{q} \text{Tr} \left( K_{\boldsymbol{X}^* \boldsymbol{X}} W_{\boldsymbol{X} \boldsymbol{X}} L_{\boldsymbol{Y} \boldsymbol{Y}'} W_{\boldsymbol{X}' \boldsymbol{X}'} K_{\boldsymbol{X}' \boldsymbol{X}^*} \right)$$

where the second line follows from $\text{Tr}(AB) = \sum_{i,j=1}^{n} a_{ij} b_{ji}$. Here we have defined $[K_{\boldsymbol{X}^* \boldsymbol{X}}]_{ij} := k(\boldsymbol{x}_i^*, \boldsymbol{x}_j)$, $W_{\boldsymbol{X} \boldsymbol{X}} := (K_{\boldsymbol{X} \boldsymbol{X}} + \lambda I)^{-1}$, $L_{\boldsymbol{Y} \boldsymbol{Y}'} := [l(\boldsymbol{y}_i, \boldsymbol{y}_j')]_{ij}$, $W_{\boldsymbol{X}' \boldsymbol{X}'} := (K_{\boldsymbol{X}' \boldsymbol{X}'} + \lambda I)^{-1}$, and $[K_{\boldsymbol{X}' \boldsymbol{X}^*}]_{ij} := k(\boldsymbol{x}_i', \boldsymbol{x}_j^*)$.

Noting that the derivation of the first and third term of (23) follow very similarly, we can easily see that

$$\widehat{\text{AMCMD}}^2 \left[ \mathbb{P}_{X^*}, \mathbb{P}_{Y|X}, \mathbb{P}_{Y'|X'} \right] := \frac{1}{q} \sum_{i=1}^{q} \left\| \hat{\mu}_{Y|X=\boldsymbol{x}_i^*} - \hat{\mu}_{Y'|X'=\boldsymbol{x}_i^*} \right\|_{\mathcal{H}_l}^2$$

$$= \frac{1}{q} \text{Tr} \left( K_{\boldsymbol{X}^* \boldsymbol{X}} W_{\boldsymbol{X} \boldsymbol{X}} L_{\boldsymbol{Y} \boldsymbol{Y}} W_{\boldsymbol{X} \boldsymbol{X}} K_{\boldsymbol{X} \boldsymbol{X}^*} \right)$$

$$- \frac{2}{q} \text{Tr} \left( K_{\boldsymbol{X}^* \boldsymbol{X}} W_{\boldsymbol{X} \boldsymbol{X}} L_{\boldsymbol{Y} \boldsymbol{Y}'} W_{\boldsymbol{X}' \boldsymbol{X}'} K_{\boldsymbol{X}' \boldsymbol{X}^*} \right)$$

$$+ \frac{1}{q} \text{Tr} \left( K_{\boldsymbol{X}^* \boldsymbol{X}'} W_{\boldsymbol{X}' \boldsymbol{X}'} L_{\boldsymbol{Y}' \boldsymbol{Y}'} W_{\boldsymbol{X}' \boldsymbol{X}'} K_{\boldsymbol{X}' \boldsymbol{X}^*} \right) .$$

∎

## B.7 Proof of Corollary 4.4

To state Corollary 4.4 in full context, we first introduce some additional definitions and notation. Let $\mathcal{L}(\mathcal{H}_l)$ denote the space of bounded linear operators from $\mathcal{H}_l$ to itself. An operator-valued kernel $\Gamma : \mathcal{X} \times \mathcal{X} \to \mathcal{L}(\mathcal{H}_l)$ induces a *vector-valued reproducing kernel Hilbert space* (vvRKHS), denoted $\mathcal{H}_\Gamma$, which consists of functions $f : \mathcal{X} \to \mathcal{H}_l$ (see [38] for further details). The vvRKHS $\mathcal{H}_\Gamma$ is $C_0$ if $\mathcal{H}_\Gamma \subseteq C_0(\mathcal{X}, \mathcal{H}_l)$, the space of continuous functions from $\mathcal{X}$ to $\mathcal{H}_l$ that vanish at infinity. Moreover, the kernel $\Gamma$ is called $C_0$-universal if it is $C_0$ and $\mathcal{H}_\Gamma$ is dense in $L^2(\mathcal{X}, \mathbb{P}_X; \mathcal{H}_l)$ for any probability measure $\mathbb{P}_X$ on $\mathcal{X}$. Denoting the identity operator on $\mathcal{H}_l$ by $\mathcal{I}_{\mathcal{H}_l}$, it is shown in [38] that the kernel $\Gamma(x, x') = k(x, x')\mathcal{I}_{\mathcal{H}_l}$ is $C_0$-universal whenever $k$ is a universal scalar kernel, such as the Gaussian or Laplacian kernel. Hence, in the statement of Corollary B.7, the assumption that $k(\cdot, \cdot)$ is universal implies that $\Gamma$ is $C_0$-universal, with $\mathcal{H}_\Gamma$ as described above.

Note that the specific choice of kernel $\Gamma(x, x') = k(x, x')\mathcal{I}_{\mathcal{H}_l}$ leads to the form of the KCME estimator given in equation (1) [19], which we adopt in this work. More generally, the corollary could be stated under the assumption that $\mathcal{H}_\Gamma$ is induced by an arbitrary $C_0$-universal operator-valued kernel $\Gamma$, thereby removing the requirement that $k$ be universal—following the more general formulation in Theorem 4.5 of [19]. However, as stated, this would yield a potentially different form of the KCME estimator than the one used in (1).

**Corollary B.7.** *Assume that $k(\cdot, \cdot)$ and $l(\cdot, \cdot)$ are bounded, $k(\cdot, \cdot)$ is universal, and let the regularisation parameters $\lambda_n$ and $\lambda_m$ decay at slower rates than $\mathcal{O}(n^{-1/2})$ and $\mathcal{O}(m^{-1/2})$ respectively. Then, $\widehat{AMCMD}\left[\mathbb{P}_X, \mathbb{P}_{Y|X}, \mathbb{P}_{Y'|X}\right] \xrightarrow{p} AMCMD\left[\mathbb{P}_X, \mathbb{P}_{Y|X}, \mathbb{P}_{Y'|X}\right]$ as $n, m, q \to \infty$.*

**Proof**: Given sets of i.i.d. samples $\{\boldsymbol{x}_i\}_{i=1}^q \sim \mathbb{P}_X$, $\mathcal{M} := \{(\boldsymbol{x}, \boldsymbol{y}_i')\}_{i=1}^m \sim \mathbb{P}_{X,Y'}$, and $\mathcal{N} := \{(\boldsymbol{x}_i, \boldsymbol{y}_i)\}_{i=1}^n \sim \mathbb{P}_{X,Y}$, the estimate of the $\text{AMCMD}[\mathbb{P}_X, \mathbb{P}_{Y|X}, \mathbb{P}_{Y'|X}]$ is defined as

$$\widehat{\text{AMCMD}}\left[\mathbb{P}_X, \mathbb{P}_{Y|X}, \mathbb{P}_{Y'|X}\right] := \sqrt{\frac{1}{q} \sum_{i=1}^q \left\| \hat{\mu}_{Y|X=\boldsymbol{x}_i}^{\mathcal{N}} - \hat{\mu}_{Y'|X=\boldsymbol{x}_i}^{\mathcal{M}} \right\|_{\mathcal{H}_l}^2},$$

with population counterpart

$$\text{AMCMD}\left[\mathbb{P}_X, \mathbb{P}_{Y|X}, \mathbb{P}_{Y'|X}\right] := \sqrt{\mathbb{E}_{\mathbb{P}_X}\left[\left\| \mu_{Y|X=\boldsymbol{x}_i} - \mu_{Y'|X=\boldsymbol{x}_i} \right\|_{\mathcal{H}_l}^2\right]}.$$

The assumptions in Corollary B.7 satisfy the assumptions of Theorem 4.5 [19], which states that

$$\frac{1}{q} \sum_{i=1}^q \left\| \hat{\mu}_{Y|X=\boldsymbol{x}_i}^{\mathcal{N}} - \hat{\mu}_{Y'|X=\boldsymbol{x}_i}^{\mathcal{M}} \right\|_{\mathcal{H}_l}^2 \xrightarrow{p} \mathbb{E}_{\mathbb{P}_X}\left[\left\| \mu_{Y|X=\boldsymbol{x}_i} - \mu_{Y'|X=\boldsymbol{x}_i} \right\|_{\mathcal{H}_l}^2\right].$$

Now, it is enough to note that convergence in probability is conserved under continuous mappings, i.e. $A_n \xrightarrow{p} A$ as $n \to \infty$ implies $\sqrt{A_n} \xrightarrow{p} \sqrt{A}$ as $n \to \infty$ by the continuity of the square root function on $[0, \infty)$ for non-negative random variables $A, A_n$. Hence, our result follows. ∎

## B.8 Proof of Lemma 4.7

**Lemma B.8.** *Let $h : \mathcal{X} \to \mathcal{H}_l$ be a vector-valued function, then*

$$\mathbb{E}_{\boldsymbol{x} \sim \mathbb{P}_X}\left[\left\langle \mu_{Y|X=\boldsymbol{x}}, h(\boldsymbol{x}) \right\rangle_{\mathcal{H}_l}\right] = \mathbb{E}_{(\boldsymbol{x}, \boldsymbol{y}) \sim \mathbb{P}_{X,Y}}\left[h(\boldsymbol{x})(\boldsymbol{y})\right].$$

**Proof**: We apply the definition of the KCME, then the tower rule to see that

$$
\begin{aligned}
\mathbb{E}_{\boldsymbol{x} \sim \mathbb{P}_X}\left[\left\langle \mu_{Y|X=\boldsymbol{x}}, h(\boldsymbol{x}) \right\rangle_{\mathcal{H}_l}\right] &= \mathbb{E}_{\boldsymbol{x} \sim \mathbb{P}_X}\left[\mathbb{E}_{\boldsymbol{y} \sim \mathbb{P}_{Y|X=\boldsymbol{x}}}\left[h(\boldsymbol{x})(\boldsymbol{y})\right]\right] \\
&= \mathbb{E}_{(\boldsymbol{x}, \boldsymbol{y}) \sim \mathbb{P}_{X,Y}}\left[h(\boldsymbol{x})(\boldsymbol{y})\right].
\end{aligned}
$$

∎

## B.9 Derivation of ACKH Objective and Gradients

In ACKH, assuming we are at the $m^{\text{th}}$ iteration, having already constructed a compressed set of size $m-1$, $\mathcal{C}^{m-1} := \{(\tilde{\boldsymbol{x}}_j, \tilde{\boldsymbol{y}}_j)\}_{j=1}^{m-1}$, and let $\mathcal{C}^m = \mathcal{C}^{m-1} \cup (\boldsymbol{x}, \boldsymbol{y})$, then we solve the optimisation problem

$$\underset{\boldsymbol{x}, \boldsymbol{y}}{\arg\min} \ \mathbb{E}_{\boldsymbol{x}' \sim \mathbb{P}_X} \left[ \left\| \tilde{\mu}_{Y|X=\boldsymbol{x}'}^{\mathcal{C}^m} \right\|_{\mathcal{H}_l}^2 \right] - 2 \mathbb{E}_{\boldsymbol{x}' \sim \mathbb{P}_X} \left[ \left\langle \mu_{Y|X=\boldsymbol{x}'}, \tilde{\mu}_{Y|X=\boldsymbol{x}'}^{\mathcal{C}^m} \right\rangle_{\mathcal{H}_l} \right]. \tag{24}$$

via gradient descent. Letting $\mathcal{G}_m^{\mathcal{D}} : \mathbb{R}^d \times \mathbb{R}^p \to \mathbb{R}$ be the estimate of the objective function in (24) computed using the entire dataset $\mathcal{D} = \{(\boldsymbol{x}_i, \boldsymbol{y}_i)\}_{i=1}^n$, i.e.

$$\mathcal{G}_m^{\mathcal{D}}(\boldsymbol{x}, \boldsymbol{y}) := \frac{1}{n} \sum_{i=1}^n \left\| \tilde{\mu}_{Y|X=\boldsymbol{x}_i}^{\mathcal{C}^m} \right\|_{\mathcal{H}_l}^2 - \frac{2}{n} \sum_{i=1}^n \tilde{\mu}_{Y|X=\boldsymbol{x}_i}^{\mathcal{C}^m}(\boldsymbol{y}_i)$$

then we have the following lemma:

**Lemma B.9.** *We have*

$$\mathcal{G}_m^{\mathcal{D}}(\boldsymbol{x}, \boldsymbol{y}) := \frac{1}{n} \sum_{i=1}^n \left\| \tilde{\mu}_{Y|X=\boldsymbol{x}_i}^{\mathcal{C}^m} \right\|_{\mathcal{H}_l}^2 - \frac{2}{n} \sum_{i=1}^n \tilde{\mu}_{Y|X=\boldsymbol{x}_i}^{\mathcal{C}^m}(\boldsymbol{y}_i)$$

$$= \frac{1}{n} Tr \left( \bar{K}_m(\boldsymbol{x}) \tilde{W}_m(\boldsymbol{x}) \tilde{L}_m(\boldsymbol{y}) \tilde{W}_m(\boldsymbol{x}) \bar{K}_m(\boldsymbol{x})^\top \right) - \frac{2}{n} Tr \left( \bar{L}_m(\boldsymbol{y}) \tilde{W}_m(\boldsymbol{x}) \bar{K}_m(\boldsymbol{x})^\top \right),$$

*where we let*

$$\bar{K}_m(\boldsymbol{x}) := \begin{bmatrix} k(\boldsymbol{x}_1, \tilde{\boldsymbol{x}}_1) & \dots & k(\boldsymbol{x}_1, \tilde{\boldsymbol{x}}_{m-1}) & k(\boldsymbol{x}_1, \boldsymbol{x}) \\ \vdots & \ddots & \vdots & \vdots \\ k(\boldsymbol{x}_n, \tilde{\boldsymbol{x}}_1) & \dots & k(\boldsymbol{x}_n, \tilde{\boldsymbol{x}}_{m-1}) & k(\boldsymbol{x}_n, \boldsymbol{x}) \end{bmatrix} \in \mathbb{R}^{n \times m}$$

$$\bar{L}_m(\boldsymbol{y}) := \begin{bmatrix} l(\boldsymbol{y}_1, \tilde{\boldsymbol{y}}_1) & \dots & l(\boldsymbol{y}_1, \tilde{\boldsymbol{y}}_{m-1}) & l(\boldsymbol{y}_1, \boldsymbol{y}) \\ \vdots & \ddots & \vdots & \vdots \\ l(\boldsymbol{y}_n, \tilde{\boldsymbol{y}}_1) & \dots & l(\boldsymbol{y}_n, \tilde{\boldsymbol{y}}_{m-1}) & l(\boldsymbol{y}_n, \boldsymbol{y}) \end{bmatrix} \in \mathbb{R}^{n \times m}$$

$$\tilde{K}_m(\boldsymbol{x}) := \begin{bmatrix} k(\tilde{\boldsymbol{x}}_1, \tilde{\boldsymbol{x}}_1) & \dots & k(\tilde{\boldsymbol{x}}_1, \tilde{\boldsymbol{x}}_{m-1}) & k(\tilde{\boldsymbol{x}}_1, \boldsymbol{x}) \\ \vdots & \ddots & \vdots & \vdots \\ k(\tilde{\boldsymbol{x}}_{m-1}, \tilde{\boldsymbol{x}}_1) & \dots & k(\tilde{\boldsymbol{x}}_{m-1}, \tilde{\boldsymbol{x}}_{m-1}) & k(\tilde{\boldsymbol{x}}_{m-1}, \boldsymbol{x}) \\ k(\boldsymbol{x}, \tilde{\boldsymbol{x}}_1) & \dots & k(\boldsymbol{x}, \tilde{\boldsymbol{x}}_{m-1}) & k(\boldsymbol{x}, \boldsymbol{x}) \end{bmatrix} \in \mathbb{R}^{m \times m}$$

$$\tilde{L}_m(\boldsymbol{y}) := \begin{bmatrix} l(\tilde{\boldsymbol{y}}_1, \tilde{\boldsymbol{y}}_1) & \dots & l(\tilde{\boldsymbol{y}}_1, \tilde{\boldsymbol{y}}_{m-1}) & l(\tilde{\boldsymbol{y}}_1, \boldsymbol{y}) \\ \vdots & \ddots & \vdots & \vdots \\ l(\tilde{\boldsymbol{y}}_{m-1}, \tilde{\boldsymbol{y}}_1) & \dots & l(\tilde{\boldsymbol{y}}_{m-1}, \tilde{\boldsymbol{y}}_{m-1}) & l(\tilde{\boldsymbol{y}}_{m-1}, \boldsymbol{y}) \\ l(\boldsymbol{y}, \tilde{\boldsymbol{y}}_1) & \dots & l(\boldsymbol{y}, \tilde{\boldsymbol{y}}_{m-1}) & l(\boldsymbol{y}, \boldsymbol{y}) \end{bmatrix} \in \mathbb{R}^{m \times m},$$

*and $\tilde{W}_m(\boldsymbol{x}) := (\tilde{K}_m(\boldsymbol{x}) + \lambda I_m)^{-1} \in \mathbb{R}^{m \times m}$. Moreover, $\mathcal{G}_m^{\mathcal{D}}(\boldsymbol{x}, \boldsymbol{y})$ can be computed with time complexity of $\mathcal{O}(m^2 n + m^3)$ and storage complexity of $\mathcal{O}(mn + m^2)$.*

**Proof**: In order to reduce the notational burden, we write $[\tilde{W}_m(\boldsymbol{x})]_{ij} = \tilde{W}_{ij}$, and $(\boldsymbol{x}, \boldsymbol{y}) = (\tilde{\boldsymbol{x}}_m, \tilde{\boldsymbol{y}}_m)$, then, using the estimate of the KCME from (1), we have

$$
\mathcal{G}_m^{\mathcal{D}}(\tilde{\boldsymbol{x}}_m, \tilde{\boldsymbol{y}}_m) := \frac{1}{n} \sum_{i=1}^n \left\| \tilde{\mu}_{Y|X=\boldsymbol{x}_i}^{\mathcal{C}^m} \right\|_{\mathcal{H}_l}^2 - \frac{2}{n} \sum_{i=1}^n \tilde{\mu}_{Y|X=\boldsymbol{x}_i}^{\mathcal{C}^m}(\boldsymbol{y}_i)
$$

$$
\underset{\text{(a)}}{=} \frac{1}{n} \sum_{r=1}^n \left\langle \sum_{i,j=1}^m l(\cdot, \tilde{\boldsymbol{y}}_i) \tilde{W}_{ij} k(\tilde{\boldsymbol{x}}_j, \boldsymbol{x}_r), \sum_{p,q=1}^m l(\cdot, \tilde{\boldsymbol{y}}_p) \tilde{W}_{pq} k(\tilde{\boldsymbol{x}}_r, \boldsymbol{x}_q) \right\rangle_{\mathcal{H}_l}
$$

$$
- \frac{2}{n} \sum_{p=1}^n \sum_{i,j=1}^m l(\boldsymbol{y}_p, \tilde{\boldsymbol{y}}_i) \tilde{W}_{ij} k(\tilde{\boldsymbol{x}}_j, \boldsymbol{x}_p)
$$

$$
\underset{\text{(b)}}{=} \frac{1}{n} \sum_{r=1}^n \sum_{i,j,p,q=1}^m k(\boldsymbol{x}_r, \tilde{\boldsymbol{x}}_j) \tilde{W}_{ji} l(\tilde{\boldsymbol{y}}_i, \tilde{\boldsymbol{y}}_p) \tilde{W}_{pq} k(\tilde{\boldsymbol{x}}, \boldsymbol{x}_r)
$$

$$
- \frac{2}{n} \sum_{p=1}^n \sum_{i,j=1}^m l(\boldsymbol{y}_p, \tilde{\boldsymbol{y}}_i) \tilde{W}_{ij} k(\tilde{\boldsymbol{x}}_j, \boldsymbol{x}_p)
$$

$$
\underset{\text{(c)}}{=} \frac{1}{n} \text{Tr}\left( \bar{K}_m(\boldsymbol{x}) \tilde{W}_m(\boldsymbol{x}) \tilde{L}_m(\boldsymbol{y}) \tilde{W}_m(\boldsymbol{x}) \bar{K}_m(\boldsymbol{x})^\top \right) - \frac{2}{n} \text{Tr}\left( \bar{L}_m(\boldsymbol{y}) \tilde{W}_m(\boldsymbol{x}) \bar{K}_m(\boldsymbol{x})^\top \right),
$$

where (a) follows from inserting the estimate of the KCME and the definition of the norm; (b) follows from the reproducing property and the symmetry of the kernels; and (c) follows from the fact that $\text{Tr}(AB) = \sum_{i,j=1}^n a_{ij} b_{ji}$ for symmetric $A, B \in \mathbb{R}^{n \times n}$.

The storage complexity of $\mathcal{O}(mn + m^2)$ comes from the fact we have to store

$$
\tilde{L}_m(\boldsymbol{y}), \tilde{K}_m(\boldsymbol{x}), \tilde{W}_m(\boldsymbol{x}) \in \mathbb{R}^{m \times m}, \quad \text{and} \quad \bar{K}_m(\boldsymbol{x}), \bar{L}_m(\boldsymbol{y}) \in \mathbb{R}^{m \times n}.
$$

The computational complexity of $\mathcal{O}(m^2 n + m^3)$ arises from solving the linear system,

$$
A(\tilde{K}_m(\boldsymbol{x}) + \lambda I) = \bar{K}_m(\boldsymbol{x})^\top \quad \text{for} \quad A \in \mathbb{R}^{n \times m}
$$

which dominates the $\mathcal{O}(m^2 n)$ cost of the singular remaining matrix multiplication, and the $\mathcal{O}(m^2 + mn)$ cost of taking Hadamard products required to compute the traces. $\blacksquare$

We now derive the gradients of $\mathcal{G}_m^{\mathcal{D}}$, and show that they are cheap to compute and store. Note that the derivative identities used in this section are similar to those in Section B.4.1, except for third-order tensors, as we deal with derivatives of matrix-valued functions with respect to vectors.

**Lemma B.10.** *We compute the gradients of the objective function* $\mathcal{G}_m^{\mathcal{D}} : \mathbb{R}^d \times \mathbb{R}^p \to \mathbb{R}$ *as*

$$
\nabla_{\boldsymbol{y}} \mathcal{G}_m^{\mathcal{D}}(\boldsymbol{x}, \boldsymbol{y}) = \frac{1}{n} Tr\left( \bar{K}_m(\boldsymbol{x}) \tilde{W}_m(\boldsymbol{x}) \nabla_{\boldsymbol{y}} \tilde{L}_m(\boldsymbol{y}) \tilde{W}_m(\boldsymbol{x}) \bar{K}_m(\boldsymbol{x})^\top \right) \tag{25}
$$

$$
- \frac{2}{n} Tr\left( \nabla_{\boldsymbol{y}} \bar{L}_m(\boldsymbol{y}) \tilde{W}_m(\boldsymbol{x}) \bar{K}_m(\boldsymbol{x})^\top \right),
$$

$$
\nabla_{\boldsymbol{x}} \mathcal{G}_m^{\mathcal{D}}(\boldsymbol{x}, \boldsymbol{y}) = \frac{2}{n} Tr\left( \nabla_{\boldsymbol{x}} \bar{K}_m(\boldsymbol{x}) \tilde{W}_m(\boldsymbol{x}) \tilde{L}_m(\boldsymbol{y}) \tilde{W}_m(\boldsymbol{x}) \bar{K}_m(\boldsymbol{x})^\top \right) \tag{26}
$$

$$
- \frac{2}{n} Tr\left( \bar{K}_m(\boldsymbol{x}) \tilde{W}_m(\boldsymbol{x}) \nabla_{\boldsymbol{x}} \tilde{K}_m(\boldsymbol{x}) \tilde{W}_m(\boldsymbol{x}) \tilde{L}_m(\boldsymbol{y}) \tilde{W}_m(\boldsymbol{x}) \bar{K}_m(\boldsymbol{x})^\top \right)
$$

$$
+ \frac{2}{n} Tr\left( \bar{L}_m(\boldsymbol{y}) \tilde{W}_m(\boldsymbol{x}) \nabla_{\boldsymbol{x}} \tilde{K}_m(\boldsymbol{x}) \tilde{W}_m(\boldsymbol{x}) \bar{K}_m(\boldsymbol{x})^\top \right)
$$

$$
- \frac{2}{n} Tr\left( \bar{L}_m(\boldsymbol{y}) \tilde{W}_m(\boldsymbol{x}) \nabla_{\boldsymbol{x}} \bar{K}_m(\boldsymbol{x})^\top \right).
$$

*Where, in order to reduce notational burden, we write* $(\boldsymbol{x}, \boldsymbol{y}) = (\tilde{\boldsymbol{x}}_m, \tilde{\boldsymbol{y}}_m)$, *then we have*

$$
\nabla_{\tilde{\boldsymbol{x}}_m} \tilde{K}_m(\tilde{\boldsymbol{x}}_m) \in \mathbb{R}^{m \times m \times d}, \quad \text{with} \quad \left[ \nabla_{\tilde{\boldsymbol{x}}_m} \tilde{K}_m(\tilde{\boldsymbol{x}}_m) \right]_{ij} := \nabla_{\tilde{\boldsymbol{x}}_m} k(\tilde{\boldsymbol{x}}_i, \tilde{\boldsymbol{x}}_j) in \mathbb{R}^d
$$

$$
\nabla_{\tilde{\boldsymbol{x}}_m} \bar{K}_m(\tilde{\boldsymbol{x}}_m) \in \mathbb{R}^{n \times m \times d}, \quad \text{with} \quad \left[ \nabla_{\tilde{\boldsymbol{x}}_m} \bar{K}_m(\tilde{\boldsymbol{x}}_m) \right]_{ij} := \nabla_{\tilde{\boldsymbol{x}}_m} k(\boldsymbol{x}_i, \tilde{\boldsymbol{x}}_j) \in \mathbb{R}^d
$$

$$
\nabla_{\tilde{\boldsymbol{y}}_m} \tilde{K}_m(\tilde{\boldsymbol{y}}_m) \in \mathbb{R}^{m \times m \times d}, \quad \text{with} \quad \left[ \nabla_{\tilde{\boldsymbol{y}}_m} \tilde{L}_m(\tilde{\boldsymbol{y}}_m) \right]_{ij} := \nabla_{\tilde{\boldsymbol{y}}_m} l(\tilde{\boldsymbol{y}}_i, \tilde{\boldsymbol{y}}_j) \in \mathbb{R}^p
$$

$$
\nabla_{\tilde{\boldsymbol{y}}_m} \bar{L}_m(\tilde{\boldsymbol{y}}_m) \in \mathbb{R}^{n \times m \times d}, \quad \text{with} \quad \left[ \nabla_{\tilde{\boldsymbol{y}}_m} \bar{L}_m(\tilde{\boldsymbol{y}}_m) \right]_{ij} := \nabla_{\tilde{\boldsymbol{y}}_m} l(\boldsymbol{y}_i, \tilde{\boldsymbol{y}}_j) \in \mathbb{R}^p
$$

with $\mathcal{O}((m^2 n + m^3)$ time and $\mathcal{O}(mn + m^2)$ storage complexity, i.e. linear with respect to the size $n$ of the full dataset $\mathcal{D}$.

**Proof**: Firstly, by applying the trace and product rule, we can immediately see that

$$\nabla_{\boldsymbol{y}} \mathcal{G}_m^{\mathcal{D}}(\boldsymbol{x}, \boldsymbol{y}) = \frac{1}{n} \mathrm{Tr} \left( \bar{K}_m(\boldsymbol{x}) \tilde{W}_m(\boldsymbol{x}) \nabla_{\boldsymbol{y}} \tilde{L}_m(\boldsymbol{y}) \tilde{W}_m(\boldsymbol{x}) \bar{K}_m(\boldsymbol{x})^\top \right)$$
$$- \frac{2}{n} \mathrm{Tr} \left( \nabla_{\boldsymbol{y}} \bar{L}_m(\boldsymbol{y}) \tilde{W}_m(\boldsymbol{x}) \bar{K}_m(\boldsymbol{x}) \right)$$

Now, we have

$$\nabla_{\boldsymbol{x}} \mathcal{G}_m^{\mathcal{D}}(\boldsymbol{x}, \boldsymbol{y}) \underset{(a)}{=} \frac{1}{n} \nabla_{\boldsymbol{x}} \mathrm{Tr} \left( \bar{K}(\boldsymbol{x}) \tilde{W}(\boldsymbol{x}) \tilde{L}(\boldsymbol{y}) \tilde{W}(\boldsymbol{x}) \bar{K}(\boldsymbol{x})^\top \right)$$
$$- \frac{2}{n} \nabla_{\boldsymbol{x}} \mathrm{Tr} \left( \bar{L}(\boldsymbol{y}) \tilde{W}(\boldsymbol{x}) \bar{K}(\boldsymbol{x})^\top \right)$$
$$\underset{(b)}{=} \frac{1}{n} \mathrm{Tr} \left( \nabla_{\boldsymbol{x}} \bar{K}(\boldsymbol{x}) \tilde{W}(\boldsymbol{x}) \tilde{L}(\boldsymbol{y}) \tilde{W}(\boldsymbol{x}) \bar{K}(\boldsymbol{x})^\top \right)$$
$$+ \frac{1}{n} \mathrm{Tr} \left( \bar{K}(\boldsymbol{x}) \nabla_{\boldsymbol{x}} \tilde{W}(\boldsymbol{x}) \tilde{L}(\boldsymbol{y}) \tilde{W}(\boldsymbol{x}) \bar{K}(\boldsymbol{x})^\top \right)$$
$$+ \frac{1}{n} \mathrm{Tr} \left( \bar{K}(\boldsymbol{x}) \tilde{W}(\boldsymbol{x}) \tilde{L}(\boldsymbol{y}) \nabla_{\boldsymbol{x}} \tilde{W}(\boldsymbol{x}) \bar{K}(\boldsymbol{x})^\top \right)$$
$$+ \frac{1}{n} \mathrm{Tr} \left( \bar{K}(\boldsymbol{x}) \tilde{W}(\boldsymbol{x}) \tilde{L}(\boldsymbol{y}) \tilde{W}(\boldsymbol{x}) \nabla_{\boldsymbol{x}} \bar{K}(\boldsymbol{x})^\top \right)$$
$$- \frac{2}{n} \mathrm{Tr} \left( \bar{L}(\boldsymbol{y}) \nabla_{\boldsymbol{x}} \tilde{W}(\boldsymbol{x}) \bar{K}(\boldsymbol{x})^\top \right)$$
$$- \frac{2}{n} \mathrm{Tr} \left( \bar{L}(\boldsymbol{y}) \tilde{W}(\boldsymbol{x}) \nabla_{\boldsymbol{x}} \bar{K}(\boldsymbol{x})^\top \right)$$
$$\underset{(c)}{=} \frac{2}{n} \mathrm{Tr} \left( \nabla_{\boldsymbol{x}} \bar{K}(\boldsymbol{x}) \tilde{W}(\boldsymbol{x}) \tilde{L}(\boldsymbol{y}) \tilde{W}(\boldsymbol{x}) \bar{K}(\boldsymbol{x})^\top \right)$$
$$+ \frac{2}{n} \mathrm{Tr} \left( \bar{K}(\boldsymbol{x}) \nabla_{\boldsymbol{x}} \tilde{W}(\boldsymbol{x}) \tilde{L}(\boldsymbol{y}) \tilde{W}(\boldsymbol{x}) \bar{K}(\boldsymbol{x})^\top \right)$$
$$- \frac{2}{n} \mathrm{Tr} \left( \bar{L}(\boldsymbol{y}) \nabla_{\boldsymbol{x}} \tilde{W}(\boldsymbol{x}) \bar{K}(\boldsymbol{x})^\top \right)$$
$$- \frac{2}{n} \mathrm{Tr} \left( \bar{L}(\boldsymbol{y}) \tilde{W}(\boldsymbol{x}) \nabla_{\boldsymbol{x}} \bar{K}(\boldsymbol{x})^\top \right),$$

where (a) follows from the linearity of the gradient operator; (b) follows from a combination of the trace and product rules; and (c) follows from the symmetry of the feature kernel $k(\cdot, \cdot)$. Then, by applying the inverse rule, we have

$$\nabla_{\boldsymbol{x}} \mathcal{G}_m^{\mathcal{D}}(\boldsymbol{x}, \boldsymbol{y}) = \frac{2}{n} \mathrm{Tr} \left( \nabla_{\boldsymbol{x}} \bar{K}(\boldsymbol{x}) \tilde{W}(\boldsymbol{x}) \tilde{L}(\boldsymbol{y}) \tilde{W}(\boldsymbol{x}) \bar{K}(\boldsymbol{x})^\top \right)$$
$$- \frac{2}{n} \mathrm{Tr} \left( \bar{K}(\boldsymbol{x}) \tilde{W}(\boldsymbol{x}) \nabla_{\boldsymbol{x}} \tilde{K}(\boldsymbol{x}) \tilde{W}(\boldsymbol{x}) \tilde{L}(\boldsymbol{y}) \tilde{W}(\boldsymbol{x}) \bar{K}(\boldsymbol{x})^\top \right)$$
$$+ \frac{2}{n} \mathrm{Tr} \left( \bar{L}(\boldsymbol{y}) \tilde{W}(\boldsymbol{x}) \nabla_{\boldsymbol{x}} \tilde{K}(\boldsymbol{x}) \tilde{W}(\boldsymbol{x}) \bar{K}(\boldsymbol{x})^\top \right)$$
$$- \frac{2}{n} \mathrm{Tr} \left( \bar{L}(\boldsymbol{y}) \tilde{W}(\boldsymbol{x}) \nabla_{\boldsymbol{x}} \bar{K}(\boldsymbol{x})^\top \right).$$

Now, to establish the cost of computing this estimate, the first thing to notice is that there is a significant amount of symmetry and shared computation between the terms. In particular, avoiding the gradients

$$\nabla_{\boldsymbol{x}} \tilde{K}(\boldsymbol{x}) \in \mathbb{R}^{m \times m \times d} \quad \text{and} \quad \nabla_{\boldsymbol{x}} \bar{K}(\boldsymbol{x}) \in \mathbb{R}^{n \times m \times d}$$

for now, we need to compute

$$A := \tilde{W}(\boldsymbol{x}) \bar{K}(\boldsymbol{x})^\top \in \mathbb{R}^{m \times n}, \quad B := \bar{L}(\boldsymbol{y}) \tilde{W}(\boldsymbol{x}) \in \mathbb{R}^{n \times m},$$
$$C := \tilde{W}(\boldsymbol{x}) \tilde{L}(\boldsymbol{y}) \tilde{W}(\boldsymbol{x}) \bar{K}(\boldsymbol{x})^\top \in \mathbb{R}^{m \times n},$$

then we have

$$\nabla_{\boldsymbol{x}}\mathcal{G}_m^{\mathcal{D}}(\boldsymbol{x},\boldsymbol{y}) = \frac{2}{n}\text{Tr}\left(\nabla_{\boldsymbol{x}}\bar{K}(\boldsymbol{x})C\right) - \frac{2}{n}\text{Tr}\left(A^{\top}\nabla_{\boldsymbol{x}}\tilde{K}(\boldsymbol{x})C\right)$$
$$+ \frac{2}{n}\text{Tr}\left(B\nabla_{\boldsymbol{x}}\tilde{K}(\boldsymbol{x})A\right) - \frac{2}{n}\text{Tr}\left(B\nabla_{\boldsymbol{x}}\bar{K}(\boldsymbol{x})^{\top}\right)$$
$$\underset{(a)}{=} \frac{2}{n}\text{Tr}\left(\nabla_{\boldsymbol{x}}\bar{K}(\boldsymbol{x})C\right) - \frac{2}{n}\text{Tr}\left(\nabla_{\boldsymbol{x}}\tilde{K}(\boldsymbol{x})CA^{\top}\right)$$
$$+ \frac{2}{n}\text{Tr}\left(\nabla_{\boldsymbol{x}}\tilde{K}(\boldsymbol{x})AB\right) - \frac{2}{n}\text{Tr}\left(\nabla_{\boldsymbol{x}}\bar{K}(\boldsymbol{x})^{\top}B\right)$$

where (a) follows from the cyclic property of the trace and the symmetry of the kernels. Now, the cost of computing $A$, $B$ and $C$ is $\mathcal{O}(nm^2 + m^3)$, and given these matrices, the cost of computing $D := CA^{\top} \in \mathbb{R}^{m \times m}$ and $E := AB \in \mathbb{R}^{m \times m}$ is $\mathcal{O}(nm^2)$. So, we are now in a position where we have to compute

$$\nabla_{\tilde{\boldsymbol{X}}}\mathcal{J}^{\mathcal{D}}(\tilde{\boldsymbol{X}},\tilde{\boldsymbol{Y}}) = \frac{2}{n}\text{Tr}\left(\nabla_{\boldsymbol{x}}\bar{K}(\boldsymbol{x})C\right) - \frac{2}{n}\text{Tr}\left(\nabla_{\boldsymbol{x}}\tilde{K}(\boldsymbol{x})D\right)$$
$$+ \frac{2}{n}\text{Tr}\left(\nabla_{\boldsymbol{x}}\tilde{K}(\boldsymbol{x})E\right) - \frac{2}{n}\text{Tr}\left(\nabla_{\boldsymbol{x}}\bar{K}(\boldsymbol{x})^{\top}B\right).$$

We can reduce the cost of computing these terms by noticing that the majority of the elements of our third-order gradient tensors will be equal to zero, that is

$$\nabla_{\boldsymbol{x}}\bar{K}_m(\boldsymbol{x}) := \begin{bmatrix} 0 & \cdots & 0 & \nabla_{\boldsymbol{x}}k(\boldsymbol{x}_1,\boldsymbol{x}) \\ \vdots & \ddots & \vdots & \vdots \\ 0 & \cdots & 0 & \nabla_{\boldsymbol{x}}k(\boldsymbol{x}_n,\boldsymbol{x}) \end{bmatrix} \in \mathbb{R}^{n \times m \times d}$$

$$\nabla_{\boldsymbol{y}}\bar{L}_m(\boldsymbol{y}) := \begin{bmatrix} 0 & \cdots & 0 & \nabla_{\boldsymbol{y}}l(\boldsymbol{y}_1,\boldsymbol{y}) \\ \vdots & \ddots & \vdots & \vdots \\ 0 & \cdots & 0 & \nabla_{\boldsymbol{y}}l(\boldsymbol{y}_n,\boldsymbol{y}) \end{bmatrix} \in \mathbb{R}^{n \times m \times p}$$

$$\nabla_{\boldsymbol{x}}\tilde{K}_m(\boldsymbol{x}) := \begin{bmatrix} 0 & \cdots & 0 & k(\tilde{\boldsymbol{x}}_1,\boldsymbol{x}) \\ \vdots & \ddots & \vdots & \vdots \\ 0 & \cdots & 0 & k(\tilde{\boldsymbol{x}}_{m-1},\boldsymbol{x}) \\ k(\boldsymbol{x},\tilde{\boldsymbol{x}}_1) & \cdots & k(\boldsymbol{x},\tilde{\boldsymbol{x}}_{m-1}) & k(\boldsymbol{x},\boldsymbol{x}) \end{bmatrix} \in \mathbb{R}^{m \times m \times d}$$

$$\nabla_{\boldsymbol{y}}\tilde{L}_m(\boldsymbol{y}) := \begin{bmatrix} 0 & \cdots & 0 & l(\tilde{\boldsymbol{y}}_1,\boldsymbol{y}) \\ \vdots & \ddots & \vdots & \vdots \\ 0 & \cdots & 0 & l(\tilde{\boldsymbol{y}}_{m-1},\boldsymbol{y}) \\ l(\boldsymbol{y},\tilde{\boldsymbol{y}}_1) & \cdots & l(\boldsymbol{y},\tilde{\boldsymbol{y}}_{m-1}) & l(\boldsymbol{y},\boldsymbol{y}) \end{bmatrix} \in \mathbb{R}^{m \times m \times p}.$$

Hence, we have

$$\left[\text{Tr}\left(\nabla_{\boldsymbol{x}}\bar{K}(\boldsymbol{x})C\right)\right]_r = \sum_{i=1}^{n}\sum_{j=1}^{m}\left[\nabla_{\boldsymbol{x}}\bar{K}(\boldsymbol{x})\right]_{ijr}C_{ji}$$
$$= \sum_{i=1}^{n}\left[\nabla_{\boldsymbol{x}}\bar{K}(\boldsymbol{x})\right]_{imr}C_{mi}$$

for $r = 1,\ldots,d$. Hence this term, given $C$, can be computed with cost $\mathcal{O}(n)$. We also have

$$\text{Tr}\left(\nabla_{\boldsymbol{x}}\tilde{K}(\boldsymbol{x})D\right) = \sum_{i,j=1}^{m}\left[\nabla_{\boldsymbol{x}}\tilde{K}(\boldsymbol{x})\right]_{ijr}D_{ji}$$
$$= \sum_{j=1}^{m}\left[\nabla_{\boldsymbol{x}}\tilde{K}(\boldsymbol{x})\right]_{mjr}D_{jm} + \sum_{i=1}^{m}\left[\nabla_{\boldsymbol{x}}\tilde{K}(\boldsymbol{x})\right]_{imr}D_{mi}$$
$$= 2\sum_{j=1}^{m}\left[\nabla_{\boldsymbol{x}}\tilde{K}(\boldsymbol{x})\right]_{mjr}D_{jm}$$

where the last equality follows by the symmetry of the kernels. Hence this term, given $D$, can be computed with cost $\mathcal{O}(m)$. Similar derivations hold for $\text{Tr}\left(\nabla_{\boldsymbol{x}}\bar{K}(\boldsymbol{x})^\top B\right)$ and $\text{Tr}\left(\nabla_{\boldsymbol{x}}\tilde{K}(\boldsymbol{x})E\right)$.

Therefore, we have an overall storage cost of $\mathcal{O}(mn + m^2)$, and time cost of $\mathcal{O}(m^3 + m^2 n)$, i.e. *linear* with respect to the size $n$ of the full dataset $\mathcal{D}$. ∎

**Remark B.11.** Above we have derived analytical gradients of the objective function, and shown they can be computed in linear time. In practice one computes the gradients using JAX's [79] auto-differentiation capabilities. The authors observed minimal slowdown from using auto-differentiation.

## B.10 Derivation of ACKIP Objective and Gradients

Noting that $\mathcal{C} = (\tilde{\boldsymbol{X}}, \tilde{\boldsymbol{Y}})$, in ACKIP we solve the optimisation problem

$$\underset{\tilde{\boldsymbol{X}}, \tilde{\boldsymbol{Y}}}{\arg\min} \ \mathbb{E}_{\boldsymbol{x}' \sim \mathbb{P}_X}\left[\left\|\tilde{\mu}_{Y|X=\boldsymbol{x}'}^{(\tilde{\boldsymbol{X}}, \tilde{\boldsymbol{Y}})}\right\|_{\mathcal{H}_l}^2\right] - 2\mathbb{E}_{(\boldsymbol{x}', \boldsymbol{y}') \sim \mathbb{P}_{X,Y}}\left[\tilde{\mu}_{Y|X=\boldsymbol{x}'}^{(\tilde{\boldsymbol{X}}, \tilde{\boldsymbol{Y}})}(\boldsymbol{y}')\right]. \tag{27}$$

via gradient descent. Letting $\mathcal{J}^{\mathcal{D}} : \mathbb{R}^{m \times d} \times \mathbb{R}^{m \times p} \to \mathbb{R}$ be the estimate of the objective function in (27) computed using the entire dataset $\mathcal{D} = \{(\boldsymbol{x}_i, \boldsymbol{y}_i)\}_{i=1}^n$, i.e.

$$\mathcal{J}^{\mathcal{D}}(\tilde{\boldsymbol{X}}, \tilde{\boldsymbol{Y}}) := \frac{1}{n}\sum_{i=1}^n \left\|\tilde{\mu}_{Y|X=\boldsymbol{x}_i}^{(\tilde{\boldsymbol{X}}, \tilde{\boldsymbol{Y}})}\right\|_{\mathcal{H}_l}^2 - \frac{2}{n}\sum_{i=1}^n \tilde{\mu}_{Y|X=\boldsymbol{x}_i}^{(\tilde{\boldsymbol{X}}, \tilde{\boldsymbol{Y}})}(\boldsymbol{y}_i)$$

we have the following lemma:

**Lemma B.12.** *We have*

$$\mathcal{J}^{\mathcal{D}}(\tilde{\boldsymbol{X}}, \tilde{\boldsymbol{Y}}) := \frac{1}{n}\sum_{i=1}^n \left\|\tilde{\mu}_{Y|X=\boldsymbol{x}_i}^{(\tilde{\boldsymbol{X}}, \tilde{\boldsymbol{Y}})}\right\|_{\mathcal{H}_l}^2 - \frac{2}{n}\sum_{i=1}^n \tilde{\mu}_{Y|X=\boldsymbol{x}_i}^{(\tilde{\boldsymbol{X}}, \tilde{\boldsymbol{Y}})}(\boldsymbol{y}_i)$$

$$= \frac{1}{n}Tr\left(K_{\boldsymbol{X}\tilde{\boldsymbol{X}}}W_{\tilde{\boldsymbol{X}}\tilde{\boldsymbol{X}}}L_{\tilde{\boldsymbol{Y}}\tilde{\boldsymbol{Y}}}W_{\tilde{\boldsymbol{X}}\tilde{\boldsymbol{X}}}K_{\tilde{\boldsymbol{X}}\boldsymbol{X}}\right) - \frac{2}{n}Tr\left(L_{\boldsymbol{Y}\tilde{\boldsymbol{Y}}}W_{\tilde{\boldsymbol{X}}\tilde{\boldsymbol{X}}}K_{\tilde{\boldsymbol{X}}\boldsymbol{X}}\right),$$

*where* $[K_{\tilde{\boldsymbol{X}}\tilde{\boldsymbol{X}}}]_{ij} := k(\tilde{\boldsymbol{x}}_i, \tilde{\boldsymbol{x}}_j)$, $[K_{\tilde{\boldsymbol{X}}\boldsymbol{X}}]_{iq} := k(\tilde{\boldsymbol{x}}_i, \boldsymbol{x}_q)$, $[L_{\tilde{\boldsymbol{Y}}\tilde{\boldsymbol{Y}}}]_{ij} := l(\tilde{\boldsymbol{y}}_i, \tilde{\boldsymbol{y}}_j)$, $[L_{\tilde{\boldsymbol{Y}}\boldsymbol{Y}}]_{iq} := l(\tilde{\boldsymbol{y}}_i, \boldsymbol{y}_q)$, *and* $[W_{\tilde{\boldsymbol{X}}\tilde{\boldsymbol{X}}}]_{ij} := (K_{\tilde{\boldsymbol{X}}\tilde{\boldsymbol{X}}} + \lambda I)^{-1}$, *for* $i, j = 1, \dots, m$ *and* $q = 1, \dots, n$.

*Moreover,* $\mathcal{J}^{\mathcal{D}}(\tilde{\boldsymbol{X}}, \tilde{\boldsymbol{Y}})$ *can be computed with time complexity of* $\mathcal{O}(m^2 n + mn + m^3)$ *and storage complexity of* $\mathcal{O}(mn + m^2)$.

**Proof**: Using the estimate of the KCME from (1), and writing $[W_{\tilde{\boldsymbol{X}}\tilde{\boldsymbol{X}}}]_{ij} = \tilde{W}_{ij}$, we have

$$\mathcal{J}^{\mathcal{D}}(\tilde{\boldsymbol{X}}, \tilde{\boldsymbol{Y}}) := \frac{1}{n}\sum_{i=1}^n \left\|\tilde{\mu}_{Y|X=\boldsymbol{x}_i}^{(\tilde{\boldsymbol{X}}, \tilde{\boldsymbol{Y}})}\right\|_{\mathcal{H}_l}^2 - \frac{2}{n}\sum_{i=1}^n \tilde{\mu}_{Y|X=\boldsymbol{x}_i}^{(\tilde{\boldsymbol{X}}, \tilde{\boldsymbol{Y}})}(\boldsymbol{y}_i)$$

$$\underset{\text{(a)}}{=} \frac{1}{n}\sum_{r=1}^n \left\langle \sum_{i,j=1}^m l(\cdot, \tilde{\boldsymbol{y}}_i)\tilde{W}_{ij}k(\tilde{\boldsymbol{x}}_j, \boldsymbol{x}_r), \sum_{p,q=1}^m l(\cdot, \tilde{\boldsymbol{y}}_p)\tilde{W}_{pq}k(\tilde{\boldsymbol{x}}_r, \boldsymbol{x}_q) \right\rangle_{\mathcal{H}_l}$$

$$- \frac{2}{n}\sum_{p=1}^n \sum_{i,j=1}^m l(\boldsymbol{y}_p, \tilde{\boldsymbol{y}}_i)\tilde{W}_{ij}k(\tilde{\boldsymbol{x}}_j, \boldsymbol{x}_p)$$

$$\underset{\text{(b)}}{=} \frac{1}{n}\sum_{r=1}^n \sum_{i,j,p,q=1}^m k(\boldsymbol{x}_r, \tilde{\boldsymbol{x}}_j)\tilde{W}_{ji}l(\tilde{\boldsymbol{y}}_i, \tilde{\boldsymbol{y}}_p)\tilde{W}_{pq}k(\tilde{\boldsymbol{x}}_q, \boldsymbol{x}_r)$$

$$- \frac{2}{n}\sum_{p=1}^n \sum_{i,j=1}^m l(\boldsymbol{y}_p, \tilde{\boldsymbol{y}}_i)\tilde{W}_{ij}k(\tilde{\boldsymbol{x}}_j, \boldsymbol{x}_p)$$

$$\underset{\text{(c)}}{=} \frac{1}{n}\text{Tr}\left(K_{\boldsymbol{X}\tilde{\boldsymbol{X}}}W_{\tilde{\boldsymbol{X}}\tilde{\boldsymbol{X}}}L_{\tilde{\boldsymbol{Y}}\tilde{\boldsymbol{Y}}}W_{\tilde{\boldsymbol{X}}\tilde{\boldsymbol{X}}}K_{\tilde{\boldsymbol{X}}\boldsymbol{X}}\right) - \frac{2}{n}\text{Tr}\left(L_{\boldsymbol{Y}\tilde{\boldsymbol{Y}}}W_{\tilde{\boldsymbol{X}}\tilde{\boldsymbol{X}}}K_{\tilde{\boldsymbol{X}}\boldsymbol{X}}\right)$$

where (a) follows from inserting the estimate of the KCME and the definition of the norm; (b) follows from the reproducing property and the symmetry of the kernels; and (c) follows from the fact that $\text{Tr}(AB) = \sum_{i,j=1}^n a_{ij}b_{ji}$ for symmetric $A, B \in \mathbb{R}^{n \times n}$.

The storage complexity of $\mathcal{O}(mn + m^2)$ comes from the fact we have to store

$$L_{\tilde{\boldsymbol{Y}}\tilde{\boldsymbol{Y}}}, K_{\tilde{\boldsymbol{X}}\tilde{\boldsymbol{X}}}, W_{\tilde{\boldsymbol{X}}\tilde{\boldsymbol{X}}} \in \mathbb{R}^{m \times m}, \text{ and } K_{\tilde{\boldsymbol{X}}\boldsymbol{X}}, L_{\boldsymbol{Y}\tilde{\boldsymbol{Y}}}^{\top} \in \mathbb{R}^{m \times n}.$$

The overall computational complexity of $\mathcal{O}(m^2 n + m^3)$ arises from the cost of solving the linear system,

$$A(K_{\tilde{\boldsymbol{X}}\tilde{\boldsymbol{X}}} + \lambda I) = K_{\boldsymbol{X}\tilde{\boldsymbol{X}}} \text{ for } A \in \mathbb{R}^{n \times m}$$

which dominates the $\mathcal{O}(m^2 n)$ cost of the singular remaining matrix multiplication, and the $\mathcal{O}(m^2 + mn)$ cost of taking Hadamard products required to compute the traces. ∎

We now derive the gradients of $\mathcal{J}^{\mathcal{D}}$, and show that they are cheap to compute and store:

**Lemma B.13.** *We compute the gradients of the objective function $\mathcal{J}^{\mathcal{D}} \to \mathbb{R}^{m \times d} \times \mathbb{R}^{m \times d} \to \mathbb{R}$ as*

$$\nabla_{\tilde{\boldsymbol{X}}} \mathcal{J}^{\mathcal{D}}(\tilde{\boldsymbol{X}}, \tilde{\boldsymbol{Y}}) = \frac{2}{n} Tr(\nabla_{\tilde{\boldsymbol{X}}} K_{\boldsymbol{X}\tilde{\boldsymbol{X}}} W_{\tilde{\boldsymbol{X}}\tilde{\boldsymbol{X}}} L_{\tilde{\boldsymbol{Y}}\tilde{\boldsymbol{Y}}} W_{\tilde{\boldsymbol{X}}\tilde{\boldsymbol{X}}} K_{\tilde{\boldsymbol{X}}\boldsymbol{X}}) \tag{28}$$
$$- \frac{2}{n} Tr(K_{\boldsymbol{X}\tilde{\boldsymbol{X}}} W_{\tilde{\boldsymbol{X}}\tilde{\boldsymbol{X}}} \nabla_{\tilde{\boldsymbol{X}}} K_{\tilde{\boldsymbol{X}}\tilde{\boldsymbol{X}}} W_{\tilde{\boldsymbol{X}}\tilde{\boldsymbol{X}}} L_{\tilde{\boldsymbol{Y}}\tilde{\boldsymbol{Y}}} W_{\tilde{\boldsymbol{X}}\tilde{\boldsymbol{X}}} K_{\tilde{\boldsymbol{X}}\boldsymbol{X}})$$
$$+ \frac{2}{n} Tr(L_{\boldsymbol{Y}\tilde{\boldsymbol{Y}}} W_{\tilde{\boldsymbol{X}}\tilde{\boldsymbol{X}}} \nabla_{\tilde{\boldsymbol{X}}} K_{\tilde{\boldsymbol{X}}\tilde{\boldsymbol{X}}} W_{\tilde{\boldsymbol{X}}\tilde{\boldsymbol{X}}} K_{\tilde{\boldsymbol{X}}\boldsymbol{X}})$$
$$- \frac{2}{n} Tr(L_{\boldsymbol{Y}\tilde{\boldsymbol{Y}}} W_{\tilde{\boldsymbol{X}}\tilde{\boldsymbol{X}}} \nabla_{\tilde{\boldsymbol{X}}} K_{\tilde{\boldsymbol{X}}\boldsymbol{X}}),$$

$$\nabla_{\tilde{\boldsymbol{Y}}} \mathcal{J}^{\mathcal{D}}(\tilde{\boldsymbol{X}}, \tilde{\boldsymbol{Y}}) = \frac{1}{n} Tr\left(K_{\boldsymbol{X},\tilde{\boldsymbol{X}}} W_{\tilde{\boldsymbol{X}},\tilde{\boldsymbol{X}}} \nabla_{\tilde{\boldsymbol{Y}}} L_{\tilde{\boldsymbol{Y}},\tilde{\boldsymbol{Y}}} W_{\tilde{\boldsymbol{X}},\tilde{\boldsymbol{X}}} K_{\tilde{\boldsymbol{X}},\boldsymbol{X}}\right) \tag{29}$$
$$- \frac{2}{n} Tr(\nabla_{\tilde{\boldsymbol{Y}}} L_{\boldsymbol{Y}\tilde{\boldsymbol{Y}}} W_{\tilde{\boldsymbol{X}}\tilde{\boldsymbol{X}}} K_{\tilde{\boldsymbol{X}}\boldsymbol{X}}),$$

*where*

$$\nabla_{\tilde{\boldsymbol{X}}} K_{\tilde{\boldsymbol{X}},\tilde{\boldsymbol{X}}} \in \mathbb{R}^{m \times m \times m \times d}, \text{ with } \left[\nabla_{\tilde{\boldsymbol{X}}} K_{\tilde{\boldsymbol{X}},\tilde{\boldsymbol{X}}}\right]_{ijq} := \nabla_{\tilde{\boldsymbol{x}}_q} k(\tilde{\boldsymbol{x}}_i, \tilde{\boldsymbol{x}}_j) \in \mathbb{R}^d,$$
$$\nabla_{\tilde{\boldsymbol{X}}} K_{\tilde{\boldsymbol{X}},\boldsymbol{X}} \in \mathbb{R}^{m \times n \times m \times d}, \text{ with } \left[\nabla_{\tilde{\boldsymbol{X}}} K_{\tilde{\boldsymbol{X}},\boldsymbol{X}}\right]_{ijq} := \nabla_{\tilde{\boldsymbol{x}}_q} k(\tilde{\boldsymbol{x}}_i, \boldsymbol{x}_j) \in \mathbb{R}^d,$$
$$\nabla_{\tilde{\boldsymbol{Y}}} L_{\tilde{\boldsymbol{Y}},\tilde{\boldsymbol{Y}}} \in \mathbb{R}^{m \times m \times m \times p}, \text{ with } \left[\nabla_{\tilde{\boldsymbol{Y}}} L_{\tilde{\boldsymbol{Y}},\tilde{\boldsymbol{Y}}}\right]_{ijq} := \nabla_{\tilde{\boldsymbol{y}}_q} l(\tilde{\boldsymbol{y}}_i, \tilde{\boldsymbol{y}}_j) \in \mathbb{R}^p,$$
$$\nabla_{\tilde{\boldsymbol{Y}}} K_{\tilde{\boldsymbol{Y}},\tilde{\boldsymbol{Y}}} \in \mathbb{R}^{m \times n \times m \times p}, \text{ with } \left[\nabla_{\tilde{\boldsymbol{Y}}} L_{\tilde{\boldsymbol{Y}},\tilde{\boldsymbol{Y}}}\right]_{ijq} := \nabla_{\tilde{\boldsymbol{y}}_q} l(\tilde{\boldsymbol{y}}_i, \boldsymbol{y}_j) \in \mathbb{R}^p,$$

*and $W_{\tilde{\boldsymbol{X}}\tilde{\boldsymbol{X}}} := (K_{\tilde{\boldsymbol{X}}\tilde{\boldsymbol{X}}} + \lambda I)^{-1}$, with $\mathcal{O}(m^2 n + m^3)$ time complexity and $\mathcal{O}(mn + m^2)$ storage complexity, i.e. linear with respect to the size of the full dataset $\mathcal{D}$.*

**Proof**: We use the matrix-by-matrix derivative identities from Section B.4.1. Firstly, by applying rules (15) and (18) we can immediately see that

$$\nabla_{\tilde{\boldsymbol{Y}}} \mathcal{J}^{\mathcal{D}}(\tilde{\boldsymbol{X}}, \tilde{\boldsymbol{Y}}) = \frac{1}{n} \text{Tr}(K_{\boldsymbol{X},\tilde{\boldsymbol{X}}} W_{\tilde{\boldsymbol{X}},\tilde{\boldsymbol{X}}} \nabla_{\tilde{\boldsymbol{Y}}} L_{\tilde{\boldsymbol{Y}},\tilde{\boldsymbol{Y}}} W_{\tilde{\boldsymbol{X}},\tilde{\boldsymbol{X}}} K_{\tilde{\boldsymbol{X}},\boldsymbol{X}})$$
$$- \frac{2}{n} \text{Tr}(\nabla_{\tilde{\boldsymbol{Y}}} L_{\boldsymbol{Y}\tilde{\boldsymbol{Y}}} W_{\tilde{\boldsymbol{X}}\tilde{\boldsymbol{X}}} K_{\tilde{\boldsymbol{X}}\boldsymbol{X}}).$$

Now, we have

$$\nabla_{\tilde{\boldsymbol{X}}} \mathcal{J}^{\mathcal{D}}(\tilde{\boldsymbol{X}}, \tilde{\boldsymbol{Y}}) \underset{(a)}{=} \frac{1}{n} \nabla_{\tilde{\boldsymbol{X}}} \mathrm{Tr}\left(K_{\boldsymbol{X}\tilde{\boldsymbol{X}}} W_{\tilde{\boldsymbol{X}}\tilde{\boldsymbol{X}}} L_{\tilde{\boldsymbol{Y}}\tilde{\boldsymbol{Y}}} W_{\tilde{\boldsymbol{X}}\tilde{\boldsymbol{X}}} K_{\tilde{\boldsymbol{X}}\boldsymbol{X}}\right)$$

$$- \frac{2}{n} \nabla_{\tilde{\boldsymbol{X}}} \mathrm{Tr}\left(L_{\boldsymbol{Y}\tilde{\boldsymbol{Y}}} W_{\tilde{\boldsymbol{X}}\tilde{\boldsymbol{X}}} K_{\tilde{\boldsymbol{X}}\boldsymbol{X}}\right)$$

$$\underset{(b)}{=} \frac{1}{n} \mathrm{Tr}\left(\nabla_{\tilde{\boldsymbol{X}}} K_{\boldsymbol{X}\tilde{\boldsymbol{X}}} W_{\tilde{\boldsymbol{X}}\tilde{\boldsymbol{X}}} L_{\tilde{\boldsymbol{Y}}\tilde{\boldsymbol{Y}}} W_{\tilde{\boldsymbol{X}}\tilde{\boldsymbol{X}}} K_{\tilde{\boldsymbol{X}}\boldsymbol{X}}\right)$$

$$+ \frac{1}{n} \mathrm{Tr}\left(K_{\boldsymbol{X}\tilde{\boldsymbol{X}}} \nabla_{\tilde{\boldsymbol{X}}} W_{\tilde{\boldsymbol{X}}\tilde{\boldsymbol{X}}} L_{\tilde{\boldsymbol{Y}}\tilde{\boldsymbol{Y}}} W_{\tilde{\boldsymbol{X}}\tilde{\boldsymbol{X}}} K_{\tilde{\boldsymbol{X}}\boldsymbol{X}}\right)$$

$$+ \frac{1}{n} \mathrm{Tr}\left(K_{\boldsymbol{X}\tilde{\boldsymbol{X}}} W_{\tilde{\boldsymbol{X}}\tilde{\boldsymbol{X}}} L_{\tilde{\boldsymbol{Y}}\tilde{\boldsymbol{Y}}} \nabla_{\tilde{\boldsymbol{X}}} W_{\tilde{\boldsymbol{X}}\tilde{\boldsymbol{X}}} K_{\tilde{\boldsymbol{X}}\boldsymbol{X}}\right)$$

$$+ \frac{1}{n} \mathrm{Tr}\left(K_{\boldsymbol{X}\tilde{\boldsymbol{X}}} W_{\tilde{\boldsymbol{X}}\tilde{\boldsymbol{X}}} L_{\tilde{\boldsymbol{Y}}\tilde{\boldsymbol{Y}}} W_{\tilde{\boldsymbol{X}}\tilde{\boldsymbol{X}}} \nabla_{\tilde{\boldsymbol{X}}} K_{\tilde{\boldsymbol{X}}\boldsymbol{X}}\right)$$

$$- \frac{2}{n} \mathrm{Tr}\left(L_{\boldsymbol{Y}\tilde{\boldsymbol{Y}}} \nabla_{\tilde{\boldsymbol{X}}} W_{\tilde{\boldsymbol{X}}\tilde{\boldsymbol{X}}} K_{\tilde{\boldsymbol{X}}\boldsymbol{X}}\right)$$

$$- \frac{2}{n} \mathrm{Tr}\left(L_{\boldsymbol{Y}\tilde{\boldsymbol{Y}}} W_{\tilde{\boldsymbol{X}}\tilde{\boldsymbol{X}}} \nabla_{\tilde{\boldsymbol{X}}} K_{\tilde{\boldsymbol{X}}\boldsymbol{X}}\right)$$

$$\underset{(c)}{=} \frac{2}{n} \mathrm{Tr}\left(\nabla_{\tilde{\boldsymbol{X}}} K_{\boldsymbol{X}\tilde{\boldsymbol{X}}} W_{\tilde{\boldsymbol{X}}\tilde{\boldsymbol{X}}} L_{\tilde{\boldsymbol{Y}}\tilde{\boldsymbol{Y}}} W_{\tilde{\boldsymbol{X}}\tilde{\boldsymbol{X}}} K_{\tilde{\boldsymbol{X}}\boldsymbol{X}}\right)$$

$$+ \frac{2}{n} \mathrm{Tr}\left(K_{\boldsymbol{X}\tilde{\boldsymbol{X}}} \nabla_{\tilde{\boldsymbol{X}}} W_{\tilde{\boldsymbol{X}}\tilde{\boldsymbol{X}}} L_{\tilde{\boldsymbol{Y}}\tilde{\boldsymbol{Y}}} W_{\tilde{\boldsymbol{X}}\tilde{\boldsymbol{X}}} K_{\tilde{\boldsymbol{X}}\boldsymbol{X}}\right)$$

$$- \frac{2}{n} \mathrm{Tr}\left(L_{\boldsymbol{Y}\tilde{\boldsymbol{Y}}} \nabla_{\tilde{\boldsymbol{X}}} W_{\tilde{\boldsymbol{X}}\tilde{\boldsymbol{X}}} K_{\tilde{\boldsymbol{X}}\boldsymbol{X}}\right)$$

$$- \frac{2}{n} \mathrm{Tr}\left(L_{\boldsymbol{Y}\tilde{\boldsymbol{Y}}} W_{\tilde{\boldsymbol{X}}\tilde{\boldsymbol{X}}} \nabla_{\tilde{\boldsymbol{X}}} K_{\tilde{\boldsymbol{X}}\boldsymbol{X}}\right),$$

where (a) follows from the linearity of the gradient operator; (b) follows from a combination of rules (15) and (18); and (c) follows from the symmetry of the feature kernel $k(\cdot, \cdot)$. Then, by applying rule (17), we have

$$\nabla_{\tilde{\boldsymbol{X}}} \mathcal{J}^{\mathcal{D}}(\tilde{\boldsymbol{X}}, \tilde{\boldsymbol{Y}}) = \frac{2}{n} \mathrm{Tr}\left(\nabla_{\tilde{\boldsymbol{X}}} K_{\boldsymbol{X}\tilde{\boldsymbol{X}}} W_{\tilde{\boldsymbol{X}}\tilde{\boldsymbol{X}}} L_{\tilde{\boldsymbol{Y}}\tilde{\boldsymbol{Y}}} W_{\tilde{\boldsymbol{X}}\tilde{\boldsymbol{X}}} K_{\tilde{\boldsymbol{X}}\boldsymbol{X}}\right)$$

$$- \frac{2}{n} \mathrm{Tr}\left(K_{\boldsymbol{X}\tilde{\boldsymbol{X}}} W_{\tilde{\boldsymbol{X}}\tilde{\boldsymbol{X}}} \nabla_{\tilde{\boldsymbol{X}}} K_{\tilde{\boldsymbol{X}}\tilde{\boldsymbol{X}}} W_{\tilde{\boldsymbol{X}}\tilde{\boldsymbol{X}}} L_{\tilde{\boldsymbol{Y}}\tilde{\boldsymbol{Y}}} W_{\tilde{\boldsymbol{X}}\tilde{\boldsymbol{X}}} K_{\tilde{\boldsymbol{X}}\boldsymbol{X}}\right)$$

$$+ \frac{2}{n} \mathrm{Tr}\left(L_{\boldsymbol{Y}\tilde{\boldsymbol{Y}}} W_{\tilde{\boldsymbol{X}}\tilde{\boldsymbol{X}}} \nabla_{\tilde{\boldsymbol{X}}} K_{\tilde{\boldsymbol{X}}\tilde{\boldsymbol{X}}} W_{\tilde{\boldsymbol{X}}\tilde{\boldsymbol{X}}} K_{\tilde{\boldsymbol{X}}\boldsymbol{X}}\right)$$

$$- \frac{2}{n} \mathrm{Tr}\left(L_{\boldsymbol{Y}\tilde{\boldsymbol{Y}}} W_{\tilde{\boldsymbol{X}}\tilde{\boldsymbol{X}}} \nabla_{\tilde{\boldsymbol{X}}} K_{\tilde{\boldsymbol{X}}\boldsymbol{X}}\right).$$

Now, in order to establish the cost of computing this estimate, the first thing to notice is that there is a significant amount of symmetry and shared computation between the terms. In particular, avoiding the gradients

$$\nabla_{\tilde{\boldsymbol{X}}} K_{\boldsymbol{X}\tilde{\boldsymbol{X}}} \in \mathbb{R}^{n \times m \times m \times d}, \quad \nabla_{\tilde{\boldsymbol{X}}} K_{\tilde{\boldsymbol{X}}\boldsymbol{X}} \in \mathbb{R}^{m \times n \times m \times d}, \quad \nabla_{\tilde{\boldsymbol{X}}} K_{\tilde{\boldsymbol{X}}\tilde{\boldsymbol{X}}} \in \mathbb{R}^{m \times m \times m \times d},$$

for now, we need to compute

$$A := K_{\boldsymbol{X}\tilde{\boldsymbol{X}}} W_{\tilde{\boldsymbol{X}}\tilde{\boldsymbol{X}}} \in \mathbb{R}^{n \times m}, \quad B := L_{\boldsymbol{Y}\tilde{\boldsymbol{Y}}} W_{\tilde{\boldsymbol{X}}\tilde{\boldsymbol{X}}} \in \mathbb{R}^{n \times m},$$

$$C := W_{\tilde{\boldsymbol{X}}\tilde{\boldsymbol{X}}} L_{\tilde{\boldsymbol{Y}}\tilde{\boldsymbol{Y}}} W_{\tilde{\boldsymbol{X}}\tilde{\boldsymbol{X}}} K_{\tilde{\boldsymbol{X}}\boldsymbol{X}} \in \mathbb{R}^{m \times n},$$

then we have

$$\nabla_{\tilde{\boldsymbol{X}}} \mathcal{J}^{\mathcal{D}}(\tilde{\boldsymbol{X}}, \tilde{\boldsymbol{Y}}) = \frac{2}{n} \mathrm{Tr}\left(\nabla_{\tilde{\boldsymbol{X}}} K_{\boldsymbol{X}\tilde{\boldsymbol{X}}} C\right) - \frac{2}{n} \mathrm{Tr}\left(B \nabla_{\tilde{\boldsymbol{X}}} K_{\tilde{\boldsymbol{X}}\tilde{\boldsymbol{X}}} C\right)$$

$$+ \frac{2}{n} \mathrm{Tr}\left(B \nabla_{\tilde{\boldsymbol{X}}} K_{\tilde{\boldsymbol{X}}\tilde{\boldsymbol{X}}} A^{\top}\right) - \frac{2}{n} \mathrm{Tr}\left(B^{\top} \nabla_{\tilde{\boldsymbol{X}}} K_{\tilde{\boldsymbol{X}}\boldsymbol{X}}\right)$$

$$\underset{(a)}{=} \frac{2}{n} \mathrm{Tr}\left(\nabla_{\tilde{\boldsymbol{X}}} K_{\boldsymbol{X}\tilde{\boldsymbol{X}}} C\right) - \frac{2}{n} \mathrm{Tr}\left(\nabla_{\tilde{\boldsymbol{X}}} K_{\tilde{\boldsymbol{X}}\tilde{\boldsymbol{X}}} C B\right)$$

$$+ \frac{2}{n} \mathrm{Tr}\left(\nabla_{\tilde{\boldsymbol{X}}} K_{\tilde{\boldsymbol{X}}\tilde{\boldsymbol{X}}} A^{\top} B\right) - \frac{2}{n} \mathrm{Tr}\left(\nabla_{\tilde{\boldsymbol{X}}} K_{\tilde{\boldsymbol{X}}\boldsymbol{X}} B^{\top}\right),$$

where (a) follows from the cyclic property of the trace and the symmetry of the kernels. Now, the cost of computing $A$, $B$ and $C$ is $\mathcal{O}(nm^2 + m^3)$, and given these matrices, the cost of computing $D := CB \in \mathbb{R}^{m \times m}$ and $E := A^\top B \in \mathbb{R}^{m \times m}$ is $\mathcal{O}(nm^2)$. So, we are now in a position where we have to compute

$$\nabla_{\tilde{\boldsymbol{X}}} \mathcal{J}^{\mathcal{D}}(\tilde{\boldsymbol{X}}, \tilde{\boldsymbol{Y}}) = \frac{2}{n} \text{Tr}\left(\nabla_{\tilde{\boldsymbol{X}}} K_{\boldsymbol{X}\tilde{\boldsymbol{X}}} C\right) - \frac{2}{n} \text{Tr}\left(\nabla_{\tilde{\boldsymbol{X}}} K_{\tilde{\boldsymbol{X}}\tilde{\boldsymbol{X}}} D\right)$$
$$+ \frac{2}{n} \text{Tr}\left(\nabla_{\tilde{\boldsymbol{X}}} K_{\tilde{\boldsymbol{X}}\tilde{\boldsymbol{X}}} E\right) - \frac{2}{n} \text{Tr}\left(\nabla_{\tilde{\boldsymbol{X}}} K_{\tilde{\boldsymbol{X}}\boldsymbol{X}} B^\top\right),$$

but, following the exact same logic as in section B.4.2, we can compute these traces as sums of products where the majority of the elements of $\nabla_{\tilde{\boldsymbol{X}}} K_{\tilde{\boldsymbol{X}}\tilde{\boldsymbol{X}}}$ and $\nabla_{\tilde{\boldsymbol{X}}} K_{\boldsymbol{X}\tilde{\boldsymbol{X}}}$ are zeros, and hence much wasted computation and storage can be avoided. Therefore, similar to equations (21) and (22), we can compute the quantities $\text{Tr}\left(\nabla_{\tilde{\boldsymbol{X}}} K_{\boldsymbol{X}\tilde{\boldsymbol{X}}} C\right)$ and $\text{Tr}\left(\nabla_{\tilde{\boldsymbol{X}}} K_{\tilde{\boldsymbol{X}}\boldsymbol{X}} B^\top\right)$ with $\mathcal{O}(mn)$ time and storage cost, and $\text{Tr}\left(\nabla_{\tilde{\boldsymbol{X}}} K_{\tilde{\boldsymbol{X}}\tilde{\boldsymbol{X}}} E\right)$ and $\text{Tr}\left(\nabla_{\tilde{\boldsymbol{X}}} K_{\tilde{\boldsymbol{X}}\tilde{\boldsymbol{X}}} D\right)$ with $\mathcal{O}(m^2)$ time and storage cost.

Therefore, we have an overall storage cost of $\mathcal{O}(mn + m^2)$, and time cost of $\mathcal{O}(m^3 + m^2n)$, i.e. *linear* with respect to the size $n$ of the full dataset $\mathcal{D}$. ∎

**Remark B.14.** Above we have derived analytical gradients of the objective function, and shown they can be computed in linear time. In practice one computes the gradients using JAX's [79] auto-differentiation capabilities. The authors observed minimal slowdown from using auto-differentiation.

## B.11 Accelerating Objective Computation for Discrete Conditional Distributions

As outlined in C.1.3, for discrete conditional distributions e.g. those encountered in classification data, the response kernel $l(\cdot, \cdot)$ is chosen to be the indicator kernel. In this case, gradient descent on the responses is no longer possible. For herding-type algorithms it is straightforward to alternate between taking a step on the feature $\boldsymbol{x}$, and then, given this new $\boldsymbol{x}$, exhaustively search over the possible values of the paired response $\boldsymbol{y}$.

For KIP-style algorithms, given a step on each of the features in the compressed set $\tilde{\boldsymbol{X}}$, it would be very expensive to exhaustively search over each possible combination of responses in $\tilde{\boldsymbol{Y}}$ to find the jointly optimal combination. To reduce this cost, we take a greedy approach and iteratively, response-by-response, exhaustively search over the possible values carrying forward the optimal value of each response to the next iteration. For pseudocode for these procedures see Section D.1.

In this section, by treating each KIP-style objective just as a function of each response $\tilde{\boldsymbol{y}}$ in $\tilde{\boldsymbol{Y}}$, we show that one can accelerate computation of the JKIP and ACKIP objective functions, reducing the cost of the corresponding exhaustive search procedure significantly.

### B.11.1 Joint Kernel Inducing Points

Defining $\tilde{\boldsymbol{X}} := [\tilde{\boldsymbol{x}}_1, \tilde{\boldsymbol{x}}_2, \ldots, \tilde{\boldsymbol{x}}_m]^\top$ and $\tilde{\boldsymbol{Y}} := [\tilde{\boldsymbol{y}}_1, \tilde{\boldsymbol{y}}_2, \ldots, \tilde{\boldsymbol{y}}_m]^\top$, in JKIP, we optimise the following objective

$$\mathcal{L}^{\mathcal{D}}(\tilde{\boldsymbol{X}}, \tilde{\boldsymbol{Y}}) := \frac{1}{m^2} \sum_{i,j=1}^{m} k(\tilde{\boldsymbol{x}}_j, \tilde{\boldsymbol{x}}_i) l(\tilde{\boldsymbol{y}}_i, \tilde{\boldsymbol{y}}_j) - \frac{2}{mn} \sum_{i,j=1}^{m,n} k(\tilde{\boldsymbol{x}}_i, \boldsymbol{x}_j) l(\boldsymbol{y}_j, \tilde{\boldsymbol{y}}_i). \tag{30}$$

Now, if one treats this objective solely as a function of a single $\tilde{\boldsymbol{y}}_t \in \tilde{\boldsymbol{Y}}$, fixing $\tilde{\boldsymbol{X}}$ and the remaining points in $\tilde{\boldsymbol{Y}}$, then it easy to see that optimising (30) is equivalent to optimising

$$\mathcal{F}^{\mathcal{D}}(\tilde{\boldsymbol{y}}_t) := \frac{2}{m^2} \sum_{j=1}^{m} k(\tilde{\boldsymbol{x}}_j, \tilde{\boldsymbol{x}}_t) l(\tilde{\boldsymbol{y}}_t, \tilde{\boldsymbol{y}}_j) - \frac{2}{mn} \sum_{j=1}^{n} k(\tilde{\boldsymbol{x}}_t, \boldsymbol{x}_j) l(\boldsymbol{y}_j, \tilde{\boldsymbol{y}}_t)$$

$$= \frac{2}{m^2} \tilde{\boldsymbol{K}}_m(\tilde{\boldsymbol{x}}_t)^\top \tilde{\boldsymbol{L}}_m(\tilde{\boldsymbol{y}}_t) - \frac{2}{mn} \boldsymbol{K}_n(\tilde{\boldsymbol{x}}_t)^\top \boldsymbol{L}_n(\tilde{\boldsymbol{y}}_t) \tag{31}$$

where $\tilde{\boldsymbol{K}}_m(\boldsymbol{x}) := [k(\boldsymbol{x}, \tilde{\boldsymbol{x}}_1), \ldots, k(\boldsymbol{x}, \tilde{\boldsymbol{x}}_m)]^\top$, and $\boldsymbol{K}_n(\boldsymbol{x}) := [k(\boldsymbol{x}, \boldsymbol{x}_1), \ldots, k(\boldsymbol{x}, \boldsymbol{x}_n)]^\top$, $\tilde{\boldsymbol{L}}_m(\boldsymbol{y}) := [l(\boldsymbol{y}, \tilde{\boldsymbol{y}}_1), \ldots, l(\boldsymbol{y}, \tilde{\boldsymbol{y}}_m)]^\top$, and $\boldsymbol{L}_n(\boldsymbol{y}) := [l(\boldsymbol{y}, \boldsymbol{y}_1), \ldots, l(\boldsymbol{y}, \boldsymbol{y}_n)]^\top$. This reduces the cost of evaluating the objective function by a factor of $m$, both in storage and time. Note that if using a non-stationary kernel $l(\cdot, \cdot)$, one must avoid double counting the diagonal term in Equation 30 by subtracting $\frac{1}{m} k(\tilde{\boldsymbol{x}}_t, \tilde{\boldsymbol{x}}_t) l(\tilde{\boldsymbol{y}}_t, \tilde{\boldsymbol{y}}_t)$.

### B.11.2 Average Conditional Kernel Inducing Points

In Section B.10, we saw that the objective of ACKIP can be written as

$$\mathcal{J}^{\mathcal{D}}(\tilde{\boldsymbol{X}}, \tilde{\boldsymbol{Y}}) = \frac{1}{n} \sum_{r=1}^{n} \sum_{i,j,p,q=1}^{m} k(\boldsymbol{x}_r, \tilde{\boldsymbol{x}}_j) \tilde{W}_{ji} l(\tilde{\boldsymbol{y}}_i, \tilde{\boldsymbol{y}}_p) \tilde{W}_{pq} k(\tilde{\boldsymbol{x}}_q, \boldsymbol{x}_r)$$

$$- \frac{2}{n} \sum_{p=1}^{n} \sum_{i,j=1}^{m} l(\boldsymbol{y}_p, \tilde{\boldsymbol{y}}_i) \tilde{W}_{ij} k(\tilde{\boldsymbol{x}}_j, \boldsymbol{x}_p).$$

Now, if one treats this objective solely as a function of a single $\tilde{\boldsymbol{y}}_t \in \tilde{\boldsymbol{Y}}$, fixing $\tilde{\boldsymbol{X}}$ and the remaining points in $\tilde{\boldsymbol{Y}}$, then it easy to see that optimising the above is equivalent to optimising

$$\mathcal{R}^{\mathcal{D}}(\tilde{\boldsymbol{y}}_t) := \frac{2}{n} \sum_{r=1}^{n} \sum_{j,p,q=1}^{m} k(\boldsymbol{x}_r, \tilde{\boldsymbol{x}}_j) \tilde{W}_{jt} l(\tilde{\boldsymbol{y}}_t, \tilde{\boldsymbol{y}}_p) \tilde{W}_{pq} k(\tilde{\boldsymbol{x}}_q, \boldsymbol{x}_r)$$

$$- \frac{2}{n} \sum_{p,j=1}^{n,m} l(\boldsymbol{y}_p, \tilde{\boldsymbol{y}}_t) \tilde{W}_{tj} k(\tilde{\boldsymbol{x}}_j, \boldsymbol{x}_p)$$

$$\underset{(a)}{=} \frac{2}{n} \sum_{r=1}^{n} \sum_{j,p,q=1}^{m} l(\tilde{\boldsymbol{y}}_t, \tilde{\boldsymbol{y}}_p) \tilde{W}_{pq} k(\tilde{\boldsymbol{x}}_q, \boldsymbol{x}_r) k(\boldsymbol{x}_r, \tilde{\boldsymbol{x}}_j) \tilde{W}_{jt}$$

$$- \frac{2}{n} \sum_{p,j=1}^{n,m} l(\tilde{\boldsymbol{y}}_t, \boldsymbol{y}_p) k(\boldsymbol{x}_p, \tilde{\boldsymbol{x}}_j) \tilde{W}_{jt}$$

$$\underset{(b)}{=} \frac{2}{n} \tilde{\boldsymbol{L}}_m(\tilde{\boldsymbol{y}}_t)^{\top} W_{\tilde{\boldsymbol{X}}\tilde{\boldsymbol{X}}} K_{\tilde{\boldsymbol{X}}\boldsymbol{X}} K_{\boldsymbol{X}\tilde{\boldsymbol{X}}} \boldsymbol{w}_t - \frac{2}{n} \boldsymbol{L}_n(\tilde{\boldsymbol{y}}_t)^{\top} K_{\boldsymbol{X}\tilde{\boldsymbol{X}}} \boldsymbol{w}_t \qquad (32)$$

where (a) follows from the symmetry of the kernel functions and a simple reordering of the terms, and (b) follows from defining $\boldsymbol{w}_t$ to be the $t^{\text{th}}$ row of $W_{\tilde{\boldsymbol{X}}\tilde{\boldsymbol{X}}}$. Note that if using a non-stationary kernel $l(\cdot, \cdot)$ one must again subtract a correction term $\frac{1}{n}(\boldsymbol{w}_t^{\top} K_{\tilde{\boldsymbol{X}}\boldsymbol{X}} K_{\boldsymbol{X}\tilde{\boldsymbol{X}}} \boldsymbol{w}_t) \cdot l(\tilde{\boldsymbol{y}}_t, \tilde{\boldsymbol{y}}_t)$ to avoid double counting the diagonal.

Now, the important observation to make is that one only needs to compute the terms involving $\tilde{\boldsymbol{X}}$ once, as we iterate through each $\tilde{\boldsymbol{y}}_t \in \tilde{\boldsymbol{Y}}$ with $\tilde{\boldsymbol{X}}$ fixed. Hence, if we ignore the one-time cost of computing $W_{\tilde{\boldsymbol{X}}\tilde{\boldsymbol{X}}} K_{\tilde{\boldsymbol{X}}\boldsymbol{X}}$, then this objective is $\mathcal{O}(m+n)$. A very similar derivation holds for ACKH.

## C   Experiment Details

All the experiments were performed on a single NVIDIA GTX 4070 Ti with 12GB of memory, CUDA 12.2 with driver 535.183.01 and JAX version 0.4.35.

As is ubiquitous in kernel methods, unless stated otherwise, we standardise the features and responses such that each dimension has zero mean and unit standard deviation.

For continuous conditional distributions, the kernel functions $k : \mathcal{X} \times \mathcal{X} \to \mathbb{R}$ and $l : \mathcal{Y} \times \mathcal{Y} \to \mathbb{R}$ are chosen to be the Gaussian kernel, defined as

$$k(\boldsymbol{x}, \boldsymbol{x}') := \exp\left(-\frac{1}{2\alpha_k^2} \|\boldsymbol{x} - \boldsymbol{x}'\|^2\right), \quad l(\boldsymbol{y}, \boldsymbol{y}') := \exp\left(-\frac{1}{2\alpha_l^2} \|\boldsymbol{y} - \boldsymbol{y}'\|^2\right)$$

where the lengthscales $\alpha_k, \alpha_l > 0$ are set via the median heuristic. That is, given a dataset $\{\boldsymbol{z}_i\}_{i=1}^{p}$, the median heuristic is defined to be

$$H_p := \text{Med}\left\{\|\boldsymbol{z}_i - \boldsymbol{z}_j\|^2 \ : \ 1 \le i \le j \le p\right\}$$

with lengthscale $\alpha := \sqrt{H_p/2}$ such that $r(\boldsymbol{z}, \boldsymbol{z}') := \exp\left(-\frac{1}{H_p}\|\boldsymbol{z} - \boldsymbol{z}'\|^2\right)$. This heuristic is a very widely used default choice that has shown strong empirical performance [80]. For discrete conditional distributions, we replace the response kernel with the indicator kernel, defined as

$$l(\boldsymbol{y}, \boldsymbol{y}') := \begin{cases} 1 & \text{if } \boldsymbol{y} = \boldsymbol{y}', \\ 0 & \text{otherwise} \end{cases} .$$

See Section C.1.3 for additional details on the impact of this change in kernel.

The regularisation parameter $\lambda > 0$ is selected using a two-stage cross-validation procedure on a validation set consisting of 10% of the data. In the first stage, a coarse grid of candidate $\lambda$ values are used to identify a preliminary range for the regularisation parameter. In the second stage, a finer grid is constructed within this range, and searched over. To avoid the $\mathcal{O}(n^3)$ cost of training the KCME, we randomly sample 10 subsets of the training data of size 1000 such that the optimal $\lambda$ value is averaged over these random training sets to determine the final regularisation parameter.

For all experiments, we use the Adam optimiser [81] as a sensible default choice. However, the implementations of ACKIP and ACKH allow for an arbitrary choice of optimiser via the Optax package [82]. We set a default learning rate of 0.01 across all experiments.

To provide reasonable seeds for minimisation across each iteration of JKH and ACKH, we follow the approach of [3] and draw 10 random auxiliary *pairs* from the training data, choosing the seed to be the auxiliary sample which achieves the smallest value of the relevant loss. In comparison, to initialise JKIP and ACKIP, we draw 10 auxiliary *sets* of size $m$, and choose the initial seed set to be the auxiliary set which achieves the smallest value of the relevant loss.

In order to compare the performance of the above approaches, we note that the primary goal of the kernel conditional mean embedding is to approximate the conditional expectation $\mathbb{E}[h(Y) \mid X = x]$ for arbitrary functions $h \in \mathcal{H}_l$ and conditioning variables $x \in \mathcal{X}$. Hence, to assess this approximation, we report the root mean square error (RMSE), where the mean is taken with respect to the distribution of the conditioning variable, $\mathbb{P}_X$.

$$\begin{aligned}
\text{RMSE}(\mathcal{C}) &:= \sqrt{\mathbb{E}_{\boldsymbol{x} \sim \mathbb{P}_X}\left[\left(\mathbb{E}[h(Y) \mid X = \boldsymbol{x}_i] - \langle \hat{\mu}^{\mathcal{C}}_{Y|X=\boldsymbol{x}_i}, h\rangle_{\mathcal{H}_l}\right)^2\right]} \\
&\approx \sqrt{\frac{1}{n}\sum_{i=1}^{n}\left(\mathbb{E}[h(Y) \mid X = \boldsymbol{x}_i] - \langle \hat{\mu}^{\mathcal{C}}_{Y|X=\boldsymbol{x}_i}, h\rangle_{\mathcal{H}_l}\right)^2}.
\end{aligned}$$

For continuous conditional distributions, we report the RMSE for the first, second and third moments, as well as the functions $h(y) = \sin(y)$, $h(y) = \cos(y)$, $h(y) = \exp(-y^2)$, $h(y) = |y|$, and $h(y) = \mathbb{1}_{y>0}$. These choices of test functions extend those chosen to evaluate Kernel Herding [3].

## C.1   Additional Figures and Experiments

In this section we include additional figures for the experiments in the main body, and further experiments on discrete conditional distributions.

### C.1.1   Matching the True Conditional Distribution

In this section we include some additional figures for the true conditional distribution compression task outlined in Section 5.

Figure 9 displays an example of a compressed set of size $m = 500$ constructed by each method. We note that JKH and JKIP have clearly constructed a representation of the joint distribution; with the JKIP construction seemingly more structured. It is interesting to note the extreme disparity between the compressed sets constructed by ACKH and ACKIP, versus the relative similarity of JKH and JKIP. We know that ACKIP achieves superior performance, hence it may be the case that the greedy heuristic is particularly poorly suited to targeting the AMCMD.

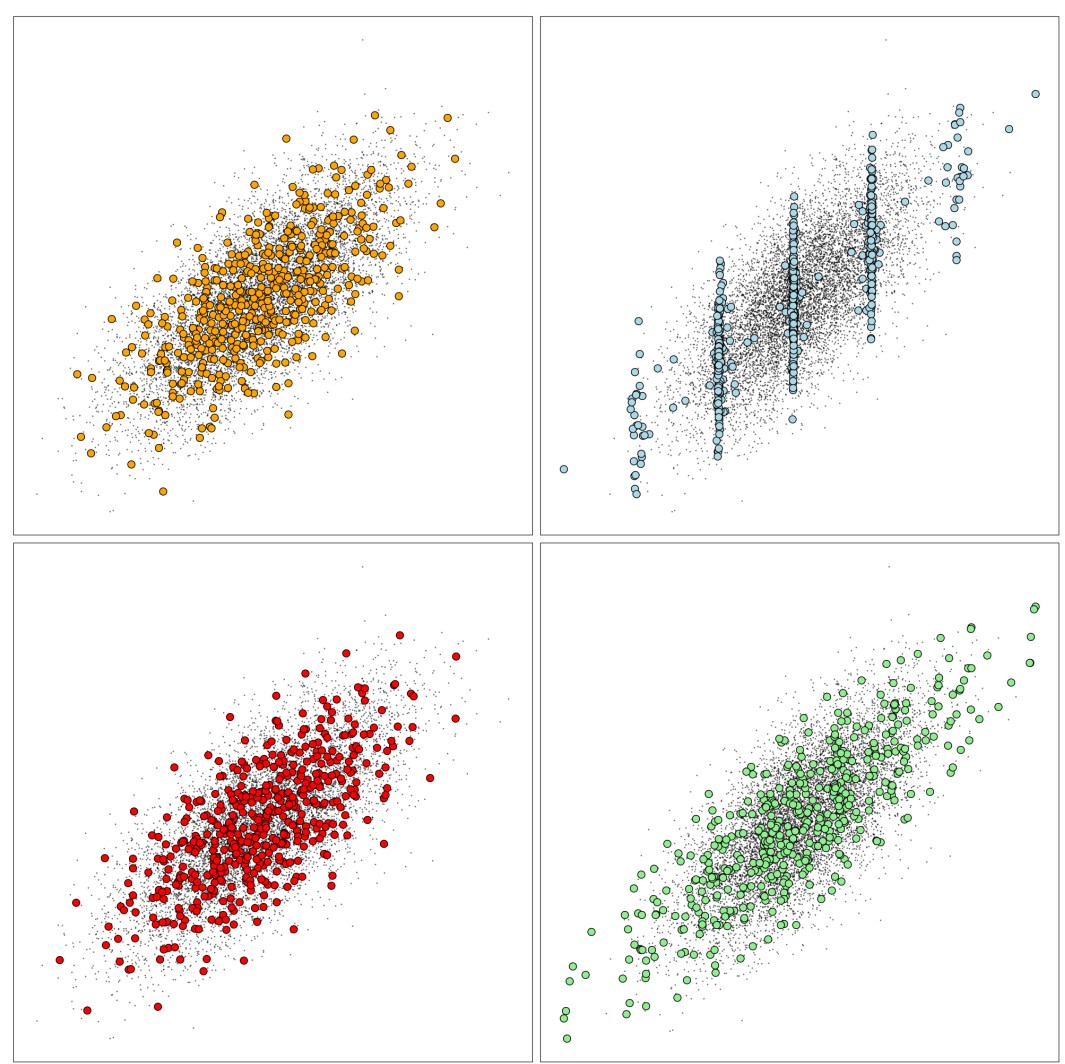

Figure 9: Compressed sets of size $m = 500$ constructed by JKH (orange), ACKH (blue), JKIP (red), and ACKIP (green), on the true conditional distribution compression task.

Figure 10 is an enlarged version of the first subfigure in Figure 2. Figure 11 is an enlarged version of Figure 2 showing results on a larger number of test functions. In Figures 11 and 12 we see that ACKIP is still dominant, achieving the best performance across all of the test functions, with ACKH achieving second best performance on six of the eight considered.

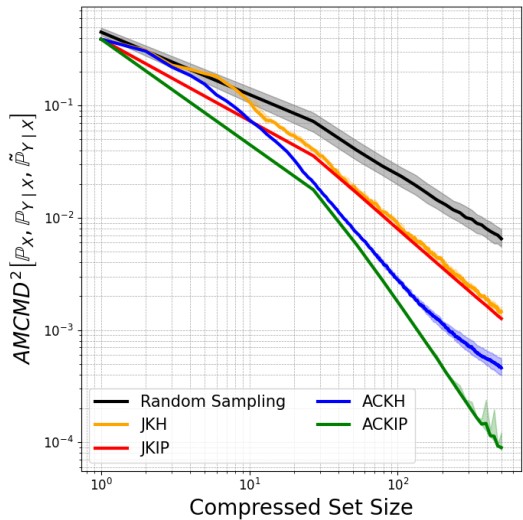

Figure 10: Results for the true conditional distribution compression task with parameters set as $a_0 = -0.5$, $a_1 = 0.5$, $\mu = 1$, $\sigma^2 = 1$, and $\sigma_\epsilon^2 = 0.5$. The $\text{AMCMD}^2 \left[ \mathbb{P}_X, \mathbb{P}_{Y|X}, \tilde{\mathbb{P}}_{Y|X} \right]$ is reported as the size of the compressed set increases. For JKH (orange), JKIP (red), ACKH (blue), and ACKIP (green), we display the median performance (bold line) with the 25th-75th percentiles (shaded region) over 20 runs. The error of random sampling (black) over 500 runs is also plotted for comparison.

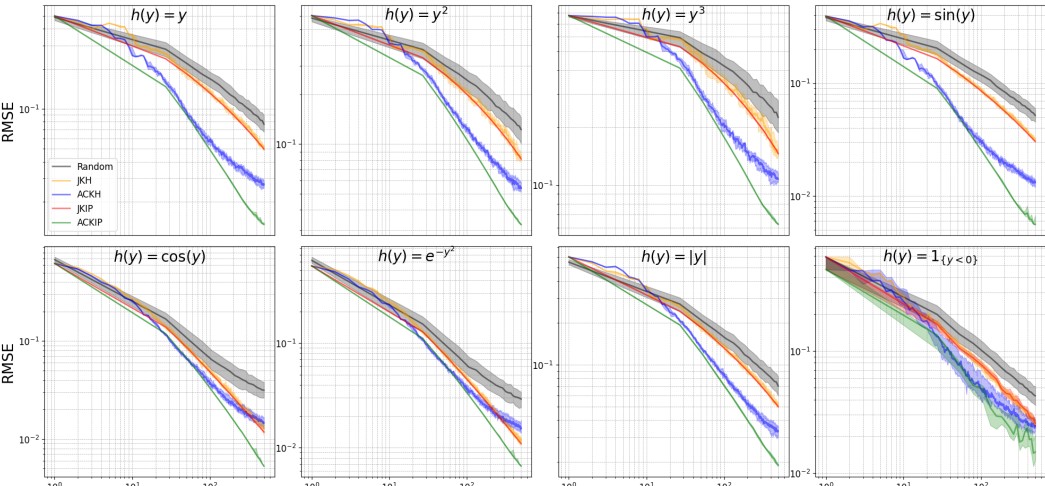

Figure 11: Results of the true conditional distribution compression task. The RMSE is reported across a variety of test functions, as the size of the compressed set increases. For JKH (orange), JKIP (red), ACKH (blue), and ACKIP (green), we display the median performance (bold line) with the 25th-75th percentiles (shaded region) over 20 runs. The error of random sampling (black) over 500 runs is also plotted for comparison.

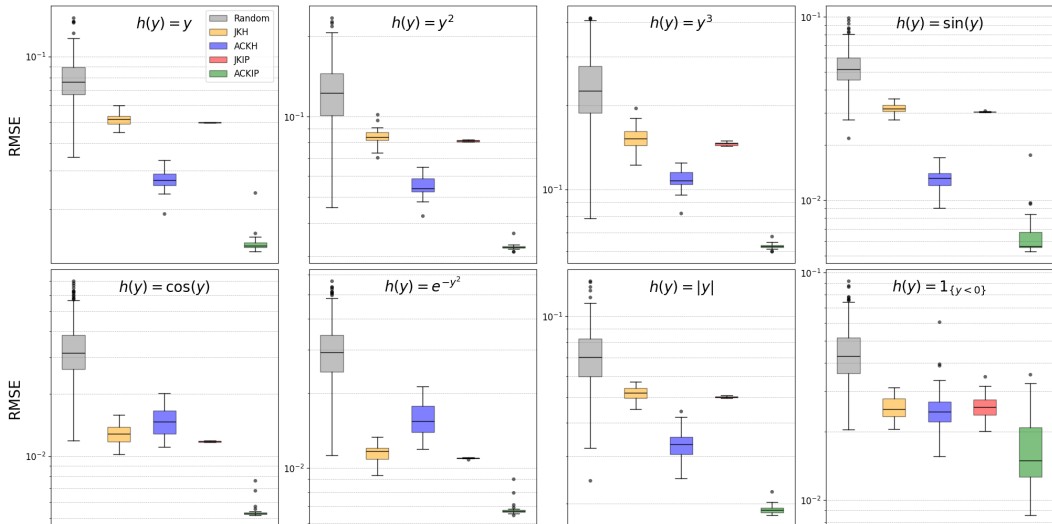

Figure 12: Results of the true conditional distribution compression task for compressed sets of size $m = 500$. The RMSE across a variety of test functions is reported, with the IQR highlighted for each method. Outliers are calculated as being above $Q_3 + 1.5\text{IQR}$ and below $Q_1 - 1.5\text{IQR}$.

### C.1.2 Matching the Empirical Conditional Distribution - Continuous / Regression

**Real**: In this section we include some additional figures for the *Superconductivity* data outlined in Section 5. Figure 13 shows the $\text{AMCMD}^2 \left[ \hat{\mathbb{P}}_X, \hat{\mathbb{P}}_{Y|X}, \tilde{\mathbb{P}}_{Y|X} \right]$ achieved by each distribution compression method as the compressed set size increases. ACKIP reaches the lowest AMCMD, followed by ACKH, JKIP, then JKH. Figures 14 and 15 are enlarged versions of 4 and 5 respectively, showing results on a larger number of test functions. We see that ACKIP achieves the lowest RMSE across each of the test functions, with ACKH in second for all but one. We also note that JKIP tends to achieve favourable performance versus JKH.

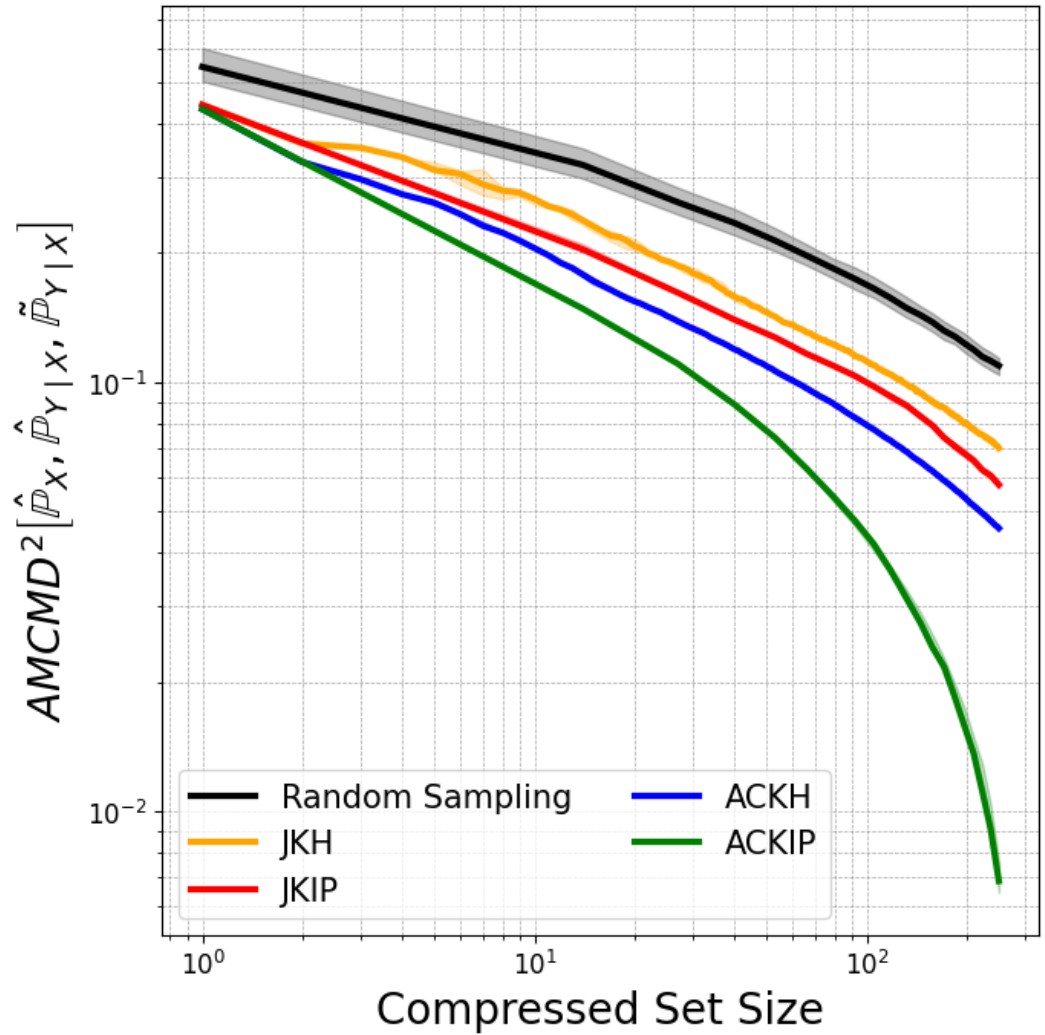

Figure 13: $\text{AMCMD}^2 \left[ \hat{\mathbb{P}}_X, \hat{\mathbb{P}}_{Y|X}, \tilde{\mathbb{P}}_{Y|X} \right]$ achieved by each method as a function of the size of the compressed sets constructed by JKH (orange), ACKH (blue), JKIP (red), and ACKIP (green), on the *Superconductivity* data. We display the median performance (bold line) with the 25th-75th percentiles (shaded region) over 20 runs. The error of random sampling (black) over 500 runs is also plotted for comparison.

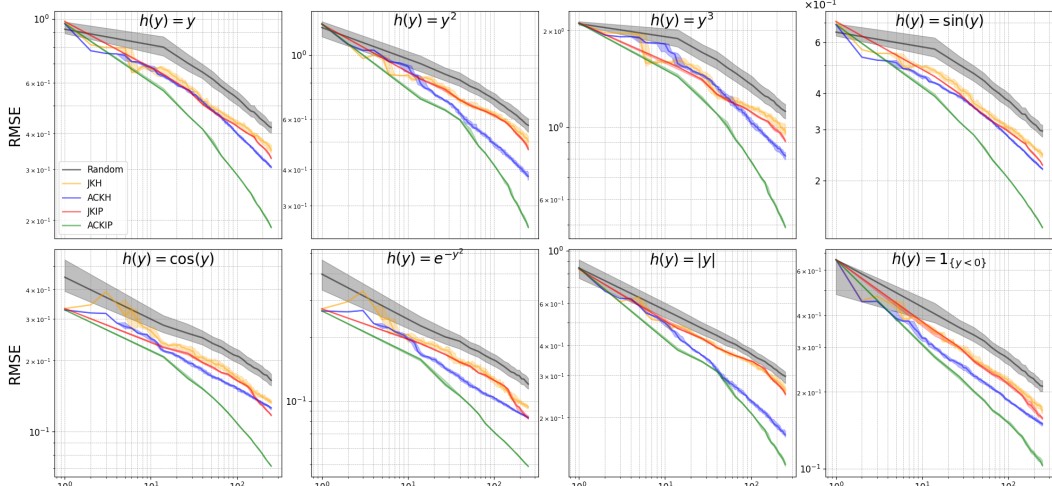

Figure 14: Results for the *Superconductivity* dataset; the RMSE is calculated against the full data estimates of $\mathbb{E}[h(Y) \mid X = \boldsymbol{x}_i]$ as the true values are not available. The RMSE is reported across a variety of test functions, as the size of the compressed set increases. For JKH (orange), JKIP (red), ACKH (blue), and ACKIP (green), wwe display the median performance (bold line) with the 25th-75th percentiles (shaded region) over 20 runs. The error of random sampling (black) over 500 runs is also plotted for comparison.

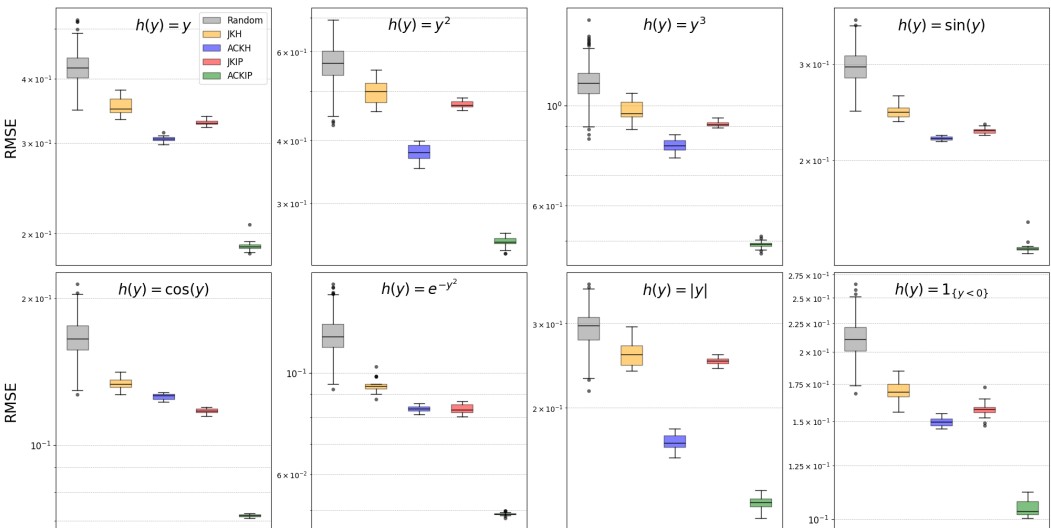

Figure 15: Results of the *Superconductivity* dataset for compressed sets of size $m = 500$. The RMSE across a variety of test functions is reported, with the IQR highlighted for each method. Outliers are calculated as being above $Q_3 + 1.5$IQR and below $Q_1 - 1.5$IQR.

**Synthetic**: In this section we include some additional figures for the *Heteroscedastic* data outlined in Section 5. Figures 16 and 17 are enlarged versions of 6 and 7 respectively, showing results on a larger number of test functions. They show that ACKIP consistently outperforms the other methods across a range of test functions, achieving the lowest RMSE as the size of the compressed set increases. In particular, Figure 7 demonstrates that with $m = 250$ pairs in the compressed set, ACKIP attains the lowest median RMSE on seven out of eight test functions. For the remaining function, all methods exhibit similar median performance. This highlights the advantage of directly compressing the conditional distribution with ACKIP, rather than targeting the joint distribution with JKH or JKIP. Finally, we note that JKIP consistently outperforms JKH across all test functions.

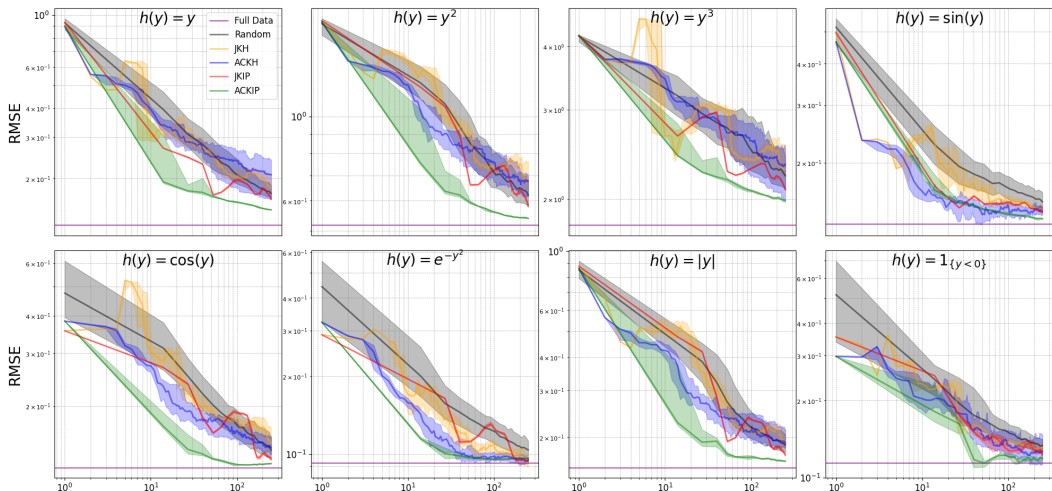

Figure 16: Results for *Heteroscedastic* data with parameters set as $\boldsymbol{a} := [3, -3, 6, -6]^{\top}$, $\boldsymbol{b} := [1, 0.1, 2, 0.5]^{\top}$, $\boldsymbol{c} := [-5, -2, 2, 5]^{\top}$, $\sigma_1^2 = 0.75$, and $\sigma_2^2 = 0.1$. RMSE is reported across a variety of test functions, as the size of the compressed set increases. For JKH (orange), JKIP (red), ACKH (blue), and ACKIP (green), we display the median performance (bold line) with the 25th-75th percentiles (shaded region) over 20 runs. The error of random sampling (black) over 500 runs is also plotted for comparison, as well as the performance of the full data (purple).

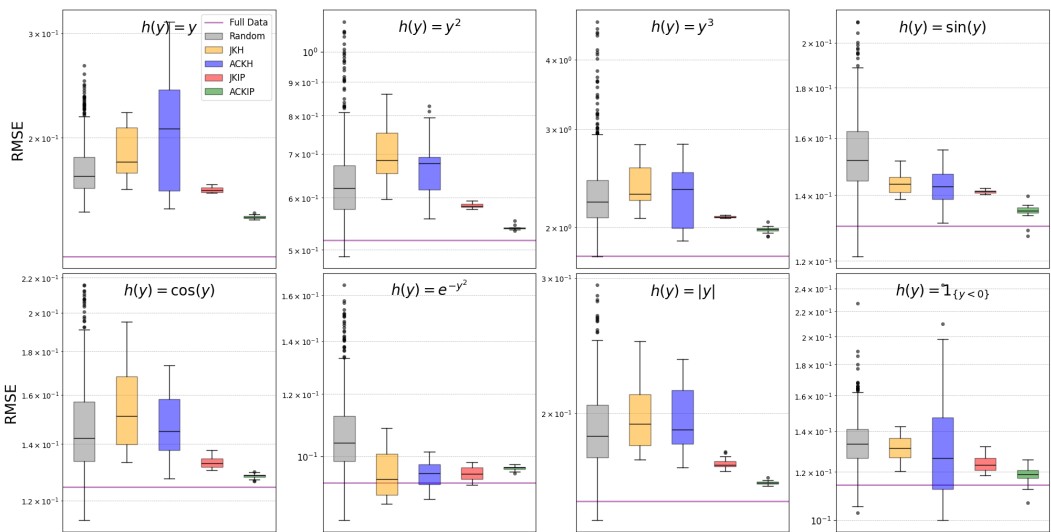

Figure 17: Results for *Heteroscedastic* data for compressed sets of size $m = 250$. The RMSE across a variety of test functions is reported, with the IQR highlighted for each method. Outliers are calculated as being above $Q_3 + 1.5\text{IQR}$ and below $Q_1 - 1.5\text{IQR}$.

Figure 18 displays an example of a compressed set of size $m = 250$ constructed by each method. We note that JKH and JKIP have clearly constructed a representation of the joint distribution, whereas ACKH and ACKIP have constructed something that is more difficult to straightforwardly interpret.

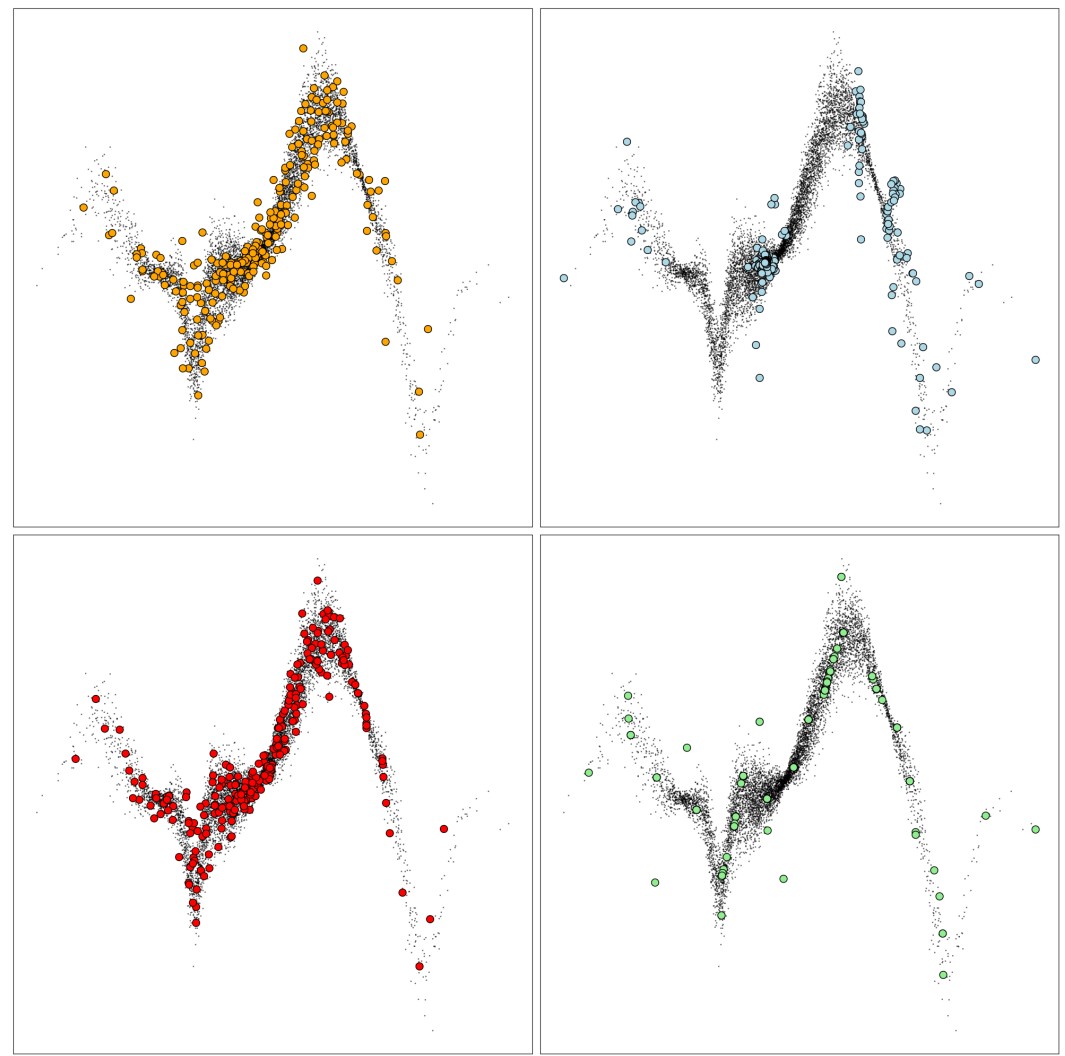

Figure 18: Compressed sets of size $m = 250$ constructed by JKH (orange), ACKH (blue), JKIP (red), and ACKIP (green), on *Heteroscedastic* data.

Figure 19 shows the $\text{AMCMD}^2 \left[ \hat{\mathbb{P}}_X, \hat{\mathbb{P}}_{Y|X}, \tilde{\mathbb{P}}_{Y|X} \right]$ achieved by each distribution compression method as the compressed set size increases. ACKIP reaches the lowest AMCMD, followed by JKIP. JKH and ACKH perform similarly to random sampling, though ACKH initially outperforms JKH and JKIP before being limited by its greedy nature, allowing JKIP to surpass it, and JKH to match it.

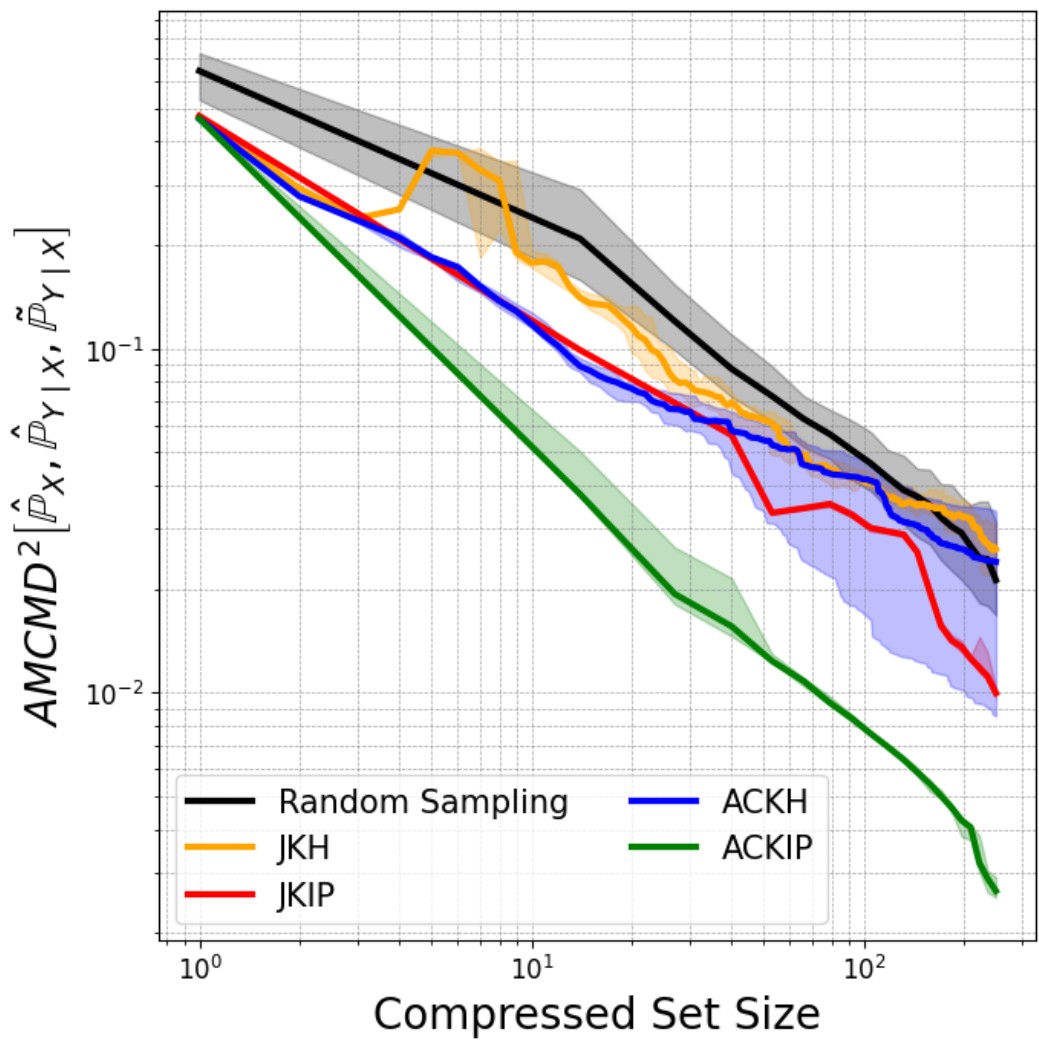

Figure 19: $\text{AMCMD}^2\left[\hat{\mathbb{P}}_X, \hat{\mathbb{P}}_{Y|X}, \tilde{\mathbb{P}}_{Y|X}\right]$ achieved by each method as a function of the size of the compressed sets constructed by JKH (orange), ACKH (blue), JKIP (red), and ACKIP (green), on the *Heteroscedastic* data. We display the median performance (bold line) with the 25th-75th percentiles (shaded region) over 20 runs. The error of random sampling (black) over 500 runs is also plotted for comparison.

**Ablation Study**: Using the same setup for the *Heteroscedastic* data, we repeat the experiment with the Gaussian kernels replaced by inverse multi-quadratic kernels. We report the results in Figures 20 and 21 where we can see that ACKIP still achieves the best results across a variety of test functions.

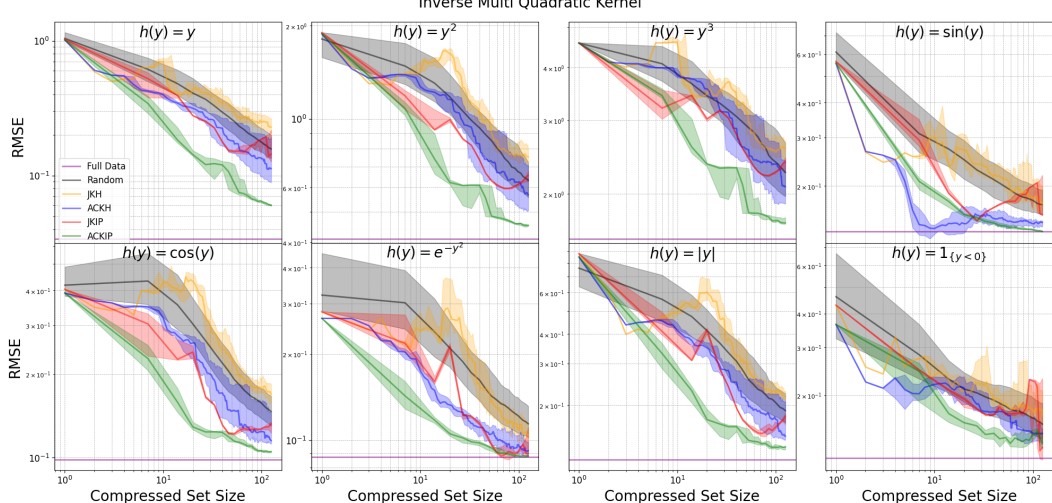

Figure 20: Results for *Heteroscedastic* data with IMQ kernels. RMSE is reported across a variety of test functions, as the size of the compressed set increases. For JKH (orange), JKIP (red), ACKH (blue), and ACKIP (green), we display the median performance (bold line) with the 25th-75th percentiles (shaded region) over 20 runs. The error of random sampling (black) over 500 runs is also plotted for comparison, as well as the performance of the full data (purple).

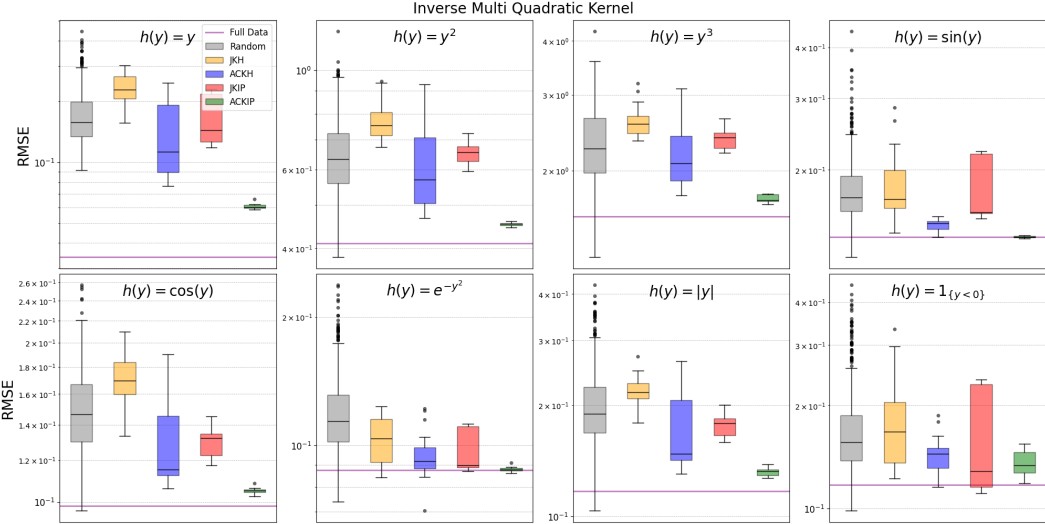

Figure 21: Results for *Heteroscedastic* data for compressed sets of size $m = 250$ with IMQ kernels. The RMSE across a variety of test functions is reported, with the IQR highlighted for each method. Outliers are calculated as being above $Q_3 + 1.5$IQR and below $Q_1 - 1.5$IQR.

**Wall-Clock Time**: In Section D.2 we derive the computational and storage complexity for each of the methods introduced in this work. In Figure 22 and Table 1 we report the wall-clock time for the *Heteroscedastic* data experiment. Despite JKIP and JKH having the same time complexity, we see that JKIP is significantly faster. As noted in Section D.2.5, the algorithms in this paper were implemented using JAX. JAX enables Just-In-Time (JIT) compilation, which significantly increases execution speed. However, to achieve this speed, JAX relies on an immutable array structure, meaning the arrays must not change shape during program execution. As a result, JKH and ACKH cannot fully leverage the speed benefits of JIT compilation, as the size of the compressed set, and hence the corresponding arrays, increases at every iteration. Conversely, the arrays considered in JKIP and

ACKIP stay the same shape throughout. This presents a notable practical advantage of JKIP and ACKIP over JKH and ACKH.

| Method | Wall Clock Time (s) |
|--------|---------------------|
| JKH    | $9.47 \pm 2.37$     |
| JKIP   | $0.84 \pm 0.081$    |
| ACKH   | $318.81 \pm 40.17$  |
| ACKIP  | $11.42 \pm 0.04$    |

Table 1: Wall clock times (in seconds) for each method over twenty runs, mean and standard deviation reported.

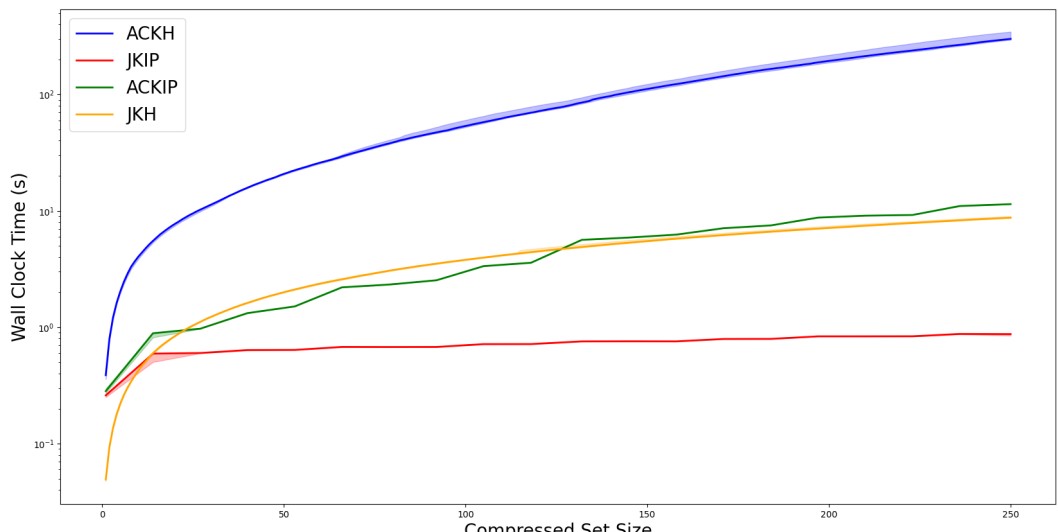

Figure 22: Timings for *Heteroscedastic* data.

### C.1.3 Matching the Empirical Conditional Distribution - Discrete / Classification

For classification tasks with $C$ possible classes, we replace the Gaussian kernel on the responses with the indicator kernel $l : \mathbb{N}^C \times \mathbb{N}^C \to [0, 1]$ defined by

$$l(\boldsymbol{y}, \boldsymbol{y}') := \begin{cases} 1 & \text{if } \boldsymbol{y} = \boldsymbol{y}' \\ 0 & \text{otherwise} \end{cases}$$

where $\mathbb{N}^C := [0, 1, \ldots, C]$. In this case, standard gradient descent on the responses is no longer possible. Moreover, solving the optimisation problems of JKH, JKIP, ACKH and ACKIP now constitute a mixed-integer programming problem, which is known to be NP-complete. Various heuristic approaches exist for problems of this type, such as relaxation-based methods (e.g., continuous relaxations followed by rounding), greedy algorithms, and metaheuristic strategies like genetic algorithms or simulated annealing. We develop a simple two-step optimisation procedure based on exhaustive search, leaving investigating the above techniques in the context of our algorithms for future work.

For JKH and ACKH, we alternate between performing a gradient step on the feature and selecting the optimal response class via exhaustive search. For JKIP and ACKIP, after each gradient step on $\tilde{\boldsymbol{X}}$, we iterate over the responses $\tilde{\boldsymbol{y}}$ in $\tilde{\boldsymbol{Y}}$, updating each in turn with the optimal class by exhaustive search, carrying forward these selections. It is important to note that one can reduce the cost of evaluating the JKIP and ACKIP objective functions significantly when $\tilde{\boldsymbol{X}}$ is fixed; see Section B.11 for a derivation of the relevant objectives and Algorithms 7 and 8 for the pseudocode.

For classification tasks with $C$ classes, we report the overall classification accuracy and F1 scores, as well as the RMSE for the indicator functions $h_i(\boldsymbol{y}) = \mathbf{1}_{\{\boldsymbol{y}=i\}}$, where $i = 1, \ldots, C$. As shown in [33], the KCME naturally functions as a multiclass classification model, since

$$\mathbb{E}[h(Y) \mid X = \boldsymbol{x}] = \mathbb{E}[\mathbf{1}_{\{Y=i\}} \mid X = \boldsymbol{x}] = \mathbb{P}(Y = i \mid X = \boldsymbol{x}),$$

which expresses the class probabilities given $X$. Moreover, the empirical decision probabilities are guaranteed to converge to the true population probabilities (Theorem 1, [33]), unlike, for example in multi-class SVCs and GPCs. However, in the finite case, the predicted probabilities are not guaranteed to lie in the range $[0, 1]$, nor form a normalised distribution. In order to produce a valid distribution, we clip-normalise the estimates (Equation 6, [33]).

**Synthetic**: We generate an unbalanced 4-class dataset using the multinomial logistic regression model, where conditional class probabilities are given by $\mathbb{P}(Y = 0 \mid X = \boldsymbol{x}) = \frac{1}{1+\sum_{j=1}^{3} \exp(\boldsymbol{\beta}_j \cdot \boldsymbol{x})}$ and $\mathbb{P}(Y = k \mid X = \boldsymbol{x}) = \frac{\exp(\boldsymbol{\beta}_k \cdot \boldsymbol{x})}{1+\sum_{j=1}^{3} \exp(\boldsymbol{\beta}_j \cdot \boldsymbol{x})}$, $1 \leq k \leq 3$, and $\mathbb{P}_X$ is a 2D Gaussian mixture model with 100 components. We assign classes using $\boldsymbol{\beta} := \begin{bmatrix} 10 & 8 & 1 & 45 \\ 40 & 45 & 40 & 10 \end{bmatrix}^{\top}$ and, to ensure class overlap, introduce additive noise $\epsilon \sim \mathcal{N}(0, 100)$ to the exponential. This setup results in class proportions of approximately 32% (class 0), 12% (class 1), 19% (class 2), and 37% (class 3).

Figures 23 and 24 show that ACKIP achieves clearly superior performance versus ACKH, JKH and JKIP, both in predicting the class probabilities, as well as in overall accuracy and F1 score, achieving parity with the full data at only 3% of the size.

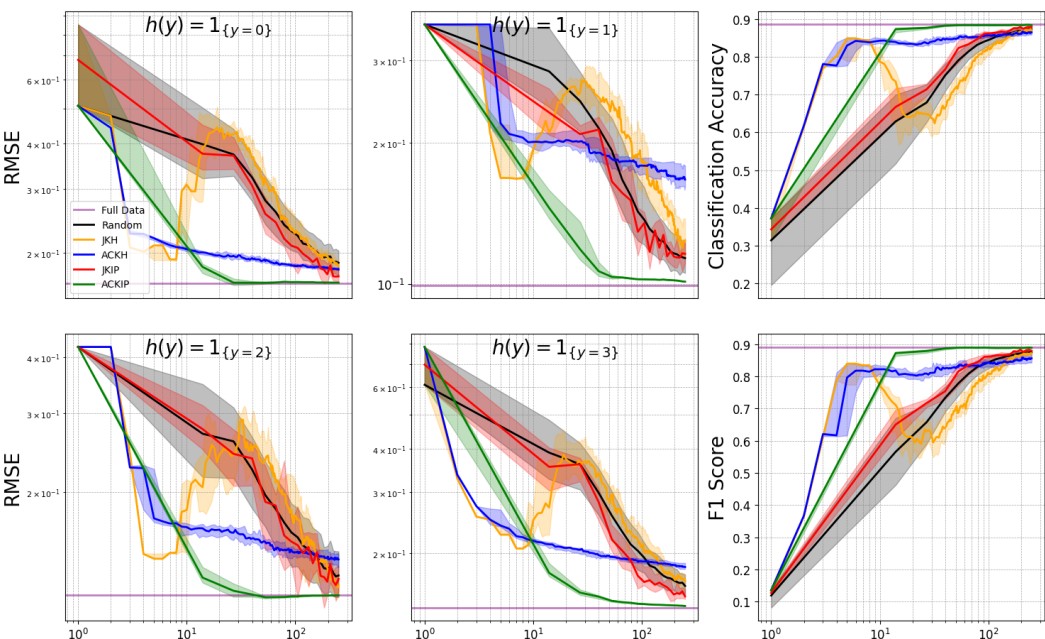

Figure 23: Results for *Imbalanced* dataset. RMSE is reported across a variety of test functions, as the size of the compressed set increases. For JKH (orange), JKIP (red), ACKH (blue), and ACKIP (green), we display the median performance (bold line) with the 25th-75th percentiles (shaded region) over 20 runs. The error of random sampling (black) over 500 runs is also plotted for comparison, as well as the performance of the full data (purple).

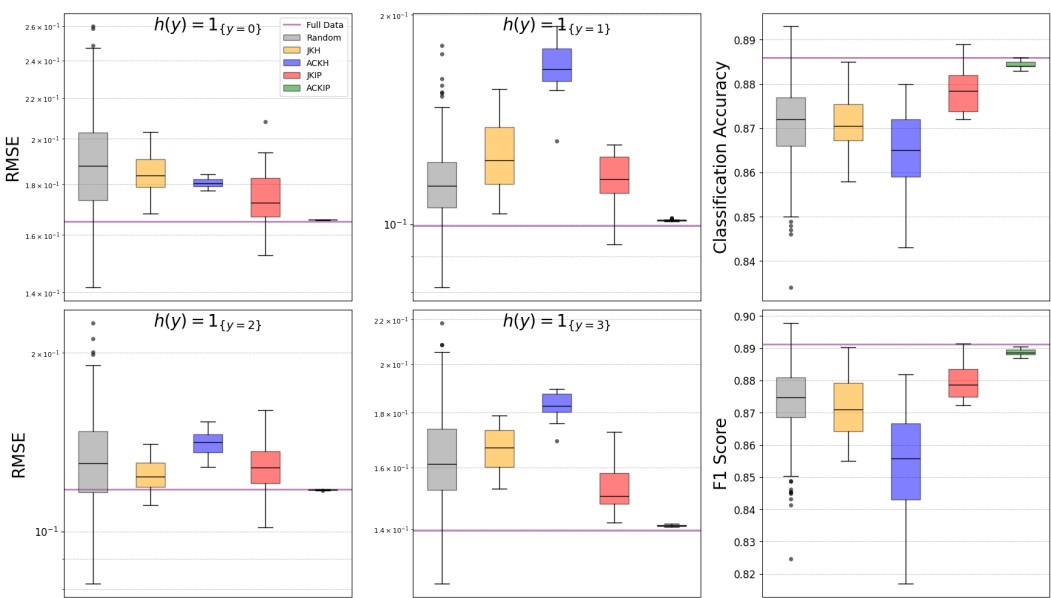

Figure 24: Results for *Imbalanced* dataset for compressed sets of size $m = 250$. The RMSE across a variety of test functions is reported, with the IQR highlighted for each method. Outliers are calculated as being above $Q_3 + 1.5\text{IQR}$ and below $Q_1 - 1.5\text{IQR}$.

In Figure 25 we see that ACKIP achieves by far the best AMCMD, noting that the final value achieved is effectively zero. Due to floating point errors in the computation of $\text{AMCMD}^2 \left[ \hat{\mathbb{P}}_X, \hat{\mathbb{P}}_{Y|X}, \tilde{\mathbb{P}}_{Y|X} \right]$, it can become slightly negative, resulting in the line leaving the log-log plot. We also see how the herding optimisation approach hinders ACKH as it initially matches ACKIP, but ends up performing worse than random. Finally, we note that JKIP outperforms JKH, which only matches random.

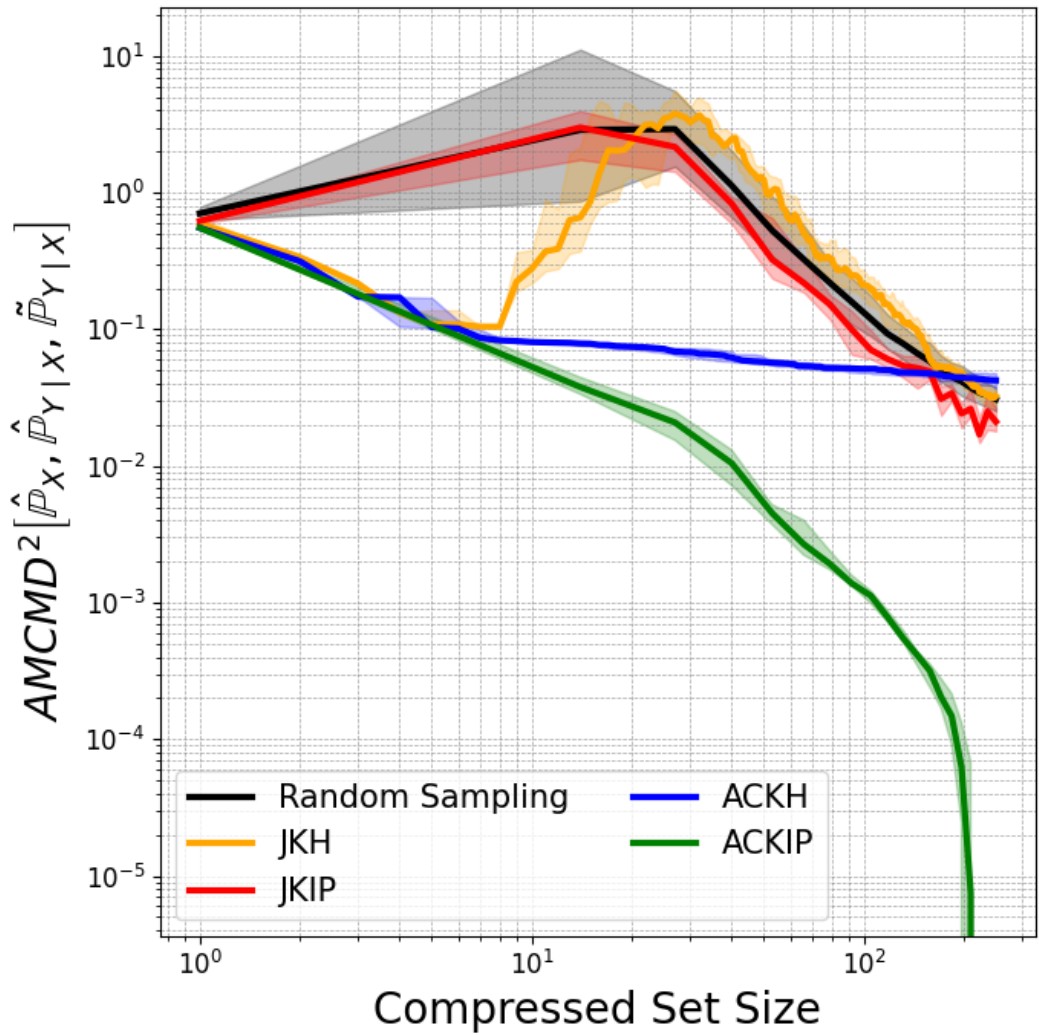

Figure 25: $\text{AMCMD}^2 \left[ \hat{\mathbb{P}}_X, \hat{\mathbb{P}}_{Y|X}, \tilde{\mathbb{P}}_{Y|X} \right]$ achieved by each method as a function of the size of the compressed sets constructed by JKH (orange), ACKH (blue), JKIP (red), and ACKIP (green), on the *Imbalanced* dataset. We display the median performance (bold line) with the 25th-75th percentiles (shaded region) over 20 runs. The error of random sampling (black) over 500 runs is also plotted for comparison.

For completeness, in Figures 26 and 27 we also include an example of the compressed set constructed by each method, as well as the corresponding decision boundary.

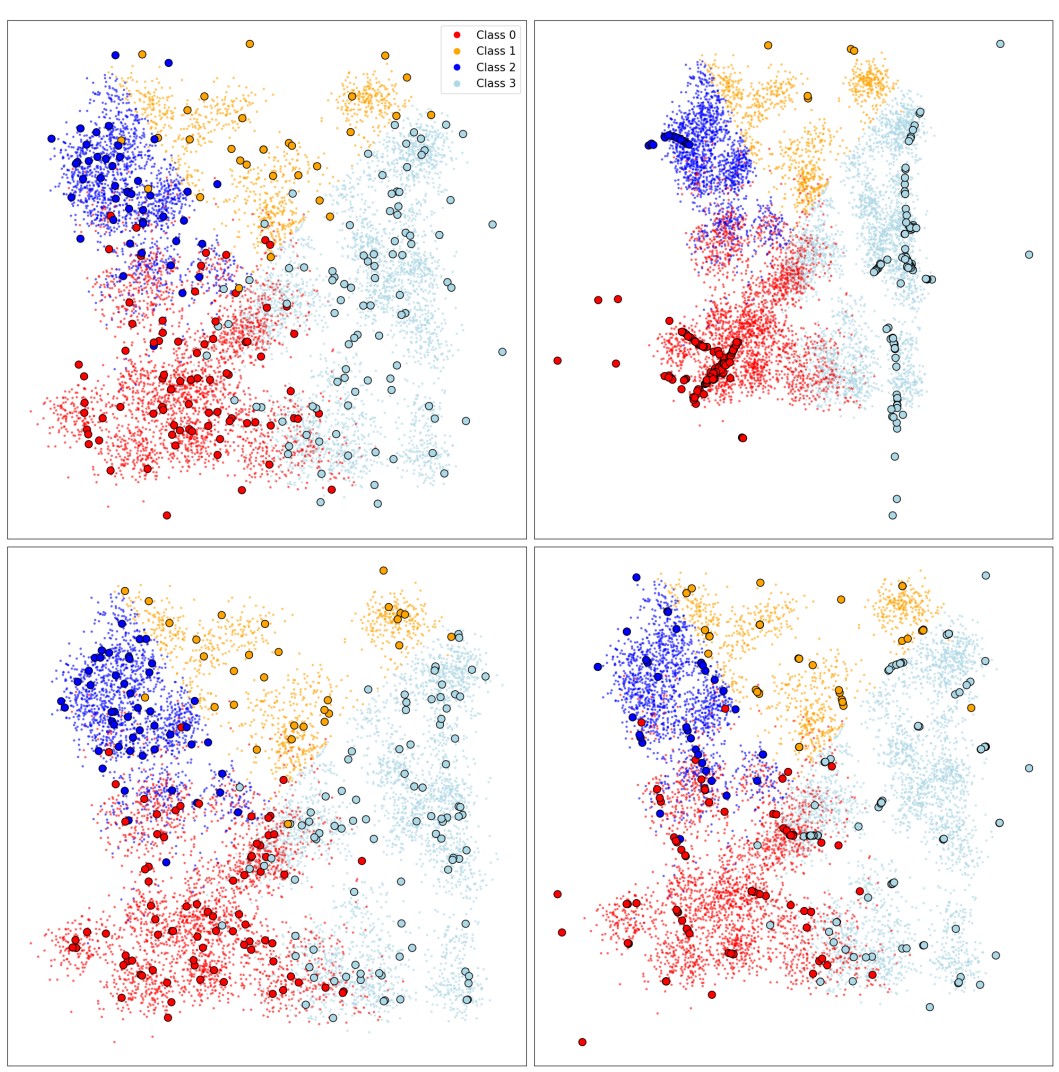

Figure 26: Compressed sets of size $m = 250$ constructed by JKH (top left), ACKH (top right), JKIP (bottom left), and ACKIP (bottom right), on multi-class classification data.

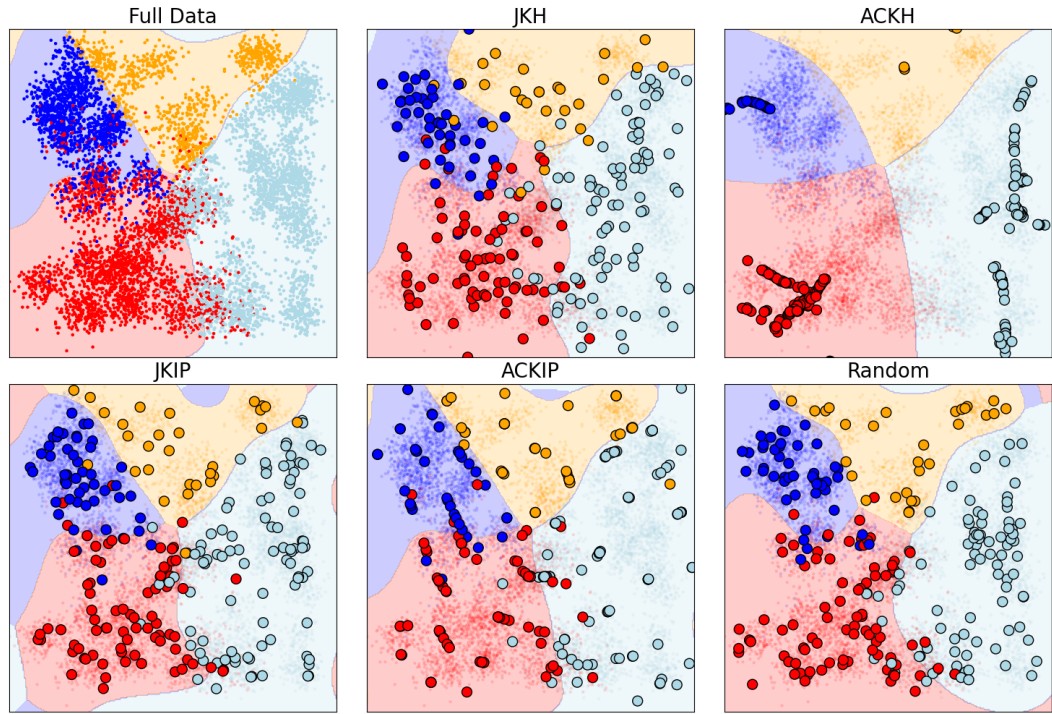

Figure 27: Decision boundaries of the KCME model estimated using compressed sets of size $M = 250$, constructed by JKH, ACKH, JKIP, and ACKIP, on the *Imbalanced* dataset. For comparison, we also include the decision boundaries of the full-data model and a model trained on a uniformly random subset.

**Real**: We use *MNIST* [83, 84], where we subsample down to $n = 10,000$ due to memory limitations, splitting off $10\%$ for validation and another $10\%$ for testing. Figure 28 shows that ACKIP achieves the lowest AMCMD, with JKIP doing second best. In Figures 29 and 30 we see that this translates to improved performance in estimating conditional expectations, with ACKIP achieving vastly superior performance versus the other methods, with similar classification accuracy and F1 score to the full data model, with just $3\%$ of the data.

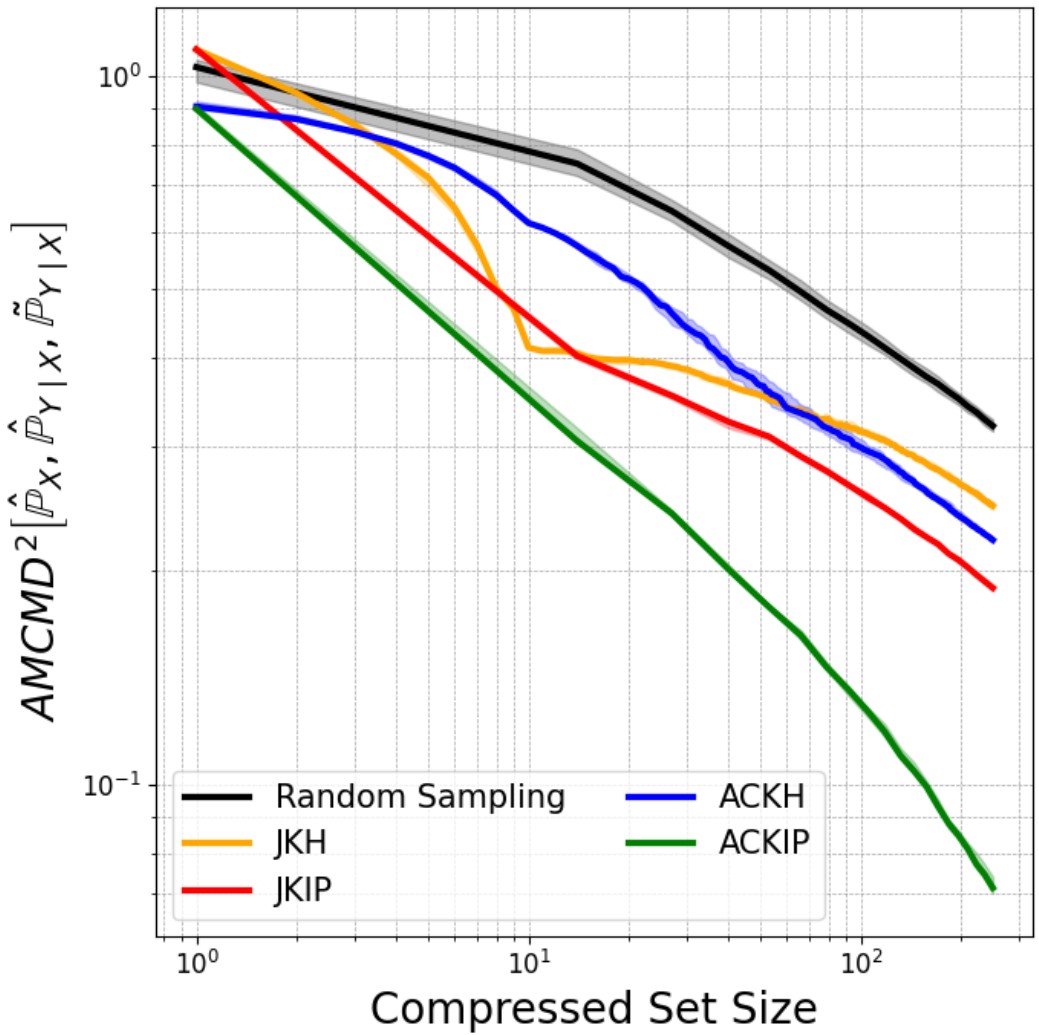

Figure 28: Results for the *MNIST* dataset. We show the AMCMD$^2\left[\hat{\mathbb{P}}_X, \hat{\mathbb{P}}_{Y|X}, \tilde{\mathbb{P}}_{Y|X}\right]$ achieved by each method as a function of the size of the compressed sets constructed by JKH (orange), ACKH (blue), JKIP (red), and ACKIP (green), on the MNIST data. We display the median performance (bold line) with the 25th-75th percentiles (shaded region) over 20 runs. The error of random sampling (black) over 500 runs is also plotted for comparison.

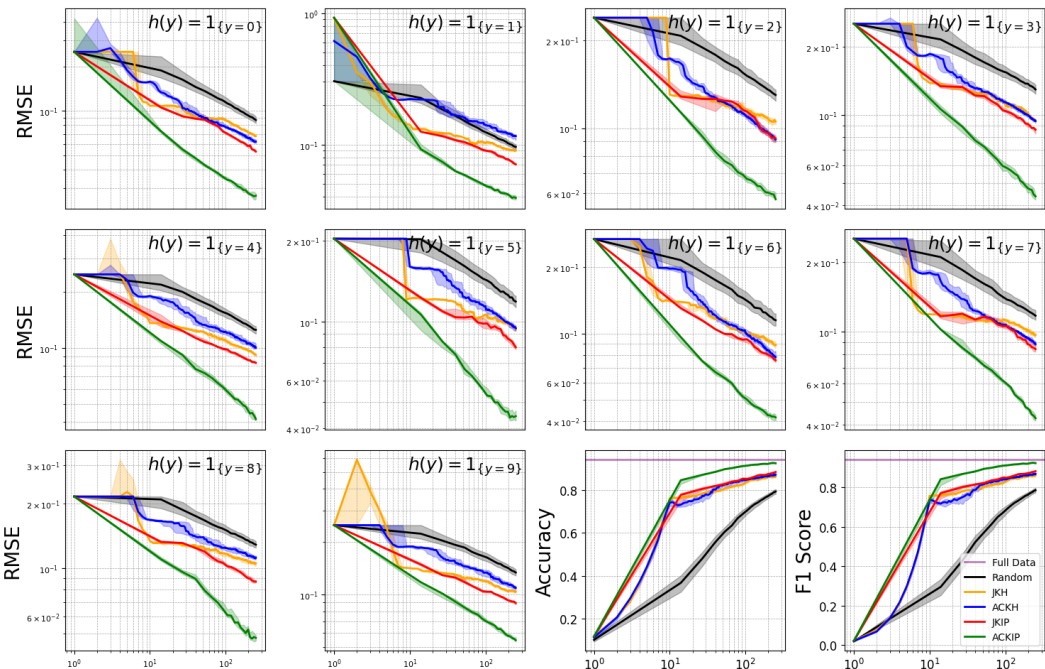

Figure 29: Results for *MNIST* data; the RMSE is calculated against the full data estimates of $\mathbb{E}[h(Y) \mid X = \boldsymbol{x}_i]$ as the true values are not available. RMSE is reported across a variety of test functions, as the size of the compressed set increases. We also report the overall classification accuracy and F1 score, comparing against the full data performance. For JKH (orange), JKIP (red), ACKH (blue), and ACKIP (green), we display the median performance (bold line) with the 25th-75th percentiles (shaded region) over 20 runs. The error of random sampling (black) over 500 runs is also plotted for comparison, as well as the performance of the full data (purple) for classification accuracy and F1 score.

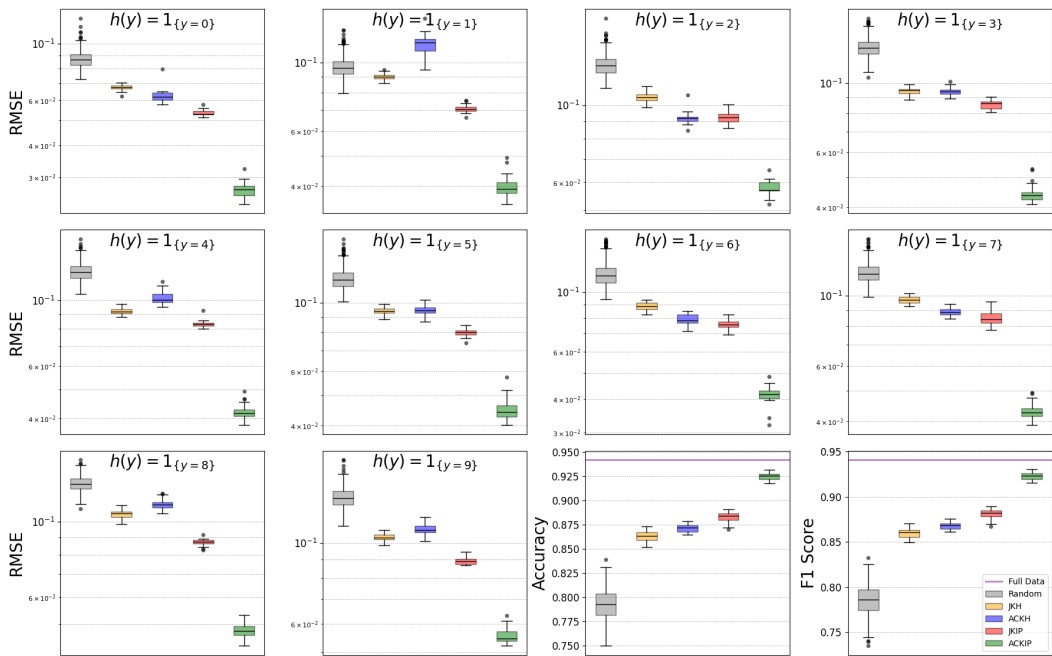

Figure 30: Results for *MNIST* data for compressed sets of size $m = 250$; the RMSE is calculated against the full data estimates of $\mathbb{E}[h(Y) \mid X = \boldsymbol{x}_i]$ as the true values are not available. The RMSE across a variety of test functions is reported, with the IQR highlighted for each method. Outliers are calculated as being above $Q_3 + 1.5\text{IQR}$ and below $Q_1 - 1.5\text{IQR}$.

## C.2 Flexibility of AMCMD versus KCD/AMMD

In this section we demonstrate how the increased flexibility of the AMCMD allows for application to tasks that AMMD/KCD are not suitable for.

Let $\mathbb{P}_X := \mathcal{N}(\mu, \sigma^2)$, $\mathbb{P}_{X'} := \mathcal{N}(-\mu, \sigma^2)$, and $\mathbb{P}_{X^*} := \mathcal{N}(0, \sigma_*^2)$ with $\mu, \sigma^2, \sigma_*^2$ chosen such that the Radon-Nikodym derivatives $\frac{d\mathbb{P}_{X^*}}{d\mathbb{P}_X}, \frac{d\mathbb{P}_{X^*}}{d\mathbb{P}_{X'}}$ are bounded. These three distributions are also clearly absolutely continuous with respect to each other, hence the conditions on the distributions in Theorem 4.1 are satisfied. Importantly, we have $\mathbb{P}_X \neq \mathbb{P}_{X'} \neq \mathbb{P}_{X^*}$, and thus the AMMD/KCD is not defined for this setup. Now, let $f_a : \mathbb{R} \to \mathbb{R}$ be a function with

$$
f_a(\boldsymbol{x}) = \begin{cases} -a + (\boldsymbol{x} + a)^2 & \text{if } \boldsymbol{x} < -a \\ x & \text{if } -a \leq \boldsymbol{x} \leq a \\ a - (\boldsymbol{x} - a)^2 & \text{if } \boldsymbol{x} > a \end{cases} ,
$$

for $a \in \mathbb{R}$, and let $\mathbb{P}_{Y|X=\boldsymbol{x}} := \mathcal{N}(\boldsymbol{x}, \sigma_\epsilon^2)$, and $\mathbb{P}_{Y'|X'=\boldsymbol{x}} := \mathcal{N}(f(\boldsymbol{x}), \sigma_\epsilon^2)$. Then, we have that $\mathbb{P}_{Y|X=\boldsymbol{x}} = \mathbb{P}_{Y'|X'=\boldsymbol{x}}$ for all $x \in [-a, a]$ and $\mathbb{P}_{Y|X=\boldsymbol{x}} \neq \mathbb{P}_{Y'|X'=\boldsymbol{x}}$ for all $x \notin [-a, a]$. The AMCMD allows us to detect this change in behaviour over regions by changing the location of the weighting distribution $\mathbb{P}_{X^*}$; see Figure 31. In fact, using Lemma 4.3, we estimate the AMCMD in Figure 31 to be approximately equal to 1e-2.

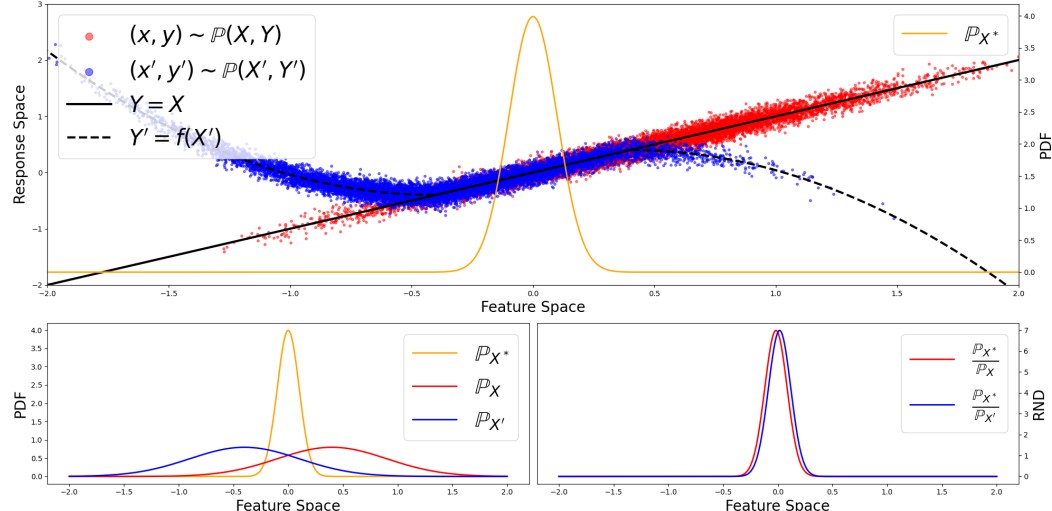

Figure 31: The top plot illustrates the data space, with $a = \mu = 0.4, \sigma^2 = 0.5$, and $\sigma_*^2 = 0.1$. Here, pairs sampled from $\mathbb{P}_{X,Y}$ (red) exhibit the same linear relationship as pairs from $\mathbb{P}_{X',Y'}$ (blue) around zero, where the density of the weighting distribution $\mathbb{P}_{X^*}$ is concentrated. Away from zero, the relationships diverge, and $\mathbb{P}_{X^*}$ is chosen to have little mass in these regions. The bottom-left plot shows the probability density functions of $\mathbb{P}_X$ (red), $\mathbb{P}_{X'}$ (blue), and $\mathbb{P}_{X^*}$ (orange). The bottom-right plot displays the Radon-Nikodym derivatives $\frac{d\mathbb{P}_{X^*}}{d\mathbb{P}_X}$ (red) and $\frac{d\mathbb{P}_{X^*}}{d\mathbb{P}_{X'}}$ (blue), which are clearly bounded in this case.

In contrast, the relative inflexibility of the KCD/AMMD would mean one would not be able to detect over which regions of the conditioning space the conditional distributions are equal; see Figure 32 for an illustration of this. Using Lemma 4.3, we estimate the AMCMD to be approximately 0.5.

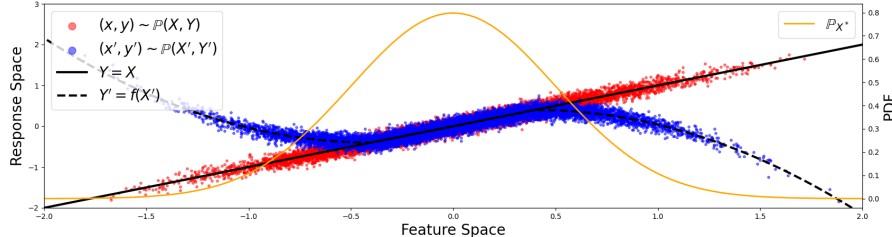

Figure 32: The data space where $\mathbb{P}_X = \mathbb{P}_{X'} = \mathbb{P}_{X^*} = \mathcal{N}(0, \sigma^2)$, with $\sigma^2 = 0.5$, $a = 0.4$.

## C.3 Targeting a Family of Conditional Distributions Exactly

In order to construct a compressed representation that *exactly* targets a family of conditional distributions $\mathbb{P}_{Y|X}$, we require access to analytical expressions for the expectations

$$\mathbb{E}_{\boldsymbol{x} \sim \mathbb{P}_X} \left[ k(\boldsymbol{x}, \boldsymbol{x}')k(\boldsymbol{x}, \boldsymbol{x}'') \right] \quad \text{and} \quad \mathbb{E}_{(\boldsymbol{x},\boldsymbol{y}) \sim \mathbb{P}_{X,Y}} \left[ k(\boldsymbol{x}, \boldsymbol{x}')l(\boldsymbol{y}, \boldsymbol{y}') \right] \tag{33}$$

for arbitrary $\boldsymbol{x}', \boldsymbol{x}'' \in \mathcal{X}$ and $\boldsymbol{y}' \in \mathcal{Y}$. Furthermore, in order to compute the exact AMCMD between the true family of conditional distributions and the family of conditional distributions generated by the compressed set, we must also be able to evaluate

$$\mathbb{E}_{\boldsymbol{x} \sim \mathbb{P}_X} [\|\mu_{Y|X=\boldsymbol{x}}\|^2_{\mathcal{H}_l}]. \tag{34}$$

In general we cannot to exactly evaluate the expectations in (33) and (34), however it is possible by restricting our attention to specific choices of the kernel functions $k : \mathcal{X} \times \mathcal{X} \to \mathbb{R}$ and $l : \mathcal{Y} \times \mathcal{Y} \to \mathbb{R}$,

and a specific data generation process. In particular, we set

$$k(\boldsymbol{x}, \boldsymbol{x}') := \exp\left(-\frac{1}{2\alpha_k^2}(\boldsymbol{x} - \boldsymbol{x}')^2\right), \quad l(\boldsymbol{y}, \boldsymbol{y}') := \exp\left(-\frac{1}{2\alpha_l^2}(\boldsymbol{y} - \boldsymbol{y}')^2\right)$$

i.e. Gaussian kernels, with $\alpha_k, \alpha_l \in \mathbb{R}_{>0}$. Moreover, we let $\mathbb{P}_X = \mathcal{N}(\mu, \sigma^2)$, and given coefficients $a_0, a_1 \in \mathbb{R}$ we let $\boldsymbol{y} = a_0 + a_1\boldsymbol{x} + \epsilon$ where $\epsilon \sim \mathcal{N}(0, \sigma_\epsilon^2)$, i.e. $\mathbb{P}_{Y|X=\boldsymbol{x}} = \mathcal{N}(a_0 + a_1\boldsymbol{x}, \sigma_\epsilon^2)$.

### C.3.1 Deriving the Marginal Expectation

In this section, we derive the marginal expectation in (33), under the conditions on the kernel and data-generating process previously laid out:

$$\mathbb{E}_{\boldsymbol{x} \sim \mathbb{P}_X}\left[k(\boldsymbol{x}, \boldsymbol{x}')k(\boldsymbol{x}, \boldsymbol{x}'')\right] = \int_{\mathcal{X}} k(\boldsymbol{x}, \boldsymbol{x}')k(\boldsymbol{x}, \boldsymbol{x}'')\mathbb{P}_X(\mathrm{d}\boldsymbol{x})$$

$$= \frac{1}{\sqrt{2\pi\sigma^2}}\int_{\mathcal{X}} \exp\left(-\frac{1}{2\alpha_k^2}[\boldsymbol{x} - \boldsymbol{x}']^2 - \frac{1}{2\alpha_k^2}[\boldsymbol{x} - \boldsymbol{x}'']^2 - \frac{1}{2\sigma^2}[\boldsymbol{x} - \mu]^2\right)\mathrm{d}\boldsymbol{x}.$$

Now,

$$-\frac{1}{2\alpha_k^2}(\boldsymbol{x} - \boldsymbol{x}')^2 - \frac{1}{2\alpha_k^2}(\boldsymbol{x} - \boldsymbol{x}'')^2 - \frac{1}{2\sigma^2}(\boldsymbol{x} - \mu)^2$$

$$= -\frac{1}{2\alpha_k^2}\left(\boldsymbol{x}^2 - 2\boldsymbol{x}\boldsymbol{x}' + \boldsymbol{x}'^2\right) - \frac{1}{2\alpha_k^2}\left(\boldsymbol{x}^2 - 2\boldsymbol{x}\boldsymbol{x}'' + \boldsymbol{x}''^2\right) - \frac{1}{2\sigma^2}\left(\boldsymbol{x}^2 - 2\boldsymbol{x}\mu + \mu^2\right)$$

$$= -\frac{\boldsymbol{x}^2}{2}\left(\frac{1}{\sigma^2} + \frac{2}{\alpha_k^2}\right) + \boldsymbol{x}\left(\frac{\mu}{\sigma^2} + \frac{(\boldsymbol{x}' + \boldsymbol{x}'')}{\alpha_k^2}\right) - \frac{\mu^2}{2\sigma^2} - \frac{(\boldsymbol{x}'^2 + \boldsymbol{x}''^2)}{2\alpha_k^2}$$

$$= -\frac{A}{2}\boldsymbol{x}^2 + B\boldsymbol{x} - \frac{\mu^2}{2\sigma^2} - \frac{(\boldsymbol{x}'^2 + \boldsymbol{x}''^2)}{2\alpha_k^2}$$

where $A := \left(\frac{1}{\sigma^2} + \frac{2}{\alpha_k^2}\right), B := \left(\frac{\mu}{\sigma^2} + \frac{(\boldsymbol{x}'+\boldsymbol{x}'')}{\alpha_k^2}\right)$. Completing the square, we have

$$-\frac{A}{2}\boldsymbol{x}^2 + B\boldsymbol{x} = -\frac{A}{2}\left(\boldsymbol{x}^2 - \frac{2B}{A}\boldsymbol{x}\right) = -\frac{A}{2}\left[\left(\boldsymbol{x} - \frac{B}{A}\right)^2 - \left(\frac{B}{A}\right)^2\right],$$

and therefore we can write that,

$$\mathbb{E}_{\boldsymbol{x} \sim \mathbb{P}_X}\left[k(\boldsymbol{x}, \boldsymbol{x}')k(\boldsymbol{x}, \boldsymbol{x}'')\right]$$

$$= \frac{1}{\sqrt{2\pi\sigma^2}}\int_{\mathcal{X}} \exp\left(-\frac{1}{2\alpha_k^2}(\boldsymbol{x} - \boldsymbol{x}')^2 - \frac{1}{2\alpha_k^2}(\boldsymbol{x} - \boldsymbol{x}'')^2 - \frac{1}{2\sigma^2}(\boldsymbol{x} - \mu)^2\right)\mathrm{d}\boldsymbol{x}$$

$$= \frac{1}{\sqrt{2\pi\sigma^2}}\int_{\mathcal{X}} \exp\left(\frac{A}{2}\left[\left(\boldsymbol{x} - \frac{B}{A}\right)^2 - \left(\frac{B}{A}\right)^2\right] - \frac{\mu^2}{2\sigma^2} - \frac{(\boldsymbol{x}'^2 + \boldsymbol{x}''^2)}{2\alpha_k^2}\right)\mathrm{d}\boldsymbol{x}$$

$$= \frac{1}{\sqrt{2\pi\sigma^2}}\exp\left(\frac{A}{2}\left(\frac{B}{A}\right)^2 - \frac{\mu^2}{2\sigma^2} - \frac{(\boldsymbol{x}'^2 + \boldsymbol{x}''^2)}{2\alpha_k^2}\right)\int_{\mathcal{X}} \exp\left(-\frac{A}{2}\left(\boldsymbol{x} - \frac{B}{A}\right)^2\right)\mathrm{d}\boldsymbol{x}$$

$$= \frac{1}{\sqrt{A\sigma^2}}\exp\left(\frac{A}{2}\left(\frac{B}{A}\right)^2 - \frac{\mu^2}{2\sigma^2} - \frac{(\boldsymbol{x}'^2 + \boldsymbol{x}''^2)}{2\alpha_k^2}\right).$$

### C.3.2 Deriving the Joint Expectation

In this section, we derive the joint expectation in (33), under the conditions on the kernels and data-generating process previously laid out. We first derive the joint distribution.

We have that

$$f_{X,Y}(\boldsymbol{x}, \boldsymbol{y}) = f_X(\boldsymbol{x})f_{Y|X=\boldsymbol{x}}(\boldsymbol{y}) = \frac{1}{2\pi\sigma\sigma_\epsilon}\exp\left(-\frac{1}{2\sigma^2}(\boldsymbol{x} - \mu)^2 - \frac{1}{2\sigma_\epsilon^2}(\boldsymbol{y} - (a_0 + a_1\boldsymbol{x}))^2\right).$$

where using

$$\mathbb{E}[X] = \mu, \ \ \mathbb{E}[Y] = \mathbb{E}[a_0 + a_1 X] = a_0 + a_1 \mu,$$
$$\mathrm{Var}(X) = \sigma^2, \ \ \mathrm{Var}(Y \mid X) = \sigma_\epsilon^2,$$
$$\mathrm{Var}(Y) = \mathbb{E}[\mathrm{Var}(Y \mid X)] + \mathrm{Var}(\mathbb{E}[Y \mid X]) = \sigma_\epsilon^2 + a_1^2 \sigma^2,$$
$$\mathbb{C}\mathrm{ov}(X, Y) = \mathbb{C}\mathrm{ov}(X, a_0 + a_1 X) = a_1 \mathrm{Var}(X) = a_1 \sigma^2,$$

we notice that $(X, Y) \sim \mathcal{N}\left( \begin{pmatrix} \mu \\ a_0 + a_1 \mu \end{pmatrix}, \begin{pmatrix} \sigma^2 & a_1 \sigma^2 \\ a_1 \sigma^2 & a_1^2 \sigma^2 + \sigma_\epsilon^2 \end{pmatrix} \right)$.

Now, we want to derive the expectation $\mathbb{E}_{(\boldsymbol{x}, \boldsymbol{y}) \sim \mathbb{P}_{X,Y}} [k(\boldsymbol{x}, \boldsymbol{x}') l(\boldsymbol{y}, \boldsymbol{y}')]$ for arbitrary $\boldsymbol{x}' \in \mathcal{X}$ and $\boldsymbol{y}' \in \mathcal{Y}$:

$$\mathbb{E}_{(\boldsymbol{x}, \boldsymbol{y}) \sim \mathbb{P}_{X,Y}} [k(\boldsymbol{x}, \boldsymbol{x}') l(\boldsymbol{y}, \boldsymbol{y}')] = \int_{\mathbb{R}} \int_{\mathbb{R}} k(\boldsymbol{x}, \boldsymbol{x}') l(\boldsymbol{y}, \boldsymbol{y}') f_{X,Y}(\boldsymbol{x}, \boldsymbol{y}) \mathrm{d}\boldsymbol{x} \mathrm{d}\boldsymbol{y}$$

where $k(\cdot, \cdot)$ and $l(\cdot, \cdot)$ are both Gaussian kernels. We need this integral to end up in the form

$$\int_{\mathbb{R}^2} \exp\left( -\frac{1}{2} \boldsymbol{\omega} A \boldsymbol{\omega} + \boldsymbol{b}^\top \boldsymbol{\omega} + c \right) \mathrm{d}\boldsymbol{\omega}$$

for $\boldsymbol{\omega} := (\boldsymbol{x}, \boldsymbol{y})^\top$ as by completing the square, it can be shown [85] that

$$\int_{\mathbb{R}^2} \exp\left( -\frac{1}{2} \boldsymbol{\omega}^\top A \boldsymbol{\omega} + \boldsymbol{b}^\top \boldsymbol{\omega} + c \right) \mathrm{d}\boldsymbol{\omega} = \frac{2\pi}{|A|^{\frac{1}{2}}} \exp\left( c + \frac{1}{2} \boldsymbol{b}^\top A^{-1} \boldsymbol{b} \right).$$

Let us first interrogate the product $k(\boldsymbol{x}, \boldsymbol{x}') l(\boldsymbol{y}, \boldsymbol{y}')$:

$$k(\boldsymbol{x}, \boldsymbol{x}') l(\boldsymbol{y}, \boldsymbol{y}')$$
$$= \exp\left( -\frac{1}{2\alpha_k^2} (\boldsymbol{x} - \boldsymbol{x}')^2 - \frac{1}{2\alpha_l^2} (\boldsymbol{y} - \boldsymbol{y}')^2 \right)$$
$$= \exp\left( -\frac{1}{2\alpha_k^2 \alpha_l^2} \left[ \alpha_l^2 (\boldsymbol{x} - \boldsymbol{x}')^2 + \alpha_k^2 (\boldsymbol{y} - \boldsymbol{y}')^2 \right] \right)$$
$$= \exp\left( -\frac{1}{2\alpha_k^2 \alpha_l^2} \left[ \alpha_l^2 (\boldsymbol{x}^2 - 2\boldsymbol{x}' \boldsymbol{x} + \boldsymbol{x}'^2) + \alpha_k^2 (\boldsymbol{y}^2 - 2\boldsymbol{y}' \boldsymbol{y} + \boldsymbol{y}'^2) \right] \right)$$
$$= \exp\left( -\frac{1}{2\alpha_k^2 \alpha_l^2} \left[ \begin{pmatrix} \boldsymbol{x} \\ \boldsymbol{y} \end{pmatrix}^\top \begin{pmatrix} \alpha_l^2 & 0 \\ 0 & \alpha_k^2 \end{pmatrix} \begin{pmatrix} \boldsymbol{x} \\ \boldsymbol{y} \end{pmatrix} + \begin{pmatrix} -2\lambda_l^2 \boldsymbol{x}' \\ -2\alpha_k^2 \boldsymbol{y}' \end{pmatrix}^\top \begin{pmatrix} \boldsymbol{x} \\ \boldsymbol{y} \end{pmatrix} + \lambda_l^2 \boldsymbol{x}'^2 + \alpha_k^2 \boldsymbol{y}'^2 \right] \right)$$
$$= \exp\left( -\frac{1}{2} \boldsymbol{\omega}^\top \begin{pmatrix} \frac{1}{\alpha_k^2} & 0 \\ 0 & \frac{1}{\alpha_l^2} \end{pmatrix} \boldsymbol{\omega} + \begin{pmatrix} \boldsymbol{x}'/\alpha_k^2 \\ \boldsymbol{y}'/\alpha_l^2 \end{pmatrix}^\top \boldsymbol{\omega} - \frac{\boldsymbol{x}'^2}{2\alpha_k^2} - \frac{\boldsymbol{y}'^2}{2\alpha_l^2} \right)$$
$$= \exp\left( -\frac{1}{2} \boldsymbol{\omega}^\top A_1 \boldsymbol{\omega} + \boldsymbol{b}_1^\top \boldsymbol{\omega} + c_1 \right)$$

where $A_1 := \begin{pmatrix} \frac{1}{\alpha_k^2} & 0 \\ 0 & \frac{1}{\alpha_l^2} \end{pmatrix}$, $\boldsymbol{b}_1 := \begin{pmatrix} \boldsymbol{x}'/\alpha_k^2 \\ \boldsymbol{y}'/\alpha_l^2 \end{pmatrix}^\top$ and $c_1 := -\frac{\boldsymbol{x}'^2}{2\alpha_k^2} - \frac{\boldsymbol{y}'^2}{2\alpha_l^2}$. Now, we need to write $f_{X,Y}(\boldsymbol{x}, y)$ in the same form:

$$f_{X,Y}(\boldsymbol{x}, \boldsymbol{y}) = f_X(\boldsymbol{x}) f_{Y|X=\boldsymbol{x}}(\boldsymbol{y}) = \frac{1}{2\pi\sigma\sigma_\epsilon} \exp\left( -\frac{1}{2\sigma^2} (\boldsymbol{x} - \mu)^2 - \frac{1}{2\sigma_\epsilon^2} (\boldsymbol{y} - (a_0 + a_1 \boldsymbol{x}))^2 \right)$$
$$= \frac{1}{2\pi\sigma\sigma_\epsilon} \exp\left( -\frac{1}{2\sigma^2 \sigma_\epsilon^2} \left[ \sigma_\epsilon^2 (\boldsymbol{x} - \mu)^2 + \sigma^2 (\boldsymbol{y} - (a_0 + a_1 \boldsymbol{x}))^2 \right] \right).$$

Now,

$$\sigma_\epsilon^2(\boldsymbol{x} - \mu)^2 + \sigma^2(\boldsymbol{y} - (a_0 + a_1\boldsymbol{x}))^2$$
$$= \sigma_\epsilon^2(\boldsymbol{x}^2 - 2\mu\boldsymbol{x} + \mu^2) + \sigma^2(\boldsymbol{y}^2 - 2\boldsymbol{y}(a_0 + a_1\boldsymbol{x}) + (a_0 + a_1\boldsymbol{x})^2)$$
$$= \sigma_\epsilon^2(\boldsymbol{x}^2 - 2\mu\boldsymbol{x} + \mu^2) + \sigma^2(\boldsymbol{y}^2 - 2a_0\boldsymbol{y} - 2a_1\boldsymbol{x}\boldsymbol{y} + a_0^2 + 2a_0a_1\boldsymbol{x} + a_1^2\boldsymbol{x}^2))$$
$$= \boldsymbol{x}^2(\sigma_\epsilon^2 + a_1^2\sigma^2) + \boldsymbol{x}(-2\mu\sigma_\epsilon^2 + 2a_0a_1\sigma^2) + \boldsymbol{y}^2(\sigma^2) + \boldsymbol{y}(-2a_0\sigma^2) + \boldsymbol{x}\boldsymbol{y}(-2a_1\sigma^2) + (a_0^2\sigma^2 + \mu^2\sigma_\epsilon^2)$$
$$= \boldsymbol{\omega}^\top \begin{pmatrix} \sigma_\epsilon^2 + a_1^2\sigma^2 & -a_1\sigma^2 \\ -a_1\sigma^2 & \sigma^2 \end{pmatrix} \boldsymbol{\omega} + \begin{pmatrix} 2a_0a_1\sigma^2 - 2\mu\sigma_\epsilon^2 \\ -2a_0\sigma^2 \end{pmatrix}^\top \boldsymbol{\omega} + a_0^2\sigma^2 + \mu^2\sigma_\epsilon^2,$$

hence,

$$f_{X,Y}(\boldsymbol{x}, \boldsymbol{y}) = \frac{1}{2\pi\sigma\sigma_\epsilon} \exp\left(-\frac{1}{2\sigma^2\sigma_\epsilon^2}\left[\sigma_\epsilon^2(\boldsymbol{x} - \mu)^2 + \sigma^2(\boldsymbol{y} - (a_0 + a_1\boldsymbol{x}))^2\right]\right)$$
$$= \frac{1}{2\pi\sigma\sigma_\epsilon} \exp\left(-\frac{1}{2\sigma^2\sigma_\epsilon^2}\left[\boldsymbol{\omega}^\top \begin{pmatrix} \sigma_\epsilon^2 + a_1^2\sigma^2 & -a_1\sigma^2 \\ -a_1\sigma^2 & \sigma^2 \end{pmatrix} \boldsymbol{\omega} + \begin{pmatrix} 2a_0a_1\sigma^2 - 2\mu\sigma_\epsilon^2 \\ -2a_0\sigma^2 \end{pmatrix}^\top \boldsymbol{\omega} + a_0^2\sigma^2 + \mu^2\sigma_\epsilon^2\right]\right)$$
$$= \frac{1}{2\pi\sigma\sigma_\epsilon} \exp\left(-\frac{1}{2}\boldsymbol{\omega}^\top \begin{pmatrix} \frac{1}{\sigma^2} + \frac{a_1^2}{\sigma_\epsilon^2} & -\frac{a_1}{\sigma_\epsilon^2} \\ -\frac{a_1}{\sigma_\epsilon^2} & \frac{1}{\sigma_\epsilon^2} \end{pmatrix} \boldsymbol{\omega} + \begin{pmatrix} \frac{\mu}{\sigma^2} - \frac{a_0a_1}{\sigma_\epsilon^2} \\ \frac{a_0}{\sigma_\epsilon^2} \end{pmatrix}^\top \boldsymbol{\omega} - \frac{a_0^2}{2\sigma_\epsilon^2} - \frac{\mu^2}{2\sigma^2}\right)$$
$$= \frac{1}{2\pi\sigma\sigma_\epsilon} \exp\left(-\frac{1}{2}\boldsymbol{\omega}^\top A_2\boldsymbol{\omega} + \boldsymbol{b}_2^\top\boldsymbol{\omega} + c_2\right)$$

where $A_2 := \begin{pmatrix} \frac{1}{\sigma^2} + \frac{a_1^2}{\sigma_\epsilon^2} & -\frac{a_1}{\sigma_\epsilon^2} \\ -\frac{a_1}{\sigma_\epsilon^2} & \frac{1}{\sigma_\epsilon^2} \end{pmatrix}$, $\boldsymbol{b}_2 := \begin{pmatrix} \frac{\mu}{\sigma^2} - \frac{a_0a_1}{\sigma_\epsilon^2} \\ \frac{a_0}{\sigma_\epsilon^2} \end{pmatrix}$ and $c_2 := -\frac{a_0^2}{2\sigma_\epsilon^2} - \frac{\mu^2}{2\sigma^2}$.

Therefore, we have that

$$\mathbb{E}_{(\boldsymbol{x},\boldsymbol{y})\sim\mathbb{P}_{X,Y}}\left[k(\boldsymbol{x}, \boldsymbol{x}')l(\boldsymbol{y}, \boldsymbol{y}')\right]$$
$$= \int_\mathbb{R}\int_\mathbb{R} k(\boldsymbol{x}, \boldsymbol{x}')l(\boldsymbol{y}, \boldsymbol{y}')f_{X,Y}(\boldsymbol{x}, \boldsymbol{y})\mathrm{d}\boldsymbol{x}\mathrm{d}\boldsymbol{y}$$
$$= \frac{1}{2\pi\sigma\sigma_\epsilon}\int_{\mathbb{R}^2} \exp\left(-\frac{1}{2}\boldsymbol{\omega}^\top A_1\boldsymbol{\omega} + \boldsymbol{b}_1^\top\boldsymbol{\omega} + c_1\right)\exp\left(-\frac{1}{2}\boldsymbol{\omega}^\top A_2\boldsymbol{\omega} + \boldsymbol{b}_2^\top\boldsymbol{\omega} + c_2\right)\mathrm{d}\boldsymbol{\omega}$$
$$= \frac{1}{2\pi\sigma\sigma_\epsilon}\int_{\mathbb{R}^2} \exp\left(-\frac{1}{2}\boldsymbol{\omega}^\top(A_1 + A_2)\boldsymbol{\omega} + (\boldsymbol{b}_1 + \boldsymbol{b}_2)^\top\boldsymbol{\omega} + c_1 + c_2\right)\mathrm{d}\boldsymbol{\omega}$$
$$= \frac{1}{2\pi\sigma\sigma_\epsilon}\int_{\mathbb{R}^2} \exp\left(-\frac{1}{2}\boldsymbol{\omega}^\top A\boldsymbol{\omega} + b^\top\boldsymbol{\omega} + c\right)\mathrm{d}\boldsymbol{\omega}$$
$$= \frac{1}{2\pi\sigma\sigma_\epsilon}\frac{2\pi}{|A|^{\frac{1}{2}}}\exp\left(c + \frac{1}{2}\boldsymbol{b}^\top A^{-1}\boldsymbol{b}\right) = \frac{1}{\sqrt{\sigma^2\sigma_\epsilon^2|A|}}\exp\left(c + \frac{1}{2}\boldsymbol{b}^\top A^{-1}\boldsymbol{b}\right)$$

where $A := A_1 + A_2$, $\boldsymbol{b} := \boldsymbol{b}_1 + \boldsymbol{b}_2$ and $c = c_1 + c_2$.

### C.3.3 Computing the AMCMD Exactly

In order to compute the AMCMD exactly, we require an analytical expression for (34). First note that we have

$$\|\mu_{Y|X=x}\|^2 = \langle\mu_{Y|X=\boldsymbol{x}}, \mu_{Y|X=\boldsymbol{x}}\rangle_{\mathcal{H}_l}$$
$$= \mathbb{E}_{\boldsymbol{y}\sim\mathbb{P}_{Y|X=\boldsymbol{x}}}\left[\mu_{Y|X=\boldsymbol{x}}(\boldsymbol{y})\right]$$
$$= \mathbb{E}_{\boldsymbol{y}\sim\mathbb{P}_{Y|X=\boldsymbol{x}}}\left[\mathbb{E}_{\boldsymbol{y}'\sim\mathbb{P}_{Y|X=\boldsymbol{x}}}\left[l(\boldsymbol{y}, \boldsymbol{y}')\right]\right],$$

where the second and third equalities follow straightforwardly from the definition of the KCME. Now, the first step is to derive an analytical expression for

$$\mathbb{E}_{\boldsymbol{y}'\sim\mathbb{P}_{Y|X=\boldsymbol{x}}}\left[l(\boldsymbol{y}, \boldsymbol{y}')\right], \quad \boldsymbol{y} \in \mathcal{Y}.$$

Writing, $f(\boldsymbol{x}) = a_0 + a_1 \boldsymbol{x}$, we have

$$
\mathbb{E}_{\boldsymbol{y}' \sim \mathbb{P}_{Y|X=\boldsymbol{x}}} [l(\boldsymbol{y}, \boldsymbol{y}')] = \int_{\mathcal{Y}} l(\boldsymbol{y}, \boldsymbol{y}') p_Y(\boldsymbol{y}') d\boldsymbol{y}'
$$

$$
= \int_{\mathcal{Y}} \exp\left(-\frac{1}{2\alpha_l^2}(\boldsymbol{y}' - \boldsymbol{y})^2\right) \frac{1}{\sqrt{2\pi\sigma_\epsilon^2}} \exp\left(-\frac{1}{2\sigma_\epsilon^2}(\boldsymbol{y}' - f(\boldsymbol{x}))^2\right) d\boldsymbol{y}
$$

$$
= \frac{1}{\sqrt{2\pi\sigma_\epsilon^2}} \int_{\mathcal{Y}} \exp\left(-\frac{1}{2\alpha_l^2}(\boldsymbol{y}' - \boldsymbol{y})^2 - \frac{1}{2\sigma_\epsilon^2}(\boldsymbol{y}' - f(\boldsymbol{x}))^2\right) d\boldsymbol{y}.
$$

Now,

$$
-\frac{1}{2\alpha_l^2}(\boldsymbol{y}' - \boldsymbol{y})^2 - \frac{1}{2\sigma_\epsilon^2}(\boldsymbol{y}' - f(\boldsymbol{x}))^2
$$

$$
= -\frac{1}{2\alpha_l^2}\left(\boldsymbol{y}^2 - 2\boldsymbol{y}\boldsymbol{y}' + \boldsymbol{y}^2\right) - \frac{1}{2\sigma_\epsilon^2}\left(\boldsymbol{y}^2 - 2\boldsymbol{y}'f(\boldsymbol{x}) + f(\boldsymbol{x})^2\right)
$$

$$
= -\frac{1}{2}\boldsymbol{y}^2\left(\frac{1}{\alpha_l^2} + \frac{1}{\sigma_\epsilon^2}\right) + \boldsymbol{y}'\left(\frac{\boldsymbol{y}}{\alpha_l^2} + \frac{f(\boldsymbol{x})}{\sigma_\epsilon^2}\right) - \frac{\boldsymbol{y}^2}{2\alpha_l^2} - \frac{f(\boldsymbol{x})^2}{2\sigma_\epsilon^2}
$$

$$
= -\frac{A}{2}\boldsymbol{y}^2 + B\boldsymbol{y}' - \frac{\boldsymbol{y}^2}{2\alpha_l^2} - \frac{f(\boldsymbol{x})^2}{2\sigma_\epsilon^2}
$$

where $A := \left(\frac{1}{\alpha_l^2} + \frac{1}{\sigma_\epsilon^2}\right)$ and $B := \left(\frac{\boldsymbol{y}}{\alpha_l^2} + \frac{f(\boldsymbol{x})}{\sigma_\epsilon^2}\right)$. Completing the square, we get

$$
-\frac{A}{2}\boldsymbol{y}^2 + B\boldsymbol{y}' = -\frac{A}{2}\left(\boldsymbol{y}'^2 - \frac{2B}{A}\boldsymbol{y}\right) = -\frac{A}{2}\left[\left(\boldsymbol{y} - \frac{B}{A}\right)^2 - \left(\frac{B}{A}\right)^2\right],
$$

and therefore we can write that,

$$
\mathbb{E}_{\boldsymbol{y}' \sim \mathbb{P}_{Y|X=\boldsymbol{x}}} [l(\boldsymbol{y}, \boldsymbol{y}')]
$$

$$
= \frac{1}{\sqrt{2\pi\sigma_\epsilon^2}} \int_{\mathcal{Y}} \exp\left(-\frac{1}{2\alpha_l^2}(\boldsymbol{y}' - \boldsymbol{y})^2 - \frac{1}{2\sigma_\epsilon^2}(\boldsymbol{y}' - f(\boldsymbol{x}))^2\right) d\boldsymbol{y}
$$

$$
= \frac{1}{\sqrt{2\pi\sigma_\epsilon^2}} \int_{\mathcal{Y}} \exp\left(-\frac{A}{2}\left[\left(\boldsymbol{y}' - \frac{B}{A}\right)^2 - \left(\frac{B}{A}\right)^2\right] - \frac{\boldsymbol{y}^2}{2\alpha_l^2} - \frac{f(\boldsymbol{x})^2}{2\sigma_\epsilon^2}\right) d\boldsymbol{y}'
$$

$$
= \frac{1}{\sqrt{2\pi\sigma_\epsilon^2}} \exp\left(\frac{A}{2}\left(\frac{B}{A}\right)^2 - \frac{\boldsymbol{y}'^2}{2\alpha_l^2} - \frac{f(\boldsymbol{x})^2}{2\sigma_\epsilon^2}\right) \int_{\mathcal{Y}} \exp\left(-\frac{A}{2}\left(\boldsymbol{y}' - \frac{B}{A}\right)^2\right) d\boldsymbol{y}'
$$

$$
= \frac{1}{\sqrt{A\sigma_\epsilon^2}} \exp\left(\frac{A}{2}\left(\frac{B}{A}\right)^2 - \frac{\boldsymbol{y}'^2}{2\alpha_l^2} - \frac{f(\boldsymbol{x})^2}{2\sigma_\epsilon^2}\right).
$$

We can further simplify by removing the unwieldy constants $A$ and $B$, writing that

$$
\mathbb{E}_{\boldsymbol{y}' \sim \mathbb{P}_{Y|X=\boldsymbol{x}}} [l(\boldsymbol{y}, \boldsymbol{y}')]
$$

$$
= \frac{1}{\sqrt{A\sigma_\epsilon^2}} \exp\left(\frac{1}{2A}B^2 - \frac{\boldsymbol{y}^2}{2\alpha_l^2} - \frac{f(\boldsymbol{x})^2}{2\sigma_\epsilon^2}\right)
$$

$$
= \frac{1}{\sqrt{A\sigma_\epsilon^2}} \exp\left(\frac{1}{2A}\left(\frac{\boldsymbol{y}^2}{\alpha_l^4} + 2\frac{\boldsymbol{y}f(\boldsymbol{x})}{\alpha_l^2\sigma_\epsilon^2} + \frac{f(\boldsymbol{x})^2}{\sigma_\epsilon^4}\right) - \frac{\boldsymbol{y}^2}{2\alpha_l^2} - \frac{f(\boldsymbol{x})^2}{2\sigma_\epsilon^2}\right)
$$

$$
= \frac{1}{\sqrt{A\sigma_\epsilon^2}} \exp\left(\frac{1}{2A}\left(\boldsymbol{y}^2\left[\frac{1}{\alpha_l^4} - \frac{A}{\alpha_l^2}\right] + 2\frac{\boldsymbol{y}f(\boldsymbol{x})}{\alpha_l^2\sigma_\epsilon^2} + f(\boldsymbol{x})^2\left[\frac{1}{\sigma_\epsilon^4} - \frac{A}{\sigma_\epsilon^2}\right]\right)\right)
$$

$$
= \frac{1}{\sqrt{A\sigma_\epsilon^2}} \exp\left(\frac{1}{2A}\left(-\frac{\boldsymbol{y}^2}{\alpha_l^2\sigma_\epsilon^2} + 2\frac{\boldsymbol{y}f(\boldsymbol{x})}{\alpha_l^2\sigma_\epsilon^2} - \frac{f(\boldsymbol{x})^2}{\alpha_l^2\sigma_\epsilon^2}\right)\right)
$$

$$
= \frac{1}{\sqrt{A\sigma_\epsilon^2}} \exp\left(-\frac{1}{2(\alpha_l^2 + \sigma_\epsilon^2)}[\boldsymbol{y} - f(\boldsymbol{x})]^2\right).
$$

The next step is therefore to compute,

$$\mathbb{E}_{\boldsymbol{y}\sim\mathbb{P}_{Y|X=\boldsymbol{x}}}\left[\mathbb{E}_{\boldsymbol{y}'\sim\mathbb{P}_{Y|X=\boldsymbol{x}}}\left[l\left(\boldsymbol{y},\boldsymbol{y}'\right)\right]\right] = \frac{1}{\sqrt{A\sigma_\epsilon^2}}\mathbb{E}_{\boldsymbol{y}\sim\mathbb{P}_{Y|X=\boldsymbol{x}}}\left[\exp\left(-\frac{1}{2(\alpha_l^2+\sigma_\epsilon^2)}\left[\boldsymbol{y}-f(\boldsymbol{x})\right]^2\right)\right]$$

$$= \frac{1}{\sqrt{A\sigma_\epsilon^2}}\frac{1}{\sqrt{2\pi\sigma_\epsilon^2}}\int_{\mathcal{Y}}\exp\left(-\frac{1}{2\sigma_\epsilon^2}[\boldsymbol{y}-f(\boldsymbol{x})]^2\right)\exp\left(-\frac{1}{2(\alpha_l^2+\sigma_\epsilon^2)}\left[\boldsymbol{y}-f(\boldsymbol{x})\right]^2\right)\mathrm{d}\boldsymbol{y}$$

$$= \frac{1}{\sigma_\epsilon^2\sqrt{2\pi A}}\int_{\mathcal{Y}}\exp\left(-\frac{1}{2}\cdot\frac{2\sigma_\epsilon^2+\alpha_l^2}{\sigma_\epsilon^2(\sigma_\epsilon^2+\alpha_l^2)}\left[\boldsymbol{y}-f(\boldsymbol{x})\right]^2\right)\mathrm{d}\boldsymbol{y}$$

$$= \frac{1}{\sigma_\epsilon^2\sqrt{2\pi A}}\cdot\sqrt{2\pi\frac{\sigma_\epsilon^2(\sigma_\epsilon^2+\alpha_l^2)}{2\sigma_\epsilon^2+\alpha_l^2}} = \sqrt{\frac{\sigma_\epsilon^2+\alpha_l^2}{A\sigma_\epsilon^2(2\sigma_\epsilon^2+\alpha_l^2)}} = \sqrt{\frac{\sigma_\epsilon^2+\alpha_l^2}{\left(1+\frac{\sigma_\epsilon^2}{\alpha_l^2}\right)(2\sigma_\epsilon^2+\alpha_l^2)}}.$$

Therefore,

$$\mathbb{E}_{\boldsymbol{x}\sim\mathbb{P}_X}\left[\|\mu_{Y|X=x}\|^2\right] = \mathbb{E}_{\boldsymbol{x}\sim\mathbb{P}_X}\left[\mathbb{E}_{\boldsymbol{y}\sim\mathbb{P}_{Y|X=\boldsymbol{x}}}\left[\mathbb{E}_{\boldsymbol{y}'\sim\mathbb{P}_{Y|X=\boldsymbol{x}}}\left[l\left(\boldsymbol{y},\boldsymbol{y}'\right)\right]\right]\right]$$

$$= \mathbb{E}_{\boldsymbol{x}\sim\mathbb{P}_X}\left[\sqrt{\frac{\sigma_\epsilon^2+\alpha_l^2}{\left(1+\frac{\sigma_\epsilon^2}{\alpha_l^2}\right)(2\sigma_\epsilon^2+\alpha_l^2)}}\right] = \sqrt{\frac{\sigma_\epsilon^2+\alpha_l^2}{\left(1+\frac{\sigma_\epsilon^2}{\alpha_l^2}\right)(2\sigma_\epsilon^2+\alpha_l^2)}},$$

where we see that the integrand with respect to the expectation over $\mathbb{P}_X$ is constant. Note that the above computations hold for arbitrary $f:\mathcal{X}\rightarrow\mathbb{R}$, however we require that the error has constant variance $\sigma_\epsilon^2$. If the error is not constant and it evolves as a function of $\boldsymbol{x}$, then the final expectation with respect to $\mathbb{P}_X$ may become very difficult to compute exactly.

# D  Algorithm Details

In this section we include additional details about the algorithms developed in this work including pseudocode and complexity analysis.

## D.1  Pseudocode

In this section we include pseudocode for the algorithms introduced in this work, including gradient-free variants of the Kernel Herding type algorithms suitable for $\mathcal{X}\neq\mathbb{R}^d$ and $\mathcal{Y}\neq\mathbb{R}^p$. In all gradient-based algorithms, the pseudocode assumes standard gradient descent. In practice, however, any gradient descent variant may be used. In our implementation, we employed the Optax [82] package, which provides access to a wide range of gradient-based optimisation methods, including ADAM [81], which we used in our experiments.

---

**Algorithm 1** Joint Kernel Herding

---

**Input**: Dataset $\mathcal{D} = \{(\boldsymbol{x}_i, \boldsymbol{y}_i)\}_{i=1}^n \subset \mathcal{X} \times \mathcal{Y}$, Coreset size $M \in \mathbb{N}$, Feature kernel $k : \mathcal{X} \times \mathcal{X} \to \mathbb{R}$, Response kernel $l : \mathcal{Y} \times \mathcal{Y} \to \mathbb{R}$, Candidate batch size $C \in \mathbb{N}$, Maximum iteration number $T$, Step size $\alpha$

**for** $t = 1$ **to** $m$ **do**
    Uniformly at random, select $C$ candidate pairs $\{(\bar{\boldsymbol{x}}_i, \bar{\boldsymbol{y}}_i)\}_{i=1}^C$ from $\mathcal{D}$
    **for** $i = 1$ **to** $C$ **do**
        Estimate $\mathcal{S}_i \leftarrow \mathcal{L}_{t-1}^{\mathcal{D}}(\bar{\boldsymbol{x}}_i, \bar{\boldsymbol{y}}_i)$ using equation (12)
    **end for**
    $i^* = \arg\min \ \mathcal{S}_i$
    $(\tilde{\boldsymbol{x}}_1, \tilde{\boldsymbol{y}}_1) \leftarrow (\bar{\boldsymbol{x}}_{i^*}, \bar{\boldsymbol{y}}_{i^*})$

    **for** $j = 1$ **to** $T$ **do**
        Compute $\nabla_{\boldsymbol{x}} \mathcal{L}_{t-1}^{\mathcal{D}}(\tilde{\boldsymbol{x}}_j, \tilde{\boldsymbol{y}}_j)$ using equation (13)
        Compute $\nabla_{\boldsymbol{y}} \mathcal{L}_{t-1}^{\mathcal{D}}(\tilde{\boldsymbol{x}}_j, \tilde{\boldsymbol{y}}_j)$ using equation (14)
        $\tilde{\boldsymbol{x}}_{j+1} \leftarrow \tilde{\boldsymbol{x}}_j - \alpha \nabla_{\boldsymbol{x}} \mathcal{L}_{t-1}^{\mathcal{D}}(\tilde{\boldsymbol{x}}_j, \tilde{\boldsymbol{y}}_j)$
        $\tilde{\boldsymbol{y}}_{j+1} \leftarrow \tilde{\boldsymbol{y}}_j - \alpha \nabla_{\boldsymbol{y}} \mathcal{L}_{t-1}^{\mathcal{D}}(\tilde{\boldsymbol{x}}_j, \tilde{\boldsymbol{y}}_j)$
        **if** converged **then**
            break
        **end if**
    **end for**
    Add the final optimised pair to the compressed set:
    $\mathcal{C}_t \leftarrow \mathcal{C}_{t-1} \cup \{(\tilde{\boldsymbol{x}}, \tilde{\boldsymbol{y}})\}$
**end for**
**return** $\mathcal{C}_M$

---

---

**Algorithm 2** Average Conditional Kernel Herding

---

**Input**: Dataset $\mathcal{D} = \{(\boldsymbol{x}_i, \boldsymbol{y}_i)\}_{i=1}^n \subset \mathcal{X} \times \mathcal{Y}$, Coreset size $M \in \mathbb{N}$, Feature kernel $k : \mathcal{X} \times \mathcal{X} \to \mathbb{R}$, Response kernel $l : \mathcal{Y} \times \mathcal{Y} \to \mathbb{R}$, Regularisation parameter $\lambda \in \mathbb{R}_{>0}$, Candidate batch size $C \in \mathbb{N}$, Maximum iteration number $T \in \mathbb{N}$, Step size $\alpha \in \mathbb{R}_{>0}$

**for** $t = 1$ **to** $m$ **do**
    Uniformly at random, select $C$ candidate pairs $\{(\bar{\boldsymbol{x}}_i, \bar{\boldsymbol{y}}_i)\}_{i=1}^C$ from $\mathcal{D}$
    **for** $i = 1$ **to** $C$ **do**
        Estimate $\mathcal{S}_i \leftarrow \mathcal{G}_{t-1}^{\mathcal{D}}(\bar{\boldsymbol{x}}_i, \bar{\boldsymbol{y}}_i)$ using equation (9)
    **end for**
    $i^* = \arg\min \ \mathcal{S}_i$
    $(\tilde{\boldsymbol{x}}_1, \tilde{\boldsymbol{y}}_1) \leftarrow (\bar{\boldsymbol{x}}_{i^*}, \bar{\boldsymbol{y}}_{i^*})$

    **for** $j = 1$ **to** $T$ **do**
        Compute $\nabla_{\boldsymbol{x}} \mathcal{G}_{t-1}^{\mathcal{D}}(\tilde{\boldsymbol{x}}_j, \tilde{\boldsymbol{y}}_j)$ using equation (26)
        Compute $\nabla_{\boldsymbol{y}} \mathcal{G}_{t-1}^{\mathcal{D}}(\tilde{\boldsymbol{x}}_j, \tilde{\boldsymbol{y}}_j)$ using equation (25)
        $\tilde{\boldsymbol{x}}_{j+1} \leftarrow \tilde{\boldsymbol{x}}_j - \alpha \nabla_{\boldsymbol{x}} \mathcal{G}_{t-1}^{\mathcal{D}}(\tilde{\boldsymbol{x}}_j, \tilde{\boldsymbol{y}}_j)$
        $\tilde{\boldsymbol{y}}_{j+1} \leftarrow \tilde{\boldsymbol{y}}_j - \alpha \nabla_{\boldsymbol{y}} \mathcal{G}_{t-1}^{\mathcal{D}}(\tilde{\boldsymbol{x}}_j, \tilde{\boldsymbol{y}}_j)$
        **if** converged **then**
            break
        **end if**
    **end for**
    Add the final optimised pair to the compressed set:
    $\mathcal{C}_t \leftarrow \mathcal{C}_{t-1} \cup \{(\tilde{\boldsymbol{x}}, \tilde{\boldsymbol{y}})\}$
**end for**
**return** $\mathcal{C}_M$

---

---
**Algorithm 3** Joint Kernel Inducing Points
---

**Input**: Dataset $\mathcal{D} = \{(\boldsymbol{x}_i, \boldsymbol{y}_i)\}_{i=1}^n \subset \mathcal{X} \times \mathcal{Y}$, Coreset size $M \in \mathbb{N}$, Feature kernel $k : \mathcal{X} \times \mathcal{X} \to \mathbb{R}$, Response kernel $l : \mathcal{Y} \times \mathcal{Y} \to \mathbb{R}$, Candidate batch size $C \in \mathbb{N}$, Maximum iteration number $T$, Step size $\alpha$

Uniformly at random, select $C$ sets of candidate sets $\{(\tilde{\boldsymbol{X}}_i, \tilde{\boldsymbol{Y}}_i)\}_{i=1}^C$ from $\mathcal{D}$, $|\tilde{\boldsymbol{X}}_i| = |\tilde{\boldsymbol{Y}}_i| = M$, $i = 1, \ldots C$
**for** $i = 1$ **to** $C$ **do**
    Estimate $\mathcal{S}_i \leftarrow \mathcal{L}^{\mathcal{D}}(\tilde{\boldsymbol{X}}_i, \tilde{\boldsymbol{Y}}_i)$ using equation (4)
**end for**
$i^* = \arg\min \ \mathcal{S}_i$
$(\tilde{\boldsymbol{X}}_1, \tilde{\boldsymbol{Y}}_1) \leftarrow (\bar{\boldsymbol{X}}_{i^*}, \bar{\boldsymbol{Y}}_{i^*})$

**for** $j = 1$ **to** $T$ **do**
    Compute $\nabla_{\tilde{\boldsymbol{X}}} \mathcal{L}^{\mathcal{D}}(\tilde{\boldsymbol{X}}_j, \tilde{\boldsymbol{Y}}_j)$ using equation (19)
    Compute $\nabla_{\tilde{\boldsymbol{Y}}} \mathcal{L}^{\mathcal{D}}(\tilde{\boldsymbol{X}}_j, \tilde{\boldsymbol{Y}}_j)$ using equation (20)
    $\tilde{\boldsymbol{X}}_{j+1} \leftarrow \tilde{\boldsymbol{X}}_j - \alpha \nabla_{\tilde{\boldsymbol{X}}} \mathcal{L}^{\mathcal{D}}(\tilde{\boldsymbol{X}}_j, \tilde{\boldsymbol{Y}}_j)$
    $\tilde{\boldsymbol{Y}}_{j+1} \leftarrow \tilde{\boldsymbol{Y}}_j - \alpha \nabla_{\tilde{\boldsymbol{Y}}} \mathcal{L}^{\mathcal{D}}(\tilde{\boldsymbol{X}}_j, \tilde{\boldsymbol{Y}}_j)$
    **if** converged **then**
        break
    **end if**
**end for**

    **return** $(\tilde{\boldsymbol{X}}, \tilde{\boldsymbol{Y}})$
---

---
**Algorithm 4** Average Conditional Kernel Inducing Points
---

**Input**: Dataset $\mathcal{D} = \{(\boldsymbol{x}_i, \boldsymbol{y}_i)\}_{i=1}^n \subset \mathcal{X} \times \mathcal{Y}$, Coreset size $M \in \mathbb{N}$, Feature kernel $k : \mathcal{X} \times \mathcal{X} \to \mathbb{R}$, Response kernel $l : \mathcal{Y} \times \mathcal{Y} \to \mathbb{R}$, Regularisation parameter $\lambda \in \mathbb{R}_{>0}$m Candidate batch size $C \in \mathbb{N}$, Maximum iteration number $T$, Step size $\alpha$

Uniformly at random, select $C$ sets of candidate sets $\{(\tilde{\boldsymbol{X}}_i, \tilde{\boldsymbol{Y}}_i)\}_{i=1}^C$ from $\mathcal{D}$, $|\tilde{\boldsymbol{X}}_i| = |\tilde{\boldsymbol{Y}}_i| = M$, $i = 1, \ldots C$
**for** $i = 1$ **to** $C$ **do**
    Estimate $\mathcal{S}_i \leftarrow \mathcal{J}^{\mathcal{D}}(\tilde{\boldsymbol{X}}_i, \tilde{\boldsymbol{Y}}_i)$ using equation (11)
**end for**
$i^* = \arg\min \ \mathcal{S}_i$
$(\tilde{\boldsymbol{X}}_1, \tilde{\boldsymbol{Y}}_1) \leftarrow (\bar{\boldsymbol{X}}_{i^*}, \bar{\boldsymbol{Y}}_{i^*})$

**for** $j = 1$ **to** $T$ **do**
    Compute $\nabla_{\tilde{\boldsymbol{X}}} \mathcal{J}^{\mathcal{D}}(\tilde{\boldsymbol{X}}_j, \tilde{\boldsymbol{Y}}_j)$ using equation (28)
    Compute $\nabla_{\tilde{\boldsymbol{Y}}} \mathcal{J}^{\mathcal{D}}(\tilde{\boldsymbol{X}}_j, \tilde{\boldsymbol{Y}}_j)$ using equation (29)
    $\tilde{\boldsymbol{X}}_{j+1} \leftarrow \tilde{\boldsymbol{X}}_j - \alpha \nabla_{\tilde{\boldsymbol{X}}} \mathcal{J}^{\mathcal{D}}(\tilde{\boldsymbol{X}}_j, \tilde{\boldsymbol{Y}}_j)$
    $\tilde{\boldsymbol{Y}}_{j+1} \leftarrow \tilde{\boldsymbol{Y}}_j - \alpha \nabla_{\tilde{\boldsymbol{Y}}} \mathcal{J}^{\mathcal{D}}(\tilde{\boldsymbol{X}}_j, \tilde{\boldsymbol{Y}}_j)$
    **if** converged **then**
        break
    **end if**
**end for**

    **return** $(\tilde{\boldsymbol{X}}, \tilde{\boldsymbol{Y}})$
---

---

**Algorithm 5** Gradient-Free Joint Kernel Herding

---

**Input**: Dataset $\mathcal{D} = \{(\boldsymbol{x}_i, \boldsymbol{y}_i)\}_{i=1}^n \subset \mathcal{X} \times \mathcal{Y}$, Coreset size $M \in \mathbb{N}$, Feature kernel $k : \mathcal{X} \times \mathcal{X} \to \mathbb{R}$, Response kernel $l : \mathcal{Y} \times \mathcal{Y} \to \mathbb{R}$, Candidate batch size $C \in \mathbb{N}$

Initialise $\mathcal{C}_0 = \emptyset$
**for** $t = 1$ **to** $m$ **do**
    Uniformly at random, select $C$ candidate pairs $\{(\bar{\boldsymbol{x}}_i, \bar{\boldsymbol{y}}_i)\}_{i=1}^C$ from $\mathcal{D}$
    **for** $i = 1$ **to** $C$ **do**
        Estimate $\mathcal{S}_i \leftarrow \mathcal{L}_{t-1}^{\mathcal{D}}(\bar{\boldsymbol{x}}_i, \bar{\boldsymbol{y}}_i)$ using equation (12)
    **end for**
    $i^* = \arg\min \ \mathcal{S}_i$
    $\mathcal{C}_t \leftarrow \mathcal{C}_{t-1} \cup \{(\bar{\boldsymbol{x}}_{i^*}, \bar{\boldsymbol{y}}_{i^*})\}$
**end for**
**return** $\mathcal{C}_M$

---

---

**Algorithm 6** Gradient-Free Average Conditional Kernel Herding

---

**Input**: Dataset $\mathcal{D} = \{(\boldsymbol{x}_i, \boldsymbol{y}_i)\}_{i=1}^n \subset \mathcal{X} \times \mathcal{Y}$, Coreset size $M \in \mathbb{N}$, Feature kernel $k : \mathcal{X} \times \mathcal{X} \to \mathbb{R}$, Response kernel $l : \mathcal{Y} \times \mathcal{Y} \to \mathbb{R}$, Regularisation parameter $\lambda \in \mathbb{R}_{>0}$, Candidate batch size $C \in \mathbb{N}$

Initialise $\mathcal{C}_0 = \emptyset$
**for** $t = 1$ **to** $m$ **do**
    Uniformly at random, select $C$ candidate pairs $\{(\bar{\boldsymbol{x}}_i, \bar{\boldsymbol{y}}_i)\}_{i=1}^C$ from $\mathcal{D}$
    **for** $i = 1$ **to** $C$ **do**
        Estimate $\mathcal{S}_i \leftarrow \mathcal{G}_{t-1}^{\mathcal{D}}(\bar{\boldsymbol{x}}_i, \bar{\boldsymbol{y}}_i)$ using equation (9)
    **end for**
    $i^* = \arg\min \ \mathcal{S}_i$
    $\mathcal{C}_t \leftarrow \mathcal{C}_{t-1} \cup \{(\bar{\boldsymbol{x}}_{i^*}, \bar{\boldsymbol{y}}_{i^*})\}$
**end for**
**return** $\mathcal{C}_M$

---

**Algorithm 7** Joint Kernel Inducing Points with Exhaustive Search

---

**Input**: Dataset $\mathcal{D} = \{(\boldsymbol{x}_i, \boldsymbol{y}_i)\}_{i=1}^n \subset \mathcal{X} \times \mathcal{Y}$, Coreset size $M \in \mathbb{N}$, Feature kernel $k : \mathcal{X} \times \mathcal{X} \to \mathbb{R}$, Indicator response kernel $l : \mathcal{Y} \times \mathcal{Y} \to \mathbb{R}$, Candidate batch size $C \in \mathbb{N}$, Maximum iteration number $T$, Step size $\alpha$, Set of possible classes $A := \{0, 1, \ldots, a\}$

Uniformly at random, select $C$ sets of candidate sets $\{(\tilde{\boldsymbol{X}}_i, \tilde{\boldsymbol{Y}}_i)\}_{i=1}^C$ from $\mathcal{D}$, $|\tilde{\boldsymbol{X}}_i| = |\tilde{\boldsymbol{Y}}_i| = M$, $i = 1, \ldots C$
**for** $i = 1$ **to** $C$ **do**
    Estimate $\mathcal{S}_i \leftarrow \mathcal{L}^{\mathcal{D}}(\tilde{\boldsymbol{X}}_i, \tilde{\boldsymbol{Y}}_i)$ using equation (4)
**end for**
$i^* = \arg\min \; \mathcal{S}_i$
$(\tilde{\boldsymbol{X}}_1, \tilde{\boldsymbol{Y}}_1) \leftarrow (\bar{\boldsymbol{X}}_{i^*}, \bar{\boldsymbol{Y}}_{i^*})$

**for** $j = 1$ **to** $T$ **do**
    Compute $\nabla_{\tilde{\boldsymbol{X}}} \mathcal{L}^{\mathcal{D}}(\tilde{\boldsymbol{X}}_j, \tilde{\boldsymbol{Y}}_j)$ using equation (19)
    $\tilde{\boldsymbol{X}}_{j+1} \leftarrow \tilde{\boldsymbol{X}}_j - \alpha \nabla_{\tilde{\boldsymbol{X}}} \mathcal{L}^{\mathcal{D}}(\tilde{\boldsymbol{X}}_j, \tilde{\boldsymbol{Y}}_j)$
    **for** $i = 1$ **to** $m$ **do**
        Using equation (31), compute $\mathcal{F}^{\mathcal{D}}(\tilde{\boldsymbol{y}}_i)$ for each possible value of $\tilde{\boldsymbol{y}}_i \in \{0, 1, \ldots, a\}$
        Update $\tilde{\boldsymbol{Y}}_j$ with the optimal choice
    **end for**
    **if** converged **then**
        break
    **end if**
**end for**

**return** $(\tilde{\boldsymbol{X}}, \tilde{\boldsymbol{Y}})$

---

**Algorithm 8** Average Conditional Kernel Inducing Points with Exhaustive Search

---

**Input**: Dataset $\mathcal{D} = \{(\boldsymbol{x}_i, \boldsymbol{y}_i)\}_{i=1}^n \subset \mathcal{X} \times \mathcal{Y}$, Coreset size $M \in \mathbb{N}$, Feature kernel $k : \mathcal{X} \times \mathcal{X} \to \mathbb{R}$, Indicator response kernel $l : \mathcal{Y} \times \mathcal{Y} \to \mathbb{R}$, Regularisation parameter $\lambda \in \mathbb{R}_{>0}$ Candidate batch size $C \in \mathbb{N}$, Maximum iteration number $T$, Step size $\alpha$, Set of possible classes $A := \{0, 1, \ldots, a\}$

Uniformly at random, select $C$ sets of candidate sets $\{(\tilde{\boldsymbol{X}}_i, \tilde{\boldsymbol{Y}}_i)\}_{i=1}^C$ from $\mathcal{D}$, $|\tilde{\boldsymbol{X}}_i| = |\tilde{\boldsymbol{Y}}_i| = M$, $i = 1, \ldots C$
**for** $i = 1$ **to** $C$ **do**
    Estimate $\mathcal{S}_i \leftarrow \mathcal{J}^{\mathcal{D}}(\tilde{\boldsymbol{X}}_i, \tilde{\boldsymbol{Y}}_i)$ using equation (11)
**end for**
$i^* = \arg\min \; \mathcal{S}_i$
$(\tilde{\boldsymbol{X}}_1, \tilde{\boldsymbol{Y}}_1) \leftarrow (\bar{\boldsymbol{X}}_{i^*}, \bar{\boldsymbol{Y}}_{i^*})$

**for** $j = 1$ **to** $T$ **do**
    Compute $\nabla_{\tilde{\boldsymbol{X}}} \mathcal{J}^{\mathcal{D}}(\tilde{\boldsymbol{X}}_j, \tilde{\boldsymbol{Y}}_j)$ using equation (28)
    $\tilde{\boldsymbol{X}}_{j+1} \leftarrow \tilde{\boldsymbol{X}}_j - \alpha \nabla_{\tilde{\boldsymbol{X}}} \mathcal{J}^{\mathcal{D}}(\tilde{\boldsymbol{X}}_j, \tilde{\boldsymbol{Y}}_j)$
    **for** $i = 1$ **to** $m$ **do**
        Using equation (32), compute $\mathcal{R}^{\mathcal{D}}(\tilde{\boldsymbol{y}}_i)$ for each possible value of $\tilde{\boldsymbol{y}}_i \in \{0, 1, \ldots, a\}$
        Update $\tilde{\boldsymbol{Y}}_j$ with the optimal choice
    **end for**
    **if** converged **then**
        break
    **end if**
**end for**

**return** $(\tilde{\boldsymbol{X}}, \tilde{\boldsymbol{Y}})$

---

## D.2  Complexity Analysis

In this section we derive the overall storage and time complexity of constructing a compressed set of size $m$, where we use $n \gg m$ datapoints to estimate the objective functions of JKH, JKIP, ACKH and ACKIP respectively. The results are summarised in Table 2.

| Algorithm | Time Complexity | Memory Complexity |
|---|---|---|
| JKH | $\mathcal{O}(m^2 + mn)$ | $\mathcal{O}(m + n)$ |
| JKIP | $\mathcal{O}(m^2 + mn)$ | $\mathcal{O}(m^2 + mn)$ |
| ACKH | $\mathcal{O}(m^4 + m^3 n)$ | $\mathcal{O}(mn + m^2)$ |
| ACKIP | $\mathcal{O}(m^3 + m^2 n)$ | $\mathcal{O}(m^2 + mn)$ |

Table 2: Time and memory complexity of different algorithms.

### D.2.1  Joint Kernel Herding

From Section B.3, we know that each gradient computation of JKH has $\mathcal{O}(m + n)$ storage and time complexity.

**Storage cost**: The overall storage cost is just the cost of storing the gradients of the last iteration, i.e. $\mathcal{O}(m + n)$, that is, linear in the size of the target dataset $n$.

**Time cost**: Assuming we take $T$ gradient steps per optimisation of each pair in the compressed set, then the cost of the $m^{\text{th}}$ iteration of JKH is $\mathcal{O}((m + n)T)$. Therefore, the final cost is

$$\sum_{i=1}^{m}(i + n)T = \left(\frac{m(m+1)}{2} + mn\right)T = \mathcal{O}((m^2 + mn)T)$$

i.e. linear in the size of the target dataset $n$.

### D.2.2  Joint Kernel Inducing Points

From Section B.4, we know that each gradient computation of JKIP has $\mathcal{O}(m^2 + mn)$ storage and time complexity.

**Storage cost**: The overall storage cost is $\mathcal{O}(m^2 + mn)$ i.e. linear in the size of the target dataset $n$.

**Time cost**: Assuming we take $J$ gradient steps, then the final cost of JKIP is simply $\mathcal{O}((m^2 + mn)J)$ i.e. linear in the size of the target dataset $n$.

### D.2.3  Average Conditional Kernel Herding

From Section B.9, we know that each gradient computation of ACKH has $\mathcal{O}(m^2 + mn)$ storage and $\mathcal{O}(m^3 + m^2 n)$ time complexity.

**Storage cost**: The overall storage cost is $\mathcal{O}(m^2 + mn)$ i.e. linear in the size of the target dataset $n$.

**Time cost**: Assuming we take $T$ gradient steps per optimisation of each pair in the compressed set, then the cost of the $m^{\text{th}}$ iteration of ACKH is $\mathcal{O}((m^3 + m^2 n)T)$. Therefore, the final cost is

$$\sum_{i=1}^{m}(i^3 + i^2 N)T = \left[\left(\frac{m(m+1)}{2}\right)^2 + \frac{m(m+1)(2m+1)}{6}n\right]T$$
$$= \mathcal{O}((m^4 + m^3 n)T),$$

that is, linear in the size of the target dataset $n$, but suffering from quartic cost in $m$.

### D.2.4  Average Conditional Kernel Inducing Points

From Section B.10, we know that each gradient computation of ACKIP has $\mathcal{O}(m^2 + mn)$ storage and $\mathcal{O}(m^3 + m^2 N)$ time complexity.

**Storage cost**: The overall storage cost is $\mathcal{O}(m^2 + mn)$ i.e. linear in the size of the target dataset $n$.

**Time cost**: Assuming we take $J$ gradient steps, then the final cost of ACKIP is simply $\mathcal{O}((m^3 + m^2 n)J)$ i.e. linear in the size of the target dataset $n$, but suffering from only cubic cost in $m$ versus quartic for ACKH.

### D.2.5 Discussion

Experimentation suggests that the number of gradient steps required by JKIP and ACKIP to achieve convergence is of the same order as JKH and ACKH, i.e., $J \approx T$. Consequently, JKIP and JKH have the same time complexity, but JKIP incurs a slightly higher storage cost due to an additional factor of $m$, which arises from the joint optimisation of pairs in the compressed set.

For ACKIP and ACKH, their storage costs are identical, as the nature of the ACKH objective prevents it from being expressed solely in terms of the newest pair in the compressed set (unlike in JKH).This same property causes ACKH to have an additional factor of $m$ in its time complexity compared to ACKIP. This difference becomes significant in complex problems which call for large $m$, or more generally when $n$ is very large.

Throughout, we have omitted the contribution of the feature dimension $d$ and response dimension $p$ from the stated time complexities, as it is implicitly assumed that in the distribution compression context, $n \gg d, p$. For commonly used kernels, evaluation is typically linear in $d$ or $p$, so at most, one should expect an additional multiplicative factor of $d + p$.

The algorithms in this paper were implemented using the free, open-source Python library JAX [79]. JAX enables Just-In-Time (JIT) compilation, which significantly increases execution speed. However, to achieve this speed, JAX relies on an immutable array structure, meaning the arrays must not change shape during program execution. As a result, JKH and ACKH cannot fully leverage the speed benefits of JIT compilation in their current form, as the size of arrays increases at every iteration. Conversely, the arrays considered in JKIP and ACKIP stay the same shape throughout. This presents a notable practical advantage of JKIP and ACKIP over JKH and ACKH in the JAX implementation.

