# OpenReview forum: "Conditional Distribution Compression via the Kernel Conditional Mean Embedding"
_NeurIPS.cc/2025/Conference — NeurIPS 2025 poster_

### Official Review · Reviewer_hGvG · 2025-06-30

**Clarity:** 3
**Significance:** 3
**Originality:** 3
**Rating:** 5
**Confidence:** 2

**Summary:**

This paper extends the distribution‐compression problem to the conditional‐distribution setting and proposes a family of kernel metrics and algorithms for selecting the best compression set in an RKHS. Specifically, the authors introduce AMCMD (Average Maximum Conditional Mean Discrepancy), prove it is a valid metric, and show it can be estimated in O(n) time—rather than O(n^3)—by leveraging the tower property.
Building on AMCMD, they present ACKH, a compression algorithm for conditional distributions, and demonstrate experimentally that ACKH outperforms the joint‐distribution approach (JKH). They also propose an inducing‐point variant, ACKIP, which achieves superior performance on both synthetic and real‐world datasets. Overall, the paper is well motivated and clearly written.

**Questions:**

1. What types of real-world data motivate this approach? In particular, under what practical conditions does ACKH outperform JKH? Clarifying this would better justify focusing on conditional distributions.
2. Could you clarify in the main text how ACKIP's complexity improves over ACKH --- specifically, why the inducing-point variant is $m$ times faster?
3. In Eq. (6), you introduced random-variable notations: X*, Y|X, Y'|X'. I understand this is for generality, but could you explain what these distributions represent? For example, is X* the empirical marginal of P(X,Y)?

**Ethical Concerns:**

["NO or VERY MINOR ethics concerns only"]

**Final Justification:**

I have read the rebuttal and comments of the other reviewers.
I decide to keep my original score.

**Limitations:**

yes

**Quality:**

4

**Strengths And Weaknesses:**

Strengths:
1. The method is theoretically sound and intuitive.
2. The benefit of modeling $P(Y|X)$ directly—rather than via the joint distribution—is clearly demonstrated through experiments.
3. Empirical results are consistent, and the improvements are substantial.
4. Computational complexity is carefully addressed, and the algorithms appear practical for real‐world use.

Weaknesses:
1. The discussion focuses narrowly on kernel herding; positioning this work relative to other distribution‐compression methods would strengthen the paper.
2. The circumstances under which one should choose ACKH over JKH are shown empirically but lack a more intuitive, conceptual explanation.

---

> ### Author Rebuttal · Authors · 2025-07-29
>
> We thank the reviewer for the time and effort dedicated to reviewing this work and for their positive feedback. In particular, we appreciate the opportunity to clarify the motivation of the work.
>
> ## **Weaknesses**
>
> ### **Weakness 1** - The discussion focuses narrowly on kernel herding; positioning this work relative to other distribution‐compression methods would strengthen the paper.
>
> We thank the reviewer for this suggestion.
>
> **The main contribution of this paper are the algorithms (ACKIP, ACKH) which compress the conditional distribution. We are not aware of any other algorithms in the literature that do this, kernel-based or otherwise**.
>
> In *non-conditional* distribution compression, the most popular algorithms are KH [R1], Kernel Thinning (KT) [R2], Support Points (SP) [R3], and gradient-flow (GF) based methods [R7]. Here we adapt the approach of KH/SP/GF where the compressed set is a free parameter. In KT they instead restrict themselves to points that already exist in the target dataset, as their original motivation was to thin the output of MCMC procedures. KH, KT, and GF are both kernel-based, targeting the MMD, whereas SP is an *energy distance (ED)* based method, where the energy distance is defined as
>
> \begin{align*}
> ED(\mathbb{P}_X, \mathbb{Q}_X) := 2\mathbb{E}[\Vert a - b \Vert] - \mathbb{E}[\Vert a - a^\prime \Vert] + \mathbb{E}[\Vert b - b^\prime \Vert]\
> \end{align*}
>
> with $a, a^\prime \sim \mathbb{P}_X$, $b,b^\prime \sim \mathbb{Q}_X$, and $\Vert \cdot \Vert$ is the 2-norm. The ED is zero iff $\mathbb{P}_X =\mathbb{Q}_X$ (Theorem 1, [R4]). While SP does not initially appear to be kernel-based, the energy distance can be shown to be equivalent to the MMD for a choice of negative-definite kernel function [R4]. Hence the existing non-conditional distribution compression methods can all be thought of as kernel-based.
>
> **We will include this expanded discussion on the existing non-conditional distribution compression methods in the final version.** In particular, we will explicitly compare these methods with our contributions, clarifying their respective scopes and limitations.
>
> ### **Weakness 2** - The circumstances under which one should choose ACKH over JKH are shown empirically but lack a more intuitive, conceptual explanation.
>
> We thank the reviewer for raising this point. In order to understand why compressing the conditional distribution directly, rather than as a by-product of joint distribution compression is more favourable, it may be helpful to make an analogy to conditional density estimation. The naive method for estimating conditional densities $p(y \mid x)$  is to first estimate the joint $p(x, y)$ and marginal $p(x)$ densities, then set
> \begin{align*}
>    \hat{p}(y \mid x) := \frac{\hat{p}(x, y)}{\hat{p}(x)},
> \end{align*}
> however this approach is error propagating, and it is better to *directly* estimate the object of interest, the conditional density, see for example [R6]. This analogy follows our case quite closely, as our empirical results suggest it is far better to compress the conditional distribution directly, rather than to compress the joint distribution.
>
> Moreover, JKH/JKIP and ACKH/ACKIP optimise different loss functions designed for different tasks, so it is not too surprising that compressing via AMCMD (ACKH, ACKIP) outperforms JMMD (JKH, JKIP) empirically.
>
> **We will include additional intuition in the final version of the paper.**
>
> ## **Questions**
>
> ### **Question 1** - What types of real-world data motivate this approach? In particular, under what practical conditions does ACKH outperform JKH? Clarifying this would better justify focusing on conditional distributions.
>
> We thank the reviewer for highlighting the importance of motivating practical use cases.
>
> Our approach is broadly applicable because the kernel conditional mean embedding (KCME) is defined on any space with a positive definite kernel. In this work, we focus on $\mathbb{R}^d$, which is both common in practice and amenable to efficient gradient-based optimisation.
>
> **Our empirical results demonstrate the method’s relevance to both regression and classification tasks**. To see this, note that the kernel conditional mean embedding (KCME) can be directly used as a regression or classification model. In particular we point the reviewers attention to Figure 4. When the test function $h(y)$ is chosen such that $h(y) = y$, the KCME estimates $\mathbb{E}[Y|X]$, which **is the object of interest in regression**. Moreover, for multi-class classification, in Figure 8 we report performance for test functions $h(y) = 1_{y=i}$, $i = 1, 2, 3, 4$. In this case we have that
> $$
>         \mathbb{E}[1_{Y = i} | X] = \mathbb{P}(Y = i | X),
> $$
> and hence Figure 8 reports the error in estimating the class conditional probability, which is the **object of interest in classification**. **Therefore wherever one may wish to do regression or classification, but do not have the availbe compute to handle a large amount of data, one may apply our methods.**
>
> Beyond these tasks, KCMEs have well-established applications (as discussed in the introduction), such as estimating conditional treatment effects in medical data [R5]. All of these applications fundamentally depend on the KCME’s ability to approximate conditional expectations of functions of random variables. Our empirical results show that this property is preserved under compression, with RMSE serving as a natural error metric, suggesting strong downstream performance in diverse domains.
>
> **We will stress this motivation mroe in the final version.**
>
> ### **Question 2** - Could you clarify in the main text how ACKIP's complexity improves over ACKH --- specifically, why the inducing-point variant is $m$ times faster?
>
> Given a dataset of size $n$, we aim to compress this down to $m$ data pairs. The cost of computing a single gradient in ACKIP is $\mathcal{O}(m^3 + m^2n)$, hence if we take $J$ total gradient steps the overall cost is simply $\mathcal{O}((m^3 + m^2n)J)$.
>
> Conversely, ACKH grows the compressed set pair by pair. That is, at the $i$th iteration, the compressed set is of size $i$, and we are solving an optimisation problem by gradient descent where each gradient computation costs $\mathcal{O}(i^3 + i^2n)$. Say we take $T$ steps to solve this optimisation problem, this then gives us a total cost for the i$th$ iteration of $\mathcal{O}((i^3 + i^2n)T)$. However, we want to output a compressed set of size $m$, hence the total cost is actually
> \begin{align*}
>         \sum_{i=1}^m(i^3 + i^2n)T = \mathcal{O}((m^4 + m^3n)T).
> \end{align*}
> See Section D.2.3 for the full derivation.
>
> In words, the additional cost comes from the fact that at each iteration of the algorithm, even though we are only interested in optimising the single new pair, keeping all previously optimised pairs the same, we still have to do an $\mathcal{O}(i^3)$ inverse, as it is not straightforward to separate the contribution of just the new pair to the loss function. Through the method of bordering, it is technically possible to update the inverse of a growing matrix, minimising re-computation. Unfortunately, bordering is a highly numerically unstable procedure, and experimental tests revealed this to be a very unstable approach.
>
> Due to reviewer guidelines, we cannot include figures in this discussion, however the below table shows the wall clock time for constructing a compressed set of size $m = 250$  targeting $n = 8000$ datapoints. **We will include a figure showing size of compressed set versus wall clock time in the final version of the paper.** We report the 25th, 50th and 75th percentiles over 20 runs, the results shows that **ACKIP is much faster than ACKH in practice**.
>
> |Method|Wall Clock Time (s)
> |-|-|
> |||
> |ACKH|294.4, 300.2, 345.8|
> |**ACKIP**| **11.40, 11.42, 11.45**|
>
> ### **Question 3** - In Eq. (6), you introduced random-variable notations: X*, Y|X, Y'|X'. I understand this is for generality, but could you explain what these distributions represent? For example, is X* the empirical marginal of P(X,Y)?
>
> Thank you for raising this. In this equation we introduced an additional distribution P(X*) on the feature space. This distribution is not necessarily related in any way to the existing joint distributions P(X, Y), P(X', Y'). The only connection between these distributions is that P(X*), P(X), and P(X') must share a measurable space, and for the purposes of the triangle inequality, in Theorem 4.1. we require that the relevant Radon-Nikodym derivatives are bounded.
>
> **To get a sense of what these distributions could be, it may be helpful to check the illustrative example in Section C.2 and in particular, Figure 28.**
>
> ## **References**
>
> [R1] - Yutian Chen et al. “Super-Samples from Kernel Herding”. 2012
>
> [R2] - Raaz Dwivedi and Lester Mackey. "Kernel Thinning". 2024
>
> [R3] - Simon Mak and V. Roshan Joseph. "Support points". 2018
>
> [R4] - Dino Sejdinovic et al. “Equivalence of distance-based and RKHS-based statistics in hypothesis testing”. In: The Annals of Statistics 41.5. 2013
>
> [R5] - Junhyung Park, Uri Shalit, Bernhard Schölkopf, Krikamol Muandet. Conditional Distributional Treatment Effect with Kernel Conditional Mean Embeddings and U-Statistic Regression.
>
> [R6] - Rafael Izbicki and Ann B. Lee. “Nonparametric Conditional Density Estimation in a High-Dimensional Regression Setting”. 206.
>
> [R7] - Michael Arbel, Anna Korba, Adil Salim, Arthur Gretton. Maximum Mean Discrepancy Gradient Flow. 2019

---

> ### Author Response · Authors · 2025-08-05
>
> Thank you for acknowledging our rebuttal. If there are any further concerns or questions do not hesitate to raise them. We hope our responses have sufficiently addressed your concerns and may lead to an improved score.

---

### Official Review · Reviewer_ris1 · 2025-07-01

**Clarity:** 3
**Significance:** 4
**Originality:** 3
**Rating:** 4
**Confidence:** 1

**Summary:**

This paper studies the problem of compressing labeled data while preserving the properties of the original distribution. The authors extend concepts in Kernel Herding, which is used for compressing unlabeled data. They also consider joint distribution where they first extend Kernel Herding to Joint Kernel Herding (JKH) and also develop a new algorithm called Joint Kernel Inducing Points (JKIP). They introduce a new metric called Average Maximum Conditional Mean Discrepancy (AMCMD) and derive a closed-form formula for it. They extend the Kernel Herding algorithm to Average Conditional Kernel Herding (ACKH). They also develop a new algorithm called Average Conditional Inducing Points (ACKIP) which, instead of sequentially constructing the compression set, works with sets of points and iteratively updates the set via gradient descent. The point-based algorithm move away from greedy optimization that sequentially constructs a set and they turn out to be more efficient. The proposed algorithms are evaluated through comprehensive set of experiments.

**Questions:**

- Why are they 20 points (instead of 25) in the Figure 1 for ACKIP, do we have points that are repeated?
- Why do the point-based algorithm converge? what is their convergence time? Also what is the effect of initialization step in these methods?
- What is the runtime of the algorithms? Can you add plots or simply report the runtime?
- Are they non-Kernel methods are used for compression? how do they perform? How do they compare to the proposed algorithms?

**Ethical Concerns:**

["NO or VERY MINOR ethics concerns only"]

**Final Justification:**

This is a solid result with comprehensive experiment set. Authors resolved my concern about comparison with non-kernel based methods. My final recommendation is accept.

**Limitations:**

yes

**Paper Formatting Concerns:**

No concerns

**Quality:**

3

**Strengths And Weaknesses:**

Strength:
- paper studies a fundamental problem which has been considered before and so many of the properties have been explored before however they gather all these findings from various resources and present a comprehensive study of previous work.
- Introduction of new way of measuring the discrepancy between original and compressed distribution. Showing that it is metric and providing a closed-formula for it.
- Proposing new algorithms with better runtime and measuring their performance in practice.
- Strong empirical evidence of superiority of the proposed algorithms.

Weakness:
-  There are no discussion in terms of convergence of the point-based methods? Similarly for initialization, they mention uniform sampling but no other discussion.
- Samples that are necessarily a part of the original training set  but appear in compression set are not thoroughly discussed, e.g., what is conditional property for these samples? how are they handled generally?


Minor:
- Typo Eq(2) l(y,.) instead of k(y,.) in the sum.
- l248 add space

---

> ### Author Rebuttal · Authors · 2025-07-29
>
> We thank the reviewer for their positive and thoughtful feedback, and we understand their cautious confidence score. To clarify, our work focuses on **conditional distribution compression, a problem which, to the best of our knowledge, has not been studied before.** We hope the additional clarifications provided here help strengthen the reviewer’s confidence in their understanding of the contributions and novelty of our work.
>
> ## **Weaknesses**
>
> ### **Weakness 1.1** - There are no discussion in terms of convergence of the point-based methods?
>
> We thank the reviewer for raising this important point. While convergence rates and sample error bounds are important, they are not our main interest, which is instead to, for the first time, establish the feasibility and practical benefit of conditional distribution compression. Our focus in this work is on algorithmic formulation and empirical validation, rather than asymptotic analysis. In particular, **the novelty lies in the adaptation to the problem of conditional distribution compression, which has not been addressed previously**.
>
> **Our extensive empirical analysis demonstrates that the proposed algorithms are robust**, consistently delivering strong performance across a diverse range of datasets—both synthetic and real—and in both regression and classification settings. Importantly, given any random sample, our algorithms can be initialised with that sample and, with standard gradient descent tuning, produce a set that achieves a lower AMCMD.
>
> ### **Weakness 1.2** - Similarly for initialization, they mention uniform sampling but no other discussion.
>
> Regarding initialisation, while uniform sampling is a natural and widely used choice in distribution compression (e.g., Kernel Herding, Support Points [R1, R2]), we agree that alternative initialisation strategies could be explored. **We will include a brief discussion in the revision highlighting that initialisation is flexible**, and more sophisticated methods can be readily incorporated into our framework. However, we note that the **provided results are given with accompanying 25th and 75th percentiles which suggest that the performance of the algorithm is stable with respect to the choice of random initialisation.**
>
> ### **Weakness 2** - Samples that are necessarily a part of the original training set but appear in compression set are not thoroughly discussed, e.g., what is conditional property for these samples? how are they handled generally?
>
> We appreciate the reviewer highlighting this technical point. The compressed sets produced by our algorithms are the result of gradient-based optimisation, where the compressed set is treated as a parameter and optimised directly with respect to the AMCMD/JMMD objectives. As a result, it is **extremely unlikely that points from the original training set would be selected verbatim for the compressed set**.
>
> Nevertheless, the kernel conditional mean embedding is well defined for any choice of $\boldsymbol{x} \in \mathcal{X}$, $\boldsymbol{y} \in \mathcal{Y}$, even those in the set used to train it, and hence **if training points did appear in the compressed set there would be no issue.** For further discussion on the properties of the kernel conditional mean embedding, see [R3].
>
> ### **Minor Points**
> We thank the reviewer for pointing out these typos.
>
> ## **Questions**
>
> ### **Question 1** - Why are they 20 points (instead of 25) in the Figure 1 for ACKIP, do we have points that are repeated?
>
> We thank the reviewer for catching this, as it raises an interesting technical point. In Figure 1, some points have been optimised such that they appear to overlap visually. However, they are not identical; rather, they lie close together in the data space and only appear to coincide due to the marker size used in the plot.
>
> This clustering of points can be interpreted as a form of weighting, indicating that certain regions of the data space are particularly important for conditional distribution compression. As a result, these regions are represented more heavily in the compressed set—something that may not be captured by random sampling.
>
> ### **Question 2.1** - Why do the point-based algorithm converge?
>
> Our algorithms optimise differentiable loss functions (e.g., AMCMD, JMMD) and therefore inherit the well-established convergence behaviour of gradient-based optimisation. In practice, we observed that the algorithms required minimal hyperparameter tuning to achieve convergence. Furthermore, our empirical analysis shows that the proposed methods are robust, consistently converging across a wide range of datasets—both synthetic and real—and in both regression and classification tasks.
>
> ### **Question 2.2** - What is their convergence time?
>
> In practice, we observe rapid convergence, often within a small number of iterations, as shown in our experimental results where the loss stabilises quickly across datasets. Providing theoretical convergence rates may be possible for specific kernels, and choice of generating distribution, but is beyond the scope of this paper.
>
> ### **Question 2.3** - Also what is the effect of initialization step in these methods?
>
> We use uniform random sampling as a simple and widely adopted starting point (e.g., kernel herding [R1]). Our empirical results show that even with this simple initialisation, the algorithm consistently outperforms the initial random sample, demonstrating robustness. We note that alternative initialisations (e.g., Gaussian initialisation, data-dependent seeding) can be easily incorporated and may further improve convergence speed, which we will briefly discuss in the revised paper.
>
> ### **Question 3** - What is the runtime of the algorithms? Can you add plots or simply report the runtime?
>
> The time complexity of the algorithms developed in this work is summarised in the following table which we will include in the final version:
>
> | Algorithm | Time Complexity  |
> |-----------|--------------------------------------------------|
> | JKH       | $O((m^2 + mn)(d+p))$                             |
> | JKIP      | $O((m^2 + mn)(d+p))$                             |
> | ACKH      | $O(m^4 + m^3n + (m + n)(d+p))$                   |
> | ACKIP     | $O(m^3 + m^2n + (m^2 + mn)(d+p))$                |
>
> Due to reviewer guidelines, we cannot include figures in this discussion, however the below table shows the wall clock time for constructing a compressed set of size $m = 250$  targeting $n = 8000$ datapoints for each of the methods introduced in this work. **We will include a figure showing size of compressed set versus wall clock time in the final version of the paper.**
>
> We report the 25th, 50th and 75th percentiles over 20 runs, the results shows that **JKIP is faster than JKH in practice**, and **ACKIP is much faster than ACKH in practice**.
>
> |Method|Wall Clock Time (s)
> |-|-|
> |||
> |JKH|8.671, 8.729, 8.996|
> |**JKIP**|**0.8476, 0.8731, 0.8773**|
> |ACKH|294.4, 300.2, 345.8|
> |**ACKIP**| **11.40, 11.42, 11.45**|
>
> The reason for JKIP being faster than JKH is fairly techincal. As noted in the appendix, the algorithms in this paper were implemented using JAX. JAX enables Just-In-Time (JIT) compilation, which significantly increases execution speed. However, to achieve this speed, JAX relies on an immutable array structure, meaning the arrays must not change shape during program execution. As a result, JKH and ACKH cannot fully leverage the speed benefits of JIT compilation, as the size of the compressed set, and hence the corresponding arrays, increases at every iteration. Conversely, the arrays considered in JKIP and ACKIP stay the same shape throughout. This presents a notable practical advantage of JKIP and ACKIP over JKH and ACKH.
>
> ### **Question 4** - Are they non-Kernel methods are used for compression? how do they perform? How do they compare to the proposed algorithms?
>
> **The main contribution of this paper are the algorithms (ACKIP, ACKH) which compress the conditional distribution. We are not aware of any other algorithms in the literature that do this, kernel-based or otherwise**. To provide a baseline comparison, we modify Kernel Herding (KH) [R1] to target the joint distribution (JKH), and introduce a gradient-flow based [R6] joint distribution compression algorithm (JKIP). We show ACKIP achieving superior empirical performance versus these baselines.
>
> In *non-conditional* distribution compression, the most popular algorithms are KH [R1], Kernel Thinning (KT) [R2], Support Points (SP) [R4] and gradient flow (GF) based approaches [R6]. Here we adapt the approach of KH/SP/GF where the compressed set is a free parameter. In KT they restrict themselves to points that already exist in the target dataset. KH, KT and GF are both kernel-based, targeting the MMD, whereas SP is an *energy distance (ED)* based method, defined as
>
> \begin{align*}
> ED(\mathbb{P}_X, \mathbb{Q}_X) := 2\mathbb{E}[\Vert a - b \Vert] - \mathbb{E}[\Vert a - a^\prime \Vert] + \mathbb{E}[\Vert b - b^\prime \Vert]\
> \end{align*}
>
> where $a, a^\prime \sim \mathbb{P}_X$, $b,b^\prime \sim \mathbb{Q}_X$, and $\Vert \cdot \Vert$ is the 2-norm. The ED is zero iff $\mathbb{P}_X =\mathbb{Q}_X$ (Theorem 1, [R5]). While SP does not initially appear to be kernel-based, the energy distance is equivalent to the MMD for a choice of negative-definite kernel function [R5].
>
> ## **References**
>
> [R1] - Yutian Chen et al. “Super-Samples from Kernel Herding”. 2012
>
> [R2] - Raaz Dwivedi and Lester Mackey. "Kernel Thinning". 2024
>
> [R3] - Junhyung Park, Krikamol Muandet. A Measure-Theoretic Approach to Kernel Conditional Mean Embeddings. 2020
>
> [R4] - Simon Mak and V. Roshan Joseph. "Support points". 2018
>
> [R5] - Dino Sejdinovic et al. “Equivalence of distance-based and RKHS-based statistics in hypothesis testing”. In: The Annals of Statistics 41.5. 2013
>
> [R6] - Michael Arbel, Anna Korba, Adil Salim, Arthur Gretton. Maximum Mean Discrepancy Gradient Flow. 2019

---

> > ### Comment · Reviewer_ris1 · 2025-08-05
> > **Re rebuttal**
> >
> > Thanks for addressing my questions and concerns!

---

> > > ### Author Response · Authors · 2025-08-05
> > >
> > > You're welcome! We hope our responses have sufficiently addressed your concerns and may lead to an improved score.

---

### Official Review · Reviewer_r3Jq · 2025-07-02

**Clarity:** 2
**Significance:** 2
**Originality:** 2
**Rating:** 4
**Confidence:** 3

**Summary:**

This paper addresses conditional distribution compression for labeled data, a problem distinct from existing joint distribution compression approaches. The authors introduce the Average Maximum Conditional Mean Discrepancy (AMCMD) as a novel metric for comparing families of conditional distributions, proving it satisfies standard metric properties (triangle inequality, identity of indiscernibles). Leveraging this metric, they propose two algorithms: (1) Average Conditional Kernel Herding (ACKH), a greedy linear-time algorithm, and (2) Average Conditional Kernel Inducing Points (ACKIP), a joint optimization alternative. For comparison, they also develop joint distribution compression methods: Joint Kernel Herding (JKH) and Joint Kernel Inducing Points (JKIP). A key theoretical contribution is reducing AMCMD estimation complexity from O(n³) to O(n) via the tower property (Lemma 4.7). Experiments across synthetic and real datasets demonstrate that (1) conditional compression outperforms joint compression, and (2) joint optimization consistently outperforms greedy approaches, with ACKIP achieving near full-data performance using only 3% of the original dataset.

**Questions:**

- Since JKH and JKIP both have the same complexity of O(mn + m^2), do you observe any empirical speed difference between the two? Is there any scenario that the greedy one is preferred?
- Is there a mistake in equation (2) - should be $k(x, \tilde{x_j})l(y, \tilde{y}_j)$ instead of $k(x, \tilde{x_j})k(y, \tilde{y}_j)$

**Ethical Concerns:**

["NO or VERY MINOR ethics concerns only"]

**Final Justification:**

The authors have addressed most of my concerns during the rebuttal period. Although I still have some concerns about the general applicability of the method, I have decided to raise my score from 3 to 4 given the technical soundness and the potential of this method.

**Limitations:**

yes

**Paper Formatting Concerns:**

No major formatting issues

**Quality:**

3

**Strengths And Weaknesses:**

### Strength

- The paper has considered both the joint and conditional kernel herding and uses experiments to prove the effectiveness of doing conditional distribution compression, rather than the joint one.
- This paper is the first direct approach to conditional distribution compression, addressing a genuine gap in the literature where existing methods compress joint distributions and hope conditional structure is preserved.
- The paper presents a clear logical progression, going from joint to conditional and from greedy to the joint optimization.
- The paper also addresses a computational bottleneck of estimating AMCMD from O(n^3) to O(n).
- Demonstrates substantial compression ratios (97% data reduction) while maintaining statistical performance, with potential to significantly expand KCME applicability.

### Weakness

- The experiments are a bit weak in terms of the downstream tasks or practical guidance. For example, the Kernel Herding paper[3]’s experiments setting that this paper claims to follow also includes the Bayesian Posterior.
- Experiments also lack the comparison to other baseline methods.
- The paper claims AMCMD is "more general" than existing metrics (KCD/AMMD) but provides no compelling use cases requiring this generality
- The writing of the paper requires substantial work. There is dense notation in Section 4 with insufficient motivation for theoretical developments
- Memory constraints limit mnIST to 10K samples, reducing confidence in large-scale applicability

---

> ### Author Rebuttal · Authors · 2025-07-29
>
> We thank the reviewer for the time and effort dedicated to reviewing this work and for their valuable feedback. In particular we appreciate the reviewer pointing out the lack of clarity in our argument for the increased generality offered by AMCMD.
>
> ## **Weaknesses**
>
> ### **Weakness 1** - The experiments are a bit weak in terms of the downstream tasks or practical guidance...
>
> We respectfully disagree with the comments on downstream tasks.
>
> The KCME has many valuable downstream applications, as discussed in the introduction. **However, crucially, these all rely on the KCME’s ability to approximate conditional expectations of functions of random variables. Our empirical results demonstrate that this capability is preserved under compression**, with the RMSE being a natural choice of error metric. These results suggests strong downstream performance in other applications.
>
> **Moreover, the empirical results presented in the paper cover both regression and classification as downstream tasks.** To see this, note that the kernel conditional mean embedding (KCME) can be directly used as a regression or classification model. In particular we point the reviewers attention to Figure 4. When the test function $h(y)$ is chosen such that $h(y) = y$, the KCME estimates $\mathbb{E}[Y|X]$, which **is the object of interest in regression**. Moreover, for multi-class classification, in Figure 8 we report performance for test functions $h(y) = 1_{y=i}$, $i = 1, 2, 3, 4$. In this case we have that
> $$
>         \mathbb{E}[1_{Y = i} | X] = \mathbb{P}(Y = i | X),
> $$
> and hence Figure 8 reports the error in estimating the class conditional probability, which is the **object of interest in classification**. In Figure 20, we also report the classification accuracy and F1 score achieved by each of the methods, with ACKIP achieving parity with the full data after 97% compression.
>
> ### **Weakness 2** - Experiments also lack the comparison to other baseline methods.
>
> **The main contribution of this paper are the algorithms (ACKIP, ACKH) which compress the conditional distribution. We are not aware of any other algorithms in the literature that do this, kernel-based or otherwise**. To provide a baseline comparison, we modify Kernel Herding (KH) [R1] to target the joint distribution (JKH), and introduce a gradient flow based [R4] joint distribution compression algorithm (JKIP). We also provide comparison to uniform random subsampling which is used frequently throughout the standard distribution compression literature [R1, R2]. We show ACKIP achieving superior empirical performance versus these baselines.
>
> ### **Weakness 3** - The paper claims AMCMD is "more general" than existing metrics (KCD/AMMD) but provides no compelling use cases requiring this generality.
>
> In the main body of the text we point towards Section C.2 in the appendix which includes an experiment illustrating the inflexibility of the KCD/AMMD compared to the AMCMD. Specifically, **the KCD/AMMD fails to detect similarity in the conditional distributions of two datasets within particular regions of the conditioning space, only detecting global similarity.** This stems from its requirement that both the distribution of the conditioning variable and the expectation measure be identical, making it unsuitable for such scenarios.
>
> **We agree that this was not obvious in the original paper as we did not have space to expand upon this. With additional space we will rectify this in the final version.**
>
> **Moreover, in Section A.4. we expand upon the possible use cases of the AMCMD that are out of scope for the work in this paper. These applications include the widely studied areas of Covariate and Conditional Shift, which would not be possible using the KCD/AMMD. We will include further discussion on this topic in the main body for the final version.**
>
> ### **Weakness 4** - The writing of the paper requires substantial work. There is dense notation in Section 4 with insufficient motivation for theoretical developments.
>
> We appreciate the reviewer’s feedback regarding the clarity of Section 4. We acknowledge that this section is notation-heavy due to the mathematical nature of our theoretical contributions. However, the notation we have used throughout the section is standard in the KCME [R3] and distribution compression literature [R1, R2].
>
> ### **Weakness 5** - Memory constraints limit mnIST to 10K samples, reducing confidence in large-scale applicability.
>
> The below table shows the memory complexity for each of the methods introduced in this work, assuming the feature space is $d$-dimensional, the response space is $p$-dimensional, the target dataset has size $n$, and the compressed set is size $m$:
>
> | Algorithm | Memory Complexity |
> |-----------|--------------------------------------------------|
> | JKH      | $O((m + n)(d+p))$|
> | JKIP      | $O((m^2 + mn)(d+p))$|
> | ACKH      | $O((m + n)(d+p) + mn + m^2)$ |
> | ACKIP     | $O((m^2 + mn)(d+p))$
>
> One should notice that the **dependence on** $n$ **and** $d, p$ **is never more than $O(nm(d+p))$** for any of the methods in this work, which is quite favourable. **We will include this discussion in the final version**.
>
> **We restrict the size of the data to 10K samples due to a lack of access to GPUs with high VRAM; the experiments were run locally with just 12GB of VRAM. However we have seen above that the dependence upon dimensionality is linear, and so this could be easily expanded with access to more compute.**
>
> ## **Questions**
>
> ### **Question 1** - Since JKH and JKIP both have the same complexity of $O(mn + m^2)$, do you observe any empirical speed difference between the two? Is there any scenario that the greedy one is preferred?
>
> Due to reviewer guidelines, we cannot include figures in this discussion, however the below table shows the wall clock time for constructing a compressed set of size $m = 250$  targeting $n = 8000$ datapoints for each of the methods introduced in this work. **We will include a figure showing size of compressed set versus wall clock time in the final version of the paper.**
>
> We report the 25th, 50th and 75th percentiles over 20 runs, the results shows that **JKIP is faster than JKH in practice**, and **ACKIP is much faster than ACKH in practice**.
>
> |Method|Wall Clock Time (s)
> |-|-|
> |||
> |JKH|8.671, 8.729, 8.996|
> |**JKIP**|**0.8476, 0.8731, 0.8773**|
> |ACKH|294.4, 300.2, 345.8|
> |**ACKIP**| **11.40, 11.42, 11.45**|
>
> The reason for JKIP being faster than JKH is fairly techincal. As noted in the appendix, the algorithms in this paper were implemented using JAX. JAX enables Just-In-Time (JIT) compilation, which significantly increases execution speed. However, to achieve this speed, JAX relies on an immutable array structure, meaning the arrays must not change shape during program execution. As a result, JKH and ACKH cannot fully leverage the speed benefits of JIT compilation, as the size of the compressed set, and hence the corresponding arrays, increases at every iteration. Conversely, the arrays considered in JKIP and ACKIP stay the same shape throughout. This presents a notable practical advantage of JKIP and ACKIP over JKH and ACKH.
>
> **JKH could, in principle, be faster than JKIP if optimising each individual pair requires far fewer iterations than optimising the entire compressed set jointly. However, this was not observed in any of our experiments.**
>
> ### **Question 2** - Is there a mistake in equation (2)...
>
> Yes, thank you for pointing this out, we will fix this in the final version.
>
> ## **References**
>
> [R1] - Yutian Chen et al. “Super-Samples from Kernel Herding”. 2012
>
> [R2] - Raaz Dwivedi and Lester Mackey. "Kernel Thinning". 2024
>
> [R3] - Junhyung Park, Krikamol Muandet. A Measure-Theoretic Approach to Kernel Conditional Mean Embeddings. 2020
>
> [R4] - Michael Arbel, Anna Korba, Adil Salim, Arthur Gretton. Maximum Mean Discrepancy Gradient Flow. 2019

---

> ### Comment · Reviewer_r3Jq · 2025-08-04
>
> Thank you for your detailed rebuttal. I appreciate your responses to the concerns raised. After considering your explanations, I have the following thoughts:
>
> Regarding Weakness 1 (Downstream Tasks):
> Your clarification about Figures 4, 8, and 20 demonstrating regression and classification applications is helpful. However, my concern was more about the diversity and depth of downstream evaluations rather than their existence. The kernel herding literature typically includes more varied applications (e.g., Bayesian posterior approximation, which you mentioned but didn't include). While RMSE preservation is important, showing performance on established ML benchmarks, especially on high-dimensional data, would strengthen confidence in practical applicability.
>
> Regarding Weakness 2 (Baseline Comparisons):
> I appreciate that you created JKH and JKIP as reasonable baselines since no direct conditional compression methods exist. This is a fair approach given the novelty of the problem.
>
> Regarding Weakness 3 (Generality Claims):
> Your reference to Section C.2 is valuable, and I'm glad you'll expand this in the main text. The covariate/conditional shift applications you mention in A.4 do represent compelling use cases that would benefit from being more prominent in the paper.
>
> Regarding Weakness 5 (Memory Constraints):
> The memory complexity analysis is very helpful. The linear dependence on dimensionality is indeed favorable, and your hardware limitation explanation is reasonable.

---

> ### Author Response · Authors · 2025-08-04
>
> Thank you for taking the time to carefully read our rebuttal and for acknowledging our responses to Weaknesses 2, 3 and 5. We greatly appreciate the thoughtful feedback and constructive nature of your comments.
>
> We agree that exploring additional downstream tasks would be valuable for future work. However, we consider the most important downstream tasks, namely regression and classification, to already be included and thoroughly evaluated. Beyond these tasks, KCMEs have other well-established applications (as discussed in the introduction), but all of them fundamentally depend on the KCME’s ability to approximate conditional expectations of functions of random variables. Our empirical results demonstrate that this property is preserved under compression, with RMSE serving as a natural error metric. Consequently, we expect our method to perform well in further downstream applications, which we see as an interesting avenue for future work.

---

### Official Review · Reviewer_YpGJ · 2025-07-03

**Clarity:** 2
**Significance:** 3
**Originality:** 2
**Rating:** 4
**Confidence:** 3

**Summary:**

Existing methods like Kernel Herding (KH) were originally designed to compress unlabelled data distributions, but no prior approach directly compresses the conditional distribution of labelled data. To fill this gap, the paper first introduce Average Maximum Conditional Mean Discrepancy (AMCMD), a metric for comparing conditional distributions, along with a closed-form estimator. Then show that constructing a compressed set targeting AMCMD can be made much more efficient—reducing the cost from O(n³) to O(n). Based on this insight, we extend KH and propose Average Conditional Kernel Herding (ACKH), a linear-time greedy algorithm for compressing conditional distributions.

To compare with joint distribution compression, the paper also introduce Joint Kernel Herding (JKH), a straightforward extension of KH for joint data. While herding methods are simple and interpretable, they rely on greedy heuristics.

To explore better alternatives, the paper  propose Joint Kernel Inducing Points (JKIP) and Average Conditional Kernel Inducing Points (ACKIP)—which jointly optimize the compressed set while still maintaining linear complexity.

**Questions:**

see weakness.

**Ethical Concerns:**

["NO or VERY MINOR ethics concerns only"]

**Final Justification:**

The authors have addressed most of my concerns during the rebuttal period. I have decided to raise my score from 3 to 4, given that the method have been successfully extended existing tools to a new application in the conditional distribution setting.

**Limitations:**

yes

**Paper Formatting Concerns:**

No obvious issue

**Quality:**

2

**Strengths And Weaknesses:**

Strengths:
1. The paper extends existing kernel compression techniques to the conditional distribution setting and proposes a linear-time optimization approach based on gradient descent. I did not identify any major issues regarding the correctness of the method.
2. Experimental results under various data distribution settings demonstrate that the proposed method consistently achieves lower RMSE.

Weaknesses:
1. Writing and Structure:
The writing of the paper needs improvement. While several method variants are introduced, including some mentioned in the abstract, it is difficult to follow the key logical connections between them. The discussion of related work is insufficient in the main text; although some references are included in the appendix, this weakens the overall motivation of the paper. For example, it is unclear why existing methods perform poorly and why compressing the joint distribution is less effective than compressing the conditional distribution. I recommend adding a dedicated section to clearly outline the main contributions and structure of the paper. In addition, many important details require flipping back and forth between the main text and the appendix, which disrupts the flow of reading.

2. Theoretical Contribution:
The theoretical novelty is relatively weak. The proposed criteria are derived based on well-studied conditional kernel mean embedding estimators, and the derivation mainly relies on direct computation. The paper lacks convergence analysis or sample error bounds, which are essential for establishing the robustness of the method.

3. Experimental Evaluation:
The experimental section lacks clarity in dataset presentation. For example, in line 255, one dataset is briefly mentioned without any description of its features or relevance. Additionally, the meaning of the x-axis in the figures is not explained, making the visualizations less informative.

4. Experimental Limitations:
Most experiments rely on the Gaussian kernel and are conducted on relatively low-dimensional variables, which may limit the generality of the conclusions. Furthermore, the evaluation metric is solely based on RMSE. While RMSE is appropriate for regression tasks, the goal of distribution compression is often to support downstream tasks more broadly. Therefore, including results on a wider range of downstream applications would make the evaluation more convincing and practical.

minor: A few minor points:
1. The notation for the separable measure space in lines 85 and 86 is inconsistent.
2. Why is it m+1​ in Equation (2) instead of m?
3. In line 219, two matrix dimensions appear to be inconsistent.
4. line 248: Figures 2 and3 ->Figures 2 and 3

---

> ### Author Rebuttal · Authors · 2025-07-28
>
> We thank the reviewer for the time and effort dedicated to reviewing this work and for their valuable feedback. In particular, we appreciate the constructive comments on the paper’s writing and structure.
>
> ## **Weaknesses**
>
> ### **Weakness 1.1** - The writing of the paper needs improvement. While several method variants are introduced ... it is difficult to follow the key logical connections between them. I recommend adding a dedicated section to outline the main contributions and structure of the paper.
>
> **In the introduction, we provide bullet points outlining the main contributions of the paper. We recognise, however, that introducing four distinct algorithms may make it difficult to see their logical connections.**
>
> The central goal of our work is to compress the conditional distribution, for which no current approach is available. To this end, we extend Kernel Herding (KH)—originally designed for marginal distributions—to the conditional setting, resulting in **ACKH**. To provide a fair baseline comparison, we also extend KH to target the joint distribution, yielding **JKH**.
>
> KH relies on a greedy, point-by-point selection heuristic, which can produce suboptimal solutions. To address this limitation, we introduce **ACKIP** and **JKIP**, which target the conditional and joint distributions respectively, but **optimise all points in the compressed set jointly rather than greedily**. Our experiments show that this leads to improved performance.
>
> **We will revise the introduction in the final version to make these connections and contributions clearer to the reader.**
>
> ### **Weakness 1.2** - The discussion of related work is insufficient in the main text; although some references are included in the appendix, this weakens the overall motivation of the paper.
>
> We agree that additional discussion of related work in the main text will strengthen the motivation of the paper. Unfortunately, space constraints meant this section was moved to the appendix. **We will include a related work section in the main body of the final paper with the additional space provided.**
>
> ### **Weakness 1.3** - It is unclear why existing methods perform poorly and why compressing the joint distribution is less effective than compressing the conditional distribution.
>
> **We stress that there are no existing methods that perform conditional distribution compression**, this work is the first to develop a conditional distribution compression procedure.
>
> In order to get an **inutitive sense** for why compressing the conditional distribution directly, rather than as a by-product of joint distribution compression is preferred, it may be helpful to make an **analogy to conditional density estimation**. The naive method for estimating conditional densities $p(y | x)$  is to first estimate the joint $p(x, y)$ and marginal $p(x)$ densities, then set
> $$
>  \hat{p}(y | x) := \frac{\hat{p}(x, y)}{\hat{p}(x)},
> $$
> however this approach is error propagating. It has been shown that stronger empirical performance is found when *directly* estimating the conditional density, see for example [R3]. Similarly, our empirical results suggest it is better to compress the conditional distribution directly.
>
> Moreover, JKH/JKIP and ACKH/ACKIP optimise different loss functions designed for different tasks, so it is unsurprising that compressing via AMCMD (ACKH, ACKIP) outperforms JMMD (JKH, JKIP) empirically.
>
> **We will include discussion around this in the introduction of the work to improve clarity.**
>
> ### **Weakness 2** - The theoretical novelty is relatively weak. The proposed criteria are derived based on well-studied conditional kernel mean embedding estimators ... The paper lacks convergence analysis or sample error bounds, which are essential for establishing the robustness of the method.
>
> While our proposed criteria build on the well-studied framework of kernel conditional mean embeddings, **the novelty lies in their adaptation to the problem of conditional distribution compression, which has not been addressed previously**. While convergence rates and sample error bounds are important, they are not our main interest, which is instead to establish the feasibility and practical benefit of conditional distribution compression. Our focus in this work is on algorithmic formulation and empirical validation, rather than asymptotic analysis.
>
> **Our extensive empirical analysis demonstrates that the proposed algorithms are robust**, consistently delivering strong performance across a diverse range of datasets—both synthetic and real—and in both regression and classification settings. Importantly, given any random sample, our algorithms can be initialised with that sample and, with standard gradient descent tuning, produce a set that achieves a lower AMCMD.
>
> ### **Weakness 3** - The experimental section lacks clarity in dataset presentation. For example, in line 255, one dataset is briefly mentioned without any description of its features or relevance. Additionally, the meaning of the x-axis in the figures is not explained, making the visualizations less informative.
>
> **We respectfully clarify that this is a misunderstanding.** Line 255 only introduces Section 5.2; for all datasets used, we consistently describe their features and explain their relevance in the main text before pointing to figures presenting results. See for example Section 5.2.1.
>
> The figure at the top of page 7 is related to the *Superconductivity* dataset, which may have caused confusion. To avoid any possible ambiguity, **we will revise the layout and figure captions in the final version to make these associations between datasets, descriptions, and figures clearer**.
>
> **Additionally, we thank the reviewer for pointing out that the x-axis of some figures are unlabelled, we will rectify this in the final version.**
>
> ### **Weakness 4.1** - Most experiments rely on the Gaussian kernel ...
>
> In our experiments, we use the Gaussian kernel, as it is a universally standard choice in the kernel embedding literature [R1 , R2], and in kernel-based methods more broadly [R4]. **However, we emphasise that our algorithms are kernel-agnostic**: *any* feature kernel $k$ and *any characteristic* response kernel $l$ can be used, provided their gradients are computable.
>
> The below table includes **new results** on the *Heteroscedastic* data using the **IMQ kernel** for $k$ and $l$. We report the 25th, 50th and 75th percentiles, with ACKIP retaining the best performance in terms of AMCMD and RMSE.
>
> |Method|AMCMD
> |-|-|
> |||
> |JKH|0.0349, 0.0422, 0.0502|
> |JKIP|0.0168, 0.0215, 0.0259|
> |ACKH|0.00443, 0.00697, 0.0191|
> |**ACKIP**| **0.00143, 0.00154, 0.00170**|
> |Random|0.0182, 0.0240, , 0.0336|
>
> | Method | **RMSE**||||
> |-|-|-|-|-|
> ||$h(y) = y$|$h(y) =\sin(y)$|$h(y) =\exp(-y^2)$|$h(y) =1_{\{y < 0\}}$|
> |JKH|0.206, 0.229, 0.266|0.161, 0.170, 0.200| 0.0913, 0.104, 0.115|0.136, 0.168, 0.205|
> |JKIP|0.127, 0.144, 0.218|0.156, 0.157, 0.219| 0.0887, 0.0899, 0.111|**0.116, 0.129**, 0.231|
> |ACKH|0.0892, 0.113, 0.192|0.143, 0.148, 0.150| 0.0882, 0.0918, 0.0988|0.132, 0.144, 0.150|
> |ACKIP|**0.0596, 0.0602, 0.0615**| **0.136, 0.137, 0.138**| **0.0874, 0.0878, 0.0886**|0.127, 0.134, **0.146**|
> |Random|0.134, 0.158, 0.199|0.158,  0.171, 0.193|0.102, 0.114, 0.131|0.138, 0.156, 0.187|
>
> Due to reviewer guidelines we cannot include figures, but **in the final version we will include plots of these new results**.
>
> ### **Weakness 4.2** - Most experiments ... are conducted on relatively low-dimensional variables, which may limit the generality of the conclusions.
>
> Our experiments cover datasets across a large range of dimensions. In the appendix we performed an experiment on mnIST data, which has **moderately high dimension of 784**. In particular, Figures 25 and 26 show that ACKIP achieves very similar classification accuracy to the full data, after 97\% compression.
>
> ### **Weakness 4.3** - The evaluation metric is solely based on RMSE. While RMSE is appropriate for regression tasks, the goal of distribution compression is often to support downstream tasks more broadly.
>
> We respectfully disagree with the comments on downstream tasks.
>
> The KCME has many valuable downstream applications, as discussed in the introduction. **However, crucially, these all rely on the KCME’s ability to approximate conditional expectations of functions of random variables. Our empirical results demonstrate that this capability is preserved under compression**, with the RMSE being a natural choice of error metric. These results suggests strong downstream performance in other applications.
>
> **Moreover, the empirical results presented in the paper cover both regression and classification as downstream tasks.** To see this, note that the kernel conditional mean embedding (KCME) can be directly used as a regression or classification model. In particular we point the reviewers attention to Figure 4. When the test function $h(y)$ is chosen such that $h(y) = y$, the KCME estimates $\mathbb{E}[Y|X]$, which **is the object of interest in regression**. Moreover, for multi-class classification, in Figure 8 we report performance for test functions $h(y) = 1_{y=i}$, $i = 1, 2, 3, 4$. In this case we have that
> $$
>         \mathbb{E}[1_{Y = i} | X] = \mathbb{P}(Y = i | X),
> $$
> and hence Figure 8 reports the error in estimating the class conditional probability, which is the **object of interest in classification**. In Figure 20, we also report the classification accuracy and F1 score achieved by each of the methods, with ACKIP achieving parity with the full data after 97% compression.
>
> ## **References**
>
> [R1] - Yutian Chen et al. “Super-Samples from Kernel Herding”. 2012
>
> [R2] - Raaz Dwivedi and Lester Mackey. "Kernel Thinning". 2024
>
> [R3] - Rafael Izbicki and Ann B. Lee. “Nonparametric Conditional Density Estimation in a High-Dimensional Regression Setting”. 206.
>
> [R4] - Thomas Hofmann et al. Kernel methods in machine learning. 2007

---

> > ### Comment · Reviewer_YpGJ · 2025-08-05
> >
> > **Regarding Weaknesses 1 & 3 (Writing Clarity and Structural Organization):**
> > Your response addresses my concerns about clarity and structure. I appreciate your plan to revise the introduction to better highlight the main contributions and clarify the relationships among the proposed algorithms.
> >
> > **Regarding Weakness 2 (Novelty):**
> > We accept your assessment that, while the proposed criteria build on established tools, extending them to the task of conditional distribution compression represents a meaningful and fair innovation. The creation of adapted baselines (JKH and JKIP) in the absence of direct alternatives is justified, and the formulation of new algorithms for this novel setting provides value.
> >
> > **Regarding Weakness 4 (Evaluation Metrics and Practical Impact):**
> > We remain somewhat cautious about relying solely on RKHS-based compression metrics to argue for practical effectiveness. While we understand the theoretical grounding in kernel conditional mean embeddings, we believe stronger empirical validation—particularly in deep learning pipelines—would further strengthen the claims. We encourage the addition of more diverse downstream evaluations, including tasks where neural network components are used, to better demonstrate the method’s applicability in modern ML contexts.
> >
> > Overall, most of my concerns have been adequately addressed through the authors’ clarifications. I appreciate the thoughtful responses and improvements. I will increase my score accordingly.

---

> > > ### Author Response · Authors · 2025-08-05
> > >
> > > Thank you for taking the time to carefully read our rebuttal and for acknowledging our responses to Weaknesses 1, 2, 3 and 4. We appreciate the thoughtful feedback and constructive nature of your comments.
> > >
> > > We agree that exploring additional downstream tasks would be valuable for future work. However, we consider the most important downstream tasks, namely regression and classification, to already be included and thoroughly evaluated. Beyond these tasks, KCMEs have other well-established applications (as discussed in the introduction), but all of them fundamentally depend on the KCME’s ability to approximate conditional expectations of functions of random variables. Our empirical results demonstrate that this property is preserved under compression, with RMSE serving as a natural error metric. Consequently, we expect our method to perform well in relevant downstream applications, which we see as an interesting avenue for future work.

---

### Official Review · Reviewer_XMB6 · 2025-07-03

**Clarity:** 3
**Significance:** 3
**Originality:** 3
**Rating:** 5
**Confidence:** 3

**Summary:**

The authors propose a novel distribution compression method specifically designed to compress conditional distributions, addressing a gap in existing techniques (e.g., Kernel Herding) that primarily focus on marginal or joint distributions. The method belongs to the kernel family, minimizing the maximum mean discrepancy (MMD) between kernel embeddings to yield a distilled dataset that best represents the original data. To measure differences between conditional distributions efficiently, they introduce the AMCMD metric. Two optimization approaches for MMD minimization are developed and compared: (a) A greedy algorithm (ACKH) with linear-time complexity (when using a specially designed estimator), (b) A non-greedy, joint-optimization method (ACKIP) with $O(m^3 + m^2 n)$ complexity, which is a factor of $m$ faster than ACKH, where $m$ is the cardinality of  a compressed dataset.

A conventional compression algorithm for marginal distributions is also extended to the joint distribution case to provide a fair comparison to the methods of the same family. An experimental evaluation is conducted to benchmark all the four approaches considered. The results suggest that the proposed conditional approach is superior.

**Questions:**

1. What are the contemporary neural approaches (or methods of other, non-kernel families) for distribution compression? How does your method performs in comparison to them?
2. Are there any kernel preferences specific to the task in question (kernel embeddings construction, distribution compression)? Why have not you tried other kernels?
3. How does your method scale with dimensionality (in theory and in practice)?

**Ethical Concerns:**

["NO or VERY MINOR ethics concerns only"]

**Final Justification:**

No more comments from my side, I will keep my rating at 5.

**Limitations:**

The authors discuss limitations of their method throughout the text. However, other, more practical limitations are overlooked; see **Weaknesses**.

**Quality:**

3

**Strengths And Weaknesses:**

**Strengths:**

This work addresses an important challenge in machine learning: reducing dataset size without significant quality degradation, which could enable faster model training and inference. The study is both comprehensive and mathematically rigorous, with clear presentation. Experiments on synthetic data and the superconductivity dataset demonstrate that the proposed conditional approach consistently outperforms joint distribution methods and random subsampling. Additionally, the Appendix provides extensive supporting details.

**Weaknesses:**

1. The background review focuses exclusively on kernel methods, omitting discussion of non-kernel approaches. The experimental comparison similarly fails to include non-kernel baselines, limiting the perspective on the method's relative advantages.
2. While distribution compression has critical applications in generative modeling, computer vision, speech, and NLP (domains characterized by high-dimensional data) the evaluation is conducted primarily in low-dimensional settings. The authors use only one real-world dataset and notably avoid addressing their method's scalability with increasing dimensionality.
3. The analysis concentrates on AMCMD and RMSE for test function expectation approximations, while downstream task performance (e.g., regression/classification accuracy using compressed data for training ML models) would provide more practical insight. This omission leaves the method's real-world utility unclear.
4. The argument for the consistency of the proposed AMCMD assumes that the distributions of $X^*$ and $X$ are equal (Corollary 4.4). However, the original definition emphasizes the use of a distribution different from $X$, rendering the argument non-rigorous.

---

> ### Author Rebuttal · Authors · 2025-07-28
>
> We thank the reviewer for their time spent reviewing this work and their valuable feedback.
>
> ## **Questions**
>
> ### **Question 1** - What are the contemporary neural or non-kernel approaches or distribution compression? How does your method perform in comparison?
>
> **The main contribution of this paper are the algorithms (ACKIP, ACKH) which compress the conditional distribution. We are not aware of any other algorithms in the literature that do this, kernel-based or otherwise**. To provide a baseline comparison, we modify Kernel Herding (KH) [R1] to target the joint distribution (JKH), and introduce a joint distribution compression algorithm (JKIP). We show ACKIP achieving superior empirical performance versus these baselines.
>
> In *non-conditional*, *non-joint* distribution compression, the most popular algorithms are KH [R1], Kernel Thinning (KT) [R2], Support Points (SP) [R3], and gradient flow methods (GF) [R6]. Here we adapt the approach of KH/SP/GF where the compressed set is a free parameter. In KT they restrict themselves to points that already exist in the target dataset. KH, KT and GF are kernel-based, targeting the MMD, whereas SP is an *energy distance (ED)* based method, defined as
>
> \begin{align*}
> ED(\mathbb{P}_X, \mathbb{Q}_X) := 2\mathbb{E}[\Vert a - b \Vert] - \mathbb{E}[\Vert a - a^\prime \Vert] + \mathbb{E}[\Vert b - b^\prime \Vert]\
> \end{align*}
>
> where $a, a^\prime \sim \mathbb{P}_X$, $b,b^\prime \sim \mathbb{Q}_X$, and $\Vert \cdot \Vert$ is the 2-norm. The ED is zero iff $\mathbb{P}_X =\mathbb{Q}_X$ (Theorem 1, [R3]). While SP does not initially appear to be kernel-based, the energy distance is equivalent to the MMD for a choice of negative-definite kernel function [R4].
>
> **The existing distribution compression methods are kernel-based. We are not aware of any conditional distribution compression approach, neural-based or otherwise.**
>
> ### **Question 2.1** - Are there any kernel preferences specific to the task in question?
>
> As stated in Theorem 4.1, the **kernel $l$ must be characteristic** [R1], this includes common choices such as the Gaussian and IMQ kernels. This property guarantees that zero AMCMD implies equality of the underlying conditional distributions.
>
> In our experiments, we use the Gaussian kernel, as it is a universally standard choice in the kernel embedding literature, and in kernel-based methods more broadly. However, we emphasise that our algorithms are **kernel-agnostic**: *any* feature kernel $k$ and *any characteristic* response kernel $l$ can be used, provided their gradients are computable.
>
> ### **Question 2.2** -  Why have not you tried other kernels?
>
> As with any kernel method, the optimal kernel choice is unknown and problem-specific, hence it should be selected with care by the practitioner to align with their inductive biases, see for example [R5]. We have used the Gaussian kernel as it is a *very common* default choice due to its ability to capture smooth similarities.
>
> The below table includes **new results** on the *Heteroscedastic* data using the **IMQ kernel** for $k$ and $l$. We report the 25th, 50th and 75th percentiles, with ACKIP retaining the best performance in terms of AMCMD and RMSE.
>
> |Method|AMCMD
> |-|-|
> |||
> |JKH|0.0349, 0.0422, 0.0502|
> |JKIP|0.0168, 0.0215, 0.0259|
> |ACKH|0.00443, 0.00697, 0.0191|
> |**ACKIP**| **0.00143, 0.00154, 0.00170**|
> |Random|0.0182, 0.0240, , 0.0336|
>
> | Method | **RMSE**||||
> |-|-|-|-|-|
> ||$h(y) = y$|$h(y) =\sin(y)$|$h(y) =\exp(-y^2)$|$h(y) =1_{\{y < 0\}}$|
> |JKH|0.206, 0.229, 0.266|0.161, 0.170, 0.200| 0.0913, 0.104, 0.115|0.136, 0.168, 0.205|
> |JKIP|0.127, 0.144, 0.218|0.156, 0.157, 0.219| 0.0887, 0.0899, 0.111|**0.116, 0.129**, 0.231|
> |ACKH|0.0892, 0.113, 0.192|0.143, 0.148, 0.150| 0.0882, 0.0918, 0.0988|0.132, 0.144, 0.150|
> |ACKIP|**0.0596, 0.0602, 0.0615**| **0.136, 0.137, 0.138**| **0.0874, 0.0878, 0.0886**|0.127, 0.134, **0.146**|
> |Random|0.134, 0.158, 0.199|0.158,  0.171, 0.193|0.102, 0.114, 0.131|0.138, 0.156, 0.187|
>
> Due to reviewer guidelines we cannot include figures, but **in the final version we will include plots of these new results**.
>
> ### **Question 3.1** - How does your method scale with dimensionality (in theory)?
>
> The below table shows the time and space complexity for each of the methods introduced in this work, assuming the feature space is $d$-dimensional, the response space is $p$-dimensional, the target dataset has size $n$, and the compressed set is size $m$:
>
> | Algorithm | Time Complexity                                   | Memory Complexity                      |
> |-----------|--------------------------------------------------|----------------------------------------|
> | JKH       | $O((m^2 + mn)(d+p))$                             | $O((m + n)(d+p))$                      |
> | JKIP      | $O((m^2 + mn)(d+p))$                             | $O((m^2 + mn)(d+p))$                   |
> | ACKH      | $O(m^4 + m^3n + (m + n)(d+p))$                   | $O((m + n)(d+p) + mn + m^2)$           |
> | ACKIP     | $O(m^3 + m^2n + (m^2 + mn)(d+p))$                | $O((m^2 + mn)(d+p))$
>
> One should notice that the **dependence on** $n$ **and** $d, p$ **is never more than $O(nm(d+p))$** for any of the methods in this work, which is quite favourable. **We will include this discussion in the final version**.
>
> ### **Question 3.2** - How does your method scale with dimensionality (in practice)?
> Our experiments cover datasets across a large range of dimensions. In the appendix we performed an experiment on mnIST data, which has **moderately high dimension of 784**. In particular, Figures 25 and 26 show that ACKIP achieves very similar classification accuracy to the full data, after 97\% compression.
>
> ## **Weaknesses**
>
> ### **Weakness 1.1** - The background review focuses exclusively on kernel methods ...
>
> We have seen in the answer to Question 1 that **there are no existing approaches to conditional distribution compression in the literature, kernel-based or otherwise**. Moreover, the existing standard distribution compression approaches of which we are aware are kernel-based [R1, R2, R3]. **We will make this clearer in the final version**.
>
> ### **Weakness 1.2** - The experimental comparison similarly fails to include non-kernel baselines, limiting the perspective on the method's relative advantages.
>
> We have tried to provide reasonable baselines, extending existing methods (KH, GF  [R1], [R6]) to target the joint distribution, and by providing metrics based on uniform random subsamples. **We are not aware of any other conditional distribution compression approach which we can compare against**.
>
> ### **Weakness 2** - While distribution compression has critical applications in ... the evaluation is conducted primarily in low-dimensional settings. The authors use only one real-world dataset and notably avoid addressing their method's scalability with increasing dimensionality.
>
> We have seen in the answer to Question 3 that **the appendix contains results for an additional real world data experiment based on mnIST, which has moderately high dimension of 784**. We have also presented a **table showing linear scaling in terms of memory and time complexity with respect to both the dimension of the feature and response space**.
>
> It is widely known that, like many other methods, kernel methods suffer from the curse of dimensionality, and so we would not expect, and do not claim, to have strong performance on data such as those involved in NLP where dimensionality can be much much higher. **We will make this point clearer in the final version**.
>
> ### **Weakness 3** - The analysis concentrates on AMCMD and RMSE for test function expectation approximations, while downstream task performance (e.g., regression/classification accuracy using compressed data for training ML models) would provide more practical insight. This omission leaves the method's real-world utility unclear.
>
> **The reviewer may not have noticed that the empirical results presented in the paper do indeed correspond to regression and classification accuracy.**
>
> To see this, note that the kernel conditional mean embedding (KCME) can be directly used as a regression or classification model. In particular we point the reviewers attention to Figure 4. When the test function $h(y)$ is chosen such that $h(y) = y$, the KCME is directly approximating the conditional expectation $\mathbb{E}[Y|X]$, which **is the object of interest in regression**. Moreover, for the setting of multi-class classification, in Figure 8 we report performance for test functions $h(y) = 1_{y=i}$, $i = 1, 2, 3, 4$. In this case we have that
> \begin{align*}
>         \mathbb{E}[1_{Y = i} \mid X] = \mathbb{P}(Y = i \mid X),
> \end{align*}
> and hence Figure 8 reports the error in estimating the class conditional probability, which is the **object of interest in classification**. In Figure 20, we further report the classification accuracy and F1 score.
>
> **We will make the connection between the KCME and regression/classification more clear in the final work**.
>
> ### **Weakness 4** - The argument for the consistency of the proposed AMCMD assumes that the distributions are equal. However, the original definition emphasizes the use of a different distribution .
>
> Corollary 4.4. is intended to show consistency exactly in the *special case* that $X = X^\*$. In particular, ACKH and ACKIP are designed for the case when $X = X^*$. **We will make this clearer in the final version**.
>
> ## **References**
> [R1] - Yutian Chen et al. “Super-Samples from Kernel Herding”. 2012
>
> [R2] - Raaz Dwivedi and Lester Mackey. "Kernel Thinning". 2024
>
> [R3] - Simon Mak and V. Roshan Joseph. "Support points". 2018
>
> [R4] - Dino Sejdinovic et al. “Equivalence of distance-based and RKHS-based statistics in hypothesis testing”.
> In: The Annals of Statistics 41.5. 2013
>
> [R5] - Thomas Hofmann et al. Kernel methods in machine learning. 2007
>
> [R6] - Michael Arbel, Anna Korba, Adil Salim, Arthur Gretton. Maximum Mean Discrepancy Gradient Flow. 2019

---

> > ### Author Response · Authors · 2025-08-06
> >
> > Should you have any further questions or concerns, please feel free to raise them. We hope our response has satisfactorily addressed your comments and will contribute positively to your overall evaluation.

---

> > ### Comment · Reviewer_XMB6 · 2025-08-07
> >
> > I would like to thank the authors for the response.
> >
> > **Q1 and W1:** In fact, I am not familiar with the current state of distribution compression methods, and can not suggest concrete examples of non-kernel approaches. However, as far as I know, there are many works on dataset distillation, which seem to address the same problem (handling conditional distributions implicitly via preserving task performance):
> >
> > > Dataset distillation (DD) is an increasingly important technique that focuses on constructing a synthetic dataset capable of capturing the core information in training data to achieve comparable performance in models trained on the latter. [c1]
> >
> > > Dataset distillation aims to create a compact dataset that retains essential information while maintaining model performance. [c2]
> >
> > > Dataset distillation (DD) aims to synthesize a small dataset whose test performance is comparable to a full dataset using the same model. [c3]
> >
> > It seems that there is at least one separate group of non-kernel methods which apply information theory to compress the distribution. Perhaps, there might be other approaches. I kindly ask the authors to clarify how their problem relates to dataset distillation. If it is indeed the same task, the literature overview and comparison should be extended significantly. If the tasks are different, this also should be clarified in the text.
> >
> > **Q3:**
> > Thank you for the computational complexity table. However, my concern is statistical sample complexity. Corollary 4.4 proves asymptotic consistency of AMCMD, but omits finite-sample error bounds. Specifically, what is the explicit convergence rate in $n$? How does dimensionality $d$ impact sample complexity, i.e., to what extent is AMCMD affected by the curse of dimensionality?
> >
> > **W3:**
> > I am interested more in regression/classification for some third variable $Z$. In other words, when $X$ and $Y$ are treated as different modalities of input features, and $Z$ is a target variable unknown at the time of distribution compression (e.g., $X$ and $Y$ are medical CT/MRI-scans and corresponding human-made text descriptions of abnormalities, and $Z$ represents a ground truth region of interest on the scan).
> >
> > Another interesting evaluation direction is training conditional generative models using compressed datasets, and assessing their performance in terms of LPIPS/FID/...
> >
> > Overall, what I meant to say is that the experiments should include metrics which can not be easily expressed as an expectation of some simple test function.
> >
> > [c1] Kungurtsev, V. et al. Dataset Distillation from First Principles: Integrating Core Information Extraction and Purposeful Learning. arXiv:2409.01410
> >
> > [c2] Ye L. et al. Information-Guided Diffusion Sampling for Dataset Distillation. arXiv:2507.04619
> >
> > [c3] Zhong X. et al. MIM4DD: Mutual Information Maximization for Dataset Distillation. Proc. of NeurIPS 2023.

---

> ### Author Response · Authors · 2025-08-07
>
> Thank you for taking the time to read our rebuttal, we appreciate the thoughtful feedback. We hope that the following will assuage some of your concerns.
>
> ### **Q1, W1**:
>
> We thank the reviewer for raising this important point. While we agree that dataset distillation and distribution compression share surface-level similarities, as both aim to reduce dataset size while preserving downstream performance, they are fundamentally distinct in goal.
>
> Dataset distillation generates a compressed set optimised to perform well on a specific model and task. These methods are *model-dependent* and preserve task-specific performance. In contrast, distribution compression is model-agnostic: a distributional discrepancy is targeted, independent of any downstream model. The aim is not to preserve performance on a particular model, but rather to preserve the distribution itself under compression.
>
> We appreciate the reviewer highlighting this point, as we recognise that the distinction may be unclear. We will add a discussion clarifying this in the related work section of the final version.
>
> ### **Q3**
>
> We appreciate your interest in the finite-sample behaviour of AMCMD. While Corollary 4.4 establishes asymptotic consistency, we agree that finite-sample error bounds are of practical importance. The convergence rate of AMCMD depends on two components: the convergence of the conditional kernel mean embeddings (CKMEs), and the convergence of the outer expectation.
>
> Under standard assumptions on the kernels, and with a suitable decay schedule for the regularisation parameter $\lambda$, CKMEs are known to converge at a dimension-free rate of $\mathcal{O}_p(n^{-1/4})$ [Theorems 4.4, 4.5, R1].
>
> The outer expectation term converges via a uniform law of large numbers. However, to the best of our knowledge, no explicit convergence rate is currently available. This remains an open question and an area of ongoing interest in our work.
>
> As with all kernel methods, performance in high-dimensional settings can degrade if the kernel becomes less discriminative, particularly if the lengthscale is poorly tuned. This issue can be mitigated through data preprocessing, such as dimensionality reduction or feature normalisation, which maintain the sensitivity of the kernel to variation in the data.
>
> Beyond this, the Manifold Hypothesis is a widely accepted principle in machine learning which suggests that high-dimensional data often lie near a low-dimensional manifold embedded in the ambient space [R2]. When data exhibit such low-dimensional structure, or are sparse or otherwise highly structured, the curse of dimensionality can be further alleviated by employing manifold-adapted kernels that respect the intrinsic geometry of the data [R3]. Such kernels can be readily integrated into our conditional distribution compression framework.
>
> Empirically, we find that the proposed method performs well in moderately high-dimensional settings, such as MNIST, where the input dimension is 784.
>
> ### **W3**
>
> We appreciate this thoughtful suggestion. If we have understood correctly, you are interested in settings where $X$ and $Y$ represent separate modalities of input features—for example, $X$ could be an image (e.g., a CT or MRI scan), $Y$ a text description (e.g., a radiologist’s report), and $Z$ the response of interest (e.g., a ground-truth segmentation or diagnosis). This scenario is indeed compatible with our framework.
>
> Specifically, one can define a kernel on the joint input space $(X, Y)$ by taking a tensor product of kernels defined on the individual spaces. This enables the conditional mean embedding $\mu_{Z|X,Y}$ to be estimated in a way that respects the structure of the multimodal feature space. Our method is then applicable here, provided that the resulting kernel is differentiable.
>
> We agree there are many interesting avenues for further evaluation. While KCMEs have well-established applications (as discussed in the introduction), all of them fundamentally rely on the KCME’s ability to estimate conditional expectations. Our empirical results show that this property is preserved under compression, suggesting strong downstream performance across diverse domains. Nonetheless, we agree that evaluating compressed sets on additional complex tasks, beyond regression and classification, which are already included, remains an important direction for future work.
>
> ### **References**
> [R1] - Junhyung Park, Krikamol Muandet. A Measure-Theoretic Approach to Kernel Conditional Mean Embeddings. 2020.
>
> [R2] - Nick Whiteley, Annie Gray, Patrick Rubin-Delanchy. Statistical exploration of the Manifold Hypothesis. 2022.
>
> [R3] - Sadeep Jayasumana, Richard Hartley, Mathieu Salzmann, Hongdong Li, Mehrtash Harandi. Kernel Methods on Riemannian Manifolds with Gaussian RBF Kernels. 2014.

---

### Note · Authors · 2025-08-11

We thank the reviewers for their time and for providing thoughtful, constructive feedback. In this final remark, we address several common concerns raised during the review proces.

### **Baseline and Comparison Methods**
Several reviewers expressed concern that the paper lacked comparison to non-kernel approaches to distribution compression. In our responses, we noted that, to the best of our knowledge, **no non-kernel methods exist for standard distribution compression**: the most widely used approaches [R1, R2, R3] are explicitly kernel-based, and the only other common method [R4] can be reformulated as a kernel-based approach [R5].

Importantly, the primary contribution of this work is the introduction of algorithms that compress **conditional distributions, and we are not aware of any other methods, kernel-based or otherwise, that address this problem**. To ensure fair comparison, we extended existing methods [R1, R3] to target the joint distribution. Our experiments show that ACKIP achieves superior empirical performance compared to these baselines, as well as to random subsampling.

### **Kernel Ablation**
Another common concern was that the experiments in the paper focused primarily on the Gaussian kernel. **We have addressed this by running an additional experiment using the IMQ kernel**. The results, given in a new table in the rebuttals, show that our methods maintain strong performance with this new kernel.

### **Downstream Tasks**
Finally, some reviewers requested additional results on regression or classification tasks. **We clarified that the empirical results in the paper already include both regression and classification as explicit downstream tasks**.

Moreover, we emphasised that other downstream tasks (as discussed in the introduction) rely on the KCME’s ability to approximate conditional expectations of functions of random variables. **Our empirical results show that this capability is preserved under compression**. These findings suggest strong downstream performance in other applications.

### **References**

[R1] - Yutian Chen et al. “Super-Samples from Kernel Herding”. 2012

[R2] - Raaz Dwivedi and Lester Mackey. "Kernel Thinning". 2024

[R3] - Michael Arbel et al. Maximum Mean Discrepancy Gradient Flow. 2019

[R4] - Simon Mak and V. Roshan Joseph. "Support points". 2018

[R5] - Dino Sejdinovic et al. “Equivalence of distance-based and RKHS-based statistics in hypothesis testing”. In: The Annals of Statistics 41.5. 2013

---

### Decision · Program_Chairs · 2025-09-17

**Decision:**

Accept (poster)

**Comment:**

The authors introduce kernel-based algorithms to produce compressed sets that approximate the joint or conditional distribution of a target dataset (extending the existing work on unlabeled datasets). Most of the concerns of the reviewers were addressed during the rebuttal (e.g., lack of non-kernel baselines, missing experiments) and now there is a consensus about accepting this paper.

I recommend acceptance, but given than the topic is niche this would be a weak accept.